# Communication-Efficient Federated Bilevel Optimization with Global and Local Lower Level Problems

**Junyi Li**
Computer Science
University of Maryland
College Park, MD 20742
junyili.ai@gmail.com

**Feihu Huang**
ECE
University of Pittsburgh
Pittsburgh, PA 15261
huangfeihu2018@gmail.com

**Heng Huang** [*]
Computer Science
University of Maryland
College Park, MD 20742
henghuanghh@gmail.com

## Abstract

Bilevel Optimization has witnessed notable progress recently with new emerging efficient algorithms. However, its application in the Federated Learning setting remains relatively underexplored, and the impact of Federated Learning's inherent challenges on the convergence of bilevel algorithms remain obscure. In this work, we investigate Federated Bilevel Optimization problems and propose a communication-efficient algorithm, named FedBiOAcc. The algorithm leverages an efficient estimation of the hyper-gradient in the distributed setting and utilizes the momentum-based variance-reduction acceleration. Remarkably, FedBiOAcc achieves a communication complexity $O(\epsilon^{-1})$, a sample complexity $O(\epsilon^{-1.5})$ and the linear speed up with respect to the number of clients. We also analyze a special case of the Federated Bilevel Optimization problems, where lower level problems are locally managed by clients. We prove that FedBiOAcc-Local, a modified version of FedBiOAcc, converges at the same rate for this type of problems. Finally, we validate the proposed algorithms through two real-world tasks: Federated Data-cleaning and Federated Hyper-representation Learning. Empirical results show superior performance of our algorithms.

## 1 Introduction

Bilevel optimization [51, 48] has increasingly drawn attention due to its wide-ranging applications in numerous machine learning tasks, including hyper-parameter optimization [42], meta-learning [61] and neural architecture search [36]. A bilevel optimization problem involves an upper problem and a lower problem, wherein the upper problem is a function of the minimizer of the lower problem. Recently, great progress has been made to solve this type of problems, particularly through the development of efficient single-loop algorithms that rely on diverse gradient approximation techniques [23]. However, the majority of existing bilevel optimization research concentrates on standard, non-distributed settings, and how to solve the bilevel optimization problems under distributed settings have received much less attention. Federated learning (FL) [40] is a recently promising distributed learning paradigm. In FL, a set of clients jointly solve a machine learning task under the coordination of a central server. To protect user privacy and mitigate communication overhead, clients perform multiple steps of local update before communicating with the server. A variety of algorithms [50, 59, 16, 26, 1] have been proposed to accelerate this training process. However, most of these algorithms primarily address standard single-level optimization problems. In this work, we study the bilevel optimization problems in the Federated Learning setting and investigate the

---

[*]This work was partially supported by NSF IIS 1838627, 1837956, 1956002, 2211492, CNS 2213701, CCF 2217003, DBI 2225775.

37th Conference on Neural Information Processing Systems (NeurIPS 2023).

Table 1: **Comparisons of the Federated/Non-federated bilevel optimization algorithms for finding an $\epsilon$-stationary point of** (1). $Gc(f, \epsilon)$ and $Gc(g, \epsilon)$ denote the number of gradient evaluations *w.r.t.* $f^{(m)}(x, y)$ and $g^{(m)}(x, y)$; $JV(g, \epsilon)$ denotes the number of Jacobian-vector products; $HV(g, \epsilon)$ is the number of Hessian-vector products; $\kappa = L/\mu$ is the condition number, $p(\kappa)$ is used when no dependence is provided. Sample complexities are measured by client.

| Setting | Algorithm | Communication | $Gc(f, \epsilon)$ | $Gc(g, \epsilon)$ | $JV(g, \epsilon)$ | $HV(g, \epsilon)$ | Heterogeneity |
|---|---|---|---|---|---|---|---|
| Non-Fed | StocBiO [24] | | $O(\kappa^5\epsilon^{-2})$ | $O(\kappa^9\epsilon^{-2})$ | $O(\kappa^5\epsilon^{-2})$ | $O(\kappa^6\epsilon^{-2})$ | |
| | MRBO [56] | | $O(p(\kappa)\epsilon^{-1.5})$ | $O(p(\kappa)\epsilon^{-1.5})$ | $O(p(\kappa)\epsilon^{-1.5})$ | $O(p(\kappa)\epsilon^{-1.5})$ | |
| Federated | CommFedBiO [33] | $O(p(\kappa)\epsilon^{-2})$ | $O(p(\kappa)\epsilon^{-2})$ | $O(p(\kappa)\epsilon^{-2})$ | $O(p(\kappa)\epsilon^{-2})$ | $O(p(\kappa)\epsilon^{-2})$ | ✓ |
| | FedNest [49] | $O(\kappa^9\epsilon^{-2})$ | $O(\kappa^5\epsilon^{-2})$ | $O(\kappa^9\epsilon^{-2})$ | $O(\kappa^5\epsilon^{-2})$ | $O(\kappa^9\epsilon^{-2})$ | ✓ |
| | AggITD [53] | $O(p(\kappa)\epsilon^{-2})$ | $O(p(\kappa)\epsilon^{-2})$ | $O(p(\kappa)\epsilon^{-2})$ | $O(p(\kappa)\epsilon^{-2})$ | $O(p(\kappa)\epsilon^{-2})$ | ✓ |
| | FedMBO [21] | $O(M^{-1}p(\kappa)\epsilon^{-2})$ | $O(M^{-1}p(\kappa)\epsilon^{-2})$ | $O(M^{-1}p(\kappa)\epsilon^{-2})$ | $O(M^{-1}p(\kappa)\epsilon^{-2})$ | $O(M^{-1}p(\kappa)\epsilon^{-2})$ | ✓ |
| | SimFBO [58] | $O(p(\kappa)\epsilon^{-1})$ | $O(M^{-1}p(\kappa)\epsilon^{-2})$ | $O(M^{-1}p(\kappa)\epsilon^{-2})$ | $O(M^{-1}p(\kappa)\epsilon^{-2})$ | $O(M^{-1}p(\kappa)\epsilon^{-2})$ | ✓ |
| | Local-BSGVR [12] | $O(p(\kappa)\epsilon^{-1})$ | $O(M^{-1}p(\kappa)\epsilon^{-1.5})$ | $O(M^{-1}p(\kappa)\epsilon^{-1.5})$ | $O(M^{-1}p(\kappa)\epsilon^{-1.5})$ | $O(M^{-1}p(\kappa)\epsilon^{-1.5})$ | ✗ |
| | **FedBiOAcc (Ours)** | $\boldsymbol{O(\kappa^{19/3}\epsilon^{-1})}$ | $\boldsymbol{O(M^{-1}\kappa^8\epsilon^{-1.5})}$ | $\boldsymbol{O(M^{-1}\kappa^8\epsilon^{-1.5})}$ | $\boldsymbol{O(M^{-1}\kappa^8\epsilon^{-1.5})}$ | $\boldsymbol{O(M^{-1}\kappa^8\epsilon^{-1.5})}$ | ✓ |

following research question: *Is it possible to develop communication-efficient federated algorithms tailored for bilevel optimization problems that also ensure a rapid convergence rate?*

More specifically, a general Federated Bilevel Optimization problem has the following form:

$$\min_{x\in\mathbb{R}^p} h(x) := \frac{1}{M}\sum_{m=1}^{M} f^{(m)}(x, y_x), \text{ s.t. } y_x = \arg\min_{y\in\mathbb{R}^d} \frac{1}{M}\sum_{m=1}^{M} g^{(m)}(x, y) \qquad (1)$$

A federated bilevel optimization problem consists of an upper and a lower level problem, the upper problem $f(x, y) := \frac{1}{M}\sum_{m=1}^{M} f^{(m)}(x, y)$ relies on the solution $y_x$ of the lower problem, and $g(x, y) := \frac{1}{M}\sum_{m=1}^{M} g^{(m)}(x, y)$. Meanwhile, both the upper and the lower level problems are federated: In Eq.(1), we have $M$ clients, and each client has a local upper problem $f^{(m)}(x, y)$ and a lower level problem $g^{(m)}(x, y)$. Compared to single-level federated optimization problems, the estimation of the hyper-gradient in federated bilevel optimization problems is much more challenging. In Eq.(1), the hyper-gradient is not linear *w.r.t* the local hyper-gradients of clients, whereas the gradient of a single-level Federated Optimization problem is the average of local gradients. Consequently, directly applying the vanilla local-sgd method [40] to federated bilevel problems results in a large bias. In the literature [49, 33, 21, 53], researchers evaluate the hyper-gradient through multiple rounds of client-server communication, however, this approach leads to high communication overhead. In contrast, we view the hyper-gradient estimation as solving a quadratic federated problem and solving it with the local-sgd method. More specifically, we formulate the solution of the federated bilevel optimization as three intertwined federated problems: the upper problem, the lower problem and the quadratic problem for the hyper-gradient estimation. Then we address the three problems using alternating gradient descent steps, furthermore, to manage the noise of the stochastic gradient and obtain the fast convergence rate, we employ a momentum-based variance reduction technique.

Beyond the standard federated bilevel optimization problem as defined in Eq. 1, another variant of Federated Bilevel Optimization problem, which entails locally managed lower-level problems, is also frequently utilized in practical applications. For this type of problem, we can get an unbiased estimate of the global hyper-gradient using local hyper-gradient, thus we can solve it with a local-SGD like algorithm, named FedBiOAcc-Local. However, it is challenging to analyze the convergence of the algorithm. In particular, we need to bound the intertwined client drift error, which is intrinsic to FL and the bilevel-related errors *e.g.* the lower level solution bias. In fact, we prove that the FedBiOAcc-Local algorithm attains the same fast rate as FedBiO algorithm.

Finally, we highlight the main **contributions** of our paper as follows:

1. We propose FedBiOAcc to solve Federated Bilevel Optimization problems, the algorithm evaluates the hypergradient of federated bilevel optimization problems efficiently and achieves optimal convergence rate through momentum-based variance reduction. FedBiOAcc has sample complexity of $O(\epsilon^{-1.5})$, communication complexity of $O(\epsilon^{-1})$ and achieves linear speed-up *w.r.t* the number of clients.
2. We study Federated Bilevel Optimization problem with local lower level problem for the first time, where we show the convergence of a modified version of FedBiOAcc, named FedBiOAcc-Local for this type of problems.
3. We validate the efficacy of the proposed FedBiOAcc algorithm through two real-world tasks: Federated Data Cleaning and Federated Hyper-representation Learning.

**Notations** $\nabla$ denotes full gradient, $\nabla_x$ denotes partial derivative for variable x, higher order derivatives follow similar rules. $[K]$ represents the sequence of integers from 1 to $K$, $\bar{x}$ represents average of the sequence of variables $\{x^{(m)}\}_{m=1}^M$. $\bar{t}_s$ represents the global communication timestamp $s$.

## 2 Related Works

Bilevel optimization dates back to at least the 1960s when [51] proposed a regularization method, and then followed by many research works [10, 48, 55, 45], while in machine learning community, similar ideas in the name of implicit differentiation were also used in Hyper-parameter Optimization [30, 3, 2, 8]. Early algorithms for Bilevel Optimization solved the accurate solution of the lower problem for each upper variable. Recently, researchers developed algorithms that solve the lower problem with a fixed number of steps, and use the 'back-propagation through time' technique to compute the hyper-gradient [9, 39, 11, 43, 47]. Very Recently, it witnessed a surge of interest in using implicit differentiation to derive single loop algorithms [14, 17, 23, 28, 4, 56, 19, 32, 7, 20, 18]. In particular, [32, 7] proposes a way to iteratively evaluate the hyper-gradients to save computation. In this work, we view the hyper-gradient estimation of Federated Bilevel Optimization as solving a quadratic federated optimization problem and use a similar iterative evaluation rule as [32, 7] in local update.

The bilevel optimization problem is also considered in the more general settings. For example, bilevel optimization with multiple lower tasks is considered in [15], furthermore, [5, 57, 38, 13] studies the bilevel optimization problem in the decentralized setting, [25] studies the bilevel optimization problem in the asynchronous setting. In contrast, we study bilevel optimization problems under Federated Learning [40] setting. Federated learning is a promising privacy-preserving learning paradigm for distributed data. Compared to traditional data-center distributed learning, Federated Learning poses new challenges including data heterogeneity, privacy concerns, high communication cost, and unfairness. To deal with these challenges, various methods [26, 35, 46, 60, 41, 34] are proposed. However, bilevel optimization problems are less investigated in the federated learning setting. [54] considered the distributed bilevel formulation, but it needs to communicate the Hessian matrix for every iteration, which is computationally infeasible. More recently, FedNest [49] has been proposed to tackle the general federated nest problems, including federated bilevel problems. However, this method evaluates the full hyper-gradient at every iteration; this leads to high communication overhead; furthermore, FedNest also uses SVRG to accelerate the training. Similar works that evaluate the hyper-gradient with multiple rounds of client-server communication are [33, 21, 53, 58]. Finally, there is a concurrent work [12] that investigates the possibility of local gradients on Federated Bilevel Optimization, however, it only considers the homogeneous case, this setting is quite constrained and much simpler than the more general heterogeneous case we considered. Furthermore, [12] only considers the case where both the upper and the lower problem are federated, and omit the equally important case where the lower level problem is not federated.

## 3 Federated Bilevel Optimization

### 3.1 Some Mild Assumptions

Note that the formulation of Eq.(1) is very general, and we consider the stochastic heterogeneous case in this work. More specifically, we assume:

$$f^{(m)}(x,y) := \mathbb{E}_{\xi \sim \mathcal{D}_f^{(m)}}[f^{(m)}(x,y,\xi)], g^{(m)}(x,y) := \mathbb{E}_{\xi \sim \mathcal{D}_g^{(m)}}[g^{(m)}(x,y;\xi)]$$

where $\mathcal{D}_f^{(m)}$ and $\mathcal{D}_g^{(m)}$ are some probability distributions. Furthermore, we assume the local objectives could be potentially different: $f^{(m)}(x,y) \neq f^{(k)}(x,y)$ or $g^{(m)}(x,y) \neq g^{(k)}(x,y)$ for $m \neq k, m, k \in [M]$. Furthermore, we assume the following assumptions in our subsequent discussion:

**Assumption 3.1.** Function $f^{(m)}(x,y)$ is possibly non-convex and $g^{(m)}(x,y)$ is $\mu$-strongly convex *w.r.t* $y$ for any given $x$.

**Assumption 3.2.** Function $f^{(m)}(x,y)$ is $L$-smooth and has $C_f$-bounded gradient;

**Assumption 3.3.** Function $g^{(m)}(x,y)$ is $L$-smooth, and $\nabla_{xy}g^{(m)}(x,y)$ and $\nabla_{y^2}g^{(m)}(x,y)$ are Lipschitz continuous with constants $L_{xy}$ and $L_{y^2}$ respectively;

---
**Algorithm 1** Accelerated Federated Bilevel Optimization (**FedBiOAcc**)
---
1: **Input:** Constants $c_\omega, c_\nu, c_u, \gamma, \eta, \tau, r$; learning rate schedule $\{\alpha_t\}$, $t \in [T]$, initial state $(x_1, y_1, u_1)$;
2: **Initialization:** Set $y_1^{(m)} = y_1, x_1^{(m)} = x_1, u_1^{(m)} = u_1, \omega_1^{(m)} = \nabla_y g^{(m)}(x_1, y_1, \mathcal{B}_y), \nu_1^{(m)} = \nabla_x f^{(m)}(x_1, y_1; \mathcal{B}_{f,1}) - \nabla_{xy} g^{(m)}(x_1, y_1; \mathcal{B}_{g,1})u_1$ and $q_1 = \nabla_{y^2} g^{(m)}(x_1^{(m)}, y_1^{(m)}; \mathcal{B}_{g,2})u_1 - \nabla_y f^{(m)}(x_1^{(m)}, y_1^{(m)}; \mathcal{B}_{f,2})$ for $m \in [M]$
3: **for** $t = 1$ **to** $T$ **do**
4:   $\hat{y}_{t+1}^{(m)} = y_t^{(m)} - \gamma \alpha_t \omega_t^{(m)}, \hat{x}_{t+1}^{(m)} = x_t^{(m)} - \eta \alpha_t \nu_t^{(m)}, \hat{u}_{t+1}^{(m)} = \mathcal{P}_r(u_t^{(m)} - \tau \alpha_t q_t^{(m)})$
5:   **if** $t \bmod I = 0$ **then**
6:     $y_{t+1}^{(m)} = \frac{1}{M} \sum_{j=1}^{M} \hat{y}_{t+1}^{(j)}; x_{t+1}^{(m)} = \frac{1}{M} \sum_{j=1}^{M} \hat{x}_{t+1}^{(j)}, u_{t+1}^{(m)} = \frac{1}{M} \sum_{j=1}^{M} \hat{u}_{t+1}^{(j)}$
7:   **else**
8:     $y_{t+1}^{(m)} = \hat{y}_{t+1}^{(m)}, x_{t+1}^{(m)} = \hat{x}_{t+1}^{(m)}, u_{t+1}^{(m)} = \hat{u}_{t+1}^{(m)}$
9:   **end if**
10:   Get $\hat{\omega}_{t+1}^{(m)}, \hat{\nu}_{t+1}^{(m)}$ and $\hat{q}_{t+1}^{(m)}$ following Eq. (7)
11:   **if** $t \bmod I = 0$ **then**
12:     $\omega_{t+1}^{(m)} = \frac{1}{M} \sum_{j=1}^{M} \hat{\omega}_{t+1}^{(j)}, \nu_{t+1}^{(m)} = \frac{1}{M} \sum_{j=1}^{M} \hat{\nu}_{t+1}^{(j)}, q_{t+1}^{(m)} = \frac{1}{M} \sum_{j=1}^{M} \hat{q}_{t+1}^{(j)},$
13:   **else**
14:     $\omega_{t+1}^{(m)} = \hat{\omega}_{t+1}^{(m)}, \nu_{t+1}^{(m)} = \hat{\nu}_{t+1}^{(m)}, q_{t+1}^{(m)} = \hat{q}_{t+1}^{(m)}$
15:   **end if**
16: **end for**
---

**Assumption 3.4.** We have unbiased stochastic first-order and second-order gradient oracle with bounded variance.

**Assumption 3.5.** For any $m, j \in [M]$ and $z = (x, y)$, we have: $\|\nabla f^{(m)}(z) - \nabla f^{(j)}(z)\| \leq \zeta_f$, $\|\nabla g^{(m)}(z) - \nabla g^{(j)}(z)\| \leq \zeta_g, \|\nabla_{xy} g^{(m)}(z) - \nabla_{xy} g^{(j)}(z)\| \leq \zeta_{g,xy}, \|\nabla_{y^2} g^{(m)}(z) - \nabla_{y^2} g^{(j)}(z)\| \leq \zeta_{g,yy}$, where $\zeta_f, \zeta_g, \zeta_{g,xy}, \zeta_{g,yy}$, are constants.

As stated in The assumption 3.1, we study the **non-convex-strongly-convex** bilevel optimization problems, this class of problems is widely studied in the non-distributed bilevel literature [22, 14]. Furthermore, Assumption 3.2 and Assumption 3.3 are also standard assumptions made in the non-distributed bilevel literature. Assumption 3.4 is widely used in the study of stochastic optimization problems. For Assumption 3.5, gradient difference is widely used in single level Federated Learning literature as a measure of client heterogeneity [28, 52]. Please refer to the full version of Assumptions in Appendix.

### 3.2 The FedBiOAcc Algorithm

A major difficulty in solving a Federated Bilevel Optimization problem Eq. (1) is **evaluating the hyper-gradient $\nabla h(x)$**. For the function class (non-convex-strongly-convex) we consider, the explicit form of hypergradient $h(x)$ exists as $\nabla h(x) = \Phi(x, y_x)$, where $\Phi(x, y)$ is denoted as:

$$\Phi(x, y) = \nabla_x f(x, y) - \nabla_{xy} g(x, y) \times [\nabla_{y^2} g(x, y)]^{-1} \nabla_y f(x, y), \tag{2}$$

Based on Assumption 3.1∼3.3, we can verify $\Phi(x, y_x)$ is the hyper-gradient [14]. But since the clients only have access to their local data, for $\forall m \in [M]$, the client evaluates:

$$\Phi^{(m)}(x, y) = \nabla_x f^{(m)}(x, y) - \nabla_{xy} g^{(m)}(x, y) \times [\nabla_{y^2} g^{(m)}(x, y)]^{-1} \nabla_y f^{(m)}(x, y), \tag{3}$$

It is straightforward to verify that $\Phi^{(m)}(x, y)$ is not an unbiased estimate of the full hyper-gradient, *i.e.* $\Phi(x, y_x) \neq \frac{1}{M} \sum_{m=1}^{M} \Phi^{(m)}(x, y_x)$. To address this difficulty, we can view the **Hyper-gradient computation as the process of solving a federated optimization problem**.

In fact, Evaluating Eq. (2) is equivalent to the following two steps: first, we solve the quadratic federated optimization problem $l(u)$:

$$\min_{u \in \mathbb{R}^d} l(u) = \frac{1}{M} \sum_{m=1}^{M} u^T (\nabla_{y^2} g^{(m)}(x, y))u - \langle \nabla_y f^{(m)}(x, y), u \rangle \tag{4}$$

Suppose that we denote the solution of the above problem as $u^*$, then we have the following linear operation to get the hypergradient:

$$\nabla h(x) = \frac{1}{M} \sum_{m=1}^{M} \left( \nabla_x f^{(m)}(x, y_x) - \nabla_{xy} g^{(m)}(x, y_x) u^* \right) \tag{5}$$

Compared to the formulation Eq. (2), Eq. (4) and Eq. (5) are more suitable for the distributed setting. In fact, both Eq. (4) and Eq. (5) have a linear structure. Eq. (4) is a (single-level) quadratic federated optimization problem, and we could solve Eq. (4) through local-sgd [40], suppose that each client maintains a variable $u_t^{(m)}$, and performs the following update:

$$u_{t+1}^{(m)} = \mathcal{P}_r(u_t^{(m)} - \tau_t \nabla l^{(m)}(u_t^{(m)}; \mathcal{B}))$$

$$\nabla l^{(m)}(u_t^{(m)}; \mathcal{B}) = \nabla_{y^2} g^{(m)}(x_t^{(m)}, y_t^{(m)}; \mathcal{B}_{g,2})) u_t^{(m)} - \nabla_y f^{(m)}(x_t^{(m)}, y_t^{(m)}; \mathcal{B}_{f,2})$$

where $\nabla l^{(m)}(u_t^{(m)}; \mathcal{B})$ is client $m$'s the stochastic gradient of Eq. (4), and $(x_t^{(m)}, y_t^{(m)})$ denotes the upper and lower variable state at the timestamp $t$, the $\mathcal{P}_r(\cdot)$ denotes the projection to a bounded ball of radius-$r$. Note that Clients perform multiple local updates of $u_t^{(m)}$ before averaging. As for Eq. (5), each client evaluates $\nabla h^{(m)}(x)$ locally: $\nabla h^{(m)}(x) = \nabla_x f^{(m)}(x, y_x) - \nabla_{xy} g^{(m)}(x, y_x) u^*$ and the server averages $\nabla h^{(m)}(x)$ to get $\nabla h(x)$. In summary, the linear structure of Eq. (4) and Eq. (5) makes it suitable for local updates, therefore, reduce the communication cost.

More specifically, we perform alternative update of upper level variable $x_t^{(m)}$, the lower level variable $y_t^{(m)}$ and hyper-gradient computation variable $u_t^{(m)}$. For example, for each client $m \in [M]$, we perform the following local updates:

$$y_{t+1}^{(m)} = y_t^{(m)} - \gamma_t \nabla_y g^{(m)}(x_t^{(m)}, y_t^{(m)}, \mathcal{B}_y), \ u_{t+1}^{(m)} = \mathcal{P}_r(u_t^{(m)} - \tau_t \nabla l^{(m)}(u_t^{(m)}; \mathcal{B}))$$

$$x_{t+1}^{(m)} = x_t^{(m)} - \eta_t \left( \nabla_x f^{(m)}(x_t^{(m)}, y_t^{(m)}; \mathcal{B}_{f,1}) - \nabla_{xy} g^{(m)}(x_t^{(m)}, y_t^{(m)}; \mathcal{B}_{g,1}) u_t^{(m)} \right) \tag{6}$$

Every $I$ steps, the server averages clients' local states, this resembles the local-sgd method for single level federated optimization problems. Note that in the update of the upper variable $x_t^{(m)}$, we use $u_t^{(m)}$ as an estimation of $u^*$ in Eq. (5). An algorithm follows Eq. (6) is shown in Algorithm 2 of Appendix and we refer to it as FedBiO.

**Comparison with FedNest.** The update rule of Eq. 6 is very different from that of FedNest [49] and its follow-ups [21, 53]. In FedNest, a sub-routine named FedIHGP is used to evaluate Eq. (2) at every global epoch. This involves multiple rounds of client-server communication and leads to higher communication overhead. In contrast, Eq. (6) formulates the hyper-gradient estimation as an quadratic federated optimization problem, and then solves three intertwined federated problems through alternative updates of $x$, $y$ and $u$.

Note that Eq. 6 updates the related variables through vanilla gradient descent steps. In the non-federated setting, gradient-based methods such as stocBiO [23] requires large-batch size ($O(\epsilon^{-1})$) to reach an $\epsilon$-stationary point, and we also analyze Algorithm 2 in Appendix to show the same dependence. To control the noise and remove the dependence over large batch size, we apply the momentum-based variance-reduction technique STORM [6]. In fact, Eq. (6) solves three intertwined optimization problems: the bilevel problem $h(x)$, the lower level problem $g(x, y)$ and the hyper-gradient computation problem Eq (4). So we control the noise in the process of solving each of the three problems. More specifically, we have $\omega_t^{(m)}$, $\nu_t^{(m)}$ and $q_t^{(m)}$ to be the momentum estimator for $x_t^{(m)}$, $y_t^{(m)}$ and $u_t^{(m)}$ respectively, and we update them following the rule of STORM [6]:

$$\hat{\omega}_{t+1}^{(m)} = \nabla_y g^{(m)}(x_{t+1}^{(m)}, y_{t+1}^{(m)}, \mathcal{B}_y) + (1 - c_\omega \alpha_t^2)(\omega_t^{(m)} - \nabla_y g^{(m)}(x_t^{(m)}, y_t^{(m)}, \mathcal{B}_y))$$

$$\hat{\nu}_{t+1}^{(m)} = \left( \nabla_x f^{(m)}(x_{t+1}^{(m)}, y_{t+1}^{(m)}; \mathcal{B}_{f,1}) - \nabla_{xy} g^{(m)}(x_{t+1}^{(m)}, y_{t+1}^{(m)}; \mathcal{B}_{g,1}) u_{t+1}^{(m)} \right)$$

$$\qquad + (1 - c_\nu \alpha_t^2) \left( \nu_t^{(m)} - \left( \nabla_x f^{(m)}(x_t^{(m)}, y_t^{(m)}; \mathcal{B}_{f,1}) - \nabla_{xy} g^{(m)}(x_t^{(m)}, y_t^{(m)}; \mathcal{B}_{g,1}) u_t^{(m)} \right) \right)$$

$$\hat{q}_{t+1}^{(m)} = \left( \nabla_{y^2} g^{(m)}(x_{t+1}^{(m)}, y_{t+1}^{(m)}; \mathcal{B}_{g,2}) u_{t+1}^{(m)} - \nabla_y f^{(m)}(x_{t+1}^{(m)}, y_{t+1}^{(m)}; \mathcal{B}_{f,2}) \right)$$

$$\qquad + (1 - c_u \alpha_t^2) \left( q_t^{(m)} - \left( \nabla_{y^2} g^{(m)}(x_t^{(m)}, y_t^{(m)}; \mathcal{B}_{g,2}) u_t^{(m)} - \nabla_y f^{(m)}(x_t^{(m)}, y_t^{(m)}; \mathcal{B}_{f,2}) \right) \right) \tag{7}$$

where $c_\omega$, $c_\nu$ and $c_u$ are constants, $\alpha_t$ is the learning rate. Then we update the $x_t^{(m)}$, $y_t^{(m)}$ and $u_t^{(m)}$ as follows:

$$\hat{y}_{t+1}^{(m)} = y_t^{(m)} - \gamma\alpha_t\omega_t^{(m)}, \hat{x}_{t+1}^{(m)} = x_t^{(m)} - \eta\alpha_t\nu_t^{(m)}, \hat{u}_{t+1}^{(m)} = \mathcal{P}_r(u_t^{(m)} - \tau\alpha_t q_t^{(m)}) \tag{8}$$

where $\gamma$, $\eta$, $\tau$ are constants and $\alpha_t$ is the learning rate. The FedBiOAcc algorithm following Eq. (8) is summarized in Algorithm 1. As shown in line 6 and 12 of Algorithm 1, Every $I$ iterations, we average both variables and the momentum.

## 3.3 Convergence Analysis

In this section, we study the convergence property for the FedBiOAcc algorithm. For any $t \in [T]$, we define the following virtual sequence:

$$\bar{x}_t = \frac{1}{M}\sum_{m=1}^{M} x_t^{(m)}, \bar{y}_t = \frac{1}{M}\sum_{m=1}^{M} y_t^{(m)}, \bar{u}_t = \frac{1}{M}\sum_{m=1}^{M} u_t^{(m)}$$

we denote the average of the momentum similarly as $\bar{\omega}_t$, $\bar{\nu}_t$ and $\bar{q}_t$. Then we consider the following Lyapunov function $\mathcal{G}_t$:

$$\mathcal{G}_t = h(\bar{x}_t) + \frac{18\eta\tilde{L}^2}{\mu\gamma}(\|\bar{y}_t - y_{\bar{x}_t}\|^2 + \|\bar{u}_t - u_{\bar{x}_t}\|^2) + \frac{9bM\eta}{64\alpha_t}\|\bar{\omega}_t - \frac{1}{M}\sum_{m=1}^{M}\nabla_y g^{(m)}(x_t^{(m)}, y_t^{(m)})\|^2$$

$$+ \frac{9bM\eta}{64\alpha_t}\|\bar{q}_t - \frac{1}{M}\sum_{m=1}^{M}(\nabla_{y^2} g^{(m)}(x_t^{(m)}, y_t^{(m)})u_t^{(m)} - \nabla_y f^{(m)}(x_t^{(m)}, y_t^{(m)})))\|^2$$

$$+ \frac{9bM\eta}{64\alpha_t}\|\bar{\nu}_t - \frac{1}{M}\sum_{m=1}^{M}(\nabla_x f^{(m)}(x_t^{(m)}, y_t^{(m)}) - \nabla_{xy} g^{(m)}(x_t^{(m)}, y_t^{(m)})u_t^{(m)})\|^2 \tag{9}$$

where $y_{\bar{x}_t}$ denotes the solution of the lower level problem $g(\bar{x}_t, \cdot)$, $u_{\bar{x}_t} = [\nabla_{y^2} g(\bar{x}, y_{\bar{x}})]^{-1}\nabla_y f(\bar{x}, y_{\bar{x}})$ denotes the solution of Eq (4) at state $\bar{x}_t$. Besides, $\gamma, \eta, \tau$ are learning rates and $L, \tilde{L}$ are constants. Note that the first three terms of $\mathcal{G}_t$: $h(\bar{x}_t)$, $\|\bar{y}_t - y_{\bar{x}_t}\|^2$, $\|\bar{u}_t - u_{\bar{x}_t}\|^2$ measures the errors of three federated problems: the upper level problem, the lower level problem and the hyper-gradient estimation. Then the last three terms measure the estimation error of the momentum variables: $\bar{\omega}_t$, $\bar{\nu}_t$ and $\bar{q}_t$. The convergence proof primarily concentrates on bounding these errors, please see Lemma C.2 - C.6 in the Appendix for more details. Meanwhile, as in the single level federated optimization problems, local updates lead to client-drift error. More specifically, we need to bound $\|x_t^{(m)} - \bar{x}_t\|^2$, $\|y_t^{(m)} - \bar{y}_t\|^2$ and $\|u_t^{(m)} - \bar{u}_t\|^2$, please see Lemma C.7 - C.11 for more details. Finally, we have the following convergence theorem:

**Theorem 3.6.** *Suppose in Algorithm 1, we choose learning rate* $\alpha_t = \frac{\delta}{(u+t)^{1/3}}, t \in [T]$, *for some constant* $\delta$ *and* $u$, *and let* $c_\nu$, $c_\omega$, $c_u$ *choose some value,* $\eta$, $\gamma$ *and* $\tau$, $r$ *be some small values decided by the Lipschitz constants of* $h(x)$, *we choose the minibatch size to be* $b_x = b_y = b$ *and the first batch to be* $b_1 = O(Ib)$, *then we have:*

$$\frac{1}{T}\sum_{t=1}^{T-1}\mathbb{E}\left[\|\nabla h(\bar{x}_t)\|^2\right] = O\left(\frac{\kappa^{19/3}I}{T} + \frac{\kappa^{16/3}}{(bMT)^{2/3}}\right)$$

*To reach an $\epsilon$-stationary point, we need* $T = O(\kappa^8 (bM)^{-1}\epsilon^{-1.5})$, $I = O(\kappa^{5/3}(bM)^{-1}\epsilon^{-0.5})$.

As stated in the Theorem, to reach an $\epsilon$-stationary point, we need $T = O(\kappa^8(bM)^{-1}\epsilon^{-1.5})$, then the sample complexity for each client is $Gc(f, \epsilon) = O(M^{-1}\kappa^8\epsilon^{-1.5})$, $Gc(g, \epsilon) = O(M^{-1}\kappa^8\epsilon^{-1.5})$, $Jv(g, \epsilon) = O(M^{-1}\kappa^8\epsilon^{-1.5})$, $Hv(g, \epsilon) = O(M^{-1}\kappa^8\epsilon^{-1.5})$. So FedBiOAcc achieves the linear speed up *w.r.t.* to the number of clients $M$. Next, suppose we choose $I = O(\kappa^{5/3}(bM)^{-1}\epsilon^{-0.5})$, then the number of communication round $E = O(\kappa^{19/3}\epsilon^{-1})$. This matches the optimal communication complexity of the single level optimization problems as in the STEM [27]. Furthermore, compared to FedNest and its variants, FedBiOAcc has improved both the communication complexity and the iteration complexity. As for LocalBSCVR [12], FedBiOAcc obtains same rate, but incorporates the heterogeneous case. Note that it is much more challenging to analyze the heterogeneous case. In fact, if we assume homogeneous clients, we have local hyper-gradient (Eq. (3)) equals the global hyper-gradient (Eq. (2)), then we do not need to use the quadratic federated optimization problem view in Section 3.2, while the theoretical analysis is also simplified significantly.

# 4   Federated Bilevel Optimization with Local Lower Level Problems

In this section, we consider an alternative formulation of the Federated Bilevel Optimization problems as follows:

$$\min_{x \in \mathbb{R}^p} h(x) \coloneqq \frac{1}{M} \sum_{m=1}^{M} f^{(m)}(x, y_x^{(m)}), \text{ s.t. } y_x^{(m)} = \arg\min_{y \in \mathbb{R}^d} g^{(m)}(x, y) \tag{10}$$

Same as Eq. (1), Eq. (10) has a federated upper level problem, however, Eq. (10) has a unique lower level problem for each client, which is different from Eq. (1). In fact, federated bilevel optimization problem Eq (10) can be viewed as a special type of standard federated learning problems. If we denote $h^{(m)}(x) = f^{(m)}(x, y_x^{(m)})$, then Eq. (10) can be written as $\min_{x \in \mathbb{R}^p} h(x) \coloneqq \frac{1}{M} \sum_{m=1}^{M} h^{(m)}(x)$. But due to the bilevel structure of $h^{(m)}(x)$, Eq. (10) is more challenging than the standard Federated Learning problems.

**Hyper-gradient Estimation.** Assume Assumption 3.1∼Assumption 3.3 hold, then the hyper-gradient is $\Phi(x, y_x) = \frac{1}{M} \sum_{m=1}^{M} \Phi^{(m)}(x, y_x)$, where $\Phi^{(m)}(x, y)$ is defined in Eq. (3), in other words, the local hyper-gradient $\Phi^{(m)}(x, y)$ is an unbiased estimate of the full hyper-gradient. This fact makes it possible to solve Eq. (10) with local-sgd like methods. More specifically, we solve the local bilevel problem $h^{(m)}(x)$ multiple steps on each client and then the server averages the local states from clients. Please refer to Algorithm 3 and the variance-reduction acceleration Algorithm 4 in the Appendix. For ease of reference, we name them FedBiO-Local and FedBiOAcc-Local, respectively.

Several challenges exist in analyzing FedBiO-Local and FedBiOAcc-Local. First, Eq. (3) involves Hessian inverse, so we only evaluate it approximately through the Neumann series [37] as:

$$\Phi^{(m)}(x, y; \xi_x) = \nabla_x f^{(m)}(x, y; \xi_f) - \tau \nabla_{xy} g^{(m)}(x, y; \xi_g)$$
$$\times \sum_{q=-1}^{Q-1} \prod_{j=Q-q}^{Q} (I - \tau \nabla_{y^2} g^{(m)}(x, y; \xi_j)) \nabla_y f^{(m)}(x, y; \xi_f) \tag{11}$$

where $\xi_x = \{\xi_j (j = 1, \ldots, Q), \xi_f, \xi_g\}$, and we assume its elements are mutually independent. $\Phi^{(m)}(x, y; \xi_x)$ is a biased estimate of $\Phi^{(m)}(x, y)$, but with bounded bias and variance (Please see Proposition D.2 for more details.) Furthermore, to reduce the computation cost, each client solves the local lower level problem approximately and we update the upper and lower level variable alternatively. The idea of alternative update is widely used in the non-distributed bilevel optimization [23, 56]. However, in the federated setting, client variables drift away when performing multiple local steps. As a result, the variable drift error and the bias caused by inexact solution of the lower level problem intertwined with each other. For example, in the local update, clients optimize the lower level variable $y^{(m)}$ towards the minimizer $y_{x^{(m)}}^{(m)}$, but after the communication step, $x^{(m)}$ is smoothed among clients, as a result, the target of $y_t^{(m)}$ changes which causes a huge bias.

In the appendix, we show the FedBiOAcc-Local algorithm achieves the same optimal convergence rate as FedBiOAcc, which has iteration complexity $O(\epsilon^{-1.5})$ and communication complexity $O(\epsilon^{-1})$. However, since the lower level problem in Eq. (10) is unique for each client, FedBiOAcc-Local does not have the property of linear speed-up *w.r.t* the number of clients as FedBiOAcc does.

# 5   Numerical Experiments

In this section, we assess the performance of the proposed FedBiOAcc algorithm through two federated bilevel tasks: Federated Data Cleaning and Federated Hyper-representation Learning. The Federated Data Cleaning task involves global lower level problems, while the Hyper-representation Learning task involves local lower level problems. The implementation is carried out using PyTorch, and the Federated Learning environment is simulated using the PyTorch.Distributed package. Our experiments were conducted on servers equipped with an AMD EPYC 7763 64-core CPU and 8 NVIDIA V100 GPUs.

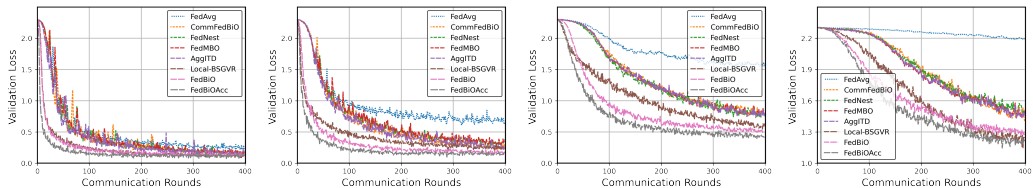

Figure 1: Validation Error vs Communication Rounds. From Left to Right: $\rho = 0.1, 0.4, 0.8, 0.95$. The local step $I$ is set as 5 for FedBiO, FedBiOAcc and FedAvg.

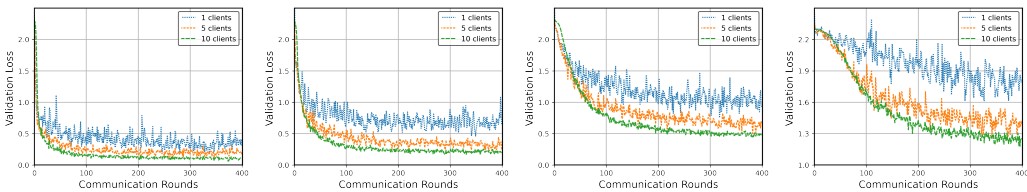

Figure 2: Validation Error vs Communication Rounds with different number of clients per epoch. From Left to Right: $\rho = 0.1, 0.4, 0.8, 0.95$. The local step $I$ is set as 5.

## 5.1 Federated Data Cleaning

In this section, we consider the Federated Data Cleaning task. In this task, we are given a noisy training dataset whose labels are corrupted by noise and a clean validation set. Then we aim to find weights for training samples such that a model that is learned over the weighted training set performs well on the validation set. This is a federated bilevel problem when the noisy training set is distributed over multiple clients. The formulation of the task is included in Appendix B.1. This task is a specialization of Eq. (1).

**Dataset and Baselines.** We create 10 clients and construct datasets based on MNIST [31]. For the training set, each client randomly samples 4500 images (no overlap among clients) from 10 classes and then randomly uniformly perturb the labels of $\rho$ ($0 \leq \rho \leq 1$) percent samples. For the validation set, each client randomly selects 50 clean images from a different class. In other words, the $m_{th}$ client only has validation samples from the $m_{th}$ class. This single-class validation setting introduces a high level of heterogeneity, such that individual clients are unable to conduct local cleaning due to they only have clean samples from one class. In our experiments, we test our FedBiOAcc algorithm, including the FedBiO algorithm (Algorithm 2 in Appendix) which does not use variance reduction; additionally, we also consider some baseline methods: a baseline that directly performs FedAvg [40] on the noisy dataset, this helps to verify the usefulness of data cleaning; Local-BSGVR [12], FedNest [49], CommFedBiO [33], AggITD [53] and FedMBO [21]. Note that Local-BSGVR is designed for the homogeneous setting, and the last four baselines all need multiple rounds of client-server communication to evaluate the hyper-gradient at each global epoch. We perform grid search to find the best hyper-parameters for each method and report the best results. Specific choices are included in Appendix B.1.

In figure 1, we compare the performance of different methods at various noise levels $\rho$. Note that the larger the $\rho$ value, the more noisy the training data are. The noise level can be illustrated by the performance of the FedAvg algorithm, which learns over the noisy data directly. As shown in the figure, FedAvg learns almost nothing when $\rho = 0.95$. Next, our algorithms are robust under various heterogeneity levels. When the noise level in the training set increases as the value of $\rho$ increases, learning relies more on the signal from the heterogeneous validation set, and our algorithms consistently outperform other baselines. Finally, in figure 2, we vary the number of clients sampled per epoch, and the experimental results show that our FedBiOAcc converges faster with more clients in the training per epoch; in figure 3, we vary the number of local steps under different noisy levels. Interestingly, the algorithm benefit more from the local training under larger noise.

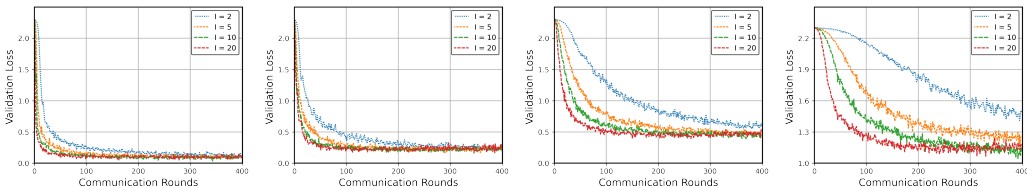

Figure 3: Validation Error vs Communication Rounds with different number of local steps $I$. From Left to Right: $\rho = 0.1, 0.4, 0.8, 0.95$.

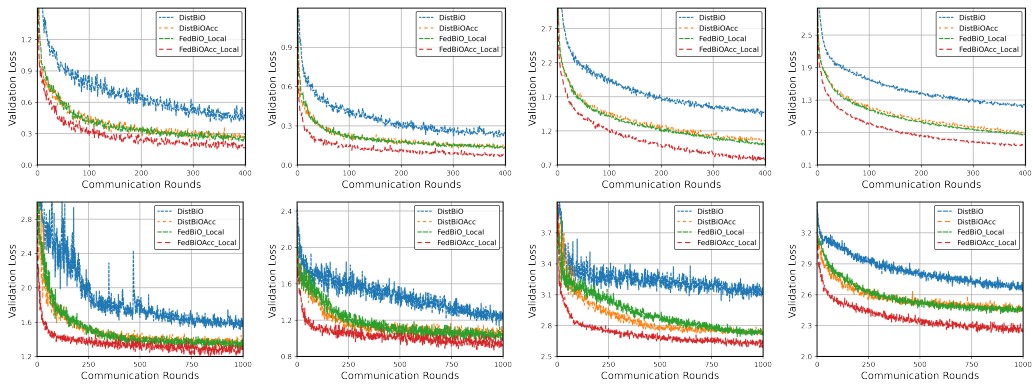

Figure 4: Validation Error vs Communication Rounds. The top row shows the result for the Omniglot Dataset and the bottom row shows MiniImageNet. From Left to Right: 5-way-1-shot, 5-way-5-shot, 20-way-1-shot, 20-way-5-shot. The local step $I$ is set to 5.

## 5.2 Federated Hyper-Representation Learning

In the Hyper-representation learning task, we learn a hyper-representation of the data such that a linear classifier can be learned quickly with a small number of data samples. A mathematical formulation of the task is included in Appendix B.2. Note that this task is an instantiation of Eq. (10), due to the fact that each client has its own tasks, and thus only the upper level problem is federated. We consider the Omniglot [29] and MiniImageNet [44] data sets. As in the non-distributed setting, we perform $N$-way-$K$-shot classification.

In this experiment, we compare FedBiOAcc-Local (Algorithm 4 in the Appendix) with three baselines FedBiO-Local (Algorithm 3 in the Appendix), DistBiO and DistBiOAcc. Note that DistBiO and DistBiOAcc are the distributed version of FedBiO-Local and FedBiOAcc-Local, respectively. In the experiments, we implement DistBiO and DistBiOAcc by setting the local steps as 1 for FedBiO-Local and FedBiOAcc-Local. We perform grid search for the hyper-parameter selection for both methods and choose the best ones, the specific choices of hyper-parameters are deferred to Appendix B.2. The results are summarized in Figure 4 (full results are included in Figure 5 and Figure 6 of Appendix. As shown by the results, FedBiOAcc converges faster than the baselines on both datasets and on all four types of classification tasks, which demonstrates the effectiveness of variance reduction and multiple steps of local training.

## 6 Conclusion

In this paper, we study the Federated Bilevel Optimization problems and introduce FedBiOAcc. In particular, FedBiOAcc evaluates the hyper-gradient by solving a federated quadratic problem, and mitigates the noise through momentum-based variance reduction technique. We provide a rigorous convergence analysis for our proposed method and show that FedBiOAcc has the optimal iteration complexity $O(\epsilon^{-1.5})$ and communication complexity $O(\epsilon^{-1})$, and it also achieves linear speed-up *w.r.t* the number of clients. Besides, we study a type of novel Federated Bilevel Optimization problems with local lower level problems. We modify FedBiO for this type of problems and propose

FedBiOAcc-Local. FedBiOAcc-Local achieves the same optimal convergence rate as FedBiOAcc. Finally, we validate our algorithms with real-world tasks.

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
