# A Assumptions

In this section, we restate all assumptions needed in our proof below:

**Assumption A.1** (Assumption 1). The function $f^{(m)}(x, y)$ is possibly non-convex and $g^{(m)}(x, y)$ is $\mu$-strongly convex *w.r.t* $y$ for any given $x$, *i.e.* for any $y_1$, $y_2 \in \mathbb{R}^d$, we have:

$$g^{(m)}(x, y_1) \geq g^{(m)}(x, y_2) + \langle \nabla_y g^{(m)}(x, y_2), y_2 - y_1 \rangle + \frac{\mu}{2} ||y_2 - y_1||^2.$$

**Assumption A.2** (Assumption 2). Function $f^{(m)}(x, y)$ is $L$-Lipschitz, *i.e.* for for any $x_1$, $x_2 \in \mathcal{X}$ and for any $y_1$, $y_2 \in \mathbb{R}^d$, and we denote $z_1 = (x_1, y_1)$, $z_2 = (x_2, y_2)$, then we have:

$$f^{(m)}(z_1) \leq f^{(m)}(z_2) + \langle \nabla f^{(m)}(z_2), z_1 - z_2 \rangle + \frac{L}{2} ||z_1 - z_2||^2.$$

or equivalently: $||\nabla f^{(m)}(z_1) - \nabla f^{(m)}(z_2)|| \leq L ||z_1 - z_2||$. We also assume and $f^{(m)}(x, y)$ has $C_f$-bounded gradient, *i.e.* for for any $x \in \mathcal{X}$ and any $y \in \mathbb{R}^d$, and we denote $z = (x, y)$, then we have $||\nabla f(z)|| \leq C_f$.

**Assumption A.3** (Assumption 3). Function $g^{(m)}(x, y)$ is $L$-Lipschitz. *i.e.* for for any $x_1$, $x_2 \in \mathcal{X}$ and for any $y_1$, $y_2 \in \mathbb{R}^d$, and we denote $z_1 = (x_1, y_1)$, $z_2 = (x_2, y_2)$, then we have:

$$g^{(m)}(z_1) \leq g^{(m)}(z_2) + \langle \nabla g^{(m)}(z_2), z_1 - z_2 \rangle + \frac{L}{2} ||z_1 - z_2||^2.$$

equivalently: $||\nabla g^{(m)}(z_1) - \nabla g^{(m)}(z_2)|| \leq L ||z_1 - z_2||$. For higher-order derivatives, we have:

a) $\nabla_{xy} g^{(m)}(x, y)$ and $\nabla_{y^2} g^{(m)}(x, y)$ are Lipschitz continuous with constant $L_{xy}$ and $L_{y^2}$ respectively, *i.e.* for for any $x_1$, $x_2 \in \mathcal{X}$ and for any $y_1$, $y_2 \in \mathbb{R}^d$, and we denote $z_1 = (x_1, y_1)$, $z_2 = (x_2, y_2)$, then we have: $||\nabla_{xy} g^{(m)}(z_1) - \nabla_{xy} g^{(m)}(z_2)|| \leq L_{xy} ||z_1 - z_2||$ and $||\nabla_{y^2} g^{(m)}(z_1) - \nabla_{y^2} g^{(m)}(z_2)|| \leq L_{y^2} ||z_1 - z_2||$.

**Assumption A.4** (Assumption 4). We have an unbiased stochastic first order and second order derivative oracle with bounded variance, more specifically, denote $z = (x, y)$, we have:

a) we have $\nabla f^{(m)}(z; \xi)$, such that: $E[\nabla f^{(m)}(z; \xi)] = \nabla f^{(m)}(z)$ and $var(\nabla f^{(m)}(z; \xi)) \leq \sigma^2$.

b) we have $\nabla g^{(m)}(z; \xi)$, such that: $E[\nabla g^{(m)}(z; \xi)] = \nabla g^{(m)}(z)$ and $var(\nabla g^{(m)}(z; \xi)) \leq \sigma^2$.

c) we have $\nabla_{y^2} g^{(m)}(z, \xi)$, such that: $E[\nabla_{y^2} g^{(m)}(z; \xi)] = \nabla_{y^2} g^{(m)}(z)$ and $var(\nabla_{y^2} g^{(m)}(z; \xi)) \leq \sigma^2$;

d) we have $\nabla_{xy} g^{(m)}(z; \xi)$, such that: $E[\nabla_{xy} g^{(m)}(z; \xi)] = \nabla_{xy} g^{(m)}(z)$ and $var(\nabla_{xy} g^{(m)}(x, y; \xi)) \leq \sigma^2$;

**Assumption A.5** (Assumption 5). For any $m, j \in [M]$ and $z = (x, y)$, we have: $||\nabla f^{(m)}(z) - \nabla f^{(j)}(z)|| \leq \zeta_f$, $||\nabla g^{(m)}(z) - \nabla g^{(j)}(z)|| \leq \zeta_g$, $||\nabla_{xy} g^{(m)}(z) - \nabla_{xy} g^{(j)}(z)|| \leq \zeta_{g,xy}$, $||\nabla_{y^2} g^{(m)}(z) - \nabla_{y^2} g^{(j)}(z)|| \leq \zeta_{g,yy}$, where $\zeta_f, \zeta_g, \zeta_{g,xy}, \zeta_{g,yy}$, are constants.

**Assumption A.6** (Assumption 6). For any $m, j \in [M]$ and $z = (x, y)$, we have: $||\nabla f^{(m)}(z) - \nabla f^{(j)}(z)|| \leq \zeta_f$, $||\nabla_{xy} g^{(m)}(z) - \nabla_{xy} g^{(j)}(z)|| \leq \zeta_{g,xy}$, $||\nabla_{y^2} g^{(m)}(z) - \nabla_{y^2} g^{(j)}(z)|| \leq \zeta_{g,yy}$, $||y_x^{(m)} - y_x^{(j)}|| \leq \zeta_{g^*}$, where $\zeta_f, \zeta_{g,xy}, \zeta_{g,yy}, \zeta_{g^*}$ are constants.

## B  More Experimental Details and Results

In this section, we introduce more details of the experiments.

### B.1  Federated Data Cleaning

The formulation of the problem is as follows:

$$\min_{x\in\mathbb{R}^p} h(x) := \frac{1}{M}\sum_{m=1}^{M} f^{(m)}(x, y_x^{(m)}) = \frac{1}{M}\sum_{m=1}^{M}\Big(\frac{1}{N_m^{(val)}}\sum_{n=1}^{N_m^{(val)}}\Theta(y_x; \xi_{m,n}^{val})\Big)$$

$$\text{s.t. } y_x = \arg\min_{y\in\mathbb{R}^d} g(x,y) = \frac{1}{M}\sum_{m=1}^{M}\sum_{n=1}^{N_m^{(tr)}} x_{m,n}\Theta(y; \xi_{m,n}^{tr})$$

In the above formulation, we have $M$ clients, each client $m \in [M]$ has a pair of (noisy) training set $\{\xi_{m,n}^{tr}\}_{n=1}^{N_m^{(tr)}}$ and validation set $\{\xi_{m,n}^{val}\}_{n=1}^{N_m^{(val)}}$, and $x_{m,n}, n \in [N_m^{(tr)}]$ are weights for training samples, $y$ is the parameter of a model, and we denote the model by $\Theta$. Note that $y_x$ is the model learned over the weighted training set. We fit a model with 3 fully connected layers for the MNIST dataset. We also use $L_2$ regularization with coefficient $10^{-3}$ to satisfy the strong convexity condition.

In the Experiments, for FedNest and CommFedBiO, we choose learning rate 1 and hyper-learning rate 10000, for FedBiO, we choose learning rate 0.5, hyper learning rate 1000, for FedBiOAcc, we choose $\delta$ as 30, $u$ as 10000, $c_\eta$ as 0.2, $C_\gamma$ as 0.2, $\tau$ as 0.01, $\eta$ as 200 and $\gamma$ as 1.

### B.2  Federated Hyper-Representation Learning

$$\min_{x\in\mathbb{R}^p} h(x) := \frac{1}{M}\sum_{m=1}^{M} f^{(m)}(x, y_x^{(m)}) = \frac{1}{M}\sum_{m=1}^{M}\Big(\frac{1}{N_m}\sum_{n=1}^{N_m}\Big(\frac{1}{N_{m,n}^{val}}\sum_{i=1}^{N_{m,n}^{val}}\Theta(x, y_x^{(\mathcal{T}_{m,n})}; \xi_i^{val})\Big)\Big)$$

$$\text{s.t. } y_x^{(\mathcal{T}_{m,n})} = \arg\min_{y\in\mathbb{R}^d} g^{(\mathcal{T}_{m,n})}(x,y) = \frac{1}{N_{m,n}^{tr}}\sum_{i=1}^{N_{m,n}^{tr}}\Theta(x, y; \xi_i^{tr})$$

In the above formulation, we have $M$ clients, each client $m \in [M]$ has $N_m$ tasks and each task $\mathcal{T}_{m,n}$ is defined by a pair of training set $\{\xi_i^{tr}\}_{i=1}^{N_{m,n}^{tr}}$ and validation set $\{\xi_i^{val}\}_{i=1}^{N_{m,n}^{val}}$. $\Theta$ defines the model, $x$ is the parameter of the backbone model and $y$ is the parameter of the linear classifier. In summary, the lower level problem is to learn the optimal linear classifier $y$ given the backbone $x$, and the upper level problem is to learn the optimal backbone parameter $x$.

The Omniglot dataset includes 1623 characters from 50 different alphabets and each character consists of 20 samples. We create the Federated version of the Omniglot dataset. Firstly, we follow the experimental protocols of [52] to divide the alphabets to train/validation/test with 33/5/12, respectively. Then we distribute three alphabets to a client, in other words, we consider 11 clients in experiments. As in the non-distributed setting, we perform $N$-way-$K$-shot classification, more specifically, for each task, we randomly sample $N$ characters from the alphabet over that client and for each character, we sample $K$ samples for training and 15 samples for validation. We augment the characters by performing rotation operations (multipliers of 90 degrees). We use a 4-layer convolutional neural network where each convolutional layer has 64 filters of 3×3 [11]. For the MiniImageNet, it has 64 training classes and 16 validation classes. We distribute the training classes into four clients, similar to Omniglot, we also perform the $N$-way-$K$-shot classification. We use a 4-layer convolutional neural network where each convolutional layer has 64 filters of 3×3 [11] for experiments.

In the Experiments for Omniglot, for FedBiO, we choose learning rate 0.4, hyper learning rate 1, $\tau$ 0.5, for FedBiOAcc, we choose $\delta$ as 2, $u$ as 10000, $C_\eta$ as 100, $\tau$ as 0.5, $eta$ as 1 and $\gamma$ as 0.4. For MiniImageNet, for FedBiO, we choose learning rate 0.05, hyper learning rate 0.1, $\tau$ 0.01, for FedBiOAcc, we choose $\delta$ as 2, $u$ as 10000, $C_\eta$ as 100, $\tau$ as 0.01, $eta$ as 1 and $\gamma$ as 0.05.

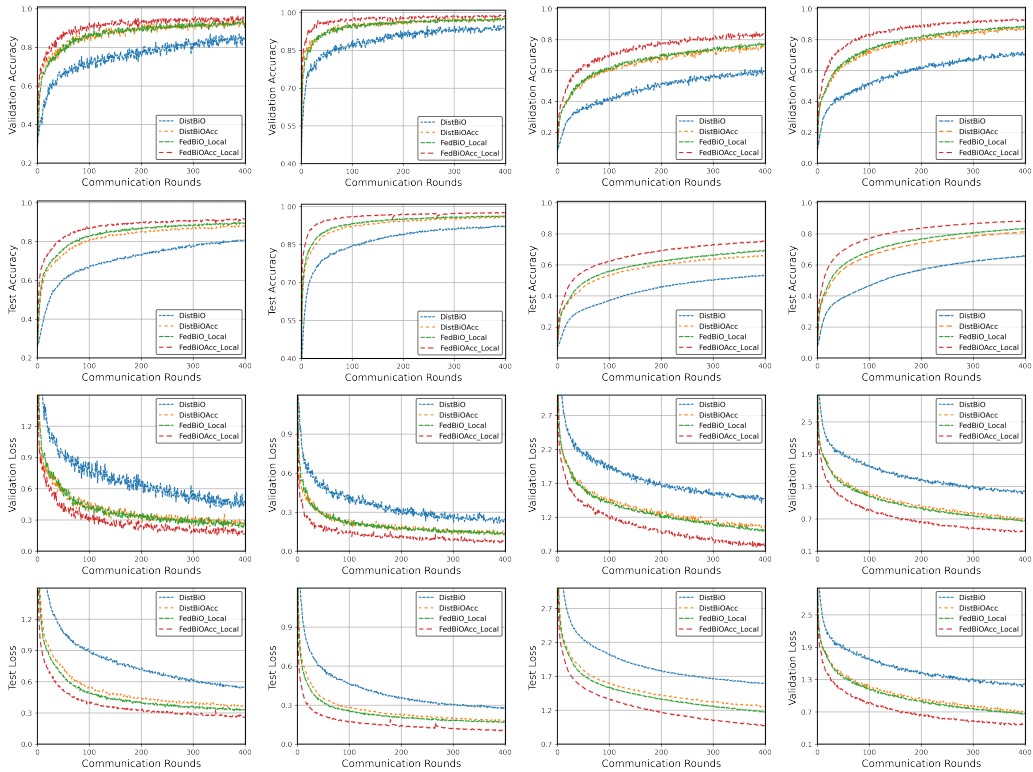

Figure 5: Results for the Omniglot Dataset. From Left to Right: 5-way-1-shot, 5-way-5-shot, 20-way-1-shot, 20-way-5-shot.

## C    Proof for Global Lower Level Problem

This section includes proofs related to the Federated Bilevel Optimization problems with global lower level problems (Eq. 1). First, we have the global and local hyper-gradient $\nabla h(x) = \Phi(x, y_x)$, $\nabla h^{(m)}(x) = \Phi^{(m)}(x, y_x)$ as defined in Eq. 2 and Eq. 3, and the following proposition:

**Proposition C.1.** *Suppose Assumptions 3.2 and 3.3 hold, the following statements hold:*

    *a) $y_x$ is Lipschitz continuous in $x$ with constant $\rho = \kappa$, where $\kappa = \frac{L}{\mu}$ is the condition number of $g(x, y)$.*

    *b) $\|\Phi(x_1; y_1) - \Phi(x_2; y_2)\|^2 \leq \hat{L}^2(\|x_1 - x_2\|^2 + \|y_1 - y_2\|^2)$, where $\hat{L} = O(\kappa^2)$.*

    *c) $h(x)$ is Lipschitz continuous in $x$ with constant $\bar{L}$ i.e., for any given $x_1, x_2 \in X$, we have $\|\nabla h(x_2) - \nabla h(x_1)\| \leq \bar{L}\|x_2 - x_1\|$ where $\bar{L} = O(\kappa^3)$.*

This is a standard results in bilevel optimization and we omit the proof here.

### C.1    Proof for the FedBiOAcc Algorithm

In this section, we prove the convergence of the FedBiOAcc Algorithm. To simplify the notation, we denote

$$\mu_{t,\xi}^{(m)} = \nabla_x f^{(m)}(x_t^{(m)}, y_t^{(m)}; \xi_{f,1}) - \nabla_{xy} g^{(m)}(x_t^{(m)}, y_t^{(m)}; \xi_{g,1})u_t^{(m)},$$

and we have:

$$\mathbb{E}_\xi[\mu_{t,\xi}^{(m)}] = \nabla_x f^{(m)}(x_t^{(m)}, y_t^{(m)}) - \nabla_{xy} g^{(m)}(x_t^{(m)}, y_t^{(m)})u_t^{(m)}$$

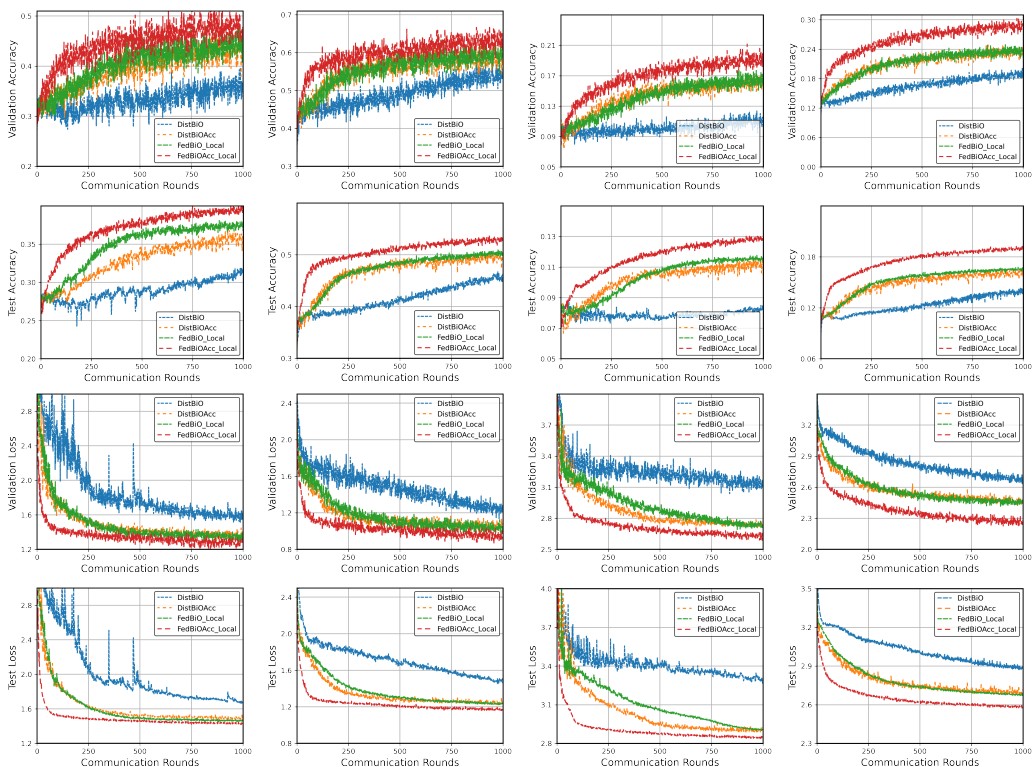

Figure 6: Results for the MiniImageNet Dataset. From Left to Right: 5-way-1-shot, 5-way-5-shot, 20-way-1-shot, 20-way-5-shot.

where the expectation is *w.r.t* $\{\xi_{f,1}, \xi_{g,1}\}$ at iteration $t$, we denote $\mu_t^{(m)} = \mathbb{E}_\xi[\mu_{t,\xi}^{(m)}]$ for short. Similarly, we denote

$$p_{t,\xi}^{(m)} = \nabla_{y^2} g^{(m)}(x_t^{(m)}, y_t^{(m)}; \xi_{g,2}) u_t^{(m)} + \nabla_y f^{(m)}(x_t^{(m)}, y_t^{(m)}; \xi_{f,2}),$$

and we have:

$$\mathbb{E}_\xi[p_{t,\xi}^{(m)}] = \nabla_{y^2} g^{(m)}(x_t^{(m)}, y_t^{(m)}) u_t^{(m)} + \nabla_y f^{(m)}(x_t^{(m)}, y_t^{(m)}).$$

where the expectation is *w.r.t* $\{\xi_{f,2}, \xi_{g,2}\}$ at iteration $t$, we denote $p_t^{(m)} = \mathbb{E}_\xi[p_{t,\xi}^{(m)}]$ for short.

### C.1.1 Hyper-Gradient Bias and Inner-Gradient Bias

**Lemma C.2.** *Suppose we have $c_u \alpha_t^2 < 1$, then we have:*

$$\mathbb{E}\big[\big\|\bar{q}_t - \bar{p}_t\big\|^2\big] \leq (1 - c_u \alpha_{t-1}^2)\mathbb{E}\big[\big\|\bar{q}_{t-1} - \bar{p}_{t-1}\big\|^2\big] + \frac{2(c_u \alpha_{t-1}^2)^2}{b_x M}\sigma^2$$

$$+ \frac{4\tilde{L}_2^2}{b_x M^2}\sum_{m=1}^M \mathbb{E}\big[\big\|x_t^{(m)} - x_{t-1}^{(m)}\big\|^2 + \big\|y_t^{(m)} - y_{t-1}^{(m)}\big\|^2\big]$$

$$+ \frac{8L^2}{b_x M^2}\sum_{m=1}^M \mathbb{E}\big[\big\|u_t^{(m)} - u_{t-1}^{(m)}\big\|^2\big]$$

*where $\tilde{L}_2^2 = \left(L^2 + \frac{2L_{y^2}^2 C_f^2}{\mu^2}\right)$ and the expectation outside is* w.r.t *all the stochasity of the algorithm.*

*Proof.* First, we have:

$$\mathbb{E}\big[\big\|\bar{q}_t - \bar{p}_t\big\|^2\big] = \mathbb{E}\big[\big\|\bar{p}_{t,\mathcal{B}_x} + (1 - c_u\alpha_{t-1}^2)(\bar{q}_{t-1} - \bar{p}_{t-1,\mathcal{B}_x}) - \bar{p}_t\big\|^2\big]$$

$$= \mathbb{E}\big[\big\|(1 - c_u\alpha_{t-1}^2)(\bar{q}_{t-1} - \bar{p}_{t-1}) + (\bar{p}_{t,\mathcal{B}_x} - \bar{p}_t + (1 - c_u\alpha_{t-1}^2)(\bar{p}_{t-1} - \bar{p}_{t-1,\mathcal{B}_x}))\big\|^2\big]$$

$$\leq (1 - c_u\alpha_{t-1}^2)\mathbb{E}\big[\big\|\bar{q}_{t-1} - \bar{p}_{t-1}\big\|^2\big] + \mathbb{E}\big[\big\|\bar{p}_{t,\mathcal{B}_x} - \bar{p}_t + (1 - c_u\alpha_{t-1}^2)(\bar{p}_{t-1} - \bar{p}_{t-1,\mathcal{B}_x})\big\|^2\big]$$

$$\leq (1 - c_u\alpha_{t-1}^2)\mathbb{E}\big[\big\|\bar{q}_{t-1} - \bar{p}_{t-1}\big\|^2\big]$$

$$+ \frac{1}{b_x^2 M^2} \sum_{m=1}^{M} \sum_{\xi_x \in \mathcal{B}_x} \mathbb{E}\big[\big\|p_{t,\xi_x}^{(m)} - p_t^{(m)} + (1 - c_u\alpha_{t-1}^2)(p_{t-1}^{(m)} - p_{t-1,\xi_x}^{(m)})\big\|^2\big]$$

where the first inequality uses the fact that the cross product term is zero in expectation, the condition that $c_\nu\alpha_t^2 < 1$ and the second inequality follows that samples are independent among clients. We denote the second term of above as $T_1$, then we have:

$$T_1 \overset{(a)}{\leq} 2(c_u\alpha_{t-1}^2)^2 \mathbb{E}\big[\big\|p_{t,\xi_x}^{(m)} - p_t^{(m)}\big\|^2\big] + 2(1 - c_u\alpha_{t-1}^2)^2 \mathbb{E}\big[\big\|p_{t,\xi_x}^{(m)} - p_{t-1,\xi_x}^{(m)} - (p_t^{(m)} - p_{t-1}^{(m)})\big\|^2\big]$$

$$\overset{(b)}{\leq} 2(c_u\alpha_{t-1}^2)^2\sigma^2 + 2\mathbb{E}\big[\big\|p_{t,\xi_x}^{(m)} - p_{t-1,\xi_x}^{(m)}\big\|^2\big]$$

where inequality (a) follows the generalized triangle inequality; (b) and the bounded variance assumption. We denote the second term above as $T_{1,2}$, we have:

$$T_{1,2} = 2\mathbb{E}\big\|\nabla_{y^2}g^{(m)}(x_t^{(m)}, y_t^{(m)}; \xi_{g,2})u_t^{(m)} + \nabla_y f^{(m)}(x_t^{(m)}, y_t^{(m)}; \xi_{f,2})$$

$$- \big(\nabla_{y^2}g^{(m)}(x_{t-1}^{(m)}, y_{t-1}^{(m)}; \xi_{g,2})u_{t-1}^{(m)} + \nabla_y f^{(m)}(x_{t-1}^{(m)}, y_{t-1}^{(m)}; \xi_{f,2})\big)\big\|^2$$

$$\leq 4\mathbb{E}\big\|\nabla_y f^{(m)}(x_t^{(m)}, y_t^{(m)}; \mathcal{B}_{f,1}) - \nabla_y f^{(m)}(x_{t-1}^{(m)}, y_{t-1}^{(m)}; \mathcal{B}_{f,1})\big\|^2$$

$$+ 4\mathbb{E}\big\|\nabla_{y^2}g^{(m)}(x_t^{(m)}, y_t^{(m)}; \mathcal{B}_{g,1})u_t^{(m)} - \nabla_{y^2}g^{(m)}(x_{t-1}^{(m)}, y_{t-1}^{(m)}; \mathcal{B}_{g,1})u_{t-1}^{(m)}\big\|^2$$

$$\leq 4\big(L^2 + \frac{2L_{y^2}^2 C_f^2}{\mu^2}\big)\mathbb{E}\big[\big\|x_t^{(m)} - x_{t-1}^{(m)}\big\|^2 + \big\|y_t^{(m)} - y_{t-1}^{(m)}\big\|^2\big] + 8L^2\mathbb{E}\big[\big\|u_t^{(m)} - u_{t-1}^{(m)}\big\|^2\big]$$

Combine everything together finishes the proof. $\qquad\square$

**Lemma C.3.** *Suppose we have $c_\nu\alpha_t^2 < 1$, then we have:*

$$\mathbb{E}\big[\big\|\bar{\nu}_t - \bar{\mu}_t\big\|^2\big] \leq (1 - c_\nu\alpha_{t-1}^2)\mathbb{E}\big[\big\|\bar{\nu}_{t-1} - \bar{\mu}_{t-1}\big\|^2\big] + \frac{2(c_\nu\alpha_{t-1}^2)^2}{b_x M}\sigma^2$$

$$+ \frac{4\tilde{L}_1^2}{b_x M^2} \sum_{m=1}^{M} \mathbb{E}\big[\big\|x_t^{(m)} - x_{t-1}^{(m)}\big\|^2 + \big\|y_t^{(m)} - y_{t-1}^{(m)}\big\|^2\big]$$

$$+ \frac{8L^2}{b_x M^2} \sum_{m=1}^{M} \mathbb{E}\big[\big\|u_t^{(m)} - u_{t-1}^{(m)}\big\|^2\big]$$

*where $\tilde{L}_1^2 = \big(L^2 + \frac{2L_{xy}^2 C_f^2}{\mu^2}\big)$ and the expectation outside is* w.r.t *all the stochasity of the algorithm.*

*Proof.* First, we have:

$$\mathbb{E}\big[\big\|\bar{\nu}_t - \bar{\mu}_t\big\|^2\big] = \mathbb{E}\big[\big\|\bar{\mu}_{t,\mathcal{B}_x} + (1 - c_\nu\alpha_{t-1}^2)(\bar{\nu}_{t-1} - \bar{\mu}_{t-1,\mathcal{B}_x}) - \bar{\mu}_t\big\|^2\big]$$

$$= \mathbb{E}\big[\big\|(1 - c_\nu\alpha_{t-1}^2)(\bar{\nu}_{t-1} - \bar{\mu}_{t-1}) + (\bar{\mu}_{t,\mathcal{B}_x} - \bar{\mu}_t + (1 - c_\nu\alpha_{t-1}^2)(\bar{\mu}_{t-1} - \bar{\mu}_{t-1,\mathcal{B}_x}))\big\|^2\big]$$

$$\leq (1 - c_\nu\alpha_{t-1}^2)\mathbb{E}\big[\big\|\bar{\nu}_{t-1} - \bar{\mu}_{t-1}\big\|^2\big] + \mathbb{E}\big[\big\|\bar{\mu}_{t,\mathcal{B}_x} - \bar{\mu}_t + (1 - c_\nu\alpha_{t-1}^2)(\bar{\mu}_{t-1} - \bar{\mu}_{t-1,\mathcal{B}_x})\big\|^2\big]$$

$$\leq (1 - c_\nu\alpha_{t-1}^2)\mathbb{E}\big[\big\|\bar{\nu}_{t-1} - \bar{\mu}_{t-1}\big\|^2\big] + \frac{1}{b_x^2 M^2} \sum_{m=1}^{M} \sum_{\xi_x \in \mathcal{B}_x} \mathbb{E}\big[\big\|\mu_{t,\xi_x}^{(m)} - \mu_t^{(m)} + (1 - c_\nu\alpha_{t-1}^2)(\mu_{t-1}^{(m)} - \mu_{t-1,\xi_x}^{(m)})\big\|^2\big]$$

where the first inequality uses the fact that the cross product term is zero in expectation, the condition that $c_\nu\alpha_t^2 < 1$ and the second inequality follows that samples are independent among clients. We

denote the second term of above as $T_1$, then we have:

$$T_1 \overset{(a)}{\leq} 2(c_\nu \alpha_{t-1}^2)^2 \mathbb{E}\big[\big\|\mu_{t,\xi_x}^{(m)} - \mu_t^{(m)}\big\|^2\big] + 2(1 - c_\nu \alpha_{t-1}^2)^2 \mathbb{E}\big[\big\|\mu_{t,\xi_x}^{(m)} - \mu_{t-1,\xi_x}^{(m)} - (\mu_t^{(m)} - \mu_{t-1}^{(m)})\big\|^2\big]$$

$$\overset{(b)}{\leq} 2(c_\nu \alpha_{t-1}^2)^2 \sigma^2 + 2\mathbb{E}\big[\big\|\mu_{t,\xi_x}^{(m)} - \mu_{t-1,\xi_x}^{(m)}\big\|^2\big]$$

where inequality (a) follows the generalized triangle inequality; (b) and the bounded variance assumption. We denote the second term above as $T_{1,2}$, we have:

$$T_{1,2} = 2\mathbb{E}\big\|\nabla_x f^{(m)}(x_t^{(m)}, y_t^{(m)}; \mathcal{B}_{f,1}) - \nabla_{xy} g^{(m)}(x_t^{(m)}, y_t^{(m)}; \mathcal{B}_{g,1}) u_t^{(m)}$$
$$- \big(\nabla_x f^{(m)}(x_{t-1}^{(m)}, y_{t-1}^{(m)}; \mathcal{B}_{f,1}) - \nabla_{xy} g^{(m)}(x_{t-1}^{(m)}, y_{t-1}^{(m)}; \mathcal{B}_{g,1}) u_{t-1}^{(m)}\big)\big\|^2$$
$$\leq 4\mathbb{E}\big\|\nabla_x f^{(m)}(x_t^{(m)}, y_t^{(m)}; \mathcal{B}_{f,1}) - \nabla_x f^{(m)}(x_{t-1}^{(m)}, y_{t-1}^{(m)}; \mathcal{B}_{f,1})\big\|^2$$
$$+ 4\mathbb{E}\big\|\nabla_{xy} g^{(m)}(x_t^{(m)}, y_t^{(m)}; \mathcal{B}_{g,1}) u_t^{(m)} - \nabla_{xy} g^{(m)}(x_{t-1}^{(m)}, y_{t-1}^{(m)}; \mathcal{B}_{g,1}) u_{t-1}^{(m)}\big\|^2$$
$$\leq 4\big(L^2 + \frac{2L_{xy}^2 C_f^2}{\mu^2}\big) \mathbb{E}\big[\big\|x_t^{(m)} - x_{t-1}^{(m)}\big\|^2 + \big\|y_t^{(m)} - y_{t-1}^{(m)}\big\|^2\big] + 8L^2 \mathbb{E}\big[\big\|u_t^{(m)} - u_{t-1}^{(m)}\big\|^2\big]$$

Combine everything together finishes the proof. $\square$

**Lemma C.4.** *Suppose we have $c_\omega \alpha_{t-1}^2 < 1$, then for $t \neq \bar{t}_s$, with $s \in [S]$, we have:*

$$\mathbb{E}\big[\big\|\bar{\omega}_t - \frac{1}{M} \sum_{m=1}^{M} \nabla_y g^{(m)}(x_t^{(m)}, y_t^{(m)})\big\|^2\big]$$

$$\leq (1 - c_\omega \alpha_{t-1}^2) \mathbb{E}\big[\big\|\bar{\omega}_{t-1} - \frac{1}{M} \sum_{m=1}^{M} \nabla_y g^{(m)}(x_{t-1}^{(m)}, y_{t-1}^{(m)})\big\|^2\big] + \frac{2(c_\omega \alpha_{t-1}^2)^2 \sigma^2}{b_y M}$$

$$+ \frac{2L^2}{b_y M^2} \sum_{m=1}^{M} \mathbb{E}\big[\big\|x_t^{(m)} - x_{t-1}^{(m)}\big\|^2 + \big\|y_t^{(m)} - y_{t-1}^{(m)}\big\|^2\big]$$

*where the expectation is w.r.t the stochasticity of the algorithm.*

*Proof.* First, we have:

$$\mathbb{E}\big[\big\|\bar{\omega}_t - \frac{1}{M} \sum_{m=1}^{M} \nabla_y g^{(m)}(x_t^{(m)}, y_t^{(m)})\big\|^2\big]$$

$$= \mathbb{E}\big[\big\|\frac{1}{M} \sum_{m=1}^{M} \big(\nabla_y g^{(m)}(x_t^{(m)}, y_t^{(m)}, \mathcal{B}_y)$$
$$+ (1 - c_\omega \alpha_{t-1}^2)(\omega_{t-1}^{(m)} - \nabla_y g^{(m)}(x_{t-1}^{(m)}, y_{t-1}^{(m)}, \mathcal{B}_y)) - \nabla_y g^{(m)}(x_t^{(m)}, y_t^{(m)})\big)\big\|^2\big]$$

$$= \mathbb{E}\big[\big\|(1 - c_\omega \alpha_{t-1}^2)(\bar{\omega}_{t-1} - \frac{1}{M} \sum_{m=1}^{M} \nabla_y g^{(m)}(x_{t-1}^{(m)}, y_{t-1}^{(m)})$$

$$+ \frac{1}{M} \sum_{m=1}^{M} \big(\nabla_y g^{(m)}(x_t^{(m)}, y_t^{(m)}, \mathcal{B}_y) - \nabla_y g^{(m)}(x_t^{(m)}, y_t^{(m)})$$

$$+ (1 - c_\omega \alpha_{t-1}^2)(\nabla_y g^{(m)}(x_{t-1}^{(m)}, y_{t-1}^{(m)}) - \nabla_y g^{(m)}(x_{t-1}^{(m)}, y_{t-1}^{(m)}, \mathcal{B}_y)))\big\|^2\big]$$

$$\overset{(a)}{\leq} (1 - c_\omega \alpha_{t-1}^2) \mathbb{E}\big[\big\|\bar{\omega}_{t-1} - \frac{1}{M} \sum_{m=1}^{M} \nabla_y g^{(m)}(x_{t-1}^{(m)}, y_{t-1}^{(m)})\big\|^2\big]$$

$$+ \frac{1}{b_y^2 M^2} \sum_{m=1}^{M} \mathbb{E} \sum_{\xi_y \in \mathcal{B}_y} \big[\big\|(\nabla_y g^{(m)}(x_t^{(m)}, y_t^{(m)}, \xi_y) - \nabla_y g^{(m)}(x_t^{(m)}, y_t^{(m)})$$

$$+ (1 - c_\omega \alpha_{t-1}^2)(\nabla_y g^{(m)}(x_{t-1}^{(m)}, y_{t-1}^{(m)}) - \nabla_y g^{(m)}(x_{t-1}^{(m)}, y_{t-1}^{(m)}, \xi_y)))\big\|^2\big]$$

where inequality (a) uses the fact that the cross product term is zero in expectation and the condition that $c_\omega \alpha_t^2 < 1, t \in [T]$, furthermore, the samples are sampled independently on clients.

We denote the second term in the above inequality as $T_1$, we have:

$$
\begin{aligned}
T_1 &\overset{(b)}{\leq} 2(c_\omega \alpha_{t-1}^2)^2 \mathbb{E}\big[\big\|\nabla_y g^{(m)}(x_t^{(m)}, y_t^{(m)}, \xi_y) - \nabla_y g^{(m)}(x_t^{(m)}, y_t^{(m)})\big\|^2\big] \\
&\quad + 2(1 - c_\omega \alpha_{t-1}^2)^2 \mathbb{E}\big[\big\| - \nabla_y g^{(m)}(x_t^{(m)}, y_t^{(m)}) \\
&\quad + \nabla_y g^{(m)}(x_t^{(m)}, y_t^{(m)}, \xi_y) + \nabla_y g^{(m)}(x_{t-1}^{(m)}, y_{t-1}^{(m)}) - \nabla_y g^{(m)}(x_{t-1}^{(m)}, y_{t-1}^{(m)}, \xi_y)\big\|^2\big] \\
&\overset{(c)}{\leq} 2(c_\omega \alpha_{t-1}^2)^2 \sigma^2 + 2\mathbb{E}\big[\big\|\nabla_y g^{(m)}(x_t^{(m)}, y_t^{(m)}, \xi_y) - \nabla_y g^{(m)}(x_{t-1}^{(m)}, y_{t-1}^{(m)}, \xi_y)\big\|^2\big] \\
&\overset{(d)}{\leq} 2(c_\omega \alpha_{t-1}^2)^2 \sigma^2 + 2L^2 \mathbb{E}\big[\big\|x_t^{(m)} - x_{t-1}^{(m)}\big\|^2 + \big\|y_t^{(m)} - y_{t-1}^{(m)}\big\|^2\big]
\end{aligned}
$$

inequality (b) uses the generalized triangle inequality; inequality (c) follows the bounded variance assumption 3.4, Proposition E.2; inequality (d) uses the smoothness assumption 3.3. $\qquad\square$

### C.1.2 Lower Problem Solution Error

**Lemma C.5.** *Suppose we choose $\gamma \leq \frac{1}{2L}$ and $\alpha_t < 1$. Then for $t \in [T]$, we have:*

$$
\begin{aligned}
\|\bar{y}_{t+1} - y_{\bar{x}_{t+1}}\|^2 &\leq \big(1 - \frac{\mu\gamma\alpha_t}{4}\big)\|\bar{y}_t - y_{\bar{x}_t}\|^2 - \frac{\gamma^2\alpha_t}{4}\|\bar{\omega}_t\|^2 + \frac{9\kappa^2\eta^2\alpha_t}{2\mu\gamma}\|\bar{\nu}_t\|^2 \\
&\quad + \frac{9\gamma\alpha_t L^2}{\mu M}\sum_{m=1}^M \big[\|x_t^{(m)} - \bar{x}_t\|^2 + \|y_t^{(m)} - \bar{y}_t\|^2\big] + \frac{9\gamma\alpha_t}{\mu}\Big\|\frac{1}{M}\sum_{m=1}^M \nabla_y g^{(m)}(x_t^{(m)}, y_t^{(m)}) - \bar{w}_t\Big\|^2
\end{aligned}
$$

*Proof.* First, we exploit Proposition E.5, and choose the function $g(\bar{x}_t, \cdot)$, by assumption it is $L$ smooth and $\mu$ strongly convex, and we choose $\gamma < \frac{1}{2L}$ and $\alpha_t < 1$, thus:

$$
\|\bar{y}_{t+1} - y_{\bar{x}_t}\|^2 \leq (1 - \frac{\mu\gamma\alpha_t}{2})\|\bar{y}_t - y_{\bar{x}_t}\|^2 - \frac{\gamma^2\alpha_t}{4}\|\bar{\omega}_t\|^2 + \frac{4\gamma\alpha_t}{\mu}\|\nabla_y g(\bar{x}_t, \bar{y}_t) - \bar{w}_t\|^2. \quad (12)
$$

Next, we decompose the term $\|\bar{y}_{t+1} - y_{\bar{x}_{t+1}}\|^2$ as follows:

$$
\begin{aligned}
\|\bar{y}_{t+1} - y_{\bar{x}_{t+1}}\|^2 &\leq (1 + \frac{\mu\gamma\alpha_t}{4})\|\bar{y}_{t+1} - y_{\bar{x}_t}\|^2 + (1 + \frac{4}{\mu\gamma\alpha_t})\|y_{\bar{x}_t} - y_{\bar{x}_{t+1}}\|^2 \\
&\leq (1 + \frac{\mu\gamma\alpha_t}{4})\|\bar{y}_{t+1} - y_{\bar{x}_t}\|^2 + (1 + \frac{4}{\mu\gamma\alpha_t})\kappa^2\|\bar{x}_t - \bar{x}_{t+1}\|^2 \quad (13)
\end{aligned}
$$

where the second inequality is due to case a) of Proposition 3.9. Combining the above inequalities 12 and 13, we have

$$
\begin{aligned}
\|\bar{y}_{t+1} - y_{\bar{x}_{t+1}}\|^2 &\leq (1 + \frac{\mu\gamma\alpha_t}{4})(1 - \frac{\mu\gamma\alpha_t}{2})\|\bar{y}_t - y_{\bar{x}_t}\|^2 - (1 + \frac{\mu\gamma\alpha_t}{4})\frac{\gamma^2\alpha_t}{4}\|\bar{\omega}_t\|^2 \\
&\quad + (1 + \frac{\mu\gamma\alpha_t}{4})\frac{4\gamma\alpha_t}{\mu}\|\nabla_y g(\bar{x}_t, \bar{y}_t) - \bar{w}_t\|^2 + (1 + \frac{4}{\mu\gamma\alpha_t})\kappa^2\eta^2\alpha_t^2\|\bar{\nu}_t\|^2
\end{aligned}
$$

Since we choose $\gamma \leq \frac{1}{2L}, \alpha_t < 1$, we have:

$$
(1 + \frac{\mu\gamma\alpha_t}{4})(1 - \frac{\mu\gamma\alpha_t}{2}) = 1 - \frac{\mu\gamma\alpha_t}{4} - \frac{\mu^2\gamma^2\alpha_t^2}{8} \leq 1 - \frac{\mu\gamma\alpha_t}{4}
$$

and $-(1 + \frac{\mu\gamma\alpha_t}{4}) \leq -1, (1 + \frac{\mu\gamma\alpha_t}{4}) \leq \frac{9}{8}, \mu\gamma\alpha_t < \frac{1}{2}$. Thus, we have

$$
\|\bar{y}_{t+1} - y_{\bar{x}_{t+1}}\|^2 \leq \big(1 - \frac{\mu\gamma\alpha_t}{4}\big)\|\bar{y}_t - y_{\bar{x}_t}\|^2 - \frac{\gamma^2\alpha_t}{4}\|\bar{\omega}_t\|^2 + \frac{9\gamma\alpha_t}{2\mu}\underbrace{\|\nabla_y g(\bar{x}_t, \bar{y}_t) - \bar{w}_t\|^2}_{T_1} + \frac{9\kappa^2\eta^2\alpha_t}{2\mu\gamma}\|\bar{\nu}_t\|^2
$$

For the term $T_1$ in the inequality above, we have:

$$\|\nabla_y g(\bar{x}_t, \bar{y}_t) - \bar{w}_t\|^2 \le 2\|\nabla_y g(\bar{x}_t, \bar{y}_t) - \frac{1}{M}\sum_{m=1}^M \nabla_y g^{(m)}(x_t^{(m)}, y_t^{(m)})\|^2$$

$$+ 2\|\frac{1}{M}\sum_{m=1}^M \nabla_y g^{(m)}(x_t^{(m)}, y_t^{(m)}) - \bar{w}_t\|^2$$

$$\le \frac{2L^2}{M}\sum_{m=1}^M \left[\|x_t^{(m)} - \bar{x}_t\|^2 + \|y_t^{(m)} - \bar{y}_t\|^2\right]$$

$$+ 2\|\frac{1}{M}\sum_{m=1}^M \nabla_y g^{(m)}(x_t^{(m)}, y_t^{(m)}) - \bar{w}_t\|^2$$

This completes the proof. $\qquad\square$

**Lemma C.6.** *Suppose we choose $\tau \le \frac{1}{2L}$ and $\alpha_t < 1$, $r = \frac{C_f}{\mu}$. Then for $t \in [T]$, we have:*

$$\|\bar{u}_{t+1} - u_{\bar{x}_{t+1}}\|^2 \le \left(1 - \frac{\mu\tau\alpha_t}{4}\right)\|\bar{u}_t - u_{\bar{x}_t}\|^2 - \frac{\tau^2\alpha_t}{4}\|\bar{q}_t\|^2 + \frac{9\kappa^2\eta^2\alpha_t}{2\mu\tau}\|\bar{\nu}_t\|^2 + \frac{9\tau\alpha_t}{\mu}\|\bar{p} - \bar{q}_t\|^2$$

$$+ \frac{18\tau\alpha_t\tilde{L}_2^2}{\mu M}\sum_{m=1}^M \left[\|x_t^{(m)} - \bar{x}_t\|^2 + \|y_t^{(m)} - \bar{y}_t\|^2\right] + \frac{18\tau\alpha_t L^2}{M}\sum_{m=1}^M \|u_t^{(m)} - \bar{u}_t\|^2$$

*where $\tilde{L}_2^2 = (L^2 + \frac{2L_{y^2}^2 C_f^2}{\mu^2})$ is a constant.*

*Proof.* First, we exploit Proposition E.5, and choose the function $\frac{1}{2}x^T\nabla_{y^2}g(\bar{x}, y_{\bar{x}})x - \nabla_y f(\bar{x}, y_{\bar{x}})^T x$, by assumption it is $L$ smooth and $\mu$ strongly convex, and we choose $\tau < \frac{1}{2L}$ and $\alpha_t < 1$, thus:

$$\|\bar{u}_{t+1} - u_{\bar{x}_t}\|^2 \le (1 - \frac{\mu\tau\alpha_t}{2})\|\bar{u}_t - u_{\bar{x}_t}\|^2 - \frac{\tau^2\alpha_t}{4}\|\bar{q}_t\|^2 + \frac{4\tau\alpha_t}{\mu}\|\nabla_{y^2}g(\bar{x}, y_{\bar{x}})\bar{u}_t - \nabla_y f(\bar{x}, y_{\bar{x}}) - \bar{q}_t\|^2.$$

where we also use the fact that

$$\|\bar{u}_{t+1} - u_{\bar{x}_t}\|^2 \le \|\bar{u}_t - \tau\alpha_t\bar{q}_t - u_{\bar{x}_t}\|^2$$

for $r = \frac{C_f}{\mu} \ge \|u_{\bar{x}_t}\|$. Next, we decompose the term $\|\bar{u}_{t+1} - u_{\bar{x}_{t+1}}\|^2$ as follows:

$$\|\bar{u}_{t+1} - u_{\bar{x}_{t+1}}\|^2 \le (1 + \frac{\mu\tau\alpha_t}{4})\|\bar{u}_{t+1} - u_{\bar{x}_t}\|^2 + (1 + \frac{4}{\mu\tau\alpha_t})\|u_{\bar{x}_t} - u_{\bar{x}_{t+1}}\|^2$$

$$\le (1 + \frac{\mu\tau\alpha_t}{4})\|\bar{u}_{t+1} - u_{\bar{x}_t}\|^2 + (1 + \frac{4}{\mu\tau\alpha_t})\bar{L}^2\|\bar{x}_t - \bar{x}_{t+1}\|^2$$

where the second inequality is due to case a) of Proposition 3.9. Combining the above inequalities 12 and 13, we have:

$$\|\bar{u}_{t+1} - u_{\bar{x}_{t+1}}\|^2 \le (1 - \frac{\mu\tau\alpha_t}{4})\|\bar{u}_t - u_{\bar{x}_t}\|^2 - \frac{\tau^2\alpha_t}{4}\|\bar{q}_t\|^2$$

$$+ \frac{9\tau\alpha_t}{2\mu}\underbrace{\|\nabla_{y^2}g(\bar{x}, y_{\bar{x}})\bar{u}_t - \nabla_y f(\bar{x}, y_{\bar{x}}) - \bar{q}_t\|^2}_{T_1} + \frac{9\bar{L}^2\eta^2\alpha_t}{2\mu\tau}\|\bar{\nu}_t\|^2$$

where we use the fact that $\tau \le \frac{1}{2L}$, $\alpha_t < 1$ For the term $T_1$ in the inequality above, we have:

$$T_1 \le 2\|\nabla_{y^2}g(\bar{x}, y_{\bar{x}})\bar{u}_t - \nabla_y f(\bar{x}, y_{\bar{x}}) - \bar{p}_t\|^2 + 2\|\bar{p}_t - \bar{q}_t\|^2$$

$$\le 2\|\nabla_{y^2}g(\bar{x}, y_{\bar{x}})\bar{u}_t - \nabla_y f(\bar{x}, y_{\bar{x}})$$

$$- \frac{1}{M}\sum_{m=1}^M \left(\nabla_{y^2}g^{(m)}(x_t^{(m)}, y_t^{(m)})u_t^{(m)} + \nabla_y f^{(m)}(x_t^{(m)}, y_t^{(m)})\right)\|^2 + 2\|\bar{p}_t - \bar{q}_t\|^2$$

We denote the first term of the above inequality as $T_{1,1}$, we have:

$$T_{1,1} \leq 4\big\|\nabla_y f(\bar{x}, y_{\bar{x}}) - \frac{1}{M}\sum_{m=1}^{M}\big(\nabla_y f^{(m)}(x_t^{(m)}, y_t^{(m)})\big)\big\|^2$$

$$+ 4\big\|\nabla_{y^2} g(\bar{x}, y_{\bar{x}})\bar{u}_t - \frac{1}{M}\sum_{m=1}^{M}\big(\nabla_{y^2} g^{(m)}(x_t^{(m)}, y_t^{(m)})u_t^{(m)}\big)\big\|^2$$

$$\leq \big(\frac{4L^2}{M} + \frac{8L_{y^2}^2 C_f^2}{\mu^2 M}\big)\sum_{m=1}^{M}\big[\|x_t^{(m)} - \bar{x}_t\|^2 + \|y_t^{(m)} - \bar{u}_t\|^2\big] + \frac{4L^2}{M}\sum_{m=1}^{M}\|u_t^{(m)} - \bar{u}_t\|^2$$

Combine everything completes the proof. □

### C.1.3 Upper Variable Drift

**Lemma C.7.** *For any $t \neq \bar{t}_s, s \in [S]$, we have:*

$$\|x_t^{(m)} - \bar{x}_t\|^2 \leq I\eta^2 \sum_{\ell=\bar{t}_{s-1}}^{t-1} \alpha_\ell^2 \|\nu_\ell^{(m)} - \bar{\nu}_\ell\|^2$$

$$\|y_t^{(m)} - \bar{y}_t\|^2 \leq I\gamma^2 \sum_{\ell=\bar{t}_{s-1}}^{t-1} \alpha_\ell^2 \|\omega_\ell^{(m)} - \bar{\omega}_\ell\|^2$$

$$\|u_t^{(m)} - \bar{u}_t\|^2 \leq I\tau^2 \sum_{\ell=\bar{t}_{s-1}}^{t-1} \alpha_\ell^2 \|q_\ell^{(m)} - \bar{q}_\ell\|^2$$

*Proof.* Note from Algorithm and the definition of $\bar{t}_s$ that at $t = \bar{t}_s$ with $s \in [S]$, $x_t^{(m)} = \bar{x}_t$, for all $k$. For $t \neq \bar{t}_s$, with $s \in [S]$, we have: $x_t^{(m)} = x_{t-1}^{(m)} - \eta\alpha_{t-1}\nu_{t-1}^{(m)}$, this implies that: $x_t^{(m)} = x_{\bar{t}_{s-1}}^{(m)} - \sum_{\ell=\bar{t}_{s-1}}^{t-1} \eta\alpha_\ell \nu_\ell^{(m)}$ and $\bar{x}_t = \bar{x}_{\bar{t}_{s-1}} - \sum_{\ell=\bar{t}_{s-1}}^{t-1} \eta\alpha_\ell \bar{\nu}_\ell$. So for $t \neq \bar{t}_s$, with $s \in [S]$ we have:

$$\|x_t^{(m)} - \bar{x}_t\|^2 = \big\|x_{\bar{t}_{s-1}}^{(m)} - \bar{x}_{\bar{t}_{s-1}} - \big(\sum_{\ell=\bar{t}_{s-1}}^{t-1} \eta\alpha_\ell \nu_\ell^{(m)} - \sum_{\ell=\bar{t}_{s-1}}^{t-1} \eta\alpha_\ell \bar{\nu}_\ell\big)\big\|^2 = \big\|\sum_{\ell=\bar{t}_{s-1}}^{t-1} \eta\alpha_\ell\big(\nu_\ell^{(m)} - \bar{\nu}_\ell\big)\big\|^2$$

$$\leq I\eta^2 \sum_{\ell=\bar{t}_{s-1}}^{t-1} \alpha_\ell^2 \|\nu_\ell^{(m)} - \bar{\nu}_\ell\|^2$$

We can derive the bound for $\|y_t^{(m)} - \bar{y}_t\|^2$ and $\|u_t^{(m)} - \bar{u}_t\|^2$ similarly. This completes the proof. □

**Lemma C.8.** *Suppose $\eta\alpha_t < \frac{1}{16I\tilde{L}_1}$, then for $t \neq \bar{t}_s, s \in [S]$, we have:*

$$\sum_{m=1}^{M} \mathbb{E}\|\hat{\nu}_t^{(m)} - \bar{\nu}_t\|^2$$

$$\leq \big(1 + \frac{17}{16I}\big)\sum_{m=1}^{M} \mathbb{E}\|\nu_{t-1}^{(m)} - \bar{\nu}_{t-1}\|^2 + 8I\tilde{L}_1^2\alpha_{t-1}^2 \sum_{m=1}^{M} \mathbb{E}\big[2\|\eta\bar{\nu}_{t-1}\|^2 + \|\gamma\omega_{t-1}^{(m)}\|^2\big] + 16IL^2\alpha_{t-1}^2 \sum_{m=1}^{M} \mathbb{E}\|\tau q_{t-1}^{(m)}\|^2$$

$$+ 128I(c_\nu\alpha_{t-1}^2)^2\tilde{L}_1^2 \sum_{m=1}^{M} \mathbb{E}\big[\|x_t^{(m)} - \bar{x}_t\|^2 + \|y_t^{(m)} - \bar{y}_t\|^2\big] + 32I(c_\nu\alpha_{t-1}^2)^2 L^2 \sum_{m=1}^{M} \mathbb{E}\|u_t^{(m)} - \bar{u}_t\|^2$$

$$+ 8IM(c_\nu\alpha_{t-1}^2)^2\frac{\sigma^2}{b_x} + 32IM(c_\nu\alpha_{t-1}^2)^2\zeta_f^2 + 64I(c_\nu\alpha_{t-1}^2)^2 M\frac{C_f^2\zeta_{g,xy}^2}{\mu^2}$$

*where the expectation is w.r.t the stochasticity of the algorithm.*

*Proof.* For $t \neq \bar{t}_s$, we have:

$$
\begin{aligned}
\mathbb{E}\|\hat{\nu}_t^{(m)} - \bar{\nu}_t\|^2 &= \mathbb{E}\big\|\mu_{t,\mathcal{B}_x}^{(m)} + (1 - c_\nu\alpha_{t-1}^2)\big(\nu_{t-1}^{(m)} - \mu_{t-1,\mathcal{B}_x}^{(m)}\big) - \big(\bar{\mu}_{t,\mathcal{B}_x} + (1 - c_\nu\alpha_{t-1}^2)\big(\bar{\nu}_{t-1} - \bar{\mu}_{t-1,\mathcal{B}_x}\big)\big)\big\|^2 \\
&= \mathbb{E}\big\|(1 - c_\nu\alpha_{t-1}^2)\big(\nu_{t-1}^{(m)} - \bar{\nu}_{t-1}\big) + \mu_{t,\mathcal{B}_x}^{(m)} - \bar{\mu}_{t,\mathcal{B}_x} - (1 - c_\nu\alpha_{t-1}^2)\big(\mu_{t-1,\mathcal{B}_x}^{(m)} - \bar{\mu}_{t-1,\mathcal{B}_x}\big)\big\|^2 \\
&\overset{(a)}{\leq} (1 + \frac{1}{I})(1 - c_\nu\alpha_{t-1}^2)^2 \mathbb{E}\|\nu_{t-1}^{(m)} - \bar{\nu}_{t-1}\|^2 \\
&\quad + (1 + I)\mathbb{E}\big\|\mu_{t,\mathcal{B}_x}^{(m)} - \bar{\mu}_{t,\mathcal{B}_x} - (1 - c_\nu\alpha_{t-1}^2)\big(\mu_{t-1,\mathcal{B}_x}^{(m)} - \bar{\mu}_{t-1,\mathcal{B}_x}\big)\big\|^2 \\
&\leq \left(1 + \frac{1}{I}\right)\mathbb{E}\|\nu_{t-1}^{(m)} - \bar{\nu}_{t-1}\|^2 + (1 + I)\mathbb{E}\big\|\mu_{t,\mathcal{B}_x}^{(m)} - \bar{\mu}_{t,\mathcal{B}_x} - (1 - c_\nu\alpha_{t-1}^2)\big(\mu_{t-1,\mathcal{B}_x}^{(m)} - \bar{\mu}_{t-1,\mathcal{B}_x}\big)\big\|^2
\end{aligned}
\tag{14}
$$

where $(a)$ follows from the the generalized triangle inequality.

Next we bound the second term of the above inequality:

$$
\begin{aligned}
\sum_{m=1}^{M} &\mathbb{E}\big\|\mu_{t,\mathcal{B}_x}^{(m)} - \bar{\mu}_{t,\mathcal{B}_x} - (1 - c_\nu\alpha_{t-1}^2)\big(\mu_{t-1,\mathcal{B}_x}^{(m)} - \bar{\mu}_{t-1,\mathcal{B}_x}\big)\big\|^2 \\
&\leq 2\sum_{m=1}^{M} \mathbb{E}\big\|\mu_{t,\mathcal{B}_x}^{(m)} - \bar{\mu}_{t,\mathcal{B}_x} - \big(\mu_{t-1,\mathcal{B}_x}^{(m)} - \bar{\mu}_{t-1,\mathcal{B}_x}\big)\big\|^2 + 2(c_\nu\alpha_{t-1}^2)^2 \sum_{m=1}^{M} \mathbb{E}\big\|\mu_{t-1,\mathcal{B}_x}^{(m)} - \bar{\mu}_{t-1,\mathcal{B}_x}\big\|^2
\end{aligned}
$$

where the inequality follows the triangle inequality. We bound the two terms separately, for the first term, we have:

$$
\begin{aligned}
\sum_{m=1}^{M} &\mathbb{E}\big\|\mu_{t,\mathcal{B}_x}^{(m)} - \bar{\mu}_{t,\mathcal{B}_x} - \big(\mu_{t-1,\mathcal{B}_x}^{(m)} - \bar{\mu}_{t-1,\mathcal{B}_x}\big)\big\|^2 \overset{(a)}{\leq} \sum_{m=1}^{M} \mathbb{E}\big\|\mu_{t,\mathcal{B}_x}^{(m)} - \mu_{t-1,\mathcal{B}_x}^{(m)}\big\|^2 \\
&\leq \sum_{m=1}^{M} \mathbb{E}\big\|\nabla_x f^{(m)}(x_t^{(m)}, y_t^{(m)}; \xi_{f,1}) - \nabla_{xy} g^{(m)}(x_t^{(m)}, y_t^{(m)}; \xi_{g,1})u_t^{(m)} \\
&\qquad - \big(\nabla_x f^{(m)}(x_{t-1}^{(m)}, y_{t-1}^{(m)}; \xi_{f,1}) - \nabla_{xy} g^{(m)}(x_{t-1}^{(m)}, y_{t-1}^{(m)}; \xi_{g,1})u_{t-1}^{(m)}\big)\big\|^2 \\
&\overset{(b)}{\leq} 2\big(L^2 + \frac{2L_{xy}^2 C_f^2}{\mu^2}\big) \sum_{m=1}^{M} \mathbb{E}\big[\|x_t^{(m)} - x_{t-1}^{(m)}\|^2 + \|y_t^{(m)} - y_{t-1}^{(m)}\|^2\big] + 4L^2 \sum_{m=1}^{M} \mathbb{E}\|u_t^{(m)} - u_{t-1}^{(m)}\|^2 \\
&\leq 2\tilde{L}_1^2\alpha_{t-1}^2 \sum_{m=1}^{M} \mathbb{E}\big[\|\eta\nu_{t-1}^{(m)}\|^2 + \|\gamma\omega_{t-1}^{(m)}\|^2\big] + 4L^2\alpha_{t-1}^2 \sum_{m=1}^{M} \mathbb{E}\|\tau q_{t-1}^{(m)}\|^2
\end{aligned}
\tag{15}
$$

where $(a)$ follows Proposition E.2; $(b)$ follows Proposition C.1 and the fact that $\hat{x}_t^{(m)} = x_t^{(m)}$ when $t \neq \bar{t}_s$; Next for the second term, we have:

$$
\begin{aligned}
\sum_{m=1}^{M} \mathbb{E}\big\|\mu_{t-1,\mathcal{B}_x}^{(m)} - \bar{\mu}_{t-1,\mathcal{B}_x}\big\|^2 &= \sum_{m=1}^{M} \mathbb{E}\big\|\mu_{t-1,\mathcal{B}_x}^{(m)} - \mu_{t-1}^{(m)} - \big(\bar{\mu}_{t-1,\mathcal{B}_x} - \bar{\mu}_{t-1}\big) + \mu_{t-1}^{(m)} - \bar{\mu}_{t-1}\big\|^2 \\
&\overset{(a)}{\leq} 2\sum_{m=1}^{M} \mathbb{E}\big\|\mu_{t-1,\mathcal{B}_x}^{(m)} - \mu_{t-1}^{(m)} - \big(\bar{\mu}_{t-1,\mathcal{B}_x} - \bar{\mu}_{t-1}\big)\big\|^2 + 2\sum_{m=1}^{M} \mathbb{E}\big\|\mu_{t-1}^{(m)} - \bar{\mu}_{t-1}\big\|^2 \\
&\overset{(b)}{\leq} 2\underbrace{\sum_{m=1}^{M} \mathbb{E}\big\|\mu_{t-1,\mathcal{B}_x}^{(m)} - \mu_{t-1}^{(m)}\big\|^2}_{T_1} + 2\underbrace{\sum_{m=1}^{M} \mathbb{E}\big\|\mu_{t-1}^{(m)} - \bar{\mu}_{t-1}\big\|^2}_{T_2}
\end{aligned}
\tag{16}
$$

Note for the term $T_1$ of Eq. 16, we have $\mathbb{E}\|\mu_{t-1,\mathcal{B}_x}^{(m)} - \mu_{t-1}^{(m)}\|^2 \le \frac{\sigma^2}{b_x}$ by the bounded variance assumption; Next for the term $T_2$, we have:

$$T_2 = \sum_{m=1}^{M} \|\nabla_x f^{(m)}(x_{t-1}^{(m)}, y_{t-1}^{(m)}) - \nabla_{xy} g^{(m)}(x_{t-1}^{(m)}, y_{t-1}^{(m)})u_{t-1}^{(m)}$$

$$- \frac{1}{M}\sum_{j=1}^{M}\left(\nabla_x f^{(j)}(x_{t-1}^{(j)}, y_{t-1}^{(j)}) - \nabla_{xy} g^{(j)}(x_{t-1}^{(j)}, y_{t-1}^{(j)})u_{t-1}^{(j)}\right)\|^2$$

$$\le 16\left(L^2 + \frac{2L_{xy}^2 C_f^2}{\mu^2}\right)\sum_{m=1}^{M}\mathbb{E}[\|x_t^{(m)} - \bar{x}_t\|^2 + \|y_t^{(m)} - \bar{y}_t\|^2] + 4L^2\sum_{m=1}^{M}\mathbb{E}\|u_t^{(m)} - \bar{u}_t\|^2 + 4M\zeta_f^2 + \frac{8MC_f^2\zeta_{g,xy}^2}{\mu^2}$$

Finally, combine Eq. 15, Eq. 16 with Eq. 14 and use the fact that $I \ge 1$, we have:

$$\sum_{m=1}^{M}\mathbb{E}\|\hat{\nu}_t^{(m)} - \bar{\nu}_t\|^2$$

$$\le \left(1 + \frac{1}{I}\right)\sum_{m=1}^{M}\mathbb{E}\|\nu_{t-1}^{(m)} - \bar{\nu}_{t-1}\|^2 + 8I\tilde{L}_1^2\alpha_{t-1}^2\sum_{m=1}^{M}\mathbb{E}\big[\underbrace{\|\eta\nu_{t-1}^{(m)}\|^2}_{T_1} + \|\gamma\omega_{t-1}^{(m)}\|^2\big] + 16IL^2\alpha_{t-1}^2\sum_{m=1}^{M}\mathbb{E}\|\tau q_{t-1}^{(m)}\|^2$$

$$+ 128I(c_\nu\alpha_{t-1}^2)^2\tilde{L}_1^2\sum_{m=1}^{M}\mathbb{E}[\|x_t^{(m)} - \bar{x}_t\|^2 + \|y_t^{(m)} - \bar{y}_t\|^2] + 32I(c_\nu\alpha_{t-1}^2)^2 L^2\sum_{m=1}^{M}\mathbb{E}\|u_t^{(m)} - \bar{u}_t\|^2$$

$$+ 8IM(c_\nu\alpha_{t-1}^2)^2\frac{\sigma^2}{b_x} + 32IM(c_\nu\alpha_{t-1}^2)^2\zeta_f^2 + 64I(c_\nu\alpha_{t-1}^2)^2 M\frac{C_f^2\zeta_{g,xy}^2}{\mu^2}$$

We separate the term $T_1$ with triangle inequality to get:

$$\sum_{m=1}^{M}\mathbb{E}\|\hat{\nu}_t^{(m)} - \bar{\nu}_t\|^2$$

$$\le \left(1 + \frac{1}{I} + 16I\tilde{L}_1^2\eta^2\alpha_{t-1}^2\right)\sum_{m=1}^{M}\mathbb{E}\|\nu_{t-1}^{(m)} - \bar{\nu}_{t-1}\|^2$$

$$+ 8I\tilde{L}_1^2\alpha_{t-1}^2\sum_{m=1}^{M}\mathbb{E}\big[2\|\eta\bar{\nu}_{t-1}\|^2 + \|\gamma\omega_{t-1}^{(m)}\|^2\big] + 16IL^2\alpha_{t-1}^2\sum_{m=1}^{M}\mathbb{E}\|\tau q_{t-1}^{(m)}\|^2$$

$$+ 128I(c_\nu\alpha_{t-1}^2)^2\tilde{L}_1^2\sum_{m=1}^{M}\mathbb{E}[\|x_t^{(m)} - \bar{x}_t\|^2 + \|y_t^{(m)} - \bar{y}_t\|^2] + 32I(c_\nu\alpha_{t-1}^2)^2 L^2\sum_{m=1}^{M}\mathbb{E}\|u_t^{(m)} - \bar{u}_t\|^2$$

$$+ 8IM(c_\nu\alpha_{t-1}^2)^2\frac{\sigma^2}{b_x} + 32IM(c_\nu\alpha_{t-1}^2)^2\zeta_f^2 + 64I(c_\nu\alpha_{t-1}^2)^2 M\frac{C_f^2\zeta_{g,xy}^2}{\mu^2}$$

This completes the proof. $\qquad\square$

**Lemma C.9.** *Suppose $\gamma\alpha_t < \frac{1}{16IL}$, then for $t \ne \bar{t}_s, s \in [S]$, we have:*

$$\sum_{m=1}^{M}\mathbb{E}\|\omega_t^{(m)} - \bar{\omega}_t\|^2 \le \left(1 + \frac{33}{32I}\right)\sum_{m=1}^{M}\mathbb{E}\|\omega_{t-1}^{(m)} - \bar{\omega}_{t-1}\|^2 + 4IL^2\alpha_{t-1}^2\sum_{m=1}^{M}\mathbb{E}\big[2\|\gamma\bar{\omega}_{t-1}\|^2 + \|\eta\nu_{t-1}^{(m)}\|^2\big]$$

$$+ 8IM(c_\omega\alpha_{t-1}^2)^2\frac{\sigma^2}{b_y} + 16IM(c_\omega\alpha_{t-1}^2)^2\zeta_g^2 + 16IL^2(c_\omega\alpha_{t-1}^2)^2\sum_{m=1}^{M}\mathbb{E}\big[\|x_{t-1}^{(m)} - \bar{x}_{t-1}\|^2\big]$$

$$+ 16IL^2(c_\omega\alpha_{t-1}^2)^2\sum_{m=1}^{M}\mathbb{E}\big[\|y_{t-1}^{(m)} - \bar{y}_{t-1}\|^2\big]$$

*where the expectation is w.r.t the stochasticity of the algorithm.*

*Proof.* By the update step in Line 7 of Algorithm 1, for $t \neq \bar{t}_s$, we have:

$$
\mathbb{E}\|\hat{\omega}_t^{(m)} - \bar{\omega}_t\|^2 = \mathbb{E}\big\|(1 - c_\omega \alpha_{t-1}^2)\big(\omega_{t-1}^{(m)} - \bar{\omega}_{t-1}\big) + \nabla_y g^{(m)}(x_t^{(m)}, y_t^{(m)}, \mathcal{B}_y) - \frac{1}{M}\sum_{j=1}^M \nabla_y g^{(j)}(x_t^{(j)}, y_t^{(j)}, \mathcal{B}_y)
$$

$$
- (1 - c_\omega \alpha_{t-1}^2)\big(\nabla_y g^{(m)}(x_{t-1}^{(m)}, y_{t-1}^{(m)}, \mathcal{B}_y) - \frac{1}{M}\sum_{j=1}^M \nabla_y g^{(j)}(x_{t-1}^{(j)}, y_{t-1}^{(j)}, \mathcal{B}_y)\big)\big\|^2
$$

$$
\leq (1 + \frac{1}{I})(1 - c_\omega \alpha_{t-1}^2)^2 \mathbb{E}\|\omega_{t-1}^{(m)} - \bar{\omega}_{t-1}\|^2
$$

$$
+ (1 + I)\mathbb{E}\big\|\nabla_y g^{(m)}(x_t^{(m)}, y_t^{(m)}, \mathcal{B}_y) - \frac{1}{M}\sum_{j=1}^M \nabla_y g^{(j)}(x_t^{(j)}, y_t^{(j)}, \mathcal{B}_y)
$$

$$
- (1 - c_\omega \alpha_{t-1}^2)\big(\nabla_y g^{(m)}(x_{t-1}^{(m)}, y_{t-1}^{(m)}, \mathcal{B}_y) - \frac{1}{M}\sum_{j=1}^M \nabla_y g^{(j)}(x_{t-1}^{(j)}, y_{t-1}^{(j)}, \mathcal{B}_y)\big)\big\|^2
\tag{17}
$$

where the inequality follows from the the generalized triangle inequality and the condition that $c_\omega \alpha_t^2 < 1$.

Next we denote the second term in Eq. 17 as $T_1$, then we have:

$$
T_1 \leq 2\sum_{m=1}^M \mathbb{E}\big\|\nabla_y g^{(m)}(x_t^{(m)}, y_t^{(m)}, \mathcal{B}_y) - \frac{1}{M}\sum_{j=1}^M \nabla_y g^{(m)}(x_t^{(j)}, y_t^{(j)}, \mathcal{B}_y)
$$

$$
- \big(\nabla_y g^{(m)}(x_{t-1}^{(m)}, y_{t-1}^{(m)}, \mathcal{B}_y) - \frac{1}{M}\sum_{j=1}^M \nabla_y g^{(j)}(x_{t-1}^{(j)}, y_{t-1}^{(m)}, \mathcal{B}_y)\big)\big\|^2
$$

$$
+ 2(c_\omega \alpha_{t-1}^2)^2 \sum_{m=1}^M \mathbb{E}\big\|\nabla_y g^{(m)}(x_{t-1}^{(m)}, y_{t-1}^{(m)}, \mathcal{B}_y) - \frac{1}{M}\sum_{j=1}^M \nabla_y g^{(j)}(x_{t-1}^{(j)}, y_{t-1}^{(j)}, \mathcal{B}_y)\big\|^2
$$

We bound the two terms separately, we denote them as $T_{1,1}$ and $T_{1,2}$ separately, then we have:

$$
T_{1,1} \overset{(a)}{\leq} \sum_{m=1}^M \mathbb{E}\big\|\nabla_y g^{(m)}(x_t^{(m)}, y_t^{(m)}, \mathcal{B}_y) - \nabla_y g^{(m)}(x_{t-1}^{(m)}, y_{t-1}^{(m)}, \mathcal{B}_y)\big\|^2
$$

$$
\overset{(b)}{\leq} L^2 \sum_{m=1}^M \mathbb{E}\big[\|x_t^{(m)} - x_{t-1}^{(m)}\|^2 + \|y_t^{(m)} - y_{t-1}^{(m)}\|^2\big] \leq L^2 \alpha_{t-1}^2 \sum_{m=1}^M \mathbb{E}\big[\|\eta\nu_{t-1}^{(m)}\|^2 + \|\gamma\omega_{t-1}^{(m)}\|^2\big]
\tag{18}
$$

where $(a)$ follows Proposition E.2; $(b)$ follows Proposition C.1.b) and the fact that $\hat{x}_t^{(m)} = x_t^{(m)}$ and $\hat{y}_t^{(m)} = y_t^{(m)}$ when $t \neq \bar{t}_s$; Next for the second term, we have:

$$T_{1,2} = \sum_{m=1}^{M} \mathbb{E} \big\| \nabla_y g^{(m)}(x_{t-1}^{(m)}, y_{t-1}^{(m)}, \mathcal{B}_y) - \nabla_y g^{(m)}(x_{t-1}^{(m)}, y_{t-1}^{(m)})$$

$$- \frac{1}{M} \sum_{j=1}^{M} \big( \nabla_y g^{(j)}(x_{t-1}^{(j)}, y_{t-1}^{(j)}, \mathcal{B}_y) - \nabla_y g^{(j)}(x_{t-1}^{(j)}, y_{t-1}^{(j)}) \big)$$

$$+ \nabla_y g^{(m)}(x_{t-1}^{(m)}, y_{t-1}^{(m)}) - \frac{1}{M} \sum_{j=1}^{M} \nabla_y g^{(j)}(x_{t-1}^{(j)}, y_{t-1}^{(j)}) \big\|^2$$

$$\overset{(b)}{\leq} 2 \sum_{m=1}^{M} \mathbb{E} \big\| \nabla_y g^{(m)}(x_{t-1}^{(m)}, y_{t-1}^{(m)}, \mathcal{B}_y) - \nabla_y g^{(m)}(x_{t-1}^{(m)}, y_{t-1}^{(m)}) \big\|^2$$

$$+ 4 \sum_{m=1}^{M} \frac{1}{M} \sum_{j=1}^{M} \mathbb{E} \big\| \nabla g^{(m)}(\bar{x}_{t-1}, \bar{y}_{t-1}) - \nabla_y g^{(j)}(\bar{x}_{t-1}, \bar{y}_{t-1}) \big\|^2$$

$$+ 4 \sum_{m=1}^{M} \mathbb{E} \big\| \nabla_y g^{(m)}(x_{t-1}^{(m)}, y_{t-1}^{(m)}) - \nabla_y g^{(m)}(\bar{x}_{t-1}, \bar{y}_{t-1})$$

$$+ \frac{1}{M} \sum_{j=1}^{M} \nabla_y g^{(j)}(\bar{x}_{t-1}, \bar{y}_{t-1}) - \nabla_y g^{(j)}(x_{t-1}^{(j)}, y_{t-1}^{(j)}) \big\|^2 \tag{19}$$

We denote the three terms above as $T_{1,2,1} - T_{1,2,3}$ respectively. For the term $T_{1,2,1}$ of Eq. 19, we have $T_{1,2,1} \leq 2M\sigma^2/b_y$ by the bounded variance assumption; For the term $T_{1,2,2}$ of Eq. 19, by the bounded intra-node heterogeneity assumption we have $T_{1,2,2} \leq 4M\zeta_g^2$. Finally, For the term $T_{1,2,3}$ of Eq. 19:

$$T_{1,2,3} \leq 4 \sum_{m=1}^{M} \mathbb{E} \big\| \nabla_y g^{(m)}(x_{t-1}^{(m)}, y_{t-1}^{(m)}) - \nabla_y g^{(m)}(\bar{x}_{t-1}, \bar{y}_{t-1}) \big\|^2$$

$$\leq 4L^2 \sum_{m=1}^{M} \mathbb{E} \big[ \|x_{t-1}^{(m)} - \bar{x}_{t-1}\|^2 \big] + 4L^2 \sum_{m=1}^{M} \mathbb{E} \big[ \|y_{t-1}^{(m)} - \bar{y}_{t-1}\|^2 \big]$$

Finally, combine Eq. 17, Eq. 18 with Eq. 19 and use the fact that $I \geq 1$, we have:

$$\sum_{m=1}^{M} \mathbb{E} \|\hat{\omega}_t^{(m)} - \bar{\omega}_t\|^2 \leq \big(1 + \frac{1}{I}\big) \sum_{m=1}^{M} \mathbb{E} \|\omega_{t-1}^{(m)} - \bar{\omega}_{t-1}\|^2 + 4IL^2\alpha_{t-1}^2 \sum_{m=1}^{M} \mathbb{E} \big[ \underbrace{\|\gamma\omega_{t-1}^{(m)}\|^2}_{T_1} + \|\eta\nu_{t-1}^{(m)}\|^2 \big]$$

$$+ 8IM(c_\omega\alpha_{t-1}^2)^2 \frac{\sigma^2}{b_y}$$

$$+ 16IM(c_\omega\alpha_{t-1}^2)^2 \zeta_g^2 + 16IL^2(c_\omega\alpha_{t-1}^2)^2 \sum_{m=1}^{M} \mathbb{E} \big[ \|x_{t-1}^{(m)} - \bar{x}_{t-1}\|^2 \big]$$

$$+ 16IL^2(c_\omega\alpha_{t-1}^2)^2 \sum_{m=1}^{M} \mathbb{E} \big[ \|y_{t-1}^{(m)} - \bar{y}_{t-1}\|^2 \big]$$

We separate the term $T_1$ with triangle inequality to get:

$$\sum_{m=1}^{M} \mathbb{E}\|\hat{\omega}_t^{(m)} - \bar{\omega}_t\|^2 \leq \left(1 + \frac{1}{I} + 8IL^2\gamma^2\alpha_{t-1}^2\right) \sum_{m=1}^{M} \mathbb{E}\|\omega_{t-1}^{(m)} - \bar{\omega}_{t-1}\|^2$$

$$+ 4IL^2\alpha_{t-1}^2 \sum_{m=1}^{M} \mathbb{E}\left[2\|\gamma\bar{\omega}_{t-1}\|^2 + \|\eta\nu_{t-1}^{(m)}\|^2\right]$$

$$+ 8IM(c_\omega\alpha_{t-1}^2)^2 \frac{\sigma^2}{b_y}$$

$$+ 16IM(c_\omega\alpha_{t-1}^2)^2\zeta_g^2 + 16IL^2(c_\omega\alpha_{t-1}^2)^2 \sum_{m=1}^{M} \mathbb{E}\left[\|x_{t-1}^{(m)} - \bar{x}_{t-1}\|^2\right]$$

$$+ 16IL^2(c_\omega\alpha_{t-1}^2)^2 \sum_{m=1}^{M} \mathbb{E}\left[\|y_{t-1}^{(m)} - \bar{y}_{t-1}\|^2\right]$$

This completes the proof. $\qquad\square$

**Lemma C.10.** *Suppose $\tau\alpha_t < \frac{1}{32IL}$, then for $t \neq \bar{t}_s, s \in [S]$, we have:*

$$\sum_{m=1}^{M} \mathbb{E}\|\hat{q}_t^{(m)} - \bar{q}_t\|^2 \leq \left(1 + \frac{33}{32I}\right) \sum_{m=1}^{M} \mathbb{E}\|q_{t-1}^{(m)} - \bar{q}_{t-1}\|^2 + 8I\tilde{L}_2^2\alpha_{t-1}^2 \sum_{m=1}^{M} \mathbb{E}\left[\|\gamma\omega_{t-1}^{(m)}\|^2 + \|\eta\nu_{t-1}^{(m)}\|^2\right]$$

$$+ 32IL^2\alpha_{t-1}^2 \sum_{m=1}^{M} \mathbb{E}\|\tau^2\bar{q}_{t-1}\|^2 + 8IM(c_u\alpha_{t-1}^2)^2 \frac{\sigma^2}{b_x}$$

$$+ 16IM(c_u\alpha_{t-1}^2)^2\zeta_f^2 + 32IM(c_u\alpha_{t-1}^2)^2 \frac{C_f^2\zeta_{g,yy}^2}{\mu^2}$$

$$+ 64I(c_u\alpha_{t-1}^2)^2\tilde{L}_2^2 \sum_{m=1}^{M} \mathbb{E}\left[\left\|x_t^{(m)} - \bar{x}_t\right\|^2 + \left\|y_t^{(m)} - \bar{y}_t\right\|^2\right]$$

$$+ 16I(c_u\alpha_{t-1}^2)^2L^2 \sum_{m=1}^{M} \mathbb{E}\left\|u_t^{(m)} - \bar{u}_t\right\|^2$$

*where the expectation is w.r.t the stochasticity of the algorithm.*

*Proof.* For $t \neq \bar{t}_s$, we have:

$$\mathbb{E}\|\hat{q}_t^{(m)} - \bar{q}_t\|^2 = \mathbb{E}\left\|(1 - c_u\alpha_{t-1}^2)(q_{t-1}^{(m)} - \bar{q}_{t-1}) + p_{t,\mathcal{B}_x}^{(m)} - \bar{p}_{t,\mathcal{B}_x} - (1 - c_u\alpha_{t-1}^2)(p_{t-1,\mathcal{B}_x}^{(m)} - \bar{p}_{t-1,\mathcal{B}_x})\right\|^2$$

$$\leq (1 + \frac{1}{I})(1 - c_u\alpha_{t-1}^2)^2\mathbb{E}\|q_{t-1}^{(m)} - \bar{q}_{t-1}\|^2$$

$$+ (1 + I)\mathbb{E}\left\|p_{t,\mathcal{B}_x}^{(m)} - \bar{p}_{t,\mathcal{B}_x} - (1 - c_u\alpha_{t-1}^2)(p_{t-1,\mathcal{B}_x}^{(m)} - \bar{p}_{t-1,\mathcal{B}_x})\right\|^2$$

where the inequality follows from the the generalized triangle inequality and the condition that $c_u\alpha_t^2 < 1$.

Next we sum over $M$ for the second term in Eq. 17 and denote it as $T_1$, then we have:

$$T_1 \leq 2\sum_{m=1}^{M} \mathbb{E}\left\|p_{t,\mathcal{B}_x}^{(m)} - \bar{p}_{t,\mathcal{B}_x} - (p_{t-1,\mathcal{B}_x}^{(m)} - \bar{p}_{t-1,\mathcal{B}_x})\right\|^2 + 2(c_u\alpha_{t-1}^2)^2 \sum_{m=1}^{M} \mathbb{E}\left\|p_{t-1,\mathcal{B}_x}^{(m)} - \bar{p}_{t-1,\mathcal{B}_x}\right\|^2$$

We bound the two terms separately, we denote them as $T_{1,1}$ and $T_{1,2}$ separately, then we have:

$$T_{1,1} \overset{(a)}{\leq} \sum_{m=1}^{M} \mathbb{E}\big\|p_{t,\mathcal{B}_x}^{(m)} - p_{t-1,\mathcal{B}_x}^{(m)}\big\|^2$$

$$\overset{(b)}{\leq} 2\big(L^2 + \frac{2L_{y^2}^2 C_f^2}{\mu^2}\big) \sum_{m=1}^{M} \mathbb{E}\big[\|x_t^{(m)} - x_{t-1}^{(m)}\|^2 + \|y_t^{(m)} - y_{t-1}^{(m)}\|^2\big] + 4L^2 \sum_{m=1}^{M} \mathbb{E}\|u_t^{(m)} - u_{t-1}^{(m)}\|^2$$

$$\leq 2\tilde{L}_2^2 \alpha_{t-1}^2 \sum_{m=1}^{M} \mathbb{E}\big[\|\eta\nu_{t-1}^{(m)}\|^2 + \|\gamma\omega_{t-1}^{(m)}\|^2\big] + 4L^2 \alpha_{t-1}^2 \sum_{m=1}^{M} \mathbb{E}\|\tau^2 q_{t-1}^{(m)}\|^2$$

where $(a)$ follows Proposition E.2; $(b)$ follows Proposition C.1 and the fact that $\hat{x}_t^{(m)} = x_t^{(m)}$ and $\hat{y}_t^{(m)} = y_t^{(m)}$ when $t \neq \bar{t}_s$; Next for the second term, we have:

$$T_{1,2} = \sum_{m=1}^{M} \mathbb{E}\big\|p_{t-1,\mathcal{B}_x}^{(m)} - p_{t-1}^{(m)} - (\bar{p}_{t-1,\mathcal{B}_x} - \bar{p}_{t-1}) + p_{t-1}^{(m)} - \bar{p}_{t-1}\big\|^2$$

$$\overset{(b)}{\leq} 2 \sum_{m=1}^{M} \mathbb{E}\big\|p_{t-1,\mathcal{B}_x}^{(m)} - p_{t-1}^{(m)}\big\|^2 + 2 \sum_{m=1}^{M} \mathbb{E}\big\|p_{t-1}^{(m)} - \bar{p}_{t-1}\big\|^2$$

We denote the two terms above as $T_{1,2,1}, T_{1,2,2}$ respectively. For the term $T_{1,2,1}$ of Eq. 19, we have $T_{1,2,1} \leq 2M\sigma^2/b_x$ by the bounded variance assumption; For the term $T_{1,2,2}$ of Eq. 19, we have

$$T_{1,2,2} \leq 16\big(L^2 + \frac{2L_{y^2}^2 C_f^2}{\mu^2}\big) \sum_{m=1}^{M} \mathbb{E}\big[\|x_t^{(m)} - \bar{x}_t\|^2 + \|y_t^{(m)} - \bar{y}_t\|^2\big]$$

$$+ 4L^2 \sum_{m=1}^{M} \mathbb{E}\|u_t^{(m)} - \bar{u}_t\|^2 + 4M\zeta_f^2 + \frac{8MC_f^2 \zeta_{g,yy}^2}{\mu^2}$$

Finally, combine everythin together and use the fact that $I \geq 1$, we have:

$$\sum_{m=1}^{M} \mathbb{E}\|\hat{q}_t^{(m)} - \bar{q}_t\|^2 \leq \big(1 + \frac{1}{I}\big) \sum_{m=1}^{M} \mathbb{E}\|q_{t-1}^{(m)} - \bar{q}_{t-1}\|^2 + 8I\tilde{L}_2^2 \alpha_{t-1}^2 \sum_{m=1}^{M} \mathbb{E}\big[\|\gamma\omega_{t-1}^{(m)}\|^2 + \|\eta\nu_{t-1}^{(m)}\|^2\big]$$

$$+ 16IL^2 \alpha_{t-1}^2 \sum_{m=1}^{M} \mathbb{E} \underbrace{\|\tau^2 q_{t-1}^{(m)}\|^2}_{T_1}$$

$$+ 8IM(c_u \alpha_{t-1}^2)^2 \frac{\sigma^2}{b_x} + 16IM(c_u \alpha_{t-1}^2)^2 \zeta_f^2 + 32IM(c_u \alpha_{t-1}^2)^2 \frac{C_f^2 \zeta_{g,yy}^2}{\mu^2}$$

$$+ 64I(c_u \alpha_{t-1}^2)^2 \tilde{L}_2^2 \sum_{m=1}^{M} \mathbb{E}\big[\|x_t^{(m)} - \bar{x}_t\|^2$$

$$+ \|y_t^{(m)} - \bar{y}_t\|^2\big] + 16I(c_u \alpha_{t-1}^2)^2 L^2 \sum_{m=1}^{M} \mathbb{E}\|u_t^{(m)} - \bar{u}_t\|^2$$

We separate the term $T_1$ with triangle inequality to get:

$$\sum_{m=1}^{M} \mathbb{E}\|\hat{q}_t^{(m)} - \bar{q}_t\|^2 \leq \left(1 + \frac{1}{I} + 32IL^2\tau^2\alpha_{t-1}^2\right) \sum_{m=1}^{M} \mathbb{E}\|q_{t-1}^{(m)} - \bar{q}_{t-1}\|^2$$

$$+ 8I\tilde{L}_2^2\alpha_{t-1}^2 \sum_{m=1}^{M} \mathbb{E}\left[\|\gamma\omega_{t-1}^{(m)}\|^2 + \|\eta\nu_{t-1}^{(m)}\|^2\right]$$

$$+ 32IL^2\alpha_{t-1}^2 \sum_{m=1}^{M} \mathbb{E}\|\tau^2\bar{q}_{t-1}\|^2 + 8IM(c_u\alpha_{t-1}^2)^2\frac{\sigma^2}{b_x}$$

$$+ 16IM(c_u\alpha_{t-1}^2)^2\zeta_f^2 + 32IM(c_u\alpha_{t-1}^2)^2\frac{C_f^2\zeta_{g,yy}^2}{\mu^2}$$

$$+ 64I(c_u\alpha_{t-1}^2)^2\tilde{L}_2^2 \sum_{m=1}^{M} \mathbb{E}\left[\|x_t^{(m)} - \bar{x}_t\|^2 + \|y_t^{(m)} - \bar{y}_t\|^2\right]$$

$$+ 16I(c_u\alpha_{t-1}^2)^2 L^2 \sum_{m=1}^{M} \mathbb{E}\|u_t^{(m)} - \bar{u}_t\|^2$$

This completes the proof. □

Next, to simply the notation, we denote $A_t = \mathbb{E}\|\bar{\nu}_t - \bar{\mu}_t\|^2$, $B_t = \mathbb{E}\|\bar{y}_t - y_{\bar{x}_t}\|^2$, $C_t = \mathbb{E}\|\bar{\omega}_t - \frac{1}{M}\sum_{m=1}^{M}\nabla_y g^{(m)}(x_t^{(m)}, y_t^{(m)})\|^2$, $D_t = \frac{1}{M}\sum_{m=1}^{M}\mathbb{E}\|\nu_t^{(m)} - \bar{\nu}_t\|^2$, $E_t = \mathbb{E}\|\bar{\nu}_t\|^2$, $F_t = \mathbb{E}\|\bar{\omega}_t\|^2$, $G_t = \frac{1}{M}\sum_{m=1}^{M}\mathbb{E}\|\omega_t^{(m)} - \bar{\omega}_t\|^2$, $H_t = \mathbb{E}[\|\bar{q}_t - \bar{p}_t\|^2]$, $I_t = \mathbb{E}[\|\bar{u}_t - u_{\bar{x}_t}\|^2]$, $J_t = \mathbb{E}\|q_t^{(m)} - \bar{q}_t\|^2$, $Q_t = \mathbb{E}\|\bar{q}_t\|^2$.

**Lemma C.11.** *For $\eta < \min(\frac{\tilde{L}^2}{c_\nu}, \frac{\tilde{L}^2}{c_\omega}, \frac{\tilde{L}^2}{c_u}, 1)$, $\gamma < \min(\frac{\tilde{L}^2}{c_\nu}, \frac{\tilde{L}^2}{c_\omega}, \frac{\tilde{L}^2}{c_u}, 1)$, $\tau < \min(\frac{\tilde{L}^2}{c_\nu}, \frac{\tilde{L}^2}{c_u}, \frac{1}{2})$ and $\alpha_t < \frac{1}{16\tilde{L}I}$, where $\tilde{L} = max(\tilde{L}_1, \tilde{L}_2)$, we have:*

$$\sum_{t=\bar{t}_{s-1}}^{\bar{t}_s-1} \alpha_t D_t \leq \sum_{t=\bar{t}_{s-1}}^{\bar{t}_s-1} \left(\alpha_t E_t + \alpha_t F_t + \alpha_t Q_t + \frac{c_\omega^2\alpha_t^3}{\tilde{L}^2}\frac{\sigma^2}{b_y} + \frac{c_\omega^2\alpha_t^3}{\tilde{L}^2}\zeta_g^2\right.$$

$$\left. + \frac{c_\nu^2\alpha_t^3}{\tilde{L}^2}\frac{\sigma^2}{b_x} + \frac{c_u^2\alpha_t^3}{\tilde{L}^2}\frac{\sigma^2}{b_x} + \frac{c_\nu^2\alpha_t^3\zeta_f^2}{\tilde{L}^2} + \frac{c_u^2\alpha_t^3\zeta_f^2}{\tilde{L}^2} + \frac{2c_\nu^2\alpha_t^3}{\tilde{L}^2}\frac{C_f^2\zeta_{g,xy}^2}{\mu^2} + \frac{4c_u^2\alpha_t^3}{\tilde{L}^2}\frac{C_f^2\zeta_{g,yy}^2}{\mu^2}\right)$$

$$\sum_{t=\bar{t}_{s-1}}^{\bar{t}_s-1} \alpha_t G_t \leq \sum_{t=\bar{t}_{s-1}}^{\bar{t}_s-1} \left(\alpha_t E_t + \alpha_t F_t + \alpha_t Q_t + \frac{2c_\omega^2\alpha_t^3}{\tilde{L}^2}\frac{\sigma^2}{b_y} + \frac{2c_\omega^2\alpha_t^3}{\tilde{L}^2}\zeta_g^2\right.$$

$$\left. + \frac{c_\nu^2\alpha_t^3}{\tilde{L}^2}\frac{2\sigma^2}{b_x} + \frac{c_u^2\alpha_t^3}{\tilde{L}^2}\frac{\sigma^2}{b_x} + \frac{c_\nu^2\alpha_t^3\zeta_f^2}{\tilde{L}^2} + \frac{c_u^2\alpha_t^3\zeta_f^2}{\tilde{L}^2} + \frac{c_\nu^2\alpha_t^3}{\tilde{L}^2}\frac{C_f^2\zeta_{g,xy}^2}{\mu^2} + \frac{2c_u^2\alpha_t^3}{\tilde{L}^2}\frac{C_f^2\zeta_{g,yy}^2}{\mu^2}\right)$$

$$\sum_{t=\bar{t}_{s-1}+1}^{\bar{t}_s} \alpha_t J_t \leq \sum_{t=\bar{t}_{s-1}}^{\bar{t}_s-1} \left(\alpha_t F_t + \alpha_t E_t + \alpha_t Q_t + \frac{c_\omega^2\alpha_t^3}{\tilde{L}^2}\frac{\sigma^2}{b_y} + \frac{c_\omega^2\alpha_t^3}{\tilde{L}^2}\zeta_g^2\right.$$

$$\left. + \frac{c_u^2\alpha_t^3}{\tilde{L}^2}\frac{\sigma^2}{b_x} + \frac{c_\nu^2\alpha_t^3}{\tilde{L}^2}\frac{\sigma^2}{b_x} + \frac{c_u^2\alpha_t^3\zeta_f^2}{\tilde{L}^2} + \frac{c_\nu^2\alpha_t^3\zeta_f^2}{2\tilde{L}^2} + \frac{c_\nu^2\alpha_t^3}{\tilde{L}^2}\frac{C_f^2\zeta_{g,xy}^2}{\mu^2} + \frac{40c_u^2\alpha_t^3}{\tilde{L}^2}\frac{C_f^2\zeta_{g,yy}^2}{\mu^2}\right)$$

*Proof.* Based on Lemma C.8, for $t \neq \bar{t}_s$, we have:

$$D_t \leq \left(1 + \frac{17}{16I}\right)D_{t-1} + 16I\tilde{L}_1^2\alpha_{t-1}^2\eta^2 E_{t-1} + 16I\tilde{L}_1^2\alpha_{t-1}^2\gamma^2 F_{t-1} + 16I\tilde{L}_1^2\alpha_{t-1}^2\gamma^2 G_{t-1} + 32IL^2\tau^2\alpha_{t-1}^2 J_{t-1}$$

$$+ 32IL^2\tau^2\alpha_{t-1}^2 Q_{t-1} + 8Ic_\nu^2\alpha_{t-1}^4\frac{\sigma^2}{b_x} + 32Ic_\nu^2\alpha_{t-1}^4\zeta_f^2 + 64Ic_\nu^2\alpha_{t-1}^4\frac{C_f^2\zeta_{g,xy}^2}{\mu^2}$$

$$+ 128I^2\tilde{L}_1^2\eta^2 c_\nu^2\alpha_{t-1}^4 \sum_{\ell=\bar{t}_{s-1}}^{t-2} \alpha_l^2 D_l + 128I^2\tilde{L}_1^2\gamma^2 c_\nu^2\alpha_{t-1}^4 \sum_{\ell=\bar{t}_{s-1}}^{t-2} \alpha_l^2 G_l + 32I^2L^2\tau^2 c_\nu^2\alpha_{t-1}^4 \sum_{\ell=\bar{t}_{s-1}}^{t-2} \alpha_l^2 J_l$$

while for $t = \bar{t}_s$, we have $D_{\bar{t}_s} = 1/M \sum_{m=1}^{M} \mathbb{E}\|\nu_{\bar{t}_s}^{(m)} - \bar{\nu}_{\bar{t}_s}\|^2 = 0$. Apply the above equation recursively from $\bar{t}_{s-1} + 1$ to $t$. so we have:

$$
\begin{aligned}
D_t &\leq \sum_{\ell=\bar{t}_{s-1}}^{\ell} \left(1 + \frac{17}{16I}\right)^{t-\ell} \big(16I\tilde{L}_1^2\alpha_\ell^2\eta^2 E_\ell + 16I\tilde{L}_1^2\alpha_\ell^2\gamma^2 F_\ell + 16I\tilde{L}_1^2\alpha_\ell^2\gamma^2 G_\ell + 32IL^2\tau^2\alpha_\ell^2 J_\ell \\
&\quad + 32IL^2\tau^2\alpha_\ell^2 Q_\ell + 8Ic_\nu^2\alpha_\ell^4\frac{\sigma^2}{b_x} + 32Ic_\nu^2\alpha_\ell^4\zeta_f^2 + 64Ic_\nu^2\alpha_\ell^4\frac{C_f^2\zeta_{g,xy}^2}{\mu^2} \\
&\quad + 128I^2\tilde{L}_1^2\eta^2 c_\nu^2\alpha_\ell^4 \sum_{\bar{\ell}=\bar{t}_{s-1}}^{\ell-1} \alpha_{\bar{\ell}}^2 D_{\bar{\ell}} + 128I^2\tilde{L}_1^2\gamma^2 c_\nu^2\alpha_\ell^4 \sum_{\bar{\ell}=\bar{t}_{s-1}}^{\ell-1} \alpha_{\bar{\ell}}^2 G_{\bar{\ell}} + 32I^2L^2\tau^2 c_\nu^2\alpha_\ell^4 \sum_{\bar{\ell}=\bar{t}_{s-1}}^{\ell-1} \alpha_{\bar{\ell}}^2 J_{\bar{\ell}}\big) \\
&\leq \sum_{\ell=\bar{t}_{s-1}}^{t-1} \big(48I\tilde{L}_1^2\alpha_\ell^2\eta^2 E_\ell + 48I\tilde{L}_1^2\alpha_\ell^2\gamma^2 F_\ell + 48I\tilde{L}_1^2\alpha_\ell^2\gamma^2 G_\ell + 96IL^2\tau^2\alpha_\ell^2 J_\ell \\
&\quad + 96IL^2\tau^2\alpha_\ell^2 Q_\ell + 24Ic_\nu^2\alpha_\ell^4\frac{\sigma^2}{b_x} + 96Ic_\nu^2\alpha_\ell^4\zeta_f^2 + 192Ic_\nu^2\alpha_\ell^4\frac{C_f^2\zeta_{g,xy}^2}{\mu^2} \\
&\quad + 384I^2\tilde{L}_1^2\eta^2 c_\nu^2\alpha_\ell^4 \sum_{\bar{\ell}=\bar{t}_{s-1}}^{\ell-1} \alpha_{\bar{\ell}}^2 D_{\bar{\ell}} + 384I^2\tilde{L}_1^2\gamma^2 c_\nu^2\alpha_\ell^4 \sum_{\bar{\ell}=\bar{t}_{s-1}}^{\ell-1} \alpha_{\bar{\ell}}^2 G_{\bar{\ell}} + 96I^2L^2\tau^2 c_\nu^2\alpha_\ell^4 \sum_{\bar{\ell}=\bar{t}_{s-1}}^{\ell-1} \alpha_{\bar{\ell}}^2 J_{\bar{\ell}}\big)
\end{aligned}
$$

The second inequality uses the fact that $t - l \leq I$ and the inequality $log(1 + a/x) \leq a/x$ for $x > -a$, so we have $(1 + a/x)^x \leq e^a$, Then we choose $a = 17/16$ and $x = I$. Finally, we use the fact that $e^{17/16} \leq 3$.

Next we multiply $\alpha_t$ over both sides and take sum from $\bar{t}_{s-1} + 1$ to $\bar{t}_s$, we have:

$$
\begin{aligned}
&\sum_{t=\bar{t}_{s-1}+1}^{\bar{t}_s} \alpha_t D_t \\
&\leq \sum_{t=\bar{t}_{s-1}}^{\bar{t}_s-1} \alpha_t \sum_{\ell=\bar{t}_{s-1}}^{t-1} \big(48I\tilde{L}_1^2\alpha_\ell^2\eta^2 E_\ell + 48I\tilde{L}_1^2\alpha_\ell^2\gamma^2 F_\ell + 48I\tilde{L}_1^2\alpha_\ell^2\gamma^2 G_\ell + 96IL^2\tau^2\alpha_\ell^2 J_\ell \\
&\quad + 96IL^2\tau^2\alpha_\ell^2 Q_\ell + 24Ic_\nu^2\alpha_\ell^4\frac{\sigma^2}{b_x} + 96Ic_\nu^2\alpha_\ell^4\zeta_f^2 + 192Ic_\nu^2\alpha_\ell^4\frac{C_f^2\zeta_{g,xy}^2}{\mu^2} \\
&\quad + 384I^2\tilde{L}_1^2\eta^2 c_\nu^2\alpha_\ell^4 \sum_{\bar{\ell}=\bar{t}_{s-1}}^{\ell-1} \alpha_{\bar{\ell}}^2 D_{\bar{\ell}} + 384I^2\tilde{L}_1^2\gamma^2 c_\nu^2\alpha_\ell^4 \sum_{\bar{\ell}=\bar{t}_{s-1}}^{\ell-1} \alpha_{\bar{\ell}}^2 G_{\bar{\ell}} + 96I^2L^2\tau^2 c_\nu^2\alpha_\ell^4 \sum_{\bar{\ell}=\bar{t}_{s-1}}^{\ell-1} \alpha_{\bar{\ell}}^2 J_{\bar{\ell}}\big) \\
&\overset{(a)}{\leq} \sum_{t=\bar{t}_{s-1}}^{\bar{t}_s-1} \big(3I\tilde{L}_1\alpha_t^2\eta^2 E_t + 3I\tilde{L}_1\alpha_t^2\gamma^2 F_t + 3I\tilde{L}_1\alpha_t^2\gamma^2 G_t + 6IL\tau^2\alpha_t^2 J_t \\
&\quad + 6IL\tau^2\alpha_t^2 Q_t + \frac{3Ic_\nu^2\alpha_t^4}{2\tilde{L}}\frac{\sigma^2}{b_x} + \frac{6Ic_\nu^2\alpha_t^4\zeta_f^2}{\tilde{L}} + \frac{12Ic_\nu^2\alpha_t^4}{\tilde{L}}\frac{C_f^2\zeta_{g,xy}^2}{\mu^2} \\
&\quad + 32I^2\tilde{L}_1\eta^2 c_\nu^2\alpha_t^4 \sum_{\ell=\bar{t}_{s-1}}^{t-1} \alpha_\ell^2 D_\ell + 32I^2\tilde{L}_1\gamma^2 c_\nu^2\alpha_t^4 \sum_{\ell=\bar{t}_{s-1}}^{t-1} \alpha_\ell^2 G_\ell + 6I^2L\tau^2 c_\nu^2\alpha_t^4 \sum_{\ell=\bar{t}_{s-1}}^{t-1} \alpha_\ell^2 J_\ell\big) \\
&\overset{(b)}{\leq} \sum_{t=\bar{t}_{s-1}}^{\bar{t}_s-1} \Big(\frac{3\eta^2}{16}\alpha_t E_t + \frac{3\gamma^2}{16}\alpha_t F_t + \frac{3\gamma^2}{16}\alpha_t G_t + \frac{3\tau^2}{8}\alpha_t J_t + \frac{3\tau^2}{8}\alpha_t Q_t \\
&\quad + \frac{3c_\nu^2\alpha_t^3}{32\tilde{L}^2}\frac{\sigma^2}{b_x} + \frac{3c_\nu^2\alpha_t^3\zeta_f^2}{8\tilde{L}^2} + \frac{3c_\nu^2\alpha_t^3}{4\tilde{L}^2}\frac{C_f^2\zeta_{g,xy}^2}{\mu^2} + \frac{\eta^2 c_\nu^2}{8*16^3 I^2\tilde{L}^4}\alpha_t D_t \\
&\quad + \frac{\gamma^2 c_\nu^2}{8*16^3 I^2\tilde{L}^4}\alpha_t G_t + \frac{3\tau^2 c_\nu^2}{8*16^4 I^2\tilde{L}^4}\alpha_t J_t\Big)
\end{aligned}
$$

In inequalities $(a)$ and $(b)$, we use $\alpha_t < \frac{1}{16\tilde{L}I} \le \frac{1}{16\tilde{L}_1 I}$. Note that $\sum_{t=\bar{t}_{s-1}+1}^{\bar{t}_s} \alpha_t D_t = \sum_{t=\bar{t}_{s-1}}^{\bar{t}_s-1} \alpha_t D_t$ as $D_{\bar{t}_s} = D_{\bar{t}_{s-1}} = 0$.

Then if we choose $\eta < \frac{\tilde{L}^2}{c_\nu}$ and $\gamma < \frac{\tilde{L}^2}{c_\nu}, \tau < \frac{\tilde{L}^2}{c_\nu}$, we have

$$
\sum_{t=\bar{t}_{s-1}}^{\bar{t}_s-1} \alpha_t D_t \le \sum_{t=\bar{t}_{s-1}}^{\bar{t}_s-1} \Big( \frac{\eta^2}{4} \alpha_t E_t + \frac{\gamma^2}{4} \alpha_t F_t + \frac{\gamma^2}{2} \alpha_t G_t + \tau^2 \alpha_t J_t + \frac{\tau^2}{2} \alpha_t Q_t
$$
$$
+ \frac{c_\nu^2 \alpha_t^3}{8\tilde{L}^2} \frac{\sigma^2}{b_x} + \frac{c_\nu^2 \alpha_t^3 \zeta_f^2}{2\tilde{L}^2} + \frac{c_\nu^2 \alpha_t^3}{\tilde{L}^2} \frac{C_f^2 \zeta_{g,xy}^2}{\mu^2} \Big) \tag{20}
$$

Based on Lemma C.9, for $t \ne \bar{t}_s$, we have:

$$
G_t \le \Big( 1 + \frac{33}{32I} \Big) G_{t-1} + 8IL^2 \eta^2 \alpha_{t-1}^2 D_{t-1} + 8IL^2 \eta^2 \alpha_{t-1}^2 E_{t-1} + 8IL^2 \gamma^2 \alpha_{t-1}^2 F_{t-1}
$$
$$
+ 8Ic_\omega^2 \alpha_{t-1}^4 \frac{\sigma^2}{b_y} + 16Ic_\omega^2 \alpha_{t-1}^4 \zeta_g^2 + 16I^2 L^2 \eta^2 c_\omega^2 \alpha_{t-1}^4 \sum_{\ell=\bar{t}_{s-1}}^{t-2} \alpha_l^2 D_l + 16I^2 L^2 \gamma^2 c_\omega^2 \alpha_{t-1}^4 \sum_{\ell=\bar{t}_{s-1}}^{t-2} \alpha_l^2 G_l
$$

Follow similar derivation, by recursively applying the above inequality, we have:

$$
G_t \le \sum_{\ell=\bar{t}_{s-1}}^{t-1} \big( 24IL^2 \eta^2 \alpha_\ell^2 D_\ell + 24IL^2 \eta^2 \alpha_\ell^2 E_\ell + 24IL^2 \gamma^2 \alpha_\ell^2 F_\ell
$$
$$
+ 24Ic_\omega^2 \alpha_\ell^4 \frac{\sigma^2}{b_y} + 48Ic_\omega^2 \alpha_\ell^4 \zeta_g^2 + 48I^2 L^2 \eta^2 c_\omega^2 \alpha_\ell^4 \sum_{\bar{\ell}=\bar{t}_{s-1}}^{\ell-1} \alpha_{\bar{\ell}}^2 D_{\bar{\ell}} + 48I^2 L^2 \gamma^2 c_\omega^2 \alpha_\ell^4 \sum_{\bar{\ell}=\bar{t}_{s-1}}^{\ell-1} \alpha_{\bar{\ell}}^2 G_{\bar{\ell}} \big)
$$

Next we multiply $\alpha_t$ over both sides and take sum from $\bar{t}_{s-1} + 1$ to $\bar{t}_s$, use the condition that $\alpha_t < \frac{1}{16I\tilde{L}} < \frac{1}{16IL}, \eta < \frac{\tilde{L}^2}{c_\omega}$ and $\gamma < \frac{\tilde{L}^2}{c_\omega}$, we have:

$$
\sum_{t=\bar{t}_{s-1}}^{\bar{t}_s-1} \alpha_t G_t \le \sum_{t=\bar{t}_{s-1}}^{\bar{t}_s-1} \Big( \frac{1}{2} \eta^2 \alpha_t D_t + \frac{1}{8} \eta^2 \alpha_t E_t + \frac{1}{8} \gamma^2 \alpha_t F_t + \frac{c_\omega^2 \alpha_t^3}{8\tilde{L}^2} \frac{\sigma^2}{b_y} + \frac{c_\omega^2 \alpha_t^3}{8\tilde{L}^2} \zeta_g^2 \Big) \tag{21}
$$

Based on Lemma C.10, we have:

$$
J_t \le \Big( 1 + \frac{33I}{32I} \Big) J_{t-1} + 16I\tilde{L}_2^2 \tau^2 \alpha_{t-1}^2 G_{t-1} + 16I\tilde{L}_2^2 \tau^2 \alpha_{t-1}^2 F_{t-1} + 16I\tilde{L}_2^2 \eta^2 \alpha_{t-1}^2 D_{t-1} + 16I\tilde{L}_2^2 \eta^2 \alpha_{t-1}^2 E_{t-1}
$$
$$
+ 16IL^2 \tau^2 \alpha_{t-1}^2 Q_{t-1} + 8Ic_u^2 \alpha_{t-1}^4 \frac{\sigma^2}{b_x} + 16Ic_u^2 \alpha_{t-1}^4 \zeta_f^2 + 32Ic_u^2 \alpha_{t-1}^4 \frac{C_f^2 \zeta_{g,yy}^2}{\mu^2}
$$
$$
+ 64I^2 c_u^2 \eta^2 \alpha_{t-1}^4 \tilde{L}_2^2 \sum_{\ell=\bar{t}_{s-1}}^{t-2} \alpha_l^2 D_l + 64I^2 c_u^2 \gamma^2 \alpha_{t-1}^4 \tilde{L}_2^2 \sum_{\ell=\bar{t}_{s-1}}^{t-2} \alpha_l^2 G_l + 16I^2 c_u^2 \tau^2 \alpha_{t-1}^4 L^2 \sum_{\ell=\bar{t}_{s-1}}^{t-2} \alpha_l^2 J_l
$$

Suppose we have $\alpha_t < \frac{1}{16\tilde{L}I}, \eta < \frac{\tilde{L}^2}{c_u}, \gamma < \frac{\tilde{L}^2}{c_u}, \tau < \frac{\tilde{L}^2}{c_u}$

$$
\sum_{t=\bar{t}_{s-1}+1}^{\bar{t}_s} \alpha_t J_t \le \sum_{t=\bar{t}_{s-1}}^{\bar{t}_s-1} \Big( \frac{\tau^2}{2} \alpha G_t + \frac{\tau^2}{4} \alpha_t F_t + \frac{\eta^2}{2} \alpha_t D_t + \frac{\eta^2}{4} \alpha_{t-1} E_t
$$
$$
+ \frac{\tau^2}{4} \alpha_t Q_t + \frac{c_u^2 \alpha_t^3}{8\tilde{L}^2} \frac{\sigma^2}{b_x} + \frac{c_u^2 \alpha_t^3 \zeta_f^2}{4\tilde{L}^2} + \frac{3c_u^2 \alpha_t^3}{\tilde{L}^2} \frac{C_f^2 \zeta_{g,yy}^2}{\mu^2} \Big) \tag{22}
$$

Next, we combine Eq. 20, Eq. 21 and Eq. 22 to have the result in the lemma. $\qquad \square$

### C.1.4  Descent Lemma

**Lemma C.12.** *For all $t \in [\bar{t}_{s-1}, \bar{t}_s - 1]$, the iterates generated satisfy:*

$$\mathbb{E}\big\|\nabla h(\bar{x}_t) - \bar{\mu}_t\big\|^2 \leq \frac{2\tilde{L}_1^2}{M} \sum_{m=1}^{M} \mathbb{E}\big[\big\|\bar{x}_t - x_t^{(m)}\big\|^2 + 2\big\|\bar{y}_t - y_t^{(m)}\big\|^2 + 2\big\|y_{\bar{x}_t} - \bar{y}_t\big\|^2\big] + 4L^2 \mathbb{E}\big\|u_{\bar{x}_t} - \bar{u}_t\big\|^2$$

*where we denote $u_{\bar{x}_t} = [\nabla_{y^2} g(\bar{x}_t, y_{\bar{x}_t})]^{-1} \nabla_y f(\bar{x}_t, y_{\bar{x}_t})$ and $\tilde{L}_1^2 = \big(L^2 + \frac{2L_{xy}^2 C_f^2}{\mu^2}\big)$ is a constant.*

*Proof.* This lemma follows the same derivation as Lemma C.22. $\qquad\square$

**Lemma C.13.** *Suppose $\eta \alpha_t < \frac{1}{2\bar{L}}$, for all $t \in [\bar{t}_{s-1}, \bar{t}_s - 1]$ and $s \in [S]$, the iterates generated satisfy:*

$$\mathbb{E}\big[h(\bar{x}_{t+1})\big] \leq \mathbb{E}\big[h(\bar{x}_t)\big] - \frac{\eta\alpha_t}{4}\mathbb{E}\big[\big\|\bar{\nu}_t\big\|^2\big] - \frac{\eta\alpha_t}{2}\mathbb{E}\big[\|\nabla h(\bar{x}_t)\|^2\big] + \eta\alpha_t \mathbb{E}\big[\big\|\bar{u}_t - \bar{\nu}_t\big\|^2\big]$$

$$+ \frac{2\tilde{L}_1^2 \eta \alpha_t}{M} \sum_{m=1}^{M} \mathbb{E}\big[\big\|\bar{x}_t - x_t^{(m)}\big\|^2 + 2\big\|\bar{y}_t - y_t^{(m)}\big\|^2 + 2\big\|y_{\bar{x}_t} - \bar{y}_t\big\|^2\big] + 4L^2 \eta\alpha_t \mathbb{E}\big\|u_{\bar{x}_t} - \bar{u}_t\big\|^2$$

*where the expectation is w.r.t the stochasticity of the algorithm.*

*Proof.* By the smoothness of $h(x)$ we have:

$$\mathbb{E}[h(\bar{x}_{t+1})] \leq \mathbb{E}\big[h(\bar{x}_t) + \langle \nabla h(\bar{x}_t), \bar{x}_{t+1} - \bar{x}_t \rangle + \frac{\bar{L}}{2}\|\bar{x}_{t+1} - \bar{x}_t\|^2\big]$$

$$\overset{(a)}{=} \mathbb{E}\big[h(\bar{x}_t) - \eta\alpha_t \langle \nabla h(\bar{x}_t), \bar{\nu}_t \rangle + \frac{\eta^2 \alpha_t^2 \bar{L}}{2}\|\bar{\nu}_t\|^2\big]$$

$$\overset{(b)}{=} \mathbb{E}\big[h(\bar{x}_t) - \frac{\eta\alpha_t}{2}\|\bar{\nu}_t\|^2 - \frac{\eta\alpha_t}{2}\|\nabla h(\bar{x}_t)\|^2 + \frac{\eta\alpha_t}{2}\|\nabla h(\bar{x}_t) - \bar{\nu}_t\|^2 + \frac{\eta\alpha_t^2 \bar{L}}{2}\|\bar{\nu}_t\|^2\big]$$

$$= \mathbb{E}\big[h(\bar{x}_t) - \frac{\eta\alpha_t}{4}\|\bar{\nu}_t\|^2 - \frac{\eta\alpha_t}{2}\|\nabla h(\bar{x}_t)\|^2 + \frac{\eta\alpha_t}{2}\underbrace{\|\nabla h(\bar{x}_t) - \bar{\nu}_t\|^2}_{T_1}\big]$$

where equality $(a)$ follows from the iterate update given in Algorithm 1; $(b)$ uses $\langle a, b \rangle = \frac{1}{2}[\|a\|^2 + \|b\|^2 - \|a - b\|^2]$ and $\eta\alpha_t < \frac{1}{2\bar{L}}$; For the term $T_1$, we have:

$$\mathbb{E}\big[\big\|\nabla h(\bar{x}_t) - \bar{\nu}_t\big\|^2\big] \leq 2\mathbb{E}\big[\big\|\nabla h(\bar{x}_t) - \bar{u}_t\big\|^2\big] + 2\mathbb{E}\big[\big\|\bar{u}_t - \bar{\nu}_t\big\|^2\big]$$

Use Lemma C.3 for the first term and combine everything together finishes the proof. $\qquad\square$

### C.1.5  Proof of Convergence Theorem

We first denote the following potential function $\mathcal{G}(t)$:

$$\mathcal{G}_t = h(\bar{x}_t) + \frac{9bM\eta}{64\alpha_t}\big\|\bar{\nu}_t - \bar{\mu}_t\big\|^2 + \frac{18\eta\tilde{L}^2}{\mu\gamma}\big\|\bar{y}_t - y_{\bar{x}_t}\big\|^2 + \frac{9bM\eta}{64\alpha_t}\big\|\bar{q}_t - \bar{p}_t\big\|^2$$

$$+ \frac{9bM\eta}{64\alpha_t}\big\|\bar{\omega}_t - \frac{1}{M}\sum_{m=1}^{M}\nabla_y g^{(m)}(x_t^{(m)}, y_t^{(m)})\big\|^2 + \frac{18\eta L^2}{\mu\tau}\big\|\bar{u}_t - u_{\bar{x}_t}\big\|^2$$

Furthermore, we have constants $\tilde{L}_1^2 = \big(L^2 + \frac{2L_{xy}^2 C_f^2}{\mu^2}\big)$ and $\tilde{L}_2^2 = \big(L^2 + \frac{2L_{y^2}^2 C_f^2}{\mu^2}\big)$, to ease the writing, without loss of generality, we assume the second order Lipschitz constants $L_{xy} = L_{y^2}$, as a result $\tilde{L}_1^2 = \tilde{L}_2^2$, we denote it as $\tilde{L}$ in the subsequent proof.

**Theorem C.14.** *Suppose we choose $c_\nu = \frac{64}{9bM} + \frac{2}{3b^2 M^2}$, $c_\omega = \frac{48^2}{bM\mu^2} + \frac{2}{3b^2 M^2}$, $c_u = \frac{48^2}{bM\mu^2} + \frac{2}{3b^2 M^2}$ $u = (bM\sigma)^2 \bar{u}$, where $\bar{u} = \max\big(2, 16^2 I^3 \tilde{L}^2, c_\nu^{3/2}, c_\omega^{3/2}\big)$, $\delta = \frac{(bM\sigma)^{2/3}}{(16\tilde{L})^{1/3}}$, $\alpha_t = \frac{\delta}{(u+t)^{1/3}}, t \in [T]$, $\gamma < \min\big(\frac{1}{8C_1^{1/2}}, \frac{\tilde{L}}{4C_1^{1/2}}, \frac{\tilde{L}^2}{c_\nu}, \frac{\tilde{L}^2}{c_\omega}, \frac{\tilde{L}^2}{c_u}, \frac{1}{2\tilde{L}}, 1\big)$, $\eta < \min\big(\frac{\mu\gamma}{36\kappa\tilde{L}}, \frac{1}{8C_1^{1/2}}, \frac{\tilde{L}}{4C_1^{1/2}}, \frac{\tilde{L}^2}{c_\nu}, \frac{\tilde{L}^2}{c_\omega}, \frac{\tilde{L}^2}{c_u}, \frac{1}{2\tilde{L}}, 1\big)$, $\tau <$*

$\min\left(\frac{1}{8C_1^{1/2}}, \frac{\tilde{L}}{4C_1^{1/2}}, \frac{\tilde{L}^2}{c_\nu}, \frac{\tilde{L}^2}{c_u}, \frac{1}{2L}, \frac{1}{2}\right)$ *where $C_1$ is a constant, we set the mini-batch size $b_x = b_y = b$*
*and the first batch with size $b_1 = O(Ib)$, $r = \frac{C_f}{\mu}$, then we have:*

$$\frac{1}{T}\sum_{t=1}^{T-1}\mathbb{E}\left[\|\nabla h(\bar{x}_t)\|^2\right] = O\left(\frac{\kappa^{19/3}I}{T} + \frac{\kappa^{16/3}}{(bMT)^{2/3}}\right)$$

*To reach an $\epsilon$-stationary point, we need $T = O(\kappa^8(bM)^{-1}\epsilon^{-1.5})$, $I = O(\kappa^{5/3}(bM)^{-1}\epsilon^{-0.5})$.*

*Proof.* By the condition that $u \geq c_\nu^{3/2}\delta^3$, it is straightforward to verify that $c_\nu\alpha_t^2 < 1$. By Lemma C.3, we have:

$$\frac{A_t}{\alpha_{t-1}} - \frac{A_{t-1}}{\alpha_{t-2}} \leq \left(\alpha_{t-1}^{-1} - \alpha_{t-2}^{-1} - c_\nu\alpha_{t-1}\right)A_{t-1} + \frac{2c_\nu^2\alpha_{t-1}^3\sigma^2}{bM} + \frac{16L^2\tau^2\alpha_{t-1}}{bM}(J_{t-1} + Q_{t-1})$$

$$+ \frac{8\tilde{L}^2\eta^2\alpha_{t-1}}{bM}(D_{t-1} + E_{t-1}) + \frac{8\tilde{L}^2\gamma^2\alpha_{t-1}}{bM}(F_{t-1} + G_{t-1})$$

where we choose $b_x = b_y = b$. For $\alpha_{t-1}^{-1} - \alpha_{t-2}^{-1}$, we have:

$$\alpha_t^{-1} - \alpha_{t-1}^{-1} = \frac{(u + \sigma^2 t)^{1/3}}{\delta} - \frac{(u + \sigma^2(t-1))^{1/3}}{\delta} \stackrel{(a)}{\leq} \frac{\sigma^2}{3\delta(u + \sigma^2(t-1))^{2/3}}$$

$$\stackrel{(b)}{\leq} \frac{2^{2/3}\sigma^2\delta^2}{3\delta^3(u + \sigma^2 t)^{2/3}} \stackrel{(c)}{=} \frac{2^{2/3}\sigma^2}{3\delta^3}\alpha_t^2 \leq \frac{2}{3Ib^2M^2}\alpha_t \leq \frac{2^{2/3}\sigma^2}{3\delta^3}\alpha_t^2 \leq \frac{2}{3b^2M^2}\alpha_t$$

where inequality $(a)$ results from the concavity of $x^{1/3}$ as: $(x+y)^{1/3} - x^{1/3} \leq y/3x^{2/3}$, inequality $(b)$ used the fact that $u_t \geq 2\sigma^2$, inequality $(c)$ uses the definition of $\alpha_t$, By choosing $c_\nu = \frac{64}{9bM} + \frac{2}{3b^2M^2}$, we have:

$$\frac{A_t}{\alpha_{t-1}} - \frac{A_{t-1}}{\alpha_{t-2}} \leq -\frac{64}{9bM}\alpha_{t-1}A_{t-1} + \frac{2c_\nu^2\alpha_{t-1}^3\sigma^2}{bM} + \frac{16L^2\tau^2\alpha_{t-1}}{bM}(J_{t-1} + Q_{t-1})$$

$$+ \frac{8\tilde{L}^2\eta^2\alpha_{t-1}}{bM}(D_{t-1} + E_{t-1}) + \frac{8\tilde{L}^2\gamma^2\alpha_{t-1}}{bM}(F_{t-1} + G_{t-1})$$

Next, we telescope from $\bar{t}_{s-1} + 1$ to $\bar{t}_s$:

$$\left(\frac{A_{\bar{t}_s}}{\alpha_{\bar{t}_s - 1}} - \frac{A_{\bar{t}_{s-1}}}{\alpha_{\bar{t}_{s-1} - 1}}\right) \leq -\frac{64}{9bM}\sum_{t=\bar{t}_{s-1}}^{\bar{t}_s - 1}\alpha_t A_t + \frac{2c_\nu^2\sigma^2}{bM}\sum_{t=\bar{t}_{s-1}}^{\bar{t}_s - 1}\alpha_t^3 + \frac{16\tilde{L}^2\eta^2}{bM}\sum_{t=\bar{t}_{s-1}}^{\bar{t}_s - 1}\alpha_t D_t$$

$$+ \frac{8\tilde{L}^2\eta^2}{bM}\sum_{t=\bar{t}_{s-1}}^{\bar{t}_s - 1}\alpha_t E_t + \frac{8\tilde{L}^2\gamma^2}{bM}\sum_{t=\bar{t}_{s-1}}^{\bar{t}_s - 1}\alpha_t F_t + \frac{16\tilde{L}^2\gamma^2}{bM}\sum_{t=\bar{t}_{s-1}}^{\bar{t}_s - 1}\alpha_t G_t$$

$$+ \frac{32L^2\tau^2}{bM}\sum_{t=\bar{t}_{s-1}}^{\bar{t}_s - 1}\alpha_t J_t + \frac{16L^2\tau^2}{bM}\sum_{t=\bar{t}_{s-1}}^{\bar{t}_s - 1}\alpha_t Q_t \tag{23}$$

Next, we follow similar derivation as $A_t/\alpha_{t-1} - A_{t-1}/\alpha_{t-2}$. By Lemma C.4. we choose $c_\omega = \frac{48^2}{bM\mu^2} + \frac{2}{3b^2M^2}$, to obtain:

$$\frac{C_t}{\alpha_{t-1}} - \frac{C_{t-1}}{\alpha_{t-2}} \leq -\frac{48^2\alpha_{t-1}}{bM\mu^2}C_{t-1} + \frac{2c_\omega^2\alpha_{t-1}^3\sigma^2}{bM} + \frac{4L^2\eta^2\alpha_{t-1}}{bM}(D_{t-1} + E_{t-1}) + \frac{4L^2\gamma^2\alpha_{t-1}}{bM}(F_{t-1} + G_{t-1})$$

Then telescope from $\bar{t}_{s-1} + 1$ to $\bar{t}_s$, we have:

$$\frac{C_{\bar{t}_s}}{\alpha_{\bar{t}_s - 1}} - \frac{C_{\bar{t}_{s-1}}}{\alpha_{\bar{t}_{s-1} - 1}}$$

$$\leq -\frac{48^2}{bM\mu^2}\sum_{t=\bar{t}_{s-1}}^{\bar{t}_s - 1}\alpha_t C_t + \frac{2c_\omega^2\sigma^2}{bM}\sum_{t=\bar{t}_{s-1}}^{\bar{t}_s - 1}\alpha_t^3 + \frac{16L^2\eta^2}{bM}\sum_{t=\bar{t}_{s-1}}^{\bar{t}_s - 1}\alpha_t D_t$$

$$+ \frac{8L^2\eta^2}{bM}\sum_{t=\bar{t}_{s-1}}^{\bar{t}_s - 1}\alpha_t E_t + \frac{8L^2\gamma^2}{bM}\sum_{t=\bar{t}_{s-1}}^{\bar{t}_s - 1}\alpha_t F_t + \frac{16L^2\gamma^2}{bM}\sum_{t=\bar{t}_{s-1}}^{\bar{t}_s - 1}\alpha_t G_t \tag{24}$$

Next from Lemma C.2, we choose $c_u = \frac{48^2}{bM\mu^2} + \frac{2}{3b^2M^2}$, to obtain:

$$\frac{H_t}{\alpha_{t-1}} - \frac{H_{t-1}}{\alpha_{t-2}} \leq -\frac{48^2\alpha_{t-1}}{bM\mu^2}H_{t-1} + \frac{2c_u^2\alpha_{t-1}^3}{bM}\sigma^2 + \frac{8\eta^2\alpha_{t-1}\tilde{L}^2}{bM}(D_{t-1} + E_{t-1})$$
$$+ \frac{8\gamma^2\alpha_{t-1}\tilde{L}^2}{bM}(F_{t-1} + G_{t-1}) + \frac{8\tau^2\alpha_{t-1}L^2}{bM}(J_{t-1} + Q_{t-1})$$

Then telescope from $\bar{t}_{s-1} + 1$ to $\bar{t}_s$, we have:

$$\frac{H_{\bar{t}_s}}{\alpha_{\bar{t}_s-1}} - \frac{H_{\bar{t}_{s-1}}}{\alpha_{\bar{t}_{s-1}-1}} \leq -\frac{48^2}{bM\mu^2}\sum_{t=\bar{t}_{s-1}}^{\bar{t}_s-1}\alpha_t H_t + \frac{2c_u^2}{bM}\sum_{t=\bar{t}_{s-1}}^{\bar{t}_s-1}\alpha_t^3\sigma^2 + \frac{8\eta^2\tilde{L}^2}{bM}\sum_{t=\bar{t}_{s-1}}^{\bar{t}_s-1}\alpha_{t-1}(D_t + E_t)$$
$$+ \frac{8\gamma^2\tilde{L}^2}{bM}\sum_{t=\bar{t}_{s-1}}^{\bar{t}_s-1}\alpha_{t-1}(F_t + G_t) + \frac{8\tau^2L^2}{bM}\sum_{t=\bar{t}_{s-1}}^{\bar{t}_s-1}\alpha_t(J_t + Q_{t-1}) \qquad (25)$$

Next from Lemma C.5, for $t \neq \bar{t}_s$, we have:

$$B_{t+1} - B_t \leq -\frac{\mu\gamma\alpha_t B_t}{4} - \frac{\gamma^2\alpha_t F_t}{4} + \frac{9\gamma\alpha_t C_t}{\mu} + \frac{9\kappa^2\eta^2\alpha_t E_t}{2\mu\gamma}$$
$$+ \frac{9\gamma\alpha_t L^2}{\mu}\sum_{\ell=\bar{t}_{s-1}}^{t-1}I\eta^2\alpha_\ell^2 D_\ell + \frac{9\gamma\alpha_t L^2}{\mu}\sum_{\ell=\bar{t}_{s-1}}^{t-1}I\gamma^2\alpha_\ell^2 G_\ell$$

When $t = \bar{t}_s$, we do not have the last two terms in the above inequality. Next, we telescope from $\bar{t}_{s-1} + 1$ to $\bar{t}_s$ and have:

$$B_{\bar{t}_s} - B_{\bar{t}_{s-1}} \leq -\frac{\mu\gamma}{4}\sum_{t=\bar{t}_{s-1}}^{\bar{t}_s-1}\alpha_t B_t - \frac{\gamma^2}{4}\sum_{t=\bar{t}_{s-1}}^{\bar{t}_s-1}\alpha_t F_t + \frac{9\gamma}{\mu}\sum_{t=\bar{t}_{s-1}}^{\bar{t}_s-1}\alpha_t C_t + \frac{9\kappa^2\eta^2}{2\mu\gamma}\sum_{t=\bar{t}_{s-1}}^{\bar{t}_s-1}\alpha_t E_t$$
$$+ \frac{9I\eta^2\gamma L^2}{\mu}\sum_{t=\bar{t}_{s-1}+1}^{\bar{t}_s-1}\alpha_t\sum_{\ell=\bar{t}_{s-1}}^{t-1}\alpha_\ell^2 D_\ell + \frac{9I\gamma^3 L^2}{\mu}\sum_{t=\bar{t}_{s-1}+1}^{\bar{t}_s-1}\alpha_t\sum_{\ell=\bar{t}_{s-1}}^{t-1}\alpha_\ell^2 G_\ell$$
$$\leq -\frac{\mu\gamma}{4}\sum_{t=\bar{t}_{s-1}}^{\bar{t}_s-1}\alpha_t B_t - \frac{\gamma^2}{4}\sum_{t=\bar{t}_{s-1}}^{\bar{t}_s-1}\alpha_t F_t + \frac{9\gamma}{\mu}\sum_{t=\bar{t}_{s-1}}^{\bar{t}_s-1}\alpha_t C_t + \frac{9\kappa^2\eta^2}{2\mu\gamma}\sum_{t=\bar{t}_{s-1}}^{\bar{t}_s-1}\alpha_t E_t$$
$$+ \frac{9L^2\eta^2\gamma}{16^2\hat{L}^2\mu}\sum_{\ell=\bar{t}_{s-1}}^{\bar{t}_s-1}\alpha_\ell D_\ell + \frac{9L^2\gamma^3}{16^2\hat{L}^2\mu}\sum_{\ell=\bar{t}_{s-1}}^{\bar{t}_s-1}\alpha_\ell G_\ell \qquad (26)$$

where we use the fact that $\alpha_t < \frac{1}{16\hat{L}I}$. Next, from Lemma C.6, we have:

$$I_{t+1} - I_t \leq -\frac{\mu\tau\alpha_t}{4}I_t - \frac{\tau^2\alpha_t}{4}Q_t + \frac{9\kappa^2\eta^2\alpha_t}{2\mu\tau}E_t + \frac{9\tau\alpha_t}{\mu}H_t$$
$$+ \frac{18I\eta^2\tau\alpha_t\tilde{L}^2}{\mu}\sum_{\ell=\bar{t}_{s-1}}^{t-1}\alpha_\ell^2 D_\ell + \frac{18I\gamma^2\tau\alpha_t\tilde{L}^2}{\mu}\sum_{\ell=\bar{t}_{s-1}}^{t-1}\alpha_\ell^2 G_\ell + 18I\tau^3\alpha_t L^2\sum_{\ell=\bar{t}_{s-1}}^{t-1}\alpha_\ell^2 J_\ell$$

when $t = \bar{t}_s$, we do not have the last three terms in the above inequality. Next, we telescope from $\bar{t}_{s-1} + 1$ to $\bar{t}_s$ and have:

$$
\begin{aligned}
I_{\bar{t}_s} - I_{\bar{t}_{s-1}} &\leq -\frac{\mu\tau}{4} \sum_{t=\bar{t}_{s-1}}^{\bar{t}_s-1} \alpha_t I_t - \frac{\tau^2}{4} \sum_{t=\bar{t}_{s-1}}^{\bar{t}_s-1} \alpha_t Q_t + \frac{9\kappa^2\eta^2}{2\mu\tau} \sum_{t=\bar{t}_{s-1}}^{\bar{t}_s-1} \alpha_t E_t + \frac{9\tau}{\mu} \sum_{t=\bar{t}_{s-1}}^{\bar{t}_s-1} \alpha_t H_t \\
&\quad + \frac{18I\eta^2\tau\tilde{L}^2}{\mu} \sum_{t=\bar{t}_{s-1}}^{\bar{t}_s-1} \alpha_t \sum_{\ell=\bar{t}_{s-1}}^{t-1} \alpha_\ell^2 D_\ell + \frac{18I\gamma^2\tau\tilde{L}^2}{\mu} \sum_{t=\bar{t}_{s-1}}^{\bar{t}_s-1} \alpha_t \sum_{\ell=\bar{t}_{s-1}}^{t-1} \alpha_\ell^2 G_\ell \\
&\quad + 18I\tau^3 L^2 \sum_{t=\bar{t}_{s-1}}^{\bar{t}_s-1} \alpha_t \sum_{\ell=\bar{t}_{s-1}}^{t-1} \alpha_\ell^2 J_\ell \\
&\leq -\frac{\mu\tau}{4} \sum_{t=\bar{t}_{s-1}}^{\bar{t}_s-1} \alpha_t I_t - \frac{\tau^2}{4} \sum_{t=\bar{t}_{s-1}}^{\bar{t}_s-1} \alpha_t Q_t + \frac{9\kappa^2\eta^2}{2\mu\tau} \sum_{t=\bar{t}_{s-1}}^{\bar{t}_s-1} \alpha_t E_t + \frac{9\tau}{\mu} \sum_{t=\bar{t}_{s-1}}^{\bar{t}_s-1} \alpha_t H_t \\
&\quad + \frac{18\eta^2\tau}{16^2\mu} \sum_{t=\bar{t}_{s-1}}^{\bar{t}_s-1} \alpha_t D_t + \frac{18\gamma^2\tau}{16^2\mu} \sum_{t=\bar{t}_{s-1}}^{\bar{t}_s-1} \alpha_t G_t + \frac{18\tau^3 L^2}{16^2\tilde{L}^2} \sum_{t=\bar{t}_{s-1}}^{\bar{t}_s-1} \alpha_t J_t \qquad (27)
\end{aligned}
$$

Next, by Lemma C.13, when $t + 1 \neq \bar{t}_s$, we have:

$$
\begin{aligned}
\mathbb{E}[h(\bar{x}_{t+1})] &\leq \mathbb{E}[h(\bar{x}_t)] - \frac{\eta\alpha_t}{4} E_t - \frac{\eta\alpha_t}{2} \mathbb{E}[\|\nabla h(\bar{x}_t)\|^2] + \eta\alpha_t A_t + 4\tilde{L}^2\eta\alpha_t B_t + 4L^2\eta\alpha_t I_t \\
&\quad + 2\tilde{L}^2 I\eta^3\alpha_t \sum_{\ell=\bar{t}_{s-1}}^{t-1} \alpha_l^2 D_l + 4\tilde{L}^2 I\gamma^2\eta\alpha_t \sum_{\ell=\bar{t}_{s-1}}^{t-1} \alpha_l^2 G_l
\end{aligned}
$$

When $t = \bar{t}_s$, we do not have the last two terms. Next, we telescope from $\bar{t}_{s-1}$ to $\bar{t}_s - 1$ to have:

$$
\begin{aligned}
\mathbb{E}[h(\bar{x}_{\bar{t}_s}) &- h(\bar{x}_{\bar{t}_{s-1}})] \\
&\leq -\sum_{t=\bar{t}_{s-1}}^{\bar{t}_s-1} \frac{\eta\alpha_t}{4} E_t - \sum_{t=\bar{t}_{s-1}}^{\bar{t}_s-1} \frac{\eta\alpha_t}{2} \mathbb{E}[\|\nabla h(\bar{x}_t)\|^2] + 4L^2\eta \sum_{t=\bar{t}_{s-1}}^{\bar{t}_s-1} \alpha_t I_t + \sum_{t=\bar{t}_{s-1}}^{\bar{t}_s-1} \eta\alpha_t A_t \\
&\quad + \sum_{t=\bar{t}_{s-1}}^{\bar{t}_s-1} 4\tilde{L}^2\eta\alpha_t B_t + 2\tilde{L}^2 I\eta^3 \sum_{t=\bar{t}_{s-1}}^{\bar{t}_s-1} \alpha_t \sum_{\ell=\bar{t}_{s-1}}^{t-1} \alpha_l^2 D_l + 4\tilde{L}^2 I\gamma^2\eta \sum_{t=\bar{t}_{s-1}}^{\bar{t}_s-1} \alpha_t \sum_{\ell=\bar{t}_{s-1}}^{t-1} \alpha_l^2 G_l \\
&\leq -\sum_{t=\bar{t}_{s-1}}^{\bar{t}_s-1} \frac{\eta\alpha_t}{4} E_t - \sum_{t=\bar{t}_{s-1}}^{\bar{t}_s-1} \frac{\eta\alpha_t}{2} \mathbb{E}[\|\nabla h(\bar{x}_t)\|^2] + 4L^2\eta \sum_{t=\bar{t}_{s-1}}^{\bar{t}_s-1} \alpha_t I_t + \sum_{t=\bar{t}_{s-1}}^{\bar{t}_s-1} \eta\alpha_t A_t \\
&\quad + \sum_{t=\bar{t}_{s-1}}^{\bar{t}_s-1} 4\tilde{L}^2\eta\alpha_t B_t + \frac{\eta^3}{128} \sum_{t=\bar{t}_{s-1}}^{\bar{t}_s-1} \alpha_t D_t + \frac{\gamma^2\eta}{64} \sum_{t=\bar{t}_{s-1}}^{\bar{t}_s-1} \alpha_t G_t \qquad (28)
\end{aligned}
$$

In the inequality, we use the fact that $\bar{t}_s - \bar{t}_{s-1} \leq I$, $\alpha_t < \frac{1}{16\tilde{L}I}$.

Combine Eq. (23),Eq. (24), Eq. (26) and Eq. (28) and we have:

$$\mathbb{E}[\mathcal{G}_{\bar{t}_s}] - \mathbb{E}[\mathcal{G}_{\bar{t}_{s-1}}]$$

$$\leq -\sum_{t=\bar{t}_{s-1}}^{\bar{t}_s-1} \frac{\eta\alpha_t}{2}\mathbb{E}[\|\nabla h(\bar{x}_t)\|^2] + \Big(\frac{9\eta c_\omega^2\sigma^2}{32} + \frac{9\eta c_\nu^2\sigma^2}{32} + \frac{9\eta c_u^2\sigma^2}{32}\Big)\sum_{t=\bar{t}_{s-1}}^{\bar{t}_s-1}\alpha_t^3$$

$$- \frac{\tilde{L}^2\eta}{2}\sum_{t=\bar{t}_{s-1}}^{\bar{t}_s-1}\alpha_t B_t - \frac{L^2\eta}{2}\sum_{t=\bar{t}_{s-1}}^{\bar{t}_s-1}\alpha_t I_t - \sum_{t=\bar{t}_{s-1}}^{\bar{t}_s-1}\Big(\frac{9\eta\gamma\tilde{L}^2}{2\mu} - \frac{9\eta\gamma^2\tilde{L}^2}{4} - \frac{9\eta\gamma^2 L^2}{8}\Big)\alpha_t F_t$$

$$- \sum_{t=\bar{t}_{s-1}}^{\bar{t}_s-1}\Big(\frac{1}{4} - \frac{81\kappa^2\tilde{L}^2\eta^2}{\mu^2\gamma^2} - \frac{81\kappa^2 L^2\eta^2}{\mu^2\tau^2} - \frac{9L^2\eta^2}{8} - \frac{9\tilde{L}^2\eta^2}{4}\Big)\eta\alpha_t E_t$$

$$- \sum_{t=\bar{t}_{s-1}}^{\bar{t}_s-1}\Big(\frac{9\tau\eta L^2}{2\mu} - \frac{9\eta\tau^2 L^2}{4}\Big)\alpha_t Q_t + \Big(\frac{81\kappa^2}{64} + \frac{9L^2}{4} + 9\tilde{L}^2\Big)\tau^2\eta\sum_{t=\bar{t}_{s-1}}^{\bar{t}_s-1}\alpha_t J_t$$

$$+ \Big(\frac{1}{128} + \frac{81\kappa^2}{128} + \frac{81\kappa^2}{64} + \frac{9L^2}{4} + \frac{9\tilde{L}^2}{2}\Big)\eta^3\sum_{t=\bar{t}_{s-1}}^{\bar{t}_s-1}\alpha_t D_t$$

$$+ \Big(\frac{1}{64} + \frac{81\kappa^2}{128} + \frac{81\kappa^2}{64} + \frac{9\tilde{L}^2}{4} + \frac{9L^2}{2}\Big)\gamma^2\eta\sum_{t=\bar{t}_{s-1}}^{\bar{t}_s-1}\alpha_t G_t$$

By the condition that $\eta < \frac{\mu\gamma}{36\kappa\tilde{L}}$ and $\gamma \leq \frac{1}{2L} < \frac{1}{2\mu}$. Next, we denote:

$$C_1 = \frac{1}{64} + \frac{81\kappa^2}{32} + 9\tilde{L}^2 = O(\kappa^2)$$

Then, we have:

$$\mathbb{E}[\mathcal{G}_{\bar{t}_s}] - \mathbb{E}[\mathcal{G}_{\bar{t}_{s-1}}]$$

$$\leq -\sum_{t=\bar{t}_{s-1}}^{\bar{t}_s-1} \frac{\eta\alpha_t}{2}\mathbb{E}\big[\|\nabla h(\bar{x}_t)\|^2\big] + \Big(\frac{9\eta c_\omega^2\sigma^2}{32} + \frac{9\eta c_\nu^2\sigma^2}{32} + \frac{9\eta c_u^2\sigma^2}{32}\Big)\sum_{t=\bar{t}_{s-1}}^{\bar{t}_s-1}\alpha_t^3$$

$$- \frac{9\eta\gamma^2\tilde{L}^2}{8}\sum_{t=\bar{t}_{s-1}}^{\bar{t}_s-1}\alpha_t F_t - \frac{\eta}{8}\sum_{t=\bar{t}_{s-1}}^{\bar{t}_s-1}\alpha_t E_t - \frac{\tilde{L}^2\eta}{2}\sum_{t=\bar{t}_{s-1}}^{\bar{t}_s-1}\alpha_t B_t - \frac{L^2\eta}{2}\sum_{t=\bar{t}_{s-1}}^{\bar{t}_s-1}\alpha_t I_t$$

$$- \frac{9\eta\tau^2 L^2}{4}\sum_{t=\bar{t}_{s-1}}^{\bar{t}_s-1}\alpha_t Q_t + C_1\eta^3\sum_{t=\bar{t}_{s-1}}^{\bar{t}_s-1}\alpha_t D_t + C_1\gamma^2\eta\sum_{t=\bar{t}_{s-1}}^{\bar{t}_s-1}\alpha_t G_t + C_1\tau^2\eta\sum_{t=\bar{t}_{s-1}}^{\bar{t}_s-1}\alpha_t J_t$$

$$\tag{29}$$

Combine Eq. (29) with Lemma C.11, and use the condition that $\eta < \min\big(\frac{1}{8C_1^{1/2}}, \frac{\tilde{L}}{4C_1^{1/2}}, 1\big)$, $\gamma < \min\big(\frac{1}{8C_1^{1/2}}, \frac{\tilde{L}}{4C_1^{1/2}}, 1\big)$ and $\tau < \min\big(\frac{1}{8C_1^{1/2}}, \frac{\tilde{L}}{4C_1^{1/2}}, 1\big)$ we have:

$$\mathbb{E}[\mathcal{G}_{\bar{t}_s}] - \mathbb{E}[\mathcal{G}_{\bar{t}_{s-1}}] \leq -\sum_{t=\bar{t}_{s-1}}^{\bar{t}_s-1} \frac{\eta\alpha_t}{2}\mathbb{E}\big[\|\nabla h(\bar{x}_t)\|^2\big] + C_{\sigma,\zeta}\eta\sum_{t=\bar{t}_{s-1}}^{\bar{t}_s-1}\alpha_t^3$$

For ease of notation, we denote

$$C_{\sigma,\zeta} = \Big(4c_\omega^2\sigma^2 + 4c_u^2\sigma^2 + 4c_\nu^2\sigma^2 + 3c_u^2\zeta_f^2 + 3c_\nu^2\zeta_f^2 + 3c_\omega^2\zeta_g^2 + \frac{3c_\nu^2 C_f^2\zeta_{g,xy}^2}{\mu^2} + \frac{120c_u^2 C_f^2\zeta_{g,yy}^2}{\mu^2}\Big).$$

Next, sum over all $s \in [S]$ (assume $T = SI + 1$ without loss of generality), we have:

$$\mathbb{E}[\mathcal{G}_T] - \mathbb{E}[\mathcal{G}_1] \leq -\sum_{t=1}^{T-1} \frac{\eta\alpha_t}{2}\mathbb{E}\big[\|\nabla h(\bar{x}_t)\|^2\big] + \eta C_{\sigma,\zeta}\sum_{t=1}^{T-1}\alpha_t^3$$

Rearranging the terms and use the fact that $\alpha_t$ is non-increasing, we have:

$$\frac{\eta\alpha_T}{2}\sum_{t=1}^{T-1}\mathbb{E}\left[\|\nabla h(\bar{x}_t)\|^2\right] \leq \mathbb{E}[\mathcal{G}_1] - \mathbb{E}[\mathcal{G}_T] + \eta C_{\sigma,\varsigma}\sum_{t=1}^{T-1}\alpha_t^3$$

$$\leq h(x_1) - h^* + \frac{9bM\eta A_1}{64\alpha_1} + \frac{18\eta\tilde{L}^2 B_1}{\mu\gamma}$$

$$+ \frac{9bM\eta C_1}{64\alpha_1} + \frac{9bM\eta H_1}{64\alpha_1} + \frac{18\eta L^2 I_1}{\mu\tau} + \eta C_{\sigma,\varsigma}\sum_{t=1}^{T-1}\alpha_t^3$$

where we use $\mathcal{G}_T \geq h^*$ ($h^*$ is the optimal value of $h$), and for the last term, we use the following fact:

$$\sum_{t=1}^{T}\alpha_t^3 = \sum_{t=1}^{T}\frac{\delta^3}{u+\sigma^2 t} \leq \sum_{t=1}^{T}\frac{\delta^3}{\sigma^2+\sigma^2 t} = \frac{\delta^3}{\sigma^2}\sum_{t=1}^{T}\frac{1}{1+t} \leq \frac{\delta^3}{\sigma^2}\ln(T+1) = \frac{b^2 M^2\ln(T+1)}{16\tilde{L}}$$

the first inequality follows $u_t > \sigma^2$, the last inequality follows Proposition E.3.

Next, we denote the initial sub-optimality as $\Delta = h(\bar{x}_1) - h^*$, initial inner variable estimation error *i.e.* $B_1 = \|y_1 - y_{x_1}\|^2 \leq \Delta_y$ and the initial hyper-gradient computation error $I_1 = \|u_1 - [\nabla_{y^2}g(x_1, y_{x_1})]^{-1}\nabla_y f(x_1, y_{x_1})\|^2 \leq \Delta_u$.

Furthermore, we have $A_1 = \leq \frac{\sigma^2}{b_1 M}$, $C_1 \leq \frac{\sigma^2}{b_1 M}$, $H_1 \leq \frac{\sigma^2}{b_1 M}$ where $b_1$ be the size of the first batch. Then, we divide both sides by $\eta\alpha_T T/2$ to have:

$$\frac{1}{T}\sum_{t=1}^{T-1}\mathbb{E}\left[\|\nabla h(\bar{x}_t)\|^2\right] \leq \left(\frac{2\Delta}{\eta} + \frac{27b\sigma^2}{32b_1\alpha_1} + \frac{36\tilde{L}^2\Delta_y}{\mu\gamma} + \frac{36L^2\Delta_u}{\mu\tau} + \frac{b^2 M^2 C_{\sigma,\varsigma}\ln(T)}{8\tilde{L}}\right)\frac{1}{T\alpha_T}$$

Note that we have:

$$\frac{1}{\alpha_t t} = \frac{(u+\sigma^2 t)^{1/3}}{\delta t} \leq \frac{u^{1/3}}{\delta t} + \frac{\sigma^{2/3}}{\delta t^{2/3}}$$

where the inequality uses the fact that $(x+y)^{1/3} \leq x^{1/3} + y^{1/3}$. In particular, when $t = 1$, we have

$$\frac{1}{\alpha_1} \leq \frac{u^{1/3} + \sigma^{2/3}}{\delta} = \frac{(16\tilde{L})^{1/3}((bM)^{2/3}\bar{u}^{1/3} + 1)}{(bM)^{2/3}} \tag{30}$$

when $t = T$, we have:

$$\frac{1}{\alpha_T T} \leq \frac{u^{1/3}}{\delta T} + \frac{\sigma^{2/3}}{\delta T^{2/3}} = (16\tilde{L})^{1/3}\left(\frac{\bar{u}^{1/3}}{T} + \frac{1}{(bMT)^{2/3}}\right) \tag{31}$$

In summary, we have:

$$\frac{1}{T}\sum_{t=1}^{T-1}\mathbb{E}\left[\|\nabla h(\bar{x}_t)\|^2\right]$$

$$\leq \left(\frac{2\Delta}{\eta} + \frac{27b\sigma^2}{32b_1\alpha_1} + \frac{36\tilde{L}^2\Delta_y}{\mu\gamma} + \frac{36L^2\Delta_u}{\mu\tau} + \frac{b^2 M^2 C_{\sigma,\varsigma}\ln(T)}{8\tilde{L}}\right)\left(\frac{(16\tilde{L}\bar{u})^{1/3}}{T} + \frac{(16\tilde{L})^{1/3}}{(bMT)^{2/3}}\right)$$

Recall that $\tilde{L} = O(\kappa)$, $\bar{L} = O(\kappa^3)$, therefore we have $c_\nu = \Theta((bM)^{-1})$, $c_\omega = \Theta(\kappa^2(bM)^{-1})$, $c_u = \Theta(\kappa^2(bM)^{-1})$ $\bar{u} = \Theta(I^3\kappa^3)$, then for $\eta, \gamma, \tau$, we have:

$$\gamma < \min\left(\frac{1}{8C_1^{1/2}}, \frac{\tilde{L}}{4C_1^{1/2}}, \frac{\tilde{L}^2}{c_\nu}, \frac{\tilde{L}^2}{c_\omega}, \frac{\tilde{L}^2}{c_u}, \frac{1}{2L}, 1\right)$$

$$\tau < \min\left(\frac{1}{8C_1^{1/2}}, \frac{\tilde{L}}{4C_1^{1/2}}, \frac{\tilde{L}^2}{c_\nu}, \frac{\tilde{L}^2}{c_u}, \frac{1}{2L}, \frac{1}{2}\right)$$

$$\eta < \min \left( \frac{\mu\gamma}{36\kappa\tilde{L}}, \frac{1}{8C_1^{1/2}}, \frac{\tilde{L}}{4C_1^{1/2}}, \frac{\tilde{L}^2}{c_\nu}, \frac{\tilde{L}^2}{c_\omega}, \frac{\tilde{L}^2}{c_u}, \frac{1}{2\bar{L}}, 1 \right)$$

where $C_1 = O(\kappa^2)$, so we have $\gamma^{-1} = O(\kappa)$, $\eta^{-1} = O(\kappa^3)$, $\tau^{-1} = O(\kappa)$, furthermore, $\alpha_1^{-1} = O(I\kappa^{4/3})$, $C_{\sigma,\zeta} = O(\kappa^6(bM)^{-2})$, assume we choose the size of the first batch to be $b_1 = Ib$.

Combine everything together, we have:

$$\frac{1}{T}\sum_{t=1}^{T-1} \mathbb{E}\left[\|\nabla h(\bar{x}_t)\|^2\right] = O\left(\frac{\kappa^{19/3}I}{T} + \frac{\kappa^{16/3}}{(bMT)^{2/3}}\right)$$

To reach an $\epsilon$-stationary point, we need $T = O(\kappa^8(bM)^{-1}\epsilon^{-1.5})$, $I = O(\kappa^{5/3}(bM)^{-1}\epsilon^{-0.5})$. The communication cost is $E = T/I \geq \kappa^{19/3}\epsilon^{-1}$, the sample complexity is $Gc(f,\epsilon) = O(M^{-1}\kappa^8\epsilon^{-1.5})$, $Gc(g,\epsilon) = O(M^{-1}\kappa^8\epsilon^{-1.5})$, $Jv(g,\epsilon) = O(M^{-1}\kappa^8\epsilon^{-1.5})$, $Hv(g,\epsilon) = O(M^{-1}\kappa^8\epsilon^{-1.5})$  □

---

**Algorithm 2** Federated Bilevel Optimization (**FedBiO**)

---

1: **Input:** Initial states $x_1$, $y_1$ and $u_1$; learning rates $\{\gamma_t, \eta_t, \tau_t\}_{t=1}^T$
2: **Initialization:** Set $x_1^{(m)} = x_1$, $y_1^{(m)} = y_1$, $u_1^{(m)} = u_1$;
3: **for** $t = 1$ **to** $T$ **do**
4:      Randomly sample mutually independent minibatch of samples $\mathcal{B}_y$ and $\mathcal{B}_x = \{\mathcal{B}_{g,1}, \mathcal{B}_{g,2}, \mathcal{B}_{f,1}, \mathcal{B}_{f,2}\}$ of size b;
5:      $\omega_t^{(m)} = \nabla_y g^{(m)}(x_t^{(m)}, y_t^{(m)}, \mathcal{B}_y)$
6:      $\nu_t^{(m)} = \nabla_x f^{(m)}(x_t^{(m)}, y_t^{(m)}; \mathcal{B}_{f,1}) - \nabla_{xy} g^{(m)}(x_t^{(m)}, y_t^{(m)}; \mathcal{B}_{g,1}) u_t^{(m)}$;
7:      $\hat{y}_{t+1}^{(m)} = y_t^{(m)} - \gamma_t \omega_t^{(m)}$, $\hat{x}_{t+1}^{(m)} = x_t^{(m)} - \eta_t \nu_t^{(m)}$;
8:      **if** $t \bmod I = 0$ **then**
9:          $y_{t+1}^{(m)} = \frac{1}{M} \sum_{j=1}^M \hat{y}_{t+1}^{(j)}$; $x_{t+1}^{(m)} = \frac{1}{M} \sum_{j=1}^M \hat{x}_{t+1}^{(j)}$
10:    **else**
11:        $y_{t+1}^{(m)} = \hat{y}_{t+1}^{(m)}$, $x_{t+1}^{(m)} = \hat{x}_{t+1}^{(m)}$
12:    **end if**
13:    $\hat{u}_{t+1}^{(m)} = \mathcal{P}_r(\tau_t \nabla_y f^{(m)}(x_t^{(m)}, y_t^{(m)}; \mathcal{B}_{f,2}) + (I - \tau_t \nabla_{y^2} g^{(m)}(x_t^{(m)}, y_t^{(m)}; \mathcal{B}_{g,2})) u_t^{(m)})$;
14:    **if** $t \bmod I = 0$ **then**
15:        $u_{t+1}^{(m)} = \frac{1}{M} \sum_{j=1}^M \hat{u}_{t+1}^{(j)}$
16:    **else**
17:        $u_{t+1}^{(m)} = \hat{u}_{t+1}^{(m)}$
18:    **end if**
19: **end for**

---

## C.2 Proof for the FedBiO Algorithm

Algorithm 2 follows Eq. 6, and we discuss its convergence property in this subsection.

### C.2.1 Lower Problem Solution Error and Hyper-gradient Estimation Error

**Lemma C.15.** *When $\gamma < \frac{1}{2L}$, we have:*

$$
\mathbb{E}\|\bar{y}_t - y_{\bar{x}_t}\|^2 \leq (1 - \frac{\mu\gamma}{4})\mathbb{E}\|\bar{y}_{t-1} - y_{\bar{x}_{t-1}}\|^2 + \frac{9\kappa^2\eta^2}{2\mu\gamma}\mathbb{E}\|\bar{\nu}_{t-1}\|^2 - \frac{\gamma^2}{4}\mathbb{E}\|\bar{\omega}_{t-1}\|^2
$$
$$
+ \frac{9L^2\gamma}{2\mu M} \sum_{m=1}^M \mathbb{E}\big[\|x_{t-1}^{(m)} - \bar{x}_{t-1}\|^2 + \|y_{t-1}^{(m)} - \bar{y}_{t-1})\|^2\big] + \frac{4\gamma\sigma^2}{\mu b_y M}
$$

**Lemma C.16.** *Suppose we choose $\tau < \frac{1}{L}$, then we have:*

$$
\mathbb{E}\|\bar{u}_{t+1} - u_{\bar{x}_{t+1}}\|^2 \leq (1 - \frac{\mu\tau}{4})\mathbb{E}\|\bar{u}_t - u_{\bar{x}_t}\|^2 + \frac{5\tau^2\sigma^2}{4b_x M} + \frac{5\eta^2\bar{L}^2}{\mu\tau}\mathbb{E}\|\bar{\nu}_t\|^2
$$
$$
+ \frac{5}{4}\Big(\frac{3\tau L^2}{\mu M} + \frac{\tau L_{y^2}^2 C_f^2}{2\mu^3 M}\Big) \sum_{m=1}^M \mathbb{E}\big[\|\bar{x}_t - x_t^{(m)}\|^2 + 2\|\bar{y}_t - y_t^{(m)}\|^2 + 2\|y_{\bar{x}_t} - \bar{y}_t\|^2\big]
$$

We provide the proof for Lemma C.16 here and Lemma C.15 can be derived similarly.

*Proof.* First, by proposition E.5 (set $\alpha = 1$) and choose $\gamma < \frac{1}{2L}$, we have:

$$\mathbb{E}\|\bar{y}_t - y_{\bar{x}_{t-1}}\|^2 \leq (1 - \frac{\mu\gamma}{2})\mathbb{E}\|\bar{y}_{t-1} - y_{\bar{x}_{t-1}}\|^2 - \frac{\gamma^2}{4}\mathbb{E}\|\bar{\omega}_{t-1}\|^2$$

$$+ \frac{4\gamma}{\mu}\mathbb{E}\|\nabla_y g(\bar{x}_{t-1}, \bar{y}_{t-1}) - \frac{1}{M}\sum_{m=1}^{M}\nabla_y g(x_{t-1}^{(m)}, y_{t-1}^{(m)})\|^2 + \frac{4\gamma\sigma^2}{\mu b_y M}$$

$$\leq (1 - \frac{\mu\gamma}{2})\mathbb{E}\|\bar{y}_{t-1} - y_{\bar{x}_{t-1}}\|^2 - \frac{\gamma^2}{4}\mathbb{E}\|\bar{\omega}_{t-1}\|^2$$

$$+ \frac{4\gamma}{\mu M}\sum_{m=1}^{M}\mathbb{E}\|\nabla_y^{(m)} g(\bar{x}_{t-1}, \bar{y}_{t-1}) - \nabla_y g(x_{t-1}^{(m)}, y_{t-1}^{(m)})\|^2 + \frac{4\gamma\sigma^2}{\mu b_y M}$$

$$\leq (1 - \frac{\mu\gamma}{2})\mathbb{E}\|\bar{y}_{t-1} - y_{\bar{x}_{t-1}}\|^2 - \frac{\gamma^2}{4}\mathbb{E}\|\bar{\omega}_{t-1}\|^2$$

$$+ \frac{4L^2\gamma}{\mu M}\sum_{m=1}^{M}\mathbb{E}\big[\|x_{t-1}^{(m)} - \bar{x}_{t-1}\|^2 + \|y_{t-1}^{(m)} - \bar{y}_{t-1})\|^2\big] + \frac{4\gamma\sigma^2}{\mu b_y M}$$

Furthermore, by the generalized triangle inequality, we have:

$$\mathbb{E}\|\bar{y}_t - y_{\bar{x}_t}\|^2 \leq (1 - \frac{\mu\gamma}{4})\mathbb{E}\|\bar{y}_{t-1} - y_{\bar{x}_{t-1}}\|^2 + (1 + \frac{4}{\mu\gamma})\mathbb{E}\|y_{\bar{x}_t} - y_{\bar{x}_{t-1}}\|^2 - (1 + \frac{\mu\gamma}{4})\frac{\gamma^2}{4}\mathbb{E}\|\bar{\omega}_{t-1}\|^2$$

$$+ (1 + \frac{\mu\gamma}{4})\frac{4L^2\gamma}{\mu M}\sum_{m=1}^{M}\mathbb{E}\big[\|x_{t-1}^{(m)} - \bar{x}_{t-1}\|^2 + \|y_{t-1}^{(m)} - \bar{y}_{t-1})\|^2\big] + \frac{4\gamma\sigma^2}{\mu b_y M}$$

$$\leq (1 - \frac{\mu\gamma}{4})\mathbb{E}\|\bar{y}_{t-1} - y_{\bar{x}_{t-1}}\|^2 + \frac{9\kappa^2\eta^2}{2\mu\gamma}\mathbb{E}\|\bar{\nu}_{t-1}\|^2 - \frac{\gamma^2}{4}\mathbb{E}\|\bar{\omega}_{t-1}\|^2$$

$$+ \frac{9L^2\gamma}{2\mu M}\sum_{m=1}^{M}\mathbb{E}\big[\|x_{t-1}^{(m)} - \bar{x}_{t-1}\|^2 + \|y_{t-1}^{(m)} - \bar{y}_{t-1})\|^2\big] + \frac{4\gamma\sigma^2}{\mu b_y M}$$

where the second inequality is due to $\gamma < 1/2L$. This completes the proof. $\square$

### C.2.2 Local Variable Drift

**Lemma C.17.** *For any $t \neq \bar{t}_s, s \in [S]$, we have:*

$$\|x_t^{(m)} - \bar{x}_t\|^2 \leq I\eta^2 \sum_{\ell=\bar{t}_{s-1}}^{t-1}\|\nu_\ell^{(m)} - \bar{\nu}_\ell\|^2, \quad \|y_t^{(m)} - \bar{y}_t\|^2 \leq I\gamma^2 \sum_{\ell=\bar{t}_{s-1}}^{t-1}\|\omega_\ell^{(m)} - \bar{\omega}_\ell\|^2$$

*Proof.* Note from Algorithm and the definition of $\bar{t}_s$ that at $t = \bar{t}_s$ with $s \in [S]$, $x_t^{(m)} = \bar{x}_t$, for all $k$. For $t \neq \bar{t}_s$, with $s \in [S]$, we have: $x_t^{(m)} = x_{t-1}^{(m)} - \eta\nu_{t-1}^{(m)}$, this implies that: $x_t^{(m)} = x_{\bar{t}_{s-1}}^{(m)} - \sum_{\ell=\bar{t}_{s-1}}^{t-1}\eta\nu_\ell^{(m)}$ and $\bar{x}_t = \bar{x}_{\bar{t}_{s-1}} - \sum_{\ell=\bar{t}_{s-1}}^{t-1}\eta\bar{\nu}_\ell$. So for $t \neq \bar{t}_s$, with $s \in [S]$ we have:

$$\|x_t^{(m)} - \bar{x}_t\|^2 = \|x_{\bar{t}_{s-1}}^{(m)} - \bar{x}_{\bar{t}_{s-1}} - \big(\sum_{\ell=\bar{t}_{s-1}}^{t-1}\eta\nu_\ell^{(m)} - \sum_{\ell=\bar{t}_{s-1}}^{t-1}\eta\bar{\nu}_\ell\big)\|^2 = \|\sum_{\ell=\bar{t}_{s-1}}^{t-1}\eta\big(\nu_\ell^{(m)} - \bar{\nu}_\ell\big)\|^2$$

$$\leq I\eta^2 \sum_{\ell=\bar{t}_{s-1}}^{t-1}\|\nu_\ell^{(m)} - \bar{\nu}_\ell\|^2$$

We can derive the bound for $\|y_t^{(m)} - \bar{y}_t\|^2$ similarly. This completes the proof. $\square$

**Lemma C.18.** *For any $t \in [T]$, we have:*

$$\frac{1}{M}\sum_{m=1}^{M}\mathbb{E}\big\|\big(\nu_t^{(m)} - \bar{\nu}_t\big)\big\|^2 \leq \frac{4L^2}{M}\sum_{m=1}^{M}\mathbb{E}\big\|u_t^{(m)} - \bar{u}_t\big\|^2 + 4\zeta_f^2 + \frac{8C_f^2\zeta_{g,xy}^2}{\mu^2} + \frac{2\sigma^2}{b_x}$$

$$+ \big(\frac{16L^2}{M} + \frac{32L_{xy}^2C_f^2}{\mu^2 M}\big)\sum_{m=1}^{M}\mathbb{E}\big[\big\|x_t^{(m)} - \bar{x}_t\big\|^2 + \big\|y_t^{(m)} - \bar{y}_t\big\|^2\big]$$

**Lemma C.19.** *For $t \in T$, we have:*

$$\frac{1}{M}\sum_{m=1}^{M}\mathbb{E}\big\|\big(\omega_t^{(m)} - \bar{\omega}_t\big)\big\|^2 \leq \frac{2L^2}{M}\sum_{m=1}^{M}\mathbb{E}\big\|x_t^{(m)} - \bar{x}_t\big\|^2 + \frac{2L^2}{M}\sum_{m=1}^{M}\mathbb{E}\big\|y_t^{(m)} - \bar{y}_t\big\|^2 + \frac{2\sigma^2}{b_y} + 2\zeta_g^2$$

**Lemma C.20.** *For $t \in [T]$, we have:*

$$\frac{1}{M}\sum_{m=1}^{M}\mathbb{E}\big\|\big(u_{t+1}^{(m)} - \bar{u}_{t+1}\big)\big\|^2 \leq (1+\frac{1}{I})\frac{1}{M}\sum_{m=1}^{M}\mathbb{E}\big\|u_t^{(m)} - \bar{u}_t\big\|^2 + \frac{64I\tau^2C_f^2\zeta_{g,yy}^2}{\mu^2} + 32I\tau^2\zeta_f^2 + \frac{2\tau^2\sigma^2}{b_x}$$

$$+ \big(\frac{128IL^2\tau^2}{M} + \frac{256I\tau^2L_{y^2}^2C_f^2}{\mu^2 M}\big)\sum_{m=1}^{M}\mathbb{E}\big[\big\|x_t^{(m)} - \bar{x}_t\big\|^2 + \big\|y_t^{(m)} - \bar{y}_t\big\|^2\big]$$

Lemma C.18-Lemma C.20 bounds the local drift of $\nu_t^{(m)}$, $\omega_t^{(m)}$ and $u_{t+1}^{(m)}$. We provide the proof for Lemma C.18 here and the other two bounds can be derived similarly.

*Proof.* We have:

$$\frac{1}{M}\sum_{m=1}^{M}\mathbb{E}\big\|\big(\nu_t^{(m)} - \bar{\nu}_t\big)\big\|^2 \leq \frac{1}{M}\sum_{m=1}^{M}\mathbb{E}\big\|\nabla_x f^{(m)}(x_t^{(m)},y_t^{(m)}) - \nabla_{xy}g^{(m)}(x_t^{(m)},y_t^{(m)})u_t^{(m)}$$

$$- \frac{1}{M}\sum_{j=1}^{M}\nabla_x f^{(j)}(x_t^{(j)},y_t^{(j)}) - \nabla_{xy}g^{(j)}(x_t^{(j)},y_t^{(j)})u_t^{(j)}\big\|^2 + \frac{2\sigma^2}{b_x}$$

$$\leq \underbrace{\frac{2}{M}\sum_{m=1}^{M}\mathbb{E}\big\|\nabla_x f^{(m)}(x_t^{(m)},y_t^{(m)}) - \frac{1}{M}\sum_{j=1}^{M}\nabla_x f^{(j)}(x_t^{(j)},y_t^{(j)})\big\|^2}_{T_1}$$

$$+ \underbrace{\frac{2}{M}\sum_{m=1}^{M}\mathbb{E}\big\|\nabla_{xy}g^{(m)}(x_t^{(m)},y_t^{(m)})u_t^{(m)} - \frac{1}{M}\sum_{j=1}^{M}\nabla_{xy}g^{(j)}(x_t^{(j)},y_t^{(j)})u_t^{(j)}\big\|^2}_{T_2} + \frac{2\sigma^2}{b_x}$$

For the term $T_1$, we have:

$$T_1 \leq \frac{16}{M}\sum_{m=1}^{M}\mathbb{E}\big\|\nabla_x f^{(m)}(x_t^{(m)},y_t^{(m)}) - \nabla_x f^{(m)}(\bar{x}_t,\bar{y}_t)\big\|^2 + \frac{4}{M}\sum_{m=1}^{M}\mathbb{E}\big\|\nabla_x f^{(m)}(\bar{x}_t,\bar{y}_t) - \nabla_x f(\bar{x}_t,\bar{y}_t)\big\|^2$$

$$\leq \frac{16L^2}{M}\sum_{m=1}^{M}\mathbb{E}\big[\big\|x_t^{(m)} - \bar{x}_t\big\|^2 + \big\|y_t^{(m)} - \bar{y}_t\big\|^2\big] + 4\zeta_f^2$$

Next for the term $T_2$, we have:

$$T_2 \leq \frac{4L^2}{M}\sum_{m=1}^{M}\mathbb{E}\big\|u_t^{(m)} - \bar{u}_t\big\|^2 + \frac{4C_f^2}{\mu^2 M^2}\sum_{m=1}^{M}\sum_{j=1}^{M}\mathbb{E}\big\|\nabla_{xy}g^{(m)}(x_t^{(m)},y_t^{(m)}) - \nabla_{xy}g^{(j)}(x_t^{(j)},y_t^{(j)})\big\|^2$$

$$\leq \frac{4L^2}{M}\sum_{m=1}^{M}\mathbb{E}\big\|u_t^{(m)} - \bar{u}_t\big\|^2 + \frac{32L_{xy}^2C_f^2}{\mu^2 M}\sum_{m=1}^{M}\mathbb{E}\big[\big\|x_t^{(m)} - \bar{x}_t\big\|^2 + \big\|y_t^{(m)} - \bar{y}_t\big\|^2\big] + \frac{8C_f^2\zeta_{g,xy}^2}{\mu^2}$$

Combine everything together, we get the claim in the lemma. $\square$

Lemma C.18-Lemma C.20 have recursive dependence of each other. Next, we provide an un-intertwined bound for each of them. For ease of notation, we denote $D_t = \frac{1}{M}\sum_{m=1}^{M}\mathbb{E}\|\nu_t^{(m)} - \bar{\nu}_t\|^2$, $B_t = \frac{1}{M}\sum_{m=1}^{M}\mathbb{E}\|\omega_t^{(m)} - \bar{\omega}_t\|^2$, $A_t = \frac{1}{M}\sum_{m=1}^{M}\mathbb{E}\|u_t^{(m)} - \bar{u}_t\|^2$ and $C_t = \mathbb{E}\|\bar{y}_t - y_{\bar{x}_t}\|^2$.

**Lemma C.21.** *For* $\gamma \leq \frac{1}{8I\tilde{L}_1}$ *and* $\eta < \frac{1}{8I\tilde{L}_2}$, $\tau < \frac{1}{128I\tilde{L}_2}$, *where* $\tilde{L}_1^2 = \left(L^2 + \frac{2L_{xy}^2 C_f^2}{\mu^2}\right)$ *and* $\tilde{L}_2^2 = \left(L^2 + \frac{2L_{y^2}^2 C_f^2}{\mu^2}\right)$ *are constants, then we have:*

$$\sum_{t=\bar{t}_{s-1}}^{\bar{t}_s - 1} D_t \leq 96I\zeta_f^2 + 16I\zeta_g^2 + \frac{16IC_f^2\zeta_{g,yy}^2}{\mu^2} + \frac{32IC_f^2\zeta_{g,xy}^2}{\mu^2} + \frac{16I\sigma^2}{b_y} + \frac{20I\sigma^2}{b_x}$$

$$\sum_{t=\bar{t}_{s-1}}^{\bar{t}_s - 1} B_t \leq 24I\zeta_f^2 + 8I\zeta_g^2 + \frac{4IC_f^2\zeta_{g,yy}^2}{\mu^2} + \frac{8IC_f^2\zeta_{g,xy}^2}{\mu^2} + \frac{8I\sigma^2}{b_y} + \frac{5I\sigma^2}{b_x}$$

*Proof.* Based on Lemma C.18, and sum from $\bar{t}_{s-1} + 1$ to $\bar{t}_s - 1$, we have:

$$\sum_{t=\bar{t}_{s-1}+1}^{\bar{t}_s - 1} D_t \leq 16I\eta^2\tilde{L}_1^2 \sum_{t=\bar{t}_{s-1}+1}^{\bar{t}_s - 1}\sum_{\ell=\bar{t}_{s-1}}^{t-1} D_\ell + 16I\gamma^2\tilde{L}_1^2 \sum_{t=\bar{t}_{s-1}+1}^{\bar{t}_s - 1}\sum_{\ell=\bar{t}_{s-1}}^{t-1} B_\ell$$

$$+ 4L^2 \sum_{t=\bar{t}_{s-1}+1}^{\bar{t}_s - 1} A_t + 4(I-1)\zeta_f^2 + \frac{8(I-1)C_f^2\zeta_{g,xy}^2}{\mu^2} + \frac{2(I-1)\sigma^2}{b_x}$$

$$\leq 16I^2\eta^2\tilde{L}_1^2 \sum_{t=\bar{t}_{s-1}+1}^{\bar{t}_s - 1} D_t + 16I^2\gamma^2\tilde{L}_1^2 \sum_{t=\bar{t}_{s-1}+1}^{\bar{t}_s - 1} B_t$$

$$+ 4L^2 \sum_{t=\bar{t}_{s-1}+1}^{\bar{t}_s - 1} A_t + 4(I-1)\zeta_f^2 + \frac{8(I-1)C_f^2\zeta_{g,xy}^2}{\mu^2} + \frac{2(I-1)\sigma^2}{b_x}$$

where we denote $\tilde{L}_1^2 = (L^2 + \frac{2L_{xy}^2 C_f^2}{\mu^2})$. Combine with the case of $t = \bar{t}_s$ in Lemma C.18, we have:

$$\sum_{t=\bar{t}_{s-1}}^{\bar{t}_s - 1} D_t \leq 16I^2\tilde{L}_1^2\eta^2 \sum_{t=\bar{t}_{s-1}}^{\bar{t}_s - 1} D_t + 16I^2\tilde{L}_1^2\gamma^2 \sum_{t=\bar{t}_{s-1}}^{\bar{t}_s - 1} B_t + 4L^2 \sum_{t=\bar{t}_{s-1}}^{\bar{t}_s - 1} A_t + 4I\zeta_f^2 + \frac{8IC_f^2\zeta_{g,xy}^2}{\mu^2} + \frac{2I\sigma^2}{b_x}$$

$$\tag{32}$$

Based on Lemma C.19, and sum from $\bar{t}_{s-1} + 1$ to $\bar{t}_s - 1$, we have:

$$\sum_{t=\bar{t}_{s-1}+1}^{\bar{t}_s - 1} B_t \leq 2I\eta^2 L^2 \sum_{t=\bar{t}_{s-1}+1}^{\bar{t}_s - 1}\sum_{\ell=\bar{t}_{s-1}}^{t-1} D_\ell + 2I\gamma^2 L^2 \sum_{t=\bar{t}_{s-1}+1}^{\bar{t}_s - 1}\sum_{\ell=\bar{t}_{s-1}}^{t-1} B_\ell + 2(I-1)\sigma^2 + 2(I-1)\zeta_g^2$$

$$\leq 2I^2\eta^2 L^2 \sum_{\ell=\bar{t}_{s-1}}^{\bar{t}_s - 1} D_\ell + 2I^2\gamma^2 L^2 \sum_{\ell=\bar{t}_{s-1}}^{\bar{t}_s - 1} B_\ell + \frac{2(I-1)\sigma^2}{b_y} + 2(I-1)\zeta_g^2$$

Combine with the case of $t = \bar{t}_s$ in Lemma C.19, we have:

$$\sum_{t=\bar{t}_{s-1}}^{\bar{t}_s - 1} B_t \leq 2I^2\eta^2 L^2 \sum_{\ell=\bar{t}_{s-1}}^{\bar{t}_s - 1} D_\ell + 2I^2\gamma^2 L^2 \sum_{\ell=\bar{t}_{s-1}}^{\bar{t}_s - 1} B_\ell + \frac{2I\sigma^2}{b_y} + 2I\zeta_g^2 \tag{33}$$

Apply Lemma C.20 recursively, we have:

$$A_t \leq \sum_{\ell=\bar{t}_{s-1}}^{t-1}\left(1 + \frac{1}{I}\right)^{t-\ell}\left(\frac{64I\tau^2 C_f^2\zeta_{g,yy}^2}{\mu^2} + 32I\tau^2\zeta_f^2 + \frac{2\tau^2\sigma^2}{b_x} + 128I^2\eta^2\tau^2\tilde{L}_2^2 \sum_{\bar{\ell}=\bar{t}_{s-1}}^{\ell-1} D_{\bar{\ell}} + 128I^2\gamma^2\tau^2\tilde{L}_2^2 \sum_{\bar{\ell}=\bar{t}_{s-1}}^{\ell-1} B_{\bar{\ell}}\right)$$

$$\leq \sum_{\ell=\bar{t}_{s-1}}^{t-1}\left(\frac{192I\tau^2 C_f^2\zeta_{g,yy}^2}{\mu^2} + 96I\tau^2\zeta_f^2 + \frac{6\tau^2\sigma^2}{b_x} + 384I^2\eta^2\tau^2\tilde{L}_2^2 \sum_{\bar{\ell}=\bar{t}_{s-1}}^{\ell-1} D_{\bar{\ell}} + 384I^2\gamma^2\tau^2\tilde{L}_2^2 \sum_{\bar{\ell}=\bar{t}_{s-1}}^{\ell-1} B_{\bar{\ell}}\right)$$

where we denote $\tilde{L}_2^2 = (L^2 + \frac{2L_{y^2}^2 C_f^2}{\mu^2})$, and the second inequality uses the fact that $t - l \leq I$ and the inequality $log(1 + a/x) \leq a/x$ for $x > -a$, so we have $(1 + a/x)^x \leq e^a$, Then we choose $a = 1$ and $x = I$. Finally, we use the fact that $e^1 \leq 3$. Next, we sum from $\bar{t}_{s-1}$ to $\bar{t}_s - 1$ to have:

$$\sum_{t=\bar{t}_{s-1}}^{\bar{t}_s-1} A_t \leq \sum_{\ell=\bar{t}_{s-1}}^{\bar{t}_s-1} \left( \frac{48 I^2 \tau^2 C_f^2 \zeta_{g,yy}^2}{\mu^2} + 96 I^2 \tau^2 \zeta_f^2 + \frac{6 I \tau^2 \sigma^2}{b_x} + 24 I^3 \eta^2 \tau^2 \tilde{L}_2^2 \sum_{\bar{\ell}=\bar{t}_{s-1}}^{\ell-1} D_{\bar{\ell}} + 24 I^3 \gamma^2 \tau^2 \tilde{L}_2^2 \sum_{\bar{\ell}=\bar{t}_{s-1}}^{\ell-1} B_{\bar{\ell}} \right)$$

$$\leq \frac{192 I^3 \tau^2 C_f^2 \zeta_{g,yy}^2}{\mu^2} + 96 I^3 \tau^2 \zeta_f^2 + \frac{6 I^2 \tau^2 \sigma^2}{b_x} + 384 I^4 \eta^2 \tau^2 \tilde{L}_2^2 \sum_{\ell=\bar{t}_{s-1}}^{\bar{t}_s-1} D_\ell + 384 I^4 \gamma^2 \tau^2 \tilde{L}_2^2 \sum_{\ell=\bar{t}_{s-1}}^{\bar{t}_s-1} B_\ell$$

$$(34)$$

Combine Eq. 32, Eq. 33 and Eq. 34, and we choose $\eta$, $\gamma$ and $\tau$ such that $I^2 \gamma^2 L^2 < \frac{1}{4}$, $I^2 \gamma^2 \tilde{L}_1^2 < \frac{1}{64}$, $I^2 \eta^2 \tilde{L}_1^2 < \frac{1}{32}$, $I^2 \eta^2 L^2 < \frac{1}{16}$, $I^2 \tau^2 \tilde{L}_2^2 < \frac{1}{128^2}$, $I^2 \tau^2 L^2 < \frac{1}{48}$, we get the claim in the lemma, by using the fact that $\tilde{L}_1 > L$ and $\tilde{L}_2 > L$, we get the simplified condition in the lemma.

$\square$

### C.2.3 Descent Lemma

**Lemma C.22.** *For all $t \in [\bar{t}_{s-1}, \bar{t}_s - 1]$, the iterates generated satisfy:*

$$\mathbb{E}\big\|\nabla h(\bar{x}_t) - \mathbb{E}_\xi[\bar{\nu}_t]\big\|^2 \leq \frac{2\tilde{L}_1^2}{M} \sum_{m=1}^M \mathbb{E}\big[\big\|\bar{x}_t - x_t^{(m)}\big\|^2 + 2\big\|\bar{y}_t - y_t^{(m)}\big\|^2 + 2\big\|y_{\bar{x}_t} - \bar{y}_t\big\|^2\big] + 4L^2 \mathbb{E}\big\|u_{\bar{x}_t} - \bar{u}_t\big\|^2$$

*where we denote $u_{\bar{x}_t} = [\nabla_{y^2} g(\bar{x}_t, y_{\bar{x}_t})]^{-1} \nabla_y f(\bar{x}_t, y_{\bar{x}_t})$ and $\tilde{L}_1^2 = \big(L^2 + \frac{2L_{xy}^2 C_f^2}{\mu^2}\big)$ is a constant.*

*Proof.* By $\nabla h(\bar{x}_t) = \Phi(\bar{x}, y_{\bar{x}})$, we have:

$$\mathbb{E}\big\|\nabla h(\bar{x}_t) - \mathbb{E}_\xi[\bar{\nu}_t]\big\|^2 \leq \mathbb{E}\big\|\nabla_x f(\bar{x}_t, y_{\bar{x}_t}) - \nabla_{xy} g(\bar{x}_t, y_{\bar{x}_t}) \times [\nabla_{y^2} g(\bar{x}_t, y_{\bar{x}_t})]^{-1} \nabla_y f(\bar{x}_t, y_{\bar{x}_t})$$

$$- \frac{1}{M} \sum_{m=1}^M \big(\nabla_x f^{(m)}(x_t^{(m)}, y_t^{(m)}) - \nabla_{xy} g^{(m)}(x_t^{(m)}, y_t^{(m)}) u_t^{(m)}\big)\big\|^2$$

$$\leq 2\mathbb{E}\big\|\nabla_x f(\bar{x}_t, y_{\bar{x}_t}) - \frac{1}{M} \sum_{m=1}^M \nabla_x f^{(m)}(x_t^{(m)}, y_t^{(m)})\big\|^2$$

$$+ 2\mathbb{E}\big\|\nabla_{xy} g(\bar{x}_t, y_{\bar{x}_t}) \times [\nabla_{y^2} g(\bar{x}_t, y_{\bar{x}_t})]^{-1} \nabla_y f(\bar{x}_t, y_{\bar{x}_t})$$

$$- \frac{1}{M} \sum_{m=1}^M \nabla_{xy} g^{(m)}(x_t^{(m)}, y_t^{(m)}) u_t^{(m)}\big\|^2$$

We denote the two terms above as $T_1$, $T_2$ respectively. For the first term $T_1$, we have:

$$T_1 \leq \frac{2L^2}{M} \sum_{m=1}^M \mathbb{E}\big[\big\|\bar{x}_t - x_t^{(m)}\big\|^2 + \big\|y_{\bar{x}_t} - y_t^{(m)}\big\|^2\big] \leq \frac{2L^2}{M} \sum_{m=1}^M \mathbb{E}\big[\big\|\bar{x}_t - x_t^{(m)}\big\|^2 + 2\big\|\bar{y}_t - y_t^{(m)}\big\|^2 + 2\big\|y_{\bar{x}_t} - \bar{y}_t\big\|^2\big]$$

For the second term $T_2$, we have:

$$T_2 \leq \frac{4C_f^2}{\mu^2 M} \sum_{m=1}^M \mathbb{E}\big\|\nabla_{xy} g^{(m)}(\bar{x}_t, y_{\bar{x}_t}) - \nabla_{xy} g^{(m)}(x_t^{(m)}, y_t^{(m)})\big\|^2$$

$$+ 4L^2 \mathbb{E}\big\|[\nabla_{y^2} g(\bar{x}_t, y_{\bar{x}})]^{-1} \nabla_y f(\bar{x}_t, y_{\bar{x}_t}) - \bar{u}_t\big\|^2$$

We denote the first term above as $T_{2,1}$. For the term $T_{2,1}$, we have:

$$T_{2,1} \leq \frac{4L_{xy}^2 C_f^2}{\mu^2 M} \sum_{m=1}^M \mathbb{E}\big[\big\|\bar{x}_t - x_t^{(m)}\big\|^2 + 2\big\|\bar{y}_t - y_t^{(m)}\big\|^2 + 2\big\|y_{\bar{x}_t} - \bar{y}_t\big\|^2\big]$$

Combine everything together, we get the claim in the lemma. $\square$

**Lemma C.23.** *For all $t \in [\bar{t}_{s-1} + 1, \bar{t}_s - 1]$ and $s \in [S]$, suppose $\eta < \frac{1}{2\bar{L}}$, the iterates generated satisfy:*

$$\mathbb{E}[h(\bar{x}_{t+1})] \leq \mathbb{E}[h(\bar{x}_t)] - \frac{\eta}{2}\mathbb{E}\|\nabla h(\bar{x}_t)\|^2 - \frac{\eta}{4}\|\mathbb{E}_\xi[\nu_t^{(m)}]\|^2 + \frac{\eta^2 \bar{L}\sigma^2}{2b_x M} + 2\eta L^2 \mathbb{E}\|u_{\bar{x}_t} - \bar{u}_t\|^2$$
$$+ \frac{\eta \tilde{L}_1^2}{M}\sum_{m=1}^{M}\mathbb{E}\big[\|\bar{x}_t - x_t^{(m)}\|^2 + 2\|\bar{y}_t - y_t^{(m)}\|^2 + 2\|y_{\bar{x}_t} - \bar{y}_t\|^2\big]$$

*where the expectation is w.r.t the stochasticity of the algorithm.*

*Proof.* Using the smoothness of $f$ we have:

$$\mathbb{E}[h(\bar{x}_{t+1})] \leq \mathbb{E}[h(\bar{x}_t)] + \mathbb{E}\langle \nabla h(\bar{x}_t), \bar{x}_{t+1} - \bar{x}_t\rangle + \frac{\bar{L}}{2}\mathbb{E}\|\bar{x}_{t+1} - \bar{x}_t\|^2$$
$$= \mathbb{E}[h(\bar{x}_t)] - \eta\mathbb{E}\langle \nabla h(\bar{x}_t), \mathbb{E}_\xi[\bar{\nu}_t]\rangle + \frac{\eta^2 \bar{L}}{2}\mathbb{E}\|\mathbb{E}_\xi[\bar{\nu}_t]\|^2 + \frac{\eta^2 \bar{L}\sigma^2}{2b_x M}$$
$$\overset{(a)}{=} \mathbb{E}[h(\bar{x}_t)] - \frac{\eta}{2}\mathbb{E}\|\nabla h(\bar{x}_t)\|^2 + \frac{\eta}{2}\mathbb{E}\|\nabla h(\bar{x}_t) - \mathbb{E}_\xi[\bar{\nu}_t]\|^2 - \left(\frac{\eta}{2} - \frac{\eta^2 \bar{L}}{2}\right)\|\mathbb{E}_\xi[\bar{\nu}_t]\|^2 + \frac{\eta^2 \bar{L}\sigma^2}{2b_x M}$$
$$\overset{(b)}{\leq} \mathbb{E}[h(\bar{x}_t)] - \frac{\eta}{2}\mathbb{E}\|\nabla h(\bar{x}_t)\|^2 - \frac{\eta}{4}\|\mathbb{E}_\xi[\nu_t^{(m)}]\|^2 + \frac{\eta^2 \bar{L}\sigma^2}{2b_x M}$$
$$+ \frac{\eta \tilde{L}_1^2}{M}\sum_{m=1}^{M}\mathbb{E}\big[\|\bar{x}_t - x_t^{(m)}\|^2 + 2\|\bar{y}_t - y_t^{(m)}\|^2 + 2\|y_{\bar{x}_t} - \bar{y}_t\|^2\big] + 2\eta L^2\mathbb{E}\|u_{\bar{x}_t} - \bar{u}_t\|^2$$

*where equality $(a)$ uses $\langle a, b\rangle = \frac{1}{2}[\|a\|^2 + \|b\|^2 - \|a-b\|^2]$; (b) follows the assumption that $\eta < 1/2\bar{L}$ and Lemma C.22.* $\qquad\square$

### C.2.4 Proof of Convergence Theorem

We first denote the following potential function $\mathcal{G}(t)$:

$$\mathcal{G}_t = \mathbb{E}[h(\bar{x}_t)] + \frac{9\eta\tilde{L}_1^2}{\mu\gamma}\mathbb{E}\|\bar{y}_t - y_{\bar{x}_t}\|^2 + \frac{9\eta L^2}{\mu\tau}\mathbb{E}\|\bar{u}_t - u_{\bar{x}_t}\|^2$$

**Theorem C.24.** *Suppose we choose $\tau = \min(\frac{1}{128I\tilde{L}_2}, \frac{1}{144\kappa L})$, then denote $\bar{\gamma} = \min(\frac{1}{8I\tilde{L}_2}, \frac{\tau}{36\kappa\bar{L}}, \frac{1}{4L}, \frac{1}{8I\tilde{L}_1})$, if we choose $\eta = \frac{\mu\gamma}{36\kappa\tilde{L}_1}$, and $\gamma = \min\left(\bar{\gamma}, \left(\frac{\Delta'}{C'_\gamma T}\right)^{1/3}\right)$ and $r = \frac{C_f}{\mu}$ where $\Delta'$ and $C'_\gamma$ are constants denoted in Eq. 36, then we have:*

$$\frac{1}{T}\sum_{t=1}^{T}\mathbb{E}\|\nabla h(\bar{x}_t)\|^2 = O\left(\frac{\kappa^8}{T} + \left(\frac{\kappa^{12}}{T^2}\right)^{1/3} + \frac{\kappa^4\sigma^2}{b_y M} + \frac{\sigma^2}{b_x M}\right)$$

*and to reach an $\epsilon$ stationary point, we choose the inner batch size $b_y = O(M^{-1}\kappa^4\epsilon^{-1})$, upper batch size $b_x = O(M^{-1}\epsilon^{-1})$, and $T = O(\kappa^6\epsilon^{-1.5})$ number of iterations.*

*Proof.* Similar to Lemma C.21, we denote $D_t = \frac{1}{M}\sum_{m=1}^{M}\|\nu_t^{(m)} - \bar{\nu}_t\|^2$, $B_t = \frac{1}{M}\sum_{m=1}^{M}\|\omega_t^{(m)} - \bar{\omega}_t\|^2$, $C_t = \mathbb{E}\|\bar{y}_t - y_{\bar{x}_t}\|^2$ and $A_t = \frac{1}{M}\sum_{m=1}^{M}\mathbb{E}\|u_t^{(m)} - \bar{u}_t\|^2$, additionally, we denote $E_t = \|\mathbb{E}_\xi[\bar{\nu}_t]\|^2$ and $F_t = \mathbb{E}\|\bar{u}_t - u_{\bar{x}_t}\|^2$. Combine Lemma C.15, Lemma C.16 and the definition of the

potential function we have:

$$\mathcal{G}_{\bar{t}_s} - \mathcal{G}_{\bar{t}_{s-1}} \leq -\frac{\eta}{2} \sum_{t=\bar{t}_{s-1}}^{\bar{t}_s-1} \mathbb{E}\|\nabla h(\bar{x}_t)\|^2 - \frac{\eta L^2}{4} \sum_{t=\bar{t}_{s-1}}^{\bar{t}_s-1} F_t - \frac{\eta \tilde{L}_1^2}{4} \sum_{t=\bar{t}_{s-1}}^{\bar{t}_s-1} C_t$$

$$- \frac{\eta}{4} \left( 1 - \frac{162\kappa^2\eta^2\tilde{L}_1^2}{\mu^2\gamma^2} - \frac{180\kappa^2\eta^2\bar{L}^2}{\tau^2} \right) \sum_{t=\bar{t}_{s-1}}^{\bar{t}_s-1} E_t + \left( \tilde{L}_1^2 + \frac{81\kappa^2\tilde{L}_1^2}{2} + \frac{45\kappa^2\tilde{L}_2^2}{16} \right) I^2 \eta^3 \sum_{t=\bar{t}_{s-1}}^{\bar{t}_s-1} D_t$$

$$+ \left( 2\tilde{L}_1^2 + \frac{81\kappa^2\tilde{L}_1^2}{2} + \frac{45\kappa^2\tilde{L}_2^2}{16} \right) I^2 \gamma^2 \eta \sum_{t=\bar{t}_{s-1}}^{\bar{t}_s-1} B_t + \frac{81 I \kappa^2 \tilde{L}_1^2 \eta^3 \sigma^2}{2\mu^2\gamma^2 b_x M} + \frac{36 I \tilde{L}_1^2 \eta \sigma^2}{\mu^2 b_y M}$$

$$+ \frac{45 I \kappa^2 \bar{L}^2 \eta^3 \sigma^2}{\tau^2 b_x M} + \frac{45 I \kappa L \tau \eta \sigma^2}{4 b_x M} + \frac{\eta^2 I \bar{L} \sigma^2}{2 b_x M}$$

to bound the coefficients above, we choose $\eta \leq \min\left(\frac{\mu\gamma}{36\kappa\tilde{L}_1}, \frac{\tau}{36\kappa\bar{L}}, \frac{1}{4L}\right)$, $\tau < \frac{1}{144\kappa L}$ and we denote $C_1 = \left( 2\tilde{L}_1^2 + \frac{81\kappa^2\tilde{L}_1^2}{2} + \frac{45\kappa^2\tilde{L}_2^2}{16} \right) I^2$. Then we have:

$$\mathcal{G}_{\bar{t}_s} - \mathcal{G}_{\bar{t}_{s-1}} \leq -\frac{\eta}{2} \sum_{t=\bar{t}_{s-1}}^{\bar{t}_s-1} \mathbb{E}\|\nabla h(\bar{x}_t)\|^2 - \frac{\eta L^2}{4} \sum_{t=\bar{t}_{s-1}}^{\bar{t}_s-1} F_t - \frac{\eta \tilde{L}_1^2}{4} \sum_{t=\bar{t}_{s-1}}^{\bar{t}_s-1} C_t - \frac{\eta}{8} \sum_{t=\bar{t}_{s-1}}^{\bar{t}_s-1} E_t$$

$$+ C_1 \eta^3 \sum_{t=\bar{t}_{s-1}}^{\bar{t}_s-1} D_t + C_1 \gamma^2 \eta \sum_{t=\bar{t}_{s-1}}^{\bar{t}_s-1} B_t + \frac{I \eta \sigma^2}{2 b_x M} + \frac{36 I \tilde{L}_1^2 \eta \sigma^2}{\mu^2 b_y M}$$

Next, we combine with lemma C.21 to have:

$$\mathcal{G}_{\bar{t}_s} - \mathcal{G}_{\bar{t}_{s-1}} \leq -\frac{\eta}{2} \sum_{t=\bar{t}_{s-1}}^{\bar{t}_s-1} \mathbb{E}\|\nabla h(\bar{x}_t)\|^2 - \frac{\eta L^2}{4} \sum_{t=\bar{t}_{s-1}}^{\bar{t}_s-1} F_t - \frac{\eta \tilde{L}_1^2}{4} \sum_{t=\bar{t}_{s-1}}^{\bar{t}_s-1} C_t - \frac{\eta}{8} \sum_{t=\bar{t}_{s-1}}^{\bar{t}_s-1} E_t + \frac{I \eta \sigma^2}{2 b_x M} + \frac{36 I \tilde{L}_1^2 \eta \sigma^2}{\mu^2 b_y M}$$

$$+ C_1 \eta^3 \left( 96 I \zeta_f^2 + 16 I \zeta_g^2 + \frac{16 I C_f^2 \zeta_{g,yy}^2}{\mu^2} + \frac{32 I C_f^2 \zeta_{g,xy}^2}{\mu^2} + \frac{16 I \sigma^2}{b_y} + \frac{20 I \sigma^2}{b_x} \right)$$

$$+ C_1 \gamma^2 \eta \left( 24 I \zeta_f^2 + 8 I \zeta_g^2 + \frac{4 I C_f^2 \zeta_{g,yy}^2}{\mu^2} + \frac{8 I C_f^2 \zeta_{g,xy}^2}{\mu^2} + \frac{8 I \sigma^2}{b_y} + \frac{5 I \sigma^2}{b_x} \right)$$

Sum over all $s \in [S]$ (assume $T = SI + 1$ without loss of generality) to obtain:

$$\frac{\eta}{2} \sum_{t=1}^{T} \mathbb{E}\|\nabla h(\bar{x}_t)\|^2 \leq \mathcal{G}_1 - \mathcal{G}_T + \frac{T \eta \sigma^2}{2 b_x M} + \frac{36 T \tilde{L}_1^2 \eta \sigma^2}{\mu^2 b_y M}$$

$$+ C_1 \eta^3 \left( 96 T \zeta_f^2 + 16 T \zeta_g^2 + \frac{16 T C_f^2 \zeta_{g,yy}^2}{\mu^2} + \frac{32 T C_f^2 \zeta_{g,xy}^2}{\mu^2} + \frac{16 T \sigma^2}{b_y} + \frac{20 T \sigma^2}{b_x} \right)$$

$$+ C_1 \gamma^2 \eta \left( 24 T \zeta_f^2 + 8 T \zeta_g^2 + \frac{4 T C_f^2 \zeta_{g,yy}^2}{\mu^2} + \frac{8 T C_f^2 \zeta_{g,xy}^2}{\mu^2} + \frac{8 T \sigma^2}{b_y} + \frac{5 T \sigma^2}{b_x} \right)$$

$$\leq \Delta + \frac{9\eta \tilde{L}_1^2 \Delta_y}{\mu\gamma} + \frac{9\eta L^2 \Delta_u}{\mu\tau} + \frac{T \eta \sigma^2}{2 b_x M} + \frac{36 T \tilde{L}_1^2 \eta \sigma^2}{\mu^2 b_y M}$$

$$+ C_1 \eta^3 \left( 96 T \zeta_f^2 + 16 T \zeta_g^2 + \frac{16 T C_f^2 \zeta_{g,yy}^2}{\mu^2} + \frac{32 T C_f^2 \zeta_{g,xy}^2}{\mu^2} + \frac{16 T \sigma^2}{b_y} + \frac{20 T \sigma^2}{b_x} \right)$$

$$+ C_1 \gamma^2 \eta \left( 24 T \zeta_f^2 + 8 T \zeta_g^2 + \frac{4 T C_f^2 \zeta_{g,yy}^2}{\mu^2} + \frac{8 T C_f^2 \zeta_{g,xy}^2}{\mu^2} + \frac{8 T \sigma^2}{b_y} + \frac{5 T \sigma^2}{b_x} \right)$$

we define $\Delta = h(x_1) - h^*$ as the initial sub-optimality of the function, $\Delta_y = \|y_1 - y_{x_1}\|^2$ as the initial sub-optimality of the inner variable estimation, $\Delta_u = \|u_1 - u_{x_1}\|^2$ as the initial sub-optimality

of the hyper-gradient estimation. Then we divide by $\eta T/2$ on both sides and have:

$$\frac{1}{T}\sum_{t=1}^{T}\mathbb{E}\|\nabla h(\bar{x}_t)\|^2 \leq \frac{2\Delta}{\eta T} + \frac{18\tilde{L}_1^2\Delta_y}{\mu\gamma T} + \frac{18L^2\Delta_u}{\mu\tau T} + \frac{\sigma^2}{b_x M} + \frac{72\tilde{L}_1^2\sigma^2}{\mu^2 b_y M}$$

$$+ 2C_1\eta^2\big(96\zeta_f^2 + 16\zeta_g^2 + \frac{16C_f^2\zeta_{g,yy}^2}{\mu^2} + \frac{32C_f^2\zeta_{g,xy}^2}{\mu^2} + \frac{16\sigma^2}{b_y} + \frac{20\sigma^2}{b_x}\big)$$

$$+ 2C_1\gamma^2\big(24\zeta_f^2 + 8\zeta_g^2 + \frac{4C_f^2\zeta_{g,yy}^2}{\mu^2} + \frac{8C_f^2\zeta_{g,xy}^2}{\mu^2} + \frac{8\sigma^2}{b_y} + \frac{5\sigma^2}{b_x}\big)$$

For ease of notation, we denote constants $C_\eta = 2C_1(96\zeta_f^2 + 16\zeta_g^2 + \frac{16C_f^2\zeta_{g,yy}^2}{\mu^2} + \frac{32C_f^2\zeta_{g,xy}^2}{\mu^2} + \frac{16\sigma^2}{b_y} + \frac{20\sigma^2}{b_x})$ and $C_\gamma = 2C_1(24\zeta_f^2 + 8\zeta_g^2 + \frac{4C_f^2\zeta_{g,yy}^2}{\mu^2} + \frac{8C_f^2\zeta_{g,xy}^2}{\mu^2} + \frac{8\sigma^2}{b_y} + \frac{5\sigma^2}{b_x})$, we have:

$$\frac{1}{T}\sum_{t=1}^{T}\mathbb{E}\|\nabla h(\bar{x}_t)\|^2 \leq \frac{2\Delta}{\eta T} + \frac{18\tilde{L}_1^2\Delta_y}{\mu\gamma T} + \frac{18L^2\Delta_u}{\mu\tau T} + \frac{\sigma^2}{b_x M} + \frac{72\tilde{L}_1^2\sigma^2}{\mu^2 b_y M} + C_\eta\eta^2 + C_\gamma\gamma^2 \quad (35)$$

Recall that, we have the condition that $\eta \leq \min\big(\frac{1}{8I\tilde{L}_2}, \frac{\mu\gamma}{36\kappa\tilde{L}_1}, \frac{\tau}{36\kappa\bar{L}}, \frac{1}{4\bar{L}}\big)$, $\gamma \leq \frac{1}{8I\tilde{L}_1}$, $\tau \leq \min(\frac{1}{128I\tilde{L}_2}, \frac{1}{144\kappa L})$. Suppose we choose $\tau = \min(\frac{1}{128I\tilde{L}_2}, \frac{1}{144\kappa L})$, then denote

$$\bar{\gamma} = \min(\frac{1}{8I\tilde{L}_2}, \frac{\tau}{36\kappa\bar{L}}, \frac{1}{4\bar{L}}, \frac{1}{8I\tilde{L}_1}),$$

and let $\gamma \leq \bar{\gamma}$, and $\eta = \frac{\mu\gamma}{36\kappa\tilde{L}_1}$, then we have:

$$\frac{1}{T}\sum_{t=1}^{T}\mathbb{E}\|\nabla h(\bar{x}_t)\|^2 \leq \frac{72\kappa\tilde{L}_1\Delta + 18\tilde{L}_1^2\Delta_y}{\mu\gamma T} + \big(\frac{C_\eta\mu^2}{36^2\kappa^2\tilde{L}_1^2} + C_\gamma\big)\gamma^2 + \frac{18L^2\Delta_u}{\mu\tau T} + \frac{\sigma^2}{b_x M} + \frac{72\tilde{L}_1^2\sigma^2}{\mu^2 b_y M}$$

We denote

$$C_\gamma' = \big(\frac{C_\eta\mu^2}{36^2\kappa^2\tilde{L}_1^2} + C_\gamma\big), \ \Delta' = \frac{72\kappa\tilde{L}_1\Delta + 18\tilde{L}_1^2\Delta_y}{\mu}, \quad (36)$$

then we choose $\gamma$ as:

$$\gamma = \min\big(\bar{\gamma}, \big(\frac{\Delta'}{C_\gamma' T}\big)^{1/3}\big)$$

and obtain:

$$\frac{1}{T}\sum_{t=1}^{T}\mathbb{E}\|\nabla h(\bar{x}_t)\|^2 \leq \frac{\Delta'}{\bar{\gamma}T} + \big(\frac{C_\gamma'(\Delta')^2}{T^2}\big)^{1/3} + \frac{18L^2\Delta_u}{\mu\tau T} + \frac{\sigma^2}{b_x M} + \frac{72\tilde{L}_1^2\sigma^2}{\mu^2 b_y M}$$

Finally, since $\tilde{L}_1 = O(\kappa)$, $\bar{L} = O(\kappa^3)$, suppose we choose $I = O(1)$, then we have $\tau^{-1} = O(\kappa)$, $\bar{\gamma}^{-1} = O(\kappa^5)$, $\Delta' = O(\kappa^3)$, $C_1 = O(\kappa^4)$, $C_\eta = O(\kappa^6)$, $C_\gamma = O(\kappa^6)$, $C_\gamma' = O(\kappa^6)$ then we have:

$$\frac{1}{T}\sum_{t=1}^{T}\mathbb{E}\|\nabla h(\bar{x}_t)\|^2 = O\left(\frac{\kappa^8}{T} + \big(\frac{\kappa^{12}}{T^2}\big)^{1/3} + \frac{\kappa^4\sigma^2}{b_y M} + \frac{\sigma^2}{b_x M}\right)$$

and to reach an $\epsilon$ stationary point, we choose the inner batch size $b_y = O(M^{-1}\kappa^4\epsilon^{-1})$, upper batch size $b_x = O(M^{-1}\epsilon^{-1})$, and $T = O(\kappa^6\epsilon^{-1.5})$ number of iterations. $\qquad\square$

---

**Algorithm 3** FedBiO- Local Lower Level Problem

---

1: **Input:** Initial states $x_1$, $y_1$; learning rates $\{\gamma_t, \eta_t\}_{t=1}^T$
2: **Initialization:** Set $x_1^{(m)} = x_1$, $y_1^{(m)} = y_1$;
3: **for** $t = 1$ **to** $T$ **do**
4:      Randomly sample mutually independent minibatch of samples $\mathcal{B}_y$ and $\mathcal{B}_x$ of size b;
5:      $\omega_t^{(m)} = \nabla_y g^{(m)}(x_t^{(m)}, y_t^{(m)}, \mathcal{B}_y)$ and $\nu_t^{(m)} = \Phi^{(m)}(x^{(m)}, y^{(m)}; \mathcal{B}_x)$;
6:      $\hat{y}_{t+1}^{(m)} = y_t^{(m)} - \gamma_t \omega_t^{(m)}$, $\hat{x}_{t+1}^{(m)} = x_t^{(m)} - \eta_t \nu_t^{(m)}$;
7:      **if** $t \bmod I = 0$ **then**
8:          $y_{t+1}^{(m)} = \hat{y}_{t+1}^{(m)}$, $x_{t+1}^{(m)} = \frac{1}{M} \sum_{j=1}^M \hat{x}_{t+1}^{(j)}$
9:      **else**
10:         $y_{t+1}^{(m)} = \hat{y}_{t+1}^{(m)}$, $x_{t+1}^{(m)} = \hat{x}_{t+1}^{(m)}$
11:      **end if**
12: **end for**

---

---

**Algorithm 4** FedBiOAcc - Local Lower Level Problem

---

1: **Input:** Constants $c_\omega$, $c_\nu$, $\gamma$, $\eta$; learning rate schedule $\{\alpha_t\}$, $t \in [T]$, initial state $(x_1, y_1)$;
2: **Initialization:** Set $y_1^{(m)} = y_1$, $x_1^{(m)} = x_1$, $\omega_1^{(m)} = \nabla_y g^{(m)}(x_1, y_1, \mathcal{B}_y)$, $\nu_1^{(m)} = \Phi^{(m)}(x_1, y_1; \mathcal{B}_x)$ for $m \in [M]$
3: **for** $t = 1$ **to** $T$ **do**
4:      $\hat{y}_{t+1}^{(m)} = y_t^{(m)} - \gamma \alpha_t \omega_t^{(m)}$, $\hat{x}_{t+1}^{(m)} = x_t^{(m)} - \eta \alpha_t \nu_t^{(m)}$, $\hat{u}_{t+1}^{(m)} = u_t^{(m)} - \tau \alpha_t q_t^{(m)}$
5:      **if** $t \bmod I = 0$ **then**
6:          $y_{t+1}^{(m)} = \hat{y}_{t+1}^{(m)}$, $x_{t+1}^{(m)} = \frac{1}{M} \sum_{j=1}^M \hat{x}_{t+1}^{(j)}$
7:      **else**
8:         $y_{t+1}^{(m)} = \hat{y}_{t+1}^{(m)}$, $x_{t+1}^{(m)} = \hat{x}_{t+1}^{(m)}$,
9:      **end if**
10:     Randomly sample minibatches $\mathcal{B}_y$ and $\mathcal{B}_x$
11:     $\hat{\omega}_{t+1}^{(m)} = \nabla_y g^{(m)}(x_{t+1}^{(m)}, y_{t+1}^{(m)}, \mathcal{B}_y) + (1 - c_\omega \alpha_t^2)(\omega_t^{(m)} - \nabla_y g^{(m)}(x_t^{(m)}, y_t^{(m)}, \mathcal{B}_y))$
12:     $\hat{\nu}_{t+1}^{(m)} = \Phi^{(m)}(x_{t+1}^{(m)}, y_{t+1}^{(m)}; \mathcal{B}_x) + (1 - c_\nu \alpha_t^2)(\nu_t^{(m)} - \Phi^{(m)}(x_t^{(m)}, y_t^{(m)}; \mathcal{B}_x))$
13:     **if** $t \bmod I = 0$ **then**
14:        $\omega_{t+1}^{(m)} = \hat{\omega}_{t+1}^{(m)}$, $\nu_{t+1}^{(m)} = \frac{1}{M} \sum_{j=1}^M \hat{\nu}_{t+1}^{(j)}$,
15:     **else**
16:       $\omega_{t+1}^{(m)} = \hat{\omega}_{t+1}^{(m)}$, $\nu_{t+1}^{(m)} = \hat{\nu}_{t+1}^{(m)}$
17:     **end if**
18: **end for**

---

# D    Proof for Local Lower Level Problem

The FedBiOAcc-Local and FedBiO-Local are presented in Algorithm 4 and Algorithm 3, respectively. Then in this section, we discuss the convergence rate of the two algorithms. Please see Theorem D.12 and Theorem D.19 for the convergence rates.

For Eq. (10), we also assume Assumptions 3.1 -3.4, with a slightly different assumption to the heterogeneity as follows:

**Assumption D.1.** For any $m, j \in [M]$ and $z = (x, y)$, we have: $\|\nabla f^{(m)}(z) - \nabla f^{(j)}(z)\| \leq \zeta_f$, $\|\nabla_{xy} g^{(m)}(z) - \nabla_{xy} g^{(j)}(z)\| \leq \zeta_{g,xy}$, $\|\nabla_{y^2} g^{(m)}(z) - \nabla_{y^2} g^{(j)}(z)\| \leq \zeta_{g,yy}$, $\|y_x^{(m)} - y_x^{(j)}\| \leq \zeta_{g^*}$, where $\zeta_f$, $\zeta_{g,xy}$, $\zeta_{g,yy}$, $\zeta_{g^*}$ are constants.

Note that we remove the requirement of gradient dissimilarity $\zeta_g$ in Assumption 3.5 and add the dissimilarity bound $\zeta_{g^*}$ for the minimizer of the lower level problem. Note that Assumption D.1 is a sufficient condition such that the dissimilarity of local hyper-gradient is bounded by some constant $\zeta$.

**Proposition D.2.** *(Lemma 4 and 7 in [59]) Suppose Assumptions 3.2, 3.3 and 3.4 hold and $\tau < \frac{1}{L}$, the hypergradient estimator $\Phi(x, y; \mathcal{B}_x)$ w.r.t. x based on a minibatch $\mathcal{B}_x$ has bounded variance and bias:*

*a)* $\|\mathbb{E}[\Phi^{(m)}(x, y; \mathcal{B}_x)] - \Phi^{(m)}(x, y)\| \leq G_1$, *where* $G_1 = \kappa(1 - \tau\mu)^{Q+1} C_f$

b) $\mathbb{E}\|\Phi^{(m)}(x,y;\mathcal{B}_x) - \mathbb{E}[\Phi^{(m)}(x,y;\mathcal{B}_x)]\|^2 \le G_2^2$, where $G_2^2 = (2C_f^2 + 12C_f^2 L^2\tau^2(Q+1)^2 + 4C_f^2 L^2(Q+2)(Q+1)^2\tau^4\sigma^2)/b_x$

**Proposition D.3.** *Suppose Assumptions 3.2 and 3.3 hold, the following statements hold:*

a) $y_x^{(m)}$ *is Lipschitz continuous in $x$ with constant $\rho = \kappa$, where $\kappa = \frac{L}{\mu}$ is the condition number of $g^{(m)}(x,y)$.*

b) $\|\Phi^{(m)}(x_1;y_1) - \Phi^{(m)}(x_2;y_2)\|^2 \le \hat{L}^2(\|x_1-x_2\|^2 + \|y_1-y_2\|^2)$, *where $\hat{L} = O(\kappa^2)$.*

d) $h^{(m)}(x)$ *is Lipschitz continuous in $x$ with constant $\bar{L}$ i.e., for any given $x_1, x_2 \in X$, we have $\|\nabla h^{(m)}(x_2) - \nabla h^{(m)}(x_1)\| \le \bar{L}\|x_2 - x_1\|$ where $\bar{L} = O(\kappa^3)$.*

This is a standard results in bilevel optimization and we omit the proof here.

**Proposition D.4.** *In Eq. 10, suppose Assumption 3.1, 3.2, 3.3, D.1 hold, we have:*

$$\|\nabla h^{(m)}(x) - \nabla h^{(j)}(x)\| \le (1+\kappa)\zeta_f + \frac{C_f}{\mu}\zeta_{g,xy} + \frac{\kappa C_f}{\mu}\zeta_{g,yy} + \left((1+\kappa)L + \frac{C_f L_{xy}}{\mu} + \frac{\kappa C_f L_{y^2}}{\mu}\right)\zeta_{g^*} := \zeta$$

*Proof.* For $h^{(m)}(x) = f^{(m)}(x, y_x^{(m)}), m \in [M]$ in Eq. 10, we have:

$$
\begin{aligned}
\|\nabla h^{(m)}(x) - \nabla h^{(j)}(x)\| &= \|\nabla_x f^{(m)}(x, y_x^{(m)}) - \nabla_{xy}g^{(m)}(x, y_x^{(m)})[\nabla_{y^2}g^{(m)}(x, y_x^{(m)})]^{-1}\nabla_y f^{(m)}(x, y_x^{(m)}) \\
&\quad - \left(\nabla_x f^{(j)}(x, y_x^{(j)}) - \nabla_{xy}g^{(j)}(x, y_x^{(j)})[\nabla_{y^2}g^{(j)}(x, y_x^{(j)})]^{-1}\nabla_y f^{(j)}(x, y_x^{(j)})\right)\| \\
&\le \|\nabla_x f^{(m)}(x, y_x^{(m)}) - \nabla_x f^{(j)}(x, y_x^{(j)})\| + \|\nabla_{xy}g^{(m)}(x, y_x^{(m)}) \\
&\quad - \nabla_{xy}g^{(j)}(x, y_x^{(j)})\|\|\left(\nabla_{yy}g^{(m)}(x, y_x^{(m)})\right)^{-1}\nabla_y f^{(m)}(x, y_x^{(m)})\| \\
&\quad + \|\nabla_{xy}g^{(j)}(x, y_x^{(j)})\|\|\left(\nabla_{yy}g^{(m)}(x, y_x^{(m)})\right)^{-1}\nabla_y f^{(m)}(x, y_x^{(m)}) \\
&\quad - \left(\nabla_{yy}g^{(j)}(x, y_x^{(j)})\right)^{-1}\nabla_y f^{(j)}(x, y_x^{(j)})\|
\end{aligned}
$$

Next we bound the three terms separately. For the first term:

$$
\begin{aligned}
\|\nabla_x f^{(m)}(x, y_x^{(m)}) - \nabla_x f^{(j)}(x, y_x^{(j)})\| &\le \|\nabla_x f^{(m)}(x, y_x^{(m)}) - \nabla_x f^{(j)}(x, y_x^{(m)})\| \\
&\quad + \|\nabla_x f^{(j)}(x, y_x^{(m)}) - \nabla_x f^{(j)}(x, y_x^{(j)})\| \\
&\le \zeta_f + L\|y_x^{(m)} - y_x^{(j)}\| \le \zeta_f + L\zeta_{g^*} \qquad (37)
\end{aligned}
$$

where the second inequality is due to Assumption 3.2 and Assumption D.1. The last inequality also follows the Assumption D.1. Next, for the second term, we have:

$$
\begin{aligned}
&\|\nabla_{xy}g^{(m)}(x, y_x^{(m)}) - \nabla_{xy}g^{(j)}(x, y_x^{(j)})\|\|\left(\nabla_{yy}g^{(m)}(x, y_x^{(m)})\right)^{-1}\nabla_y f^{(m)}(x, y_x^{(m)})\| \\
&\le \frac{C_f}{\mu}\|\nabla_{xy}g^{(m)}(x, y_x^{(m)}) - \nabla_{xy}g^{(j)}(x, y_x^{(j)})\| \\
&\le \frac{C_f}{\mu}\|\nabla_{xy}g^{(m)}(x, y_x^{(m)}) - \nabla_{xy}g^{(j)}(x, y_x^{(m)})\| + \frac{C_f}{\mu}\|\nabla_{xy}g^{(j)}(x, y_x^{(m)}) - \nabla_{xy}g^{(j)}(x, y_x^{(j)})\| \\
&\le \frac{C_f\zeta_{g,xy}}{\mu} + \frac{C_f L_{xy}}{\mu}\|y_x^{(m)} - y_x^{(j)})\| \le \frac{C_f\zeta_{g,xy}}{\mu} + \frac{C_f L_{xy}\zeta_{g^*}}{\mu}
\end{aligned}
$$

where the first inequality follows from the Assumption 3.1, 3.2; the third inequality follows from Assumption D.1, 3.3, the last inequality follows from Assumption D.1. Next, for the third term, we

have:

$$\left\|\nabla_{xy}g^{(j)}(x,y_x^{(j)})\right\|\left\|\left(\nabla_{yy}g^{(m)}(x,y_x^{(m)})\right)^{-1}\nabla_y f^{(m)}(x,y_x^{(m)})-\left(\nabla_{yy}g^{(j)}(x,y_x^{(j)})\right)^{-1}\nabla_y f^{(j)}(x,y_x^{(j)})\right\|$$

$$\leq L\left\|\left(\nabla_{yy}g^{(m)}(x,y_x^{(m)})\right)^{-1}\nabla_y f^{(m)}(x,y_x^{(m)})-\left(\nabla_{yy}g^{(j)}(x,y_x^{(j)})\right)^{-1}\nabla_y f^{(j)}(x,y_x^{(j)})\right\|$$

$$\leq L\left\|\left(\nabla_{yy}g^{(m)}(x,y_x^{(m)})\right)^{-1}\right\|\left\|\nabla_y f^{(m)}(x,y_x^{(m)})-\nabla_y f^{(j)}(x,y_x^{(j)})\right\|$$

$$\quad + L\left\|\left(\nabla_{yy}g^{(m)}(x,y_x^{(m)})\right)^{-1}-\left(\nabla_{yy}g^{(j)}(x,y_x^{(j)})\right)^{-1}\right\|\left\|\nabla_y f^{(j)}(x,y_x^{(j)})\right\|$$

$$\leq \frac{L}{\mu}\left\|\nabla_y f^{(m)}(x,y_x^{(m)})-\nabla_y f^{(j)}(x,y_x^{(j)})\right\|$$

$$\quad + C_f L\left\|\left(\nabla_{yy}g^{(m)}(x,y_x^{(m)})\right)^{-1}-\left(\nabla_{yy}g^{(j)}(x,y_x^{(j)})\right)^{-1}\right\|$$

$$\leq \frac{L(\zeta_f+L\zeta_{g^*})}{\mu}+C_f L\left\|\left(\nabla_{yy}g^{(m)}(x,y_x^{(m)})\right)^{-1}\right\|\times$$

$$\qquad \left\|\nabla_{yy}g^{(m)}(x,y_x^{(m)})-\nabla_{yy}g^{(j)}(x,y_x^{(j)})\right\|\left\|\left(\nabla_{yy}g^{(j)}(x,y_x^{(j)})\right)^{-1}\right\|$$

$$\leq \frac{L(\zeta_f+L\zeta_{g^*})}{\mu}+\frac{C_f L(\zeta_{g,yy}+L_{y^2}\zeta_{g^*})}{\mu^2}$$

where the first inequality is by Assumption 3.3; the third inequality is by Assumption 3.3, 3.2; the fourth inequality is by Cauchy Schwartz inequality; the last inequality is by Assumption 3.1, 3.3 and the result in Eq. 37. Combine everything together, we have:

$$\left\|\nabla_x f^{(m)}(x,y_x^{(m)})-\nabla_x f^{(j)}(x,y_x^{(j)})\right\|\leq \zeta_f+L\zeta_{g^*}+\frac{C_f\zeta_{g,xy}}{\mu}+\frac{C_f L_{xy}\zeta_{g^*}}{\mu}+\frac{L(\zeta_f+L\zeta_{g^*})}{\mu}$$

$$+\frac{C_f L(\zeta_{g,yy}+L_{y^2}\zeta_{g^*})}{\mu^2}$$

which completes the proof. $\qquad\square$

## D.1 Proof for the FedBiOAcc-Local Algorithm

### D.1.1 Hyper-Gradient Bias and Inner-Gradient Bias

**Lemma D.5.** *Suppose we have $c_\nu\alpha_t^2<1$, then:*

$$\mathbb{E}\left[\left\|\bar{\nu}_t-\mathbb{E}_\xi[\bar{\mu}_{t,\mathcal{B}_x}]\right\|^2\right]\leq (1-c_\nu\alpha_{t-1}^2)\mathbb{E}\left[\left\|\bar{\nu}_{t-1}-\mathbb{E}_\xi[\bar{\mu}_{t-1,\mathcal{B}_x}]\right\|^2\right]+\frac{2c_\nu^2\alpha_{t-1}^4}{b_x M}G_2^2$$

$$+\frac{2\hat{L}^2}{b_x M^2}\sum_{m=1}^M\mathbb{E}\left[\left\|x_t^{(m)}-x_{t-1}^{(m)}\right\|^2+\left\|y_t^{(m)}-y_{t-1}^{(m)}\right\|^2\right]$$

*where $\mu_{t,\xi}^{(m)}=\Phi^{(m)}(x_t^{(m)},y_t^{(m)};\mathcal{B}_x)$ and the expectation outside is* w.r.t *all the stochasity of the algorithm.*

*Proof.* For ease of notation, we denote $\mu_{t,\xi}^{(m)}=\Phi^{(m)}(x_t^{(m)},y_t^{(m)};\xi_x)$, and $\mu_t^{(m)}=\Phi^{(m)}(x_t^{(m)},y_t^{(m)})$, then by the definition of $\bar{\nu}_t$ we have:

$$\mathbb{E}\left[\left\|\bar{\nu}_t-\mathbb{E}_\xi[\bar{\mu}_{t,\mathcal{B}_x}]\right\|^2\right]=\mathbb{E}\left[\left\|\frac{1}{M}\sum_{m=1}^M\left(\hat{\nu}_t^{(m)}-\mathbb{E}_\xi[\mu_{t,\mathcal{B}_x}^{(m)}]\right)\right\|^2\right]$$

$$=\mathbb{E}\left[\left\|\frac{1}{M}\sum_{m=1}^M\left(\mu_{t,\mathcal{B}_x}^{(m)}+(1-c_\nu\alpha_{t-1}^2)(\nu_{t-1}^{(m)}-\mu_{t-1,\mathcal{B}_x}^{(m)})-\mathbb{E}_\xi[\mu_{t,\mathcal{B}_x}^{(m)}]\right)\right\|^2\right]$$

$$=\mathbb{E}\left[\left\|(1-c_\nu\alpha_{t-1}^2)\left(\bar{\nu}_{t-1}-\mathbb{E}_\xi[\bar{\mu}_{t-1,\mathcal{B}_x}]\right)+\left(\bar{\mu}_{t,\mathcal{B}_x}-\mathbb{E}_\xi[\bar{\mu}_{t,\mathcal{B}_x}]+(1-c_\nu\alpha_{t-1}^2)(\mathbb{E}_\xi[\bar{\mu}_{t-1,\mathcal{B}_x}]-\bar{\mu}_{t-1,\mathcal{B}_x})\right)\right\|^2\right]$$

$$\leq (1-c_\nu\alpha_{t-1}^2)\mathbb{E}\left[\left\|\bar{\nu}_{t-1}-\mathbb{E}_\xi[\bar{\mu}_{t-1,\mathcal{B}_x}]\right\|^2\right]$$

$$+\frac{1}{b_x^2 M^2}\sum_{m=1}^M\sum_{\xi_x\in\mathcal{B}_x}\mathbb{E}\left[\left\|\mu_{t,\xi_x}^{(m)}-\mathbb{E}_\xi[\mu_{t,\xi_x}^{(m)}]+(1-c_\nu\alpha_{t-1}^2)(\mathbb{E}_\xi[\mu_{t-1,\xi_x}^{(m)}]-\mu_{t-1,\xi_x}^{(m)})\right\|^2\right]$$

where inequality $(a)$ uses the fact that the cross product term is zero in expectation, the condition that $c_\nu \alpha_t^2 < 1$ and the fact that clients independently choose samples.

We denote the second term above as $T_1$, then we have:

$$T_1 \overset{(a)}{\leq} 2(c_\nu \alpha_{t-1}^2)^2 \mathbb{E}\big[\big\|\mu_{t,\xi_x}^{(m)} - \mathbb{E}_\xi[\mu_{t,\xi_x}^{(m)}]\big\|^2\big] + 2(1 - c_\nu \alpha_{t-1}^2)^2 \mathbb{E}\big[\big\|\mu_{t,\xi_x}^{(m)} - \mu_{t-1,\xi_x}^{(m)} - (\mathbb{E}_\xi[\mu_t^{(m)}] - \mathbb{E}_\xi[\mu_{t-1}^{(m)}])\big\|^2\big]$$

$$\overset{(b)}{\leq} 2(c_\nu \alpha_{t-1}^2)^2 \mathbb{E}\big[\big\|\mu_{t,\xi_x}^{(m)} - \mathbb{E}_\xi[\mu_t^{(m)}]\big\|^2\big] + 2\mathbb{E}\big[\big\|\mu_{t,\xi_x}^{(m)} - \mu_{t-1,\xi_x}^{(m)}\big\|^2\big]$$

$$\overset{(c)}{\leq} 2(c_\nu \alpha_{t-1}^2)^2 G_2^2 + 2\hat{L}^2 \mathbb{E}\big[\big\|x_t^{(m)} - x_{t-1}^{(m)}\big\|^2 + \big\|y_t^{(m)} - y_{t-1}^{(m)}\big\|^2\big]$$

where inequality (a) follows the generalized triangle inequality; (b) follows Proposition E.2 due to the definition of $\mu_t^{(m)}$; (c) follows the smoothness property of $\hat{L}$ and the bounded variance assumption; This completes the proof. $\qquad\square$

**Lemma D.6.** *Suppose we have $c_\omega \alpha_{t-1}^2 < 1$, then we have:*

$$\frac{1}{M} \sum_{m=1}^M \mathbb{E}\big[\big\|\omega_t^{(m)} - \nabla_y g^{(m)}(x_t^{(m)}, y_t^{(m)})\big\|^2\big]$$

$$\leq \frac{(1 - c_\omega \alpha_{t-1}^2)}{M} \sum_{m=1}^M \mathbb{E}\big[\big\|\omega_{t-1}^{(m)} - \nabla_y g^{(m)}(x_{t-1}^{(m)}, y_{t-1}^{(m)})\big\|^2\big] + \frac{2(c_\omega \alpha_{t-1}^2)^2 \sigma^2}{b_y}$$

$$+ \frac{2L^2}{b_y M} \sum_{m=1}^M \mathbb{E}\big[\big\|x_t^{(m)} - x_{t-1}^{(m)}\big\|^2 + \big\|y_t^{(m)} - y_{t-1}^{(m)}\big\|^2\big]$$

*where the expectation is w.r.t the stochasticity of the algorithm.*

The proof of Lemma D.6 can be derived similar as Lemma D.5

### D.1.2 Lower Problem Solution Error

**Lemma D.7.** *Suppose we choose $\gamma \leq \frac{1}{2L}$ and $\alpha_t < 1$. Then for $t \neq \bar{t}_s$, we have:*

$$\mathbb{E}\big[\big\|y_t^{(m)} - y_{x_t^{(m)}}^{(m)}\big\|^2\big] \leq (1 - \frac{\mu\gamma\alpha_{t-1}}{4})\mathbb{E}\big[\big\|y_{t-1}^{(m)} - y_{x_{t-1}^{(m)}}^{(m)}\big\|^2\big] - \frac{\gamma^2\alpha_{t-1}}{4}\mathbb{E}\big[\big\|\omega_{t-1}^{(m)}\big\|^2\big]$$

$$+ \frac{9\gamma\alpha_{t-1}}{2\mu}\mathbb{E}\big[\big\|\omega_{t-1}^{(m)} - \nabla_y g^{(m)}(x_{t-1}^{(m)}, y_{t-1}^{(m)})\big\|^2\big] + \frac{9\kappa^2\eta^2\alpha_{t-1}}{2\mu\gamma}\mathbb{E}\big[\big\|\nu_{t-1}^{(m)}\big\|^2\big]$$

*for $t = \bar{t}_s$, we have:*

$$\mathbb{E}\big[\big\|y_t^{(m)} - y_{x_t^{(m)}}^{(m)}\big\|^2\big] \leq (1 - \frac{\mu\gamma\alpha_{t-1}}{4})\mathbb{E}\big[\big\|y_{t-1}^{(m)} - y_{x_{t-1}^{(m)}}^{(m)}\big\|^2\big] - \frac{\gamma^2\alpha_{t-1}}{4}\mathbb{E}\big[\big\|\omega_{t-1}^{(m)}\big\|^2\big]$$

$$+ \frac{9\gamma\alpha_{t-1}}{2\mu}\mathbb{E}\big[\big\|\omega_{t-1}^{(m)} - \nabla_y g^{(m)}(x_{t-1}^{(m)}, y_{t-1}^{(m)})\big\|^2\big]$$

$$+ \frac{9\kappa^2\eta^2\alpha_{t-1}}{\mu\gamma}\mathbb{E}\big[\big\|\nu_{t-1}^{(m)}\big\|^2\big] + \frac{9\kappa^2}{\mu\gamma\alpha_{t-1}}\mathbb{E}\big[\big\|\hat{x}_t^{(m)}) - \bar{x}_t\big\|^2\big]$$

*Proof.* First, we exploit Proposition E.5, and choose the function $g^{(m)}(x_t^{(m)}, \cdot)$, by assumption it is $L$ smooth and $\mu$ strongly convex, and we choose $\gamma < \frac{1}{2L}$ and $\alpha_t < 1$, thus:

$$\|y_{t+1}^{(m)} - y_{x_t^{(m)}}^{(m)}\|^2 \leq (1 - \frac{\mu\gamma\alpha_t}{2})\|y_{x_t^{(m)}}^{(m)} - y_t^{(m)}\|^2 - \frac{\gamma^2\alpha_t}{4}\|\omega_t^{(m)}\|^2 + \frac{4\gamma\alpha_t}{\mu}\|\nabla_y g(x_t^{(m)}, y_t^{(m)}) - w_t^{(m)}\|^2. \tag{38}$$

Next, we decompose the term $\|y_{t+1}^{(m)} - y_{x_{t+1}^{(m)}}^{(m)}\|^2$ as follows:

$$\|y_{t+1}^{(m)} - y_{x_{t+1}^{(m)}}^{(m)}\|^2 \leq (1 + \frac{\mu\gamma\alpha_t}{4})\|y_{t+1}^{(m)} - y_{x_t^{(m)}}^{(m)}\|^2 + (1 + \frac{4}{\mu\gamma\alpha_t})\|y_{x_t^{(m)}}^{(m)} - y_{x_{t+1}^{(m)}}^{(m)})\|^2$$

$$\leq (1 + \frac{\mu\gamma\alpha_t}{4})\|y_{t+1}^{(m)} - y_{x_t^{(m)}}^{(m)}\|^2 + (1 + \frac{4}{\mu\gamma\alpha_t})\kappa^2\|x_t^{(m)} - x_{t+1}^{(m)}\|^2 \tag{39}$$

where the first inequality holds by the generalized triangle inequality, and the second inequality is due to case a) of Proposition 3.9. Combining the above inequalities 38 and 39, we have

$$\|y_{t+1}^{(m)} - y_{\hat{x}_{t+1}^{(m)}}^{(m)})\|^2 \leq (1 + \frac{\mu\gamma\alpha_t}{4})(1 - \frac{\mu\gamma\alpha_t}{2})\|y_t^{(m)} - y_{x_t^{(m)}}^{(m)})\|^2 - (1 + \frac{\mu\gamma\alpha_t}{4})\frac{\gamma^2\alpha_t}{4}\|\omega_t^{(m)}\|^2$$

$$+ (1 + \frac{\mu\gamma\alpha_t}{4})\frac{4\gamma\alpha_t}{\mu}\|\nabla_y g(x_t^{(m)}, y_t^{(m)}) - w_t^{(m)}\|^2 + (1 + \frac{4}{\mu\gamma\alpha_t})\kappa^2\|x_t^{(m)} - x_{t+1}^{(m)}\|^2$$

Since we choose $\gamma \leq \frac{1}{2L}, \alpha_t < 1$, we have:

$$(1 + \frac{\mu\gamma\alpha_t}{4})(1 - \frac{\mu\gamma\alpha_t}{2}) = 1 - \frac{\mu\gamma\alpha_t}{4} - \frac{\mu^2\gamma^2\alpha_t^2}{8} \leq 1 - \frac{\mu\gamma\alpha_t}{4}$$

and $-(1 + \frac{\mu\gamma\alpha_t}{4}) \leq -1, (1 + \frac{\mu\gamma\alpha_t}{4}) \leq \frac{9}{8}, \mu\gamma\alpha_t < \frac{1}{2}$. Thus, we have

$$\|y_{t+1}^{(m)} - y_{\hat{x}_{t+1}^{(m)}}^{(m)})\|^2 \leq (1 - \frac{\mu\gamma\alpha_t}{4})\|y_t^{(m)} - y_{x_t^{(m)}}^{(m)})\|^2 - \frac{\gamma^2\alpha_t}{4}\|\omega_t^{(m)}\|^2$$

$$+ \frac{9\gamma\alpha_t}{2\mu}\|\nabla_y g(x_t^{(m)}, y_t^{(m)}) - w_t\|^2 + \frac{9\kappa^2}{2\mu\gamma\alpha_t}\underbrace{\|x_t^{(m)} - x_{t+1}^{(m)}\|^2}_{T_1}$$

Note for the term $T_1$ we have $T_1 = \|\eta\alpha_t\nu_t^{(m)}\|^2$ for $t + 1 \neq \bar{t}_s$ and $T_1 = \|\bar{x}_{t+1} - x_t^{(m)}\|^2 \leq 2\|\hat{x}_{t+1}^{(m)} - \bar{x}_{t+1}\|^2 + 2\|\eta\alpha_t\nu_t^{(m)}\|^2$ for $t + 1 = \bar{t}_s$. This completes the proof. $\qquad\square$

### D.1.3  Upper Variable Drift

**Lemma D.8.** *For $t \in [\bar{t}_{s-1} + 1, \bar{t}_s]$, with $s \in [S]$ we have:*

$$\|\hat{x}_t^{(m)} - \bar{x}_t\|^2 \leq \sum_{\ell=\bar{t}_{s-1}}^{t-1} I\eta^2\alpha_l^2\|(\nu_\ell^{(m)} - \bar{\nu}_\ell)\|^2$$

*Proof.* Since we have $\hat{x}_t^{(m)} = x_{t-1}^{(m)} - \eta\alpha_{t-1}\nu_{t-1}^{(m)}$, this implies that:

$$\hat{x}_t^{(m)} = x_{\bar{t}_{s-1}}^{(m)} - \sum_{\ell=\bar{t}_{s-1}}^{t-1} \eta\alpha_\ell\nu_\ell^{(m)} \quad \text{and} \quad \bar{x}_t = \bar{x}_{\bar{t}_{s-1}} - \sum_{\ell=\bar{t}_{s-1}}^{t-1} \eta\alpha_\ell\bar{\nu}_\ell.$$

So for $t \in [\bar{t}_{s-1} + 1, \bar{t}_s]$, with $s \in [S]$ we have:

$$\|\hat{x}_t^{(m)} - \bar{x}_t\|^2 = \|x_{\bar{t}_{s-1}}^{(m)} - \bar{x}_{\bar{t}_{s-1}} - \Big(\sum_{\ell=\bar{t}_{s-1}}^{t-1} \eta\alpha_\ell\nu_\ell^{(m)} - \sum_{\ell=\bar{t}_{s-1}}^{t-1} \eta\alpha_\ell\bar{\nu}_\ell\Big)\|^2$$

$$\overset{(a)}{=} \|\sum_{\ell=\bar{t}_{s-1}}^{t-1} \eta\alpha_\ell(\nu_\ell^{(m)} - \bar{\nu}_\ell)\|^2 \overset{(b)}{\leq} \sum_{\ell=\bar{t}_{s-1}}^{t-1} I\eta^2\alpha_l^2\|(\nu_\ell^{(m)} - \bar{\nu}_\ell)\|^2$$

where the equality $(a)$ follows from the fact that $x_{\bar{t}_{s-1}}^{(m)} = \bar{x}_{\bar{t}_{s-1}}$; inequality (b) is due to $t - \bar{t}_{s-1} \leq I$ and the generalized triangle inequality. $\qquad\square$

**Lemma D.9.** *Suppose $\alpha_t < \frac{1}{16I\hat{L}}, \eta < 1$, then for $t \neq \bar{t}_s, s \in [S]$, we have:*

$$\sum_{m=1}^{M} \mathbb{E}\|\nu_t^{(m)} - \bar{\nu}_t\|^2 \leq (1 + \frac{33}{32I}) \sum_{m=1}^{M} \mathbb{E}\|\nu_{t-1}^{(m)} - \bar{\nu}_{t-1}\|^2 + 4I\hat{L}^2\alpha_{t-1}^2 \sum_{m=1}^{M} \mathbb{E}[2\|\eta\bar{\nu}_{t-1}\|^2 + \|\gamma\omega_{t-1}^{(m)}\|^2]$$

$$+ 8IM(c_\nu\alpha_{t-1}^2)^2G_1^2 + \frac{8IM(c_\nu\alpha_{t-1}^2)^2G_2^2}{b_x} + 16IM(c_\nu\alpha_{t-1}^2)^2\zeta^2$$

$$+ 128I\hat{L}^2(c_\nu\alpha_{t-1}^2)^2 \sum_{m=1}^{M} \mathbb{E}[\|y_{t-1}^{(m)} - y_{x_{t-1}^{(m)}}^{(m)}\|^2]$$

$$+ 128I\bar{L}^2(c_\nu\alpha_{t-1}^2)^2 \sum_{m=1}^{M} \sum_{\ell=\bar{t}_{s-1}}^{t-2} I\eta^2\alpha_l^2\mathbb{E}\|(\nu_\ell^{(m)} - \bar{\nu}_\ell)\|^2$$

*where the expectation is w.r.t the stochasticity of the algorithm.*

*Proof.* By the update step in Line 7 of Algorithm 1, for $t \neq \bar{t}_s$, we have:

$$\mathbb{E}\|\hat{\nu}_t^{(m)} - \bar{\nu}_t\|^2 = \mathbb{E}\big\|\mu_{t,\mathcal{B}_x}^{(m)} + (1 - c_\nu \alpha_{t-1}^2)\big(\nu_{t-1}^{(m)} - \mu_{t-1,\mathcal{B}_x}^{(m)}\big) - \big(\bar{\mu}_{t,\mathcal{B}_x} + (1 - c_\nu \alpha_{t-1}^2)\big(\bar{\nu}_{t-1} - \bar{\mu}_{t-1,\mathcal{B}_x}\big)\big)\big\|^2$$

$$= \mathbb{E}\big\|(1 - c_\nu \alpha_{t-1}^2)\big(\nu_{t-1}^{(m)} - \bar{\nu}_{t-1}\big) + \mu_{t,\mathcal{B}_x}^{(m)} - \bar{\mu}_{t,\mathcal{B}_x} - (1 - c_\nu \alpha_{t-1}^2)\big(\mu_{t-1,\mathcal{B}_x}^{(m)} - \bar{\mu}_{t-1,\mathcal{B}_x}\big)\big\|^2$$

$$\overset{(a)}{\leq} (1 + \frac{1}{I})(1 - c_\nu \alpha_{t-1}^2)^2 \mathbb{E}\|\nu_{t-1}^{(m)} - \bar{\nu}_{t-1}\|^2$$

$$+ (1 + I)\mathbb{E}\big\|\mu_{t,\mathcal{B}_x}^{(m)} - \bar{\mu}_{t,\mathcal{B}_x} - (1 - c_\nu \alpha_{t-1}^2)\big(\mu_{t-1,\mathcal{B}_x}^{(m)} - \bar{\mu}_{t-1,\mathcal{B}_x}\big)\big\|^2$$

$$\leq \left(1 + \frac{1}{I}\right)\mathbb{E}\|\nu_{t-1}^{(m)} - \bar{\nu}_{t-1}\|^2 + (1 + I)\mathbb{E}\big\|\mu_{t,\mathcal{B}_x}^{(m)} - \bar{\mu}_{t,\mathcal{B}_x} - (1 - c_\nu \alpha_{t-1}^2)\big(\mu_{t-1,\mathcal{B}_x}^{(m)} - \bar{\mu}_{t-1,\mathcal{B}_x}\big)\big\|^2$$

$$\tag{40}$$

where $(a)$ follows from the the generalized triangle inequality.

Next we bound the second term of the above inequality (denoted as $T_1$):

$$\sum_{m=1}^{M} \mathbb{E}\big\|\mu_{t,\mathcal{B}_x}^{(m)} - \bar{\mu}_{t,\mathcal{B}_x} - (1 - c_\nu \alpha_{t-1}^2)\big(\mu_{t-1,\mathcal{B}_x}^{(m)} - \bar{\mu}_{t-1,\mathcal{B}_x}\big)\big\|^2$$

$$\leq 2\sum_{m=1}^{M} \mathbb{E}\big\|\mu_{t,\mathcal{B}_x}^{(m)} - \bar{\mu}_{t,\mathcal{B}_x} - \big(\mu_{t-1,\mathcal{B}_x}^{(m)} - \bar{\mu}_{t-1,\mathcal{B}_x}\big)\big\|^2 + 2(c_\nu \alpha_{t-1}^2)^2 \sum_{m=1}^{M} \mathbb{E}\big\|\mu_{t-1,\mathcal{B}_x}^{(m)} - \bar{\mu}_{t-1,\mathcal{B}_x}\big\|^2$$

$$\leq 2\sum_{m=1}^{M} \mathbb{E}\big\|\mu_{t,\mathcal{B}_x}^{(m)} - \mu_{t-1,\mathcal{B}_x}^{(m)}\big\|^2 + 2(c_\nu \alpha_{t-1}^2)^2 \sum_{m=1}^{M} \mathbb{E}\big\|\mu_{t-1,\mathcal{B}_x}^{(m)} - \bar{\mu}_{t-1,\mathcal{B}_x}\big\|^2$$

where the second inequality follows Proposition E.2. We bound the two terms separately, for the first term, we have:

$$\sum_{m=1}^{M} \mathbb{E}\big\|\mu_{t,\mathcal{B}_x}^{(m)} - \mu_{t-1,\mathcal{B}_x}^{(m)}\big\|^2 \leq \hat{L}^2 \sum_{m=1}^{M} \mathbb{E}\big[\|x_t^{(m)} - x_{t-1}^{(m)}\|^2 + \|y_t^{(m)} - y_{t-1}^{(m)}\|^2\big]$$

$$\leq \hat{L}^2 \alpha_{t-1}^2 \sum_{m=1}^{M} \mathbb{E}\big[\|\eta \nu_{t-1}^{(m)}\|^2 + \|\gamma \omega_{t-1}^{(m)}\|^2\big] \tag{41}$$

where the inequalities follow Proposition D.3.b) and the fact that $\hat{x}_t^{(m)} = x_t^{(m)}$ when $t \neq \bar{t}_s$;

Next for the second term, we have:

$$\sum_{m=1}^{M} \mathbb{E}\big\|\mu_{t-1,\mathcal{B}_x}^{(m)} - \bar{\mu}_{t-1,\mathcal{B}_x}\big\|^2 = \sum_{m=1}^{M} \mathbb{E}\big\|\mu_{t-1,\mathcal{B}_x}^{(m)} - \mu_{t-1}^{(m)} - \big(\bar{\mu}_{t-1,\mathcal{B}_x} - \bar{\mu}_{t-1}\big) + \mu_{t-1}^{(m)} - \bar{\mu}_{t-1}\big\|^2$$

$$\overset{(a)}{\leq} 2\sum_{m=1}^{M} \mathbb{E}\big\|\mu_{t-1,\mathcal{B}_x}^{(m)} - \mu_{t-1}^{(m)} - \big(\bar{\mu}_{t-1,\mathcal{B}_x} - \bar{\mu}_{t-1}\big)\big\|^2 + 2\sum_{m=1}^{M} \mathbb{E}\big\|\mu_{t-1}^{(m)} - \bar{\mu}_{t-1}\big\|^2$$

$$\overset{(b)}{\leq} 2\underbrace{\sum_{m=1}^{M} \mathbb{E}\big\|\mu_{t-1,\mathcal{B}_x}^{(m)} - \mu_{t-1}^{(m)}\big\|^2}_{T_1} + 4\underbrace{\sum_{m=1}^{M} \mathbb{E}\big\|\nabla h^{(m)}(\bar{x}_{t-1}) - \nabla h(\bar{x}_{t-1})\big\|^2}_{T_2}$$

$$+ 4\underbrace{\sum_{m=1}^{M} \mathbb{E}\big\|\mu_{t-1}^{(m)} - \nabla h^{(m)}(\bar{x}_{t-1}) + \nabla h(\bar{x}_{t-1}) - \bar{\mu}_{t-1}\big\|^2}_{T_3} \tag{42}$$

Note for the term $T_1$ of Eq. 42, we have $\mathbb{E}\big\|\mu_{t-1,\mathcal{B}_x}^{(m)} - \mu_{t-1}^{(m)}\big\|^2 \leq G_1^2 + \frac{G_2^2}{b_x}$; For the term $T_2$ of Eq. 42, by the bounded intra-node heterogeneity assumption we have:

$$T_2 \leq 4\sum_{m=1}^{M} \frac{1}{M} \sum_{j=1}^{M} \mathbb{E}\|\nabla h^{(m)}(\bar{x}_{t-1}) - \nabla h^{(j)}(\bar{x}_{t-1})\|^2 \leq 4M\zeta^2$$

Finally, For the term $T_3$ of Eq. 42

$$T_3 \leq 8 \sum_{m=1}^{M} \mathbb{E} \big\| \mu_{t-1}^{(m)} - \nabla h^{(m)}(\bar{x}_{t-1}) \big\|^2 + 8 \sum_{m=1}^{M} \mathbb{E} \big\| \nabla h(\bar{x}_{t-1}) - \bar{\mu}_{t-1} \big\|^2 \leq 16 \sum_{m=1}^{M} \mathbb{E} \big\| \mu_{t-1}^{(m)} - \nabla h^{(m)}(\bar{x}_{t-1}) \big\|^2$$

$$\leq 32 \sum_{m=1}^{M} \mathbb{E} \big[ \big\| \mu_{t-1}^{(m)} - \nabla h^{(m)}(x_{t-1}^{(m)}) \big\|^2 + \big\| \nabla h^{(m)}(x_{t-1}^{(m)}) - \nabla h^{(m)}(\bar{x}_{t-1}) \big\|^2 \big]$$

$$\overset{(a)}{\leq} 32 \bar{L}^2 \sum_{m=1}^{M} \mathbb{E} \big[ \| x_{t-1}^{(m)} - \bar{x}_{t-1} \|^2 \big] + 32 \hat{L}^2 \sum_{m=1}^{M} \mathbb{E} \big[ \| y_{t-1}^{(m)} - y_{x_{t-1}^{(m)}}^{(m)} \|^2 \big]$$

where inequality (b) follows Proposition D.3.c) and d).

Combine Eq. 40, Eq. 41 and Eq. 42, use the fact that $I \geq 1$, we have:

$$\sum_{m=1}^{M} \mathbb{E} \| \nu_t^{(m)} - \bar{\nu}_t \|^2$$

$$\leq \Big(1 + \frac{1}{I}\Big) \sum_{m=1}^{M} \mathbb{E} \| \nu_{t-1}^{(m)} - \bar{\nu}_{t-1} \|^2 + 4 I \hat{L}^2 \alpha_{t-1}^2 \sum_{m=1}^{M} \mathbb{E} \big[ \underbrace{\| \eta \nu_{t-1}^{(m)} \|^2}_{T_1} + \| \gamma \omega_{t-1}^{(m)} \|^2 \big]$$

$$+ 8 I M (c_\nu \alpha_{t-1}^2)^2 G_1^2 + \frac{8 I M (c_\nu \alpha_{t-1}^2)^2 G_2^2}{b_x} + 16 I M (c_\nu \alpha_{t-1}^2)^2 \zeta^2$$

$$+ 128 I \bar{L}^2 (c_\nu \alpha_{t-1}^2)^2 \sum_{m=1}^{M} \mathbb{E} \big[ \| x_{t-1}^{(m)} - \bar{x}_{t-1} \|^2 \big] + 128 I \hat{L}^2 (c_\nu \alpha_{t-1}^2)^2 \sum_{m=1}^{M} \mathbb{E} \big[ \| y_{t-1}^{(m)} - y_{x_{t-1}^{(m)}}^{(m)} \|^2 \big]$$

We separate the term $T_1$ with triangle inequality to get:

$$\sum_{m=1}^{M} \mathbb{E} \| \nu_t^{(m)} - \bar{\nu}_t \|^2$$

$$\leq \Big(1 + \frac{1}{I} + 8 I \hat{L}^2 \eta^2 \alpha_{t-1}^2 \Big) \sum_{m=1}^{M} \mathbb{E} \| \nu_{t-1}^{(m)} - \bar{\nu}_{t-1} \|^2 + 4 I \hat{L}^2 \alpha_{t-1}^2 \sum_{m=1}^{M} \mathbb{E} \big[ 2 \| \eta \bar{\nu}_{t-1} \|^2 + \| \gamma \omega_{t-1}^{(m)} \|^2 \big]$$

$$+ 8 I M (c_\nu \alpha_{t-1}^2)^2 G_1^2 + \frac{8 I M (c_\nu \alpha_{t-1}^2)^2 G_2^2}{b_x} + 16 I M (c_\nu \alpha_{t-1}^2)^2 \zeta^2$$

$$+ \underbrace{128 I \bar{L}^2 (c_\nu \alpha_{t-1}^2)^2 \sum_{m=1}^{M} \mathbb{E} \big[ \| x_{t-1}^{(m)} - \bar{x}_{t-1} \|^2 \big]}_{T_1} + 128 I \hat{L}^2 (c_\nu \alpha_{t-1}^2)^2 \sum_{m=1}^{M} \mathbb{E} \big[ \| y_{t-1}^{(m)} - y_{x_{t-1}^{(m)}}^{(m)} \|^2 \big]$$

Finally, choose $\eta \alpha_t < \frac{1}{16 \hat{L} I}$ and combine with Lemma D.8 to bound the term $T_1$, we get the bound in the lemma. This completes the proof. $\qquad \square$

Next, to simply the notation, we denote $A_t = \mathbb{E} \| \bar{\nu}_t - \mathbb{E}_\xi [ \bar{\mu}_{t, \mathcal{B}_x} ] \|^2$, $B_t = \frac{1}{M} \sum_{m=1}^{M} \mathbb{E} \| y_t^{(m)} - y_{x_t^{(m)}}^{(m)} \|^2$, $C_t = \frac{1}{M} \sum_{m=1}^{M} \mathbb{E} \| \omega_t^{(m)} - \nabla_y g^{(m)}(x_t^{(m)}, y_t^{(m)}) \|^2$, $D_t = \frac{1}{M} \sum_{m=1}^{M} \mathbb{E} \| \nu_t^{(m)} - \bar{\nu}_t \|^2$, $E_t = \mathbb{E} \| \bar{\nu}_t \|^2$, $F_t = \frac{1}{M} \sum_{m=1}^{M} \mathbb{E} \| \omega_t^{(m)} \|^2$.

**Lemma D.10.** *For $\alpha_t < \frac{1}{16 \hat{L} I}$, we have:*

$$\Big(1 - \frac{3 \kappa^2 \eta^2 c_\nu^2}{4 * 16^3 I^5 \hat{L}^4}\Big) \sum_{t=\bar{t}_{s-1}}^{\bar{t}_s - 1} \alpha_t D_t \leq \frac{3 c_\nu^2}{2 * 16^2 I^4 \hat{L}^2} \sum_{\ell=\bar{t}_{s-1}}^{\bar{t}_s - 1} \alpha_\ell B_\ell + \frac{3 \eta^2}{32 I} \sum_{\ell=\bar{t}_{s-1}}^{\bar{t}_s - 1} \alpha_\ell E_\ell + \frac{3 \gamma^2}{64 I} \sum_{\ell=\bar{t}_{s-1}}^{\bar{t}_s - 1} \alpha_\ell F_\ell$$

$$+ \Big( \frac{3 c_\nu^2 G_1^2}{32 I \hat{L}^2} + \frac{3 c_\nu^2 G_2^2}{32 I b_x \hat{L}^2} + \frac{3 c_\nu^2 \zeta^2}{16 I \hat{L}^2} \Big) \sum_{\ell=\bar{t}_{s-1}}^{\bar{t}_s - 1} \alpha_\ell^3$$

*where the terms $D_t$, $E_t$ and $F_t$ are denoted above.*

*Proof.* Based on Lemma D.9, for $t \neq \bar{t}_s$, we have:

$$D_t \leq \left(1 + \frac{33}{32I}\right)D_{t-1} + 128I\hat{L}^2 c_\nu^2 \alpha_{t-1}^4 B_{t-1} + 8I\hat{L}^2 \alpha_{t-1}^2 \eta^2 E_{t-1} + 4I\hat{L}^2 \alpha_{t-1}^2 \gamma^2 F_{t-1}$$

$$+ 8Ic_\nu^2 \alpha_{t-1}^4 G_1^2 + \frac{8Ic_\nu^2 \alpha_{t-1}^4 G_2^2}{b_x} + 16Ic_\nu^2 \alpha_{t-1}^4 \zeta^2 + 128I^2 \bar{L}^2 \eta^2 c_\nu^2 \alpha_{t-1}^4 \sum_{\ell=\bar{t}_{s-1}}^{t-2} \alpha_l^2 D_l$$

while for $t = \bar{t}_s$, we have $D_{\bar{t}_s} = 1/M \sum_{m=1}^{M} \mathbb{E}\|\nu_{\bar{t}_s}^{(m)} - \bar{\nu}_{\bar{t}_s}\|^2 = 0$. Apply the above equation recursively from $\bar{t}_{s-1} + 1$ to $t$. so we have:

$$D_t \leq \sum_{\ell=\bar{t}_{s-1}}^{t-1} \left(1 + \frac{33}{32I}\right)^{t-\ell} \left(128I\hat{L}^2 c_\nu^2 \alpha_\ell^4 B_\ell + 8I\hat{L}^2 \eta^2 \alpha_\ell^2 E_\ell + 4I\hat{L}^2 \gamma^2 \alpha_\ell^2 F_\ell + 8Ic_\nu^2 G_1^2 \alpha_\ell^4 + \frac{8Ic_\nu^2 G_2^2 \alpha_\ell^4}{b_x}\right.$$

$$\left. + 16Ic_\nu^2 \zeta^2 \alpha_\ell^4 + 128I^2 \bar{L}^2 \eta^2 c_\nu^2 \alpha_\ell^4 \sum_{\bar{\ell}=\bar{t}_{s-1}}^{\ell} \alpha_{\bar{\ell}}^2 D_{\bar{\ell}}\right)$$

$$\leq \sum_{\ell=\bar{t}_{s-1}}^{t-1} \left(384I\hat{L}^2 c_\nu^2 \alpha_\ell^4 B_\ell + 24I\hat{L}^2 \eta^2 \alpha_\ell^2 E_\ell + 12I\hat{L}^2 \gamma^2 \alpha_\ell^2 F_\ell + 24Ic_\nu^2 G_1^2 \alpha_\ell^4 + \frac{24Ic_\nu^2 G_2^2 \alpha_\ell^4}{b_x}\right.$$

$$\left. + 16Ic_\nu^2 \zeta^2 \alpha_\ell^4 + 384I^2 \bar{L}^2 \eta^2 c_\nu^2 \alpha_\ell^4 \sum_{\bar{\ell}=\bar{t}_{s-1}}^{\ell} \alpha_{\bar{\ell}}^2 D_{\bar{\ell}}\right)$$

The second inequality uses the fact that $t - l \leq I$ and the inequality $log(1 + a/x) \leq a/x$ for $x > -a$, so we have $(1 + a/x)^x \leq e^a$, Then we choose $a = 33/32$ and $x = I$. Finally, we use the fact that $e^{33/32} \leq 3$.

Next we multiply $\alpha_t$ over both sides and take sum from $\bar{t}_{s-1} + 1$ to $\bar{t}_s$, we have:

$$\sum_{t=\bar{t}_{s-1}+1}^{\bar{t}_s} \alpha_t D_t \leq \sum_{t=\bar{t}_{s-1}}^{\bar{t}_s-1} \alpha_t \sum_{\ell=\bar{t}_{s-1}}^{t-1} \left(384I\hat{L}^2 c_\nu^2 \alpha_\ell^4 B_\ell + 24I\hat{L}^2 \eta^2 \alpha_\ell^2 E_\ell + 12I\hat{L}^2 \gamma^2 \alpha_\ell^2 F_\ell\right.$$

$$\left. + \left(24Ic_\nu^2 G_1^2 + \frac{24Ic_\nu^2 G_1^2}{b_x} + 48Ic_\nu^2 \zeta^2\right)\alpha_\ell^4 + 384I^2 \bar{L}^2 \eta^2 c_\nu^2 \alpha_\ell^4 \sum_{\bar{\ell}=\bar{t}_{s-1}}^{\ell} \alpha_{\bar{\ell}}^2 D_{\bar{\ell}}\right)$$

$$\overset{(a)}{\leq} \sum_{\ell=\bar{t}_{s-1}}^{\bar{t}_s-1} \left(24I^{1/2}\hat{L} c_\nu^2 \alpha_\ell^4 B_\ell + \frac{3I^{1/2}\hat{L}\eta^2}{2}\alpha_\ell^2 E_\ell + \frac{3I^{1/2}\hat{L}\gamma^2}{4}\alpha_\ell^2 F_\ell\right.$$

$$\left. + \left(\frac{3I^{1/2}c_\nu^2 G_1^2}{2\hat{L}} + \frac{3I^{1/2}c_\nu^2 G_2^2}{2b_x\hat{L}} + \frac{3I^{1/2}c_\nu^2 \zeta^2}{\hat{L}}\right)\alpha_\ell^4 + \frac{24I^{3/2}\bar{L}^2 \eta^2 c_\nu^2}{\hat{L}}\alpha_\ell^4 \sum_{\bar{\ell}=\bar{t}_{s-1}}^{\ell} \alpha_{\bar{\ell}}^2 D_{\bar{\ell}}\right)$$

$$\overset{(b)}{\leq} \frac{3c_\nu^2}{2 * 16^2 I^4 \hat{L}^2} \sum_{\ell=\bar{t}_{s-1}}^{\bar{t}_s-1} \alpha_\ell B_\ell + \frac{3\eta^2}{32I} \sum_{\ell=\bar{t}_{s-1}}^{\bar{t}_s-1} \alpha_\ell E_\ell + \frac{3\gamma^2}{64I} \sum_{\ell=\bar{t}_{s-1}}^{\bar{t}_s-1} \alpha_\ell F_\ell$$

$$+ \left(\frac{3c_\nu^2 G_1^2}{32I\hat{L}^2} + \frac{3c_\nu^2 G_2^2}{32Ib_x\hat{L}^2} + \frac{3c_\nu^2 \zeta^2}{16I\hat{L}^2}\right) \sum_{\ell=\bar{t}_{s-1}}^{\bar{t}_s-1} \alpha_\ell^3 + \frac{3\kappa^2 \eta^2 c_\nu^2}{4 * 16^3 I^5 \hat{L}^4} \sum_{t=\bar{t}_{s-1}}^{\bar{t}_s-1} \alpha_t D_t$$

In inequalities $(a)$ and $(b)$, we use $\alpha_t < \frac{1}{16\hat{L}I^{3/2}}$. Note that $\sum_{t=\bar{t}_{s-1}+1}^{\bar{t}_s} \alpha_t D_t = \sum_{t=\bar{t}_{s-1}}^{\bar{t}_s-1} \alpha_t D_t$ as $D_{\bar{t}_s} = D_{\bar{t}_{s-1}} = 0$, so we have:

$$\left(1 - \frac{3\kappa^2 \eta^2 c_\nu^2}{4 * 16^3 I^5 \hat{L}^4}\right) \sum_{t=\bar{t}_{s-1}}^{\bar{t}_s-1} \alpha_t D_t \leq \frac{3c_\nu^2}{2 * 16^2 I^4 \hat{L}^2} \sum_{\ell=\bar{t}_{s-1}}^{\bar{t}_s-1} \alpha_\ell B_\ell + \frac{3\eta^2}{32I} \sum_{\ell=\bar{t}_{s-1}}^{\bar{t}_s-1} \alpha_\ell E_\ell + \frac{3\gamma^2}{64I} \sum_{\ell=\bar{t}_{s-1}}^{\bar{t}_s-1} \alpha_\ell F_\ell$$

$$+ \left(\frac{3c_\nu^2 G_1^2}{32I\hat{L}^2} + \frac{3c_\nu^2 G_2^2}{32Ib_x\hat{L}^2} + \frac{3c_\nu^2 \zeta^2}{16I\hat{L}^2}\right) \sum_{\ell=\bar{t}_{s-1}}^{\bar{t}_s-1} \alpha_\ell^3$$

This completes the proof. $\qquad\square$

### D.1.4 Descent Lemma

**Lemma D.11.** *Suppose $\eta < \frac{1}{2\bar{L}}$, $\alpha_t < 1$, for all $t \in [\bar{t}_{s-1}, \bar{t}_s - 1]$ and $s \in [S]$, the iterates generated satisfy:*

$$\mathbb{E}[h(\bar{x}_{t+1})] \leq \mathbb{E}[h(\bar{x}_t)] - \frac{\eta\alpha_t}{4}\mathbb{E}[\|\bar{\nu}_t\|^2] - \frac{\eta\alpha_t}{2}\mathbb{E}[\|\nabla h(\bar{x}_t)\|^2] + 2\eta\alpha_t\mathbb{E}[\|\mathbb{E}_\xi[\bar{\mu}_{t,\mathcal{B}_x}] - \bar{\nu}_t\|^2] + 4\eta\alpha_t G_1^2$$

$$+ \frac{\bar{L}^2 I\eta^3\alpha_t}{M}\sum_{\ell=\bar{t}_{s-1}}^{t-1}\alpha_l^2\sum_{m=1}^{M}\mathbb{E}\|(\nu_\ell^{(m)} - \bar{\nu}_\ell)\|^2 + \frac{4\hat{L}^2\eta\alpha_t}{M}\sum_{m=1}^{M}\mathbb{E}\|y_{x_t^{(m)}}^{(m)} - y_t^{(m)}\|^2$$

*where the expectation is w.r.t the stochasticity of the algorithm.*

*Proof.* By the smoothness of $h(x)$ we have:

$$\mathbb{E}[h(\bar{x}_{t+1})] \leq \mathbb{E}\left[h(\bar{x}_t) + \langle\nabla h(\bar{x}_t), \bar{x}_{t+1} - \bar{x}_t\rangle + \frac{\bar{L}}{2}\|\bar{x}_{t+1} - \bar{x}_t\|^2\right]$$

$$\overset{(a)}{=} \mathbb{E}\left[h(\bar{x}_t) - \eta\alpha_t\langle\nabla h(\bar{x}_t), \bar{\nu}_t\rangle + \frac{\eta^2\alpha_t^2\bar{L}}{2}\|\bar{\nu}_t\|^2\right]$$

$$\overset{(b)}{=} \mathbb{E}\left[h(\bar{x}_t) - \frac{\eta\alpha_t}{2}\|\bar{\nu}_t\|^2 - \frac{\eta\alpha_t}{2}\|\nabla h(\bar{x}_t)\|^2 + \frac{\eta\alpha_t}{2}\|\nabla h(\bar{x}_t) - \bar{\nu}_t\|^2 + \frac{\eta\alpha_t^2\bar{L}}{2}\|\bar{\nu}_t\|^2\right]$$

$$= \mathbb{E}\left[h(\bar{x}_t) - \frac{\eta\alpha_t}{4}\|\bar{\nu}_t\|^2 - \frac{\eta\alpha_t}{2}\|\nabla h(\bar{x}_t)\|^2 + \frac{\eta\alpha_t}{2}\underbrace{\|\nabla h(\bar{x}_t) - \bar{\nu}_t\|^2}_{T_1}\right]$$

where equality $(a)$ follows from the iterate update given in Algorithm 1; $(b)$ uses $\langle a, b\rangle = \frac{1}{2}[\|a\|^2 + \|b\|^2 - \|a - b\|^2]$ and $\eta\alpha_t < \frac{1}{2\bar{L}}$; For the term $T_1$, we have:

$$\mathbb{E}\left[\|\nabla h(\bar{x}_t) - \bar{\nu}_t\|^2\right] \leq 2\mathbb{E}\left[\|\nabla h(\bar{x}_t) - \frac{1}{M}\sum_{m=1}^{M}\nabla h(x_t^{(m)})\|^2\right] + 4\mathbb{E}\left[\|\frac{1}{M}\sum_{m=1}^{M}\nabla h(x_t^{(m)}) - \mathbb{E}_\xi[\bar{\mu}_{t,\mathcal{B}_x}]\|^2\right]$$

$$+ 4\mathbb{E}\left[\|\mathbb{E}_\xi[\bar{\mu}_{t,\mathcal{B}_x}] - \bar{\nu}_t\|^2\right]$$

For the first term, we have:

$$2\mathbb{E}\left[\|\nabla h(\bar{x}_t) - \frac{1}{M}\sum_{m=1}^{M}\nabla h(x_t^{(m)})\|^2\right] \leq \frac{2}{M}\sum_{m=1}^{M}\mathbb{E}\left[\|\nabla h(\bar{x}_t) - \nabla h(x_t^{(m)})\|^2\right] \leq \frac{2\bar{L}^2}{M}\sum_{m=1}^{M}\mathbb{E}\left[\|\bar{x}_t - x_t^{(m)}\|^2\right]$$

$$\leq \frac{2\bar{L}^2 I\eta^2}{M}\sum_{\ell=\bar{t}_{s-1}}^{t-1}\alpha_l^2\sum_{m=1}^{M}\mathbb{E}\|(\nu_\ell^{(m)} - \bar{\nu}_\ell)\|^2$$

where the last inequality uses Lemma D.8. For the second term, we have:

$$4\mathbb{E}\left[\|\frac{1}{M}\sum_{m=1}^{M}\nabla h(x_t^{(m)}) - \mathbb{E}_\xi[\bar{\mu}_{t,\xi}]\|^2\right] \leq \frac{8}{M}\sum_{m=1}^{M}\mathbb{E}\left[\|\nabla h(x_t^{(m)}) - \mu_t^{(m)}\|^2\right] + \frac{8}{M}\sum_{m=1}^{M}\mathbb{E}\left[\|\mu_t^{(m)} - \mathbb{E}_\xi[\mu_{t,\mathcal{B}_x}^{(m)}]\|^2\right]$$

$$\leq \frac{8\hat{L}^2}{M}\sum_{m=1}^{M}\mathbb{E}\left[\|y_{x_t^{(m)}}^{(m)} - y_t^{(m)}\|^2\right] + 8G_1^2$$

Plug the bound for term $T_1$ back gets the claim in the lemma. $\qquad\square$

### D.1.5 Proof of Convergence Theorem

We first denote the following potential function $\mathcal{G}(t)$:

$$\mathcal{G}_t = h(\bar{x}_t) + \frac{9bM\eta}{16\alpha_t}\|\bar{\nu}_t - \frac{1}{M}\sum_{m=1}^{M}\nabla h(x_t^{(m)})\|^2 + \frac{18\eta\hat{L}^2}{\mu\gamma}\times\frac{1}{M}\sum_{m=1}^{M}\|y_t^{(m)} - y_{x_t^{(m)}}^{(m)}\|^2$$

$$+ \frac{9bM\hat{L}^2\eta}{16L^2\alpha_t}\times\frac{1}{M}\sum_{m=1}^{M}\|\omega_t^{(m)} - \nabla_y g^{(m)}(x_t^{(m)}, y_t^{(m)})\|^2$$

**Theorem D.12.** *Suppose* $\gamma \leq \frac{1}{2L}$, $\eta < \min\left(\frac{\mu\gamma}{144\kappa\hat{L}}, \frac{\hat{L}^2}{C_1^{1/2}c_\nu}, \frac{1}{(C_1 I)^{1/2}}, \frac{\hat{L}}{(C_1 I)^{1/2}}, \frac{I\hat{L}^2}{\kappa c_\nu}, 1\right)$, $c_\nu = \frac{32}{9bM} +$ $\frac{\hat{L}}{24Ib^2M^2}$, $c_\omega = \frac{144L^2}{bM\mu^2} + \frac{\hat{L}}{24Ib^2M^2}$, $u = (bM\sigma)^2\bar{u}$, *where* $\bar{u} = \max\left(2, 16^3 I^{9/2}\hat{L}, c_\nu^{3/2}, c_\omega^{3/2}\right)$, $\delta = \frac{(bM\sigma)^{2/3}}{\hat{L}^{2/3}}$, *then we have:*

$$\frac{1}{T}\sum_{t=1}^{T-1}\mathbb{E}\left[\|\nabla h(\bar{x}_t)\|^2\right] = O\left(\frac{\kappa^{19/3}I^{3/2}}{T} + \frac{\kappa^{16/3}}{(bMT)^{2/3}} + \kappa^3 b^2 M^2 I^{9/2} G_1^2\right)$$

*To reach an $\epsilon$-stationary point, we need* $T = O(\kappa^8(bM)^{-1}\epsilon^{-1.5})$, $I = O(\kappa^{10/9}(bM)^{-2/3}\epsilon^{-1/3})$ *and* $Q = O(\kappa\log(\frac{\kappa}{bM\epsilon}))$.

*Proof.* By the condition that $u \geq c_\nu^{3/2}\delta^3$, it is straightforward to verify that $c_\nu\alpha_t^2 < 1$. By Lemma D.5 (in new notation), when $t \neq \bar{t}_s$, we have:

$$\frac{A_t}{\alpha_{t-1}} - \frac{A_{t-1}}{\alpha_{t-2}} \leq \left(\alpha_{t-1}^{-1} - \alpha_{t-2}^{-1} - c_\nu\alpha_{t-1}\right)A_{t-1} + \frac{2c_\nu^2\alpha_{t-1}^3 G_2^2}{bM} + \frac{4\hat{L}^2\eta^2\alpha_{t-1}}{bM}(D_{t-1} + E_{t-1}) + \frac{2\hat{L}^2\gamma^2\alpha_{t-1}F_{t-1}}{bM}$$

Note we choose $b_x = b$. For $\alpha_{t-1}^{-1} - \alpha_{t-2}^{-1}$, we have:

$$\alpha_t^{-1} - \alpha_{t-1}^{-1} = \frac{(u+\sigma^2 t)^{1/3}}{\delta} - \frac{(u+\sigma^2(t-1))^{1/3}}{\delta} \overset{(a)}{\leq} \frac{\sigma^2}{3\delta(u+\sigma^2(t-1))^{2/3}}$$

$$\overset{(b)}{\leq} \frac{2^{2/3}\sigma^2\delta^2}{3\delta^3(u+\sigma^2 t)^{2/3}} \overset{(c)}{=} \frac{2^{2/3}\sigma^2}{3\delta^3}\alpha_t^2 \leq \frac{2\hat{L}^2}{3M^2}\alpha_t^2 \leq \frac{\hat{L}}{24Ib^2M^2}\alpha_t$$

where inequality $(a)$ results from the concavity of $x^{1/3}$ as: $(x+y)^{1/3} - x^{1/3} \leq y/3x^{2/3}$, inequality $(b)$ used the fact that $u_t \geq 2\sigma^2$, inequality $(c)$ uses the definition of $\alpha_t$. By choosing $c_\nu = \frac{32}{9bM} + \frac{\hat{L}}{24Ib^2M^2}$, we have:

$$\frac{A_t}{\alpha_{t-1}} - \frac{A_{t-1}}{\alpha_{t-2}} \leq -\frac{32}{9bM}\alpha_{t-1}A_{t-1} + \frac{2c_\nu^2\alpha_{t-1}^3 G_2^2}{bM} + \frac{4\hat{L}^2\eta^2\alpha_{t-1}}{bM}(D_{t-1} + E_{t-1}) + \frac{2\hat{L}^2\gamma^2\alpha_{t-1}F_{t-1}}{bM}$$

When $t = \bar{t}_s$, by Lemma D.5 and Lemma D.8, we have:

$$\frac{A_t}{\alpha_{t-1}} - \frac{A_{t-1}}{\alpha_{t-2}} \leq -\frac{32}{9bM}\alpha_{t-1}A_{t-1} + \frac{2c_\nu^2\alpha_{t-1}^3 G_2^2}{bM} + \frac{8\hat{L}^2\eta^2\alpha_{t-1}}{bM}(D_{t-1} + E_{t-1})$$

$$+ \frac{2\hat{L}^2\gamma^2\alpha_{t-1}}{bM}F_{t-1} + \frac{8\hat{L}^2}{bM}\sum_{\ell=\bar{t}_{s-1}}^{t-1} I\eta^2\alpha_\ell D_\ell$$

Note we use the fact $\alpha_t/\alpha_{\bar{t}_s-1} < 2$ in the last term, which is due to:

$$\frac{\alpha_t}{\alpha_{\bar{t}_s-1}} = \frac{(u_{\bar{t}_s-1} + \sigma^2(\bar{t}_s-1))^{1/3}}{(u_t + \sigma^2 t)^{1/3}} = \left(1 + \frac{u_{\bar{t}_s-1} - u_t + \sigma^2(\bar{t}_s-1-t)}{u_t + \sigma^2 t}\right)^{1/3}$$

$$\leq \left(1 + \frac{(I-1)\sigma^2}{u_t + \sigma^2 t}\right)^{1/3} \leq 1 + \frac{(I-1)}{3(t+I+1)} \leq 2$$

where we use the condition $u_t \geq (I+1)\sigma^2$. Next, we telescope from $\bar{t}_{s-1} + 1$ to $\bar{t}_s$:

$$\left(\frac{A_{\bar{t}_s}}{\alpha_{\bar{t}_s-1}} - \frac{A_{\bar{t}_{s-1}}}{\alpha_{\bar{t}_{s-1}-1}}\right) \leq -\frac{32}{9bM}\sum_{t=\bar{t}_{s-1}}^{\bar{t}_s-1}\alpha_t A_t + \frac{2c_\nu^2 G_2^2}{bM}\sum_{t=\bar{t}_{s-1}}^{\bar{t}_s-1}\alpha_t^3 + \frac{16I\hat{L}^2\eta^2}{bM}\sum_{t=\bar{t}_{s-1}}^{\bar{t}_s-1}\alpha_t D_t$$

$$+ \frac{8\hat{L}^2\eta^2}{bM}\sum_{t=\bar{t}_{s-1}}^{\bar{t}_s-1}\alpha_t E_t + \frac{2\hat{L}^2\gamma^2}{bM}\sum_{t=\bar{t}_{s-1}}^{\bar{t}_s-1}\alpha_t F_t \qquad (43)$$

Next, we follow similar derivation as $A_t/\alpha_{t-1} - A_{t-1}/\alpha_{t-2}$. By Lemma D.6, For $t \neq \bar{t}_s$, we choose $c_\omega = \frac{144L^2}{bM\mu^2} + \frac{\hat{L}}{24Ib^2M^2}$, to obtain:

$$\frac{C_t}{\alpha_{t-1}} - \frac{C_{t-1}}{\alpha_{t-2}} \leq -\frac{144L^2\alpha_{t-1}}{bM\mu^2}C_{t-1} + \frac{2c_\omega^2\alpha_{t-1}^3\sigma^2}{bM} + \frac{4L^2\eta^2\alpha_{t-1}}{bM}(D_{t-1} + E_{t-1}) + \frac{2L^2\gamma^2}{bM}\alpha_{t-1}F_{t-1}$$

Note we choose $b_y = bM$. When $t = \bar{t}_s$, by Lemma D.5 and Lemma D.8, we have:

$$\frac{C_t}{\alpha_{t-1}} - \frac{C_{t-1}}{\alpha_{t-2}} \leq -\frac{144L^2\alpha_{t-1}}{bM\mu^2}C_{t-1} + \frac{2c_\omega^2\alpha_{t-1}^3\sigma^2}{bM} + \frac{8L^2\eta^2\alpha_{t-1}}{bM}(D_{t-1} + E_{t-1})$$

$$+ \frac{2L^2\gamma^2\alpha_{t-1}}{bM}F_{t-1} + \frac{4L^2}{bM}\sum_{\ell=\bar{t}_{s-1}}^{t-1} I\eta^2\alpha_\ell D_\ell$$

Divide $\hat{c}_\omega$ for both sides and then telescope from $\bar{t}_{s-1} + 1$ to $\bar{t}_s$, we have:

$$\left(\frac{C_{\bar{t}_s}}{\alpha_{\bar{t}_s-1}} - \frac{C_{\bar{t}_{s-1}}}{\alpha_{\bar{t}_{s-1}-1}}\right) \leq -\frac{144L^2}{2bM\mu^2}\sum_{t=\bar{t}_{s-1}}^{\bar{t}_s-1}\alpha_t C_t + \frac{2c_\omega^2\sigma^2}{bM}\sum_{t=\bar{t}_{s-1}}^{\bar{t}_s-1}\alpha_t^3 + \frac{16IL^2\eta^2}{bM}\sum_{t=\bar{t}_{s-1}}^{\bar{t}_s-1}\alpha_t D_t$$

$$+ \frac{8L^2\eta^2}{bM}\sum_{t=\bar{t}_{s-1}}^{\bar{t}_s-1}\alpha_t E_t + \frac{2L^2\gamma^2}{bM}\sum_{t=\bar{t}_{s-1}}^{\bar{t}_s-1}\alpha_t F_t. \tag{44}$$

Next from Lemma D.7, for $t \neq \bar{t}_s$, we have:

$$B_t - B_{t-1} \leq -\frac{\mu\gamma\alpha_{t-1}B_{t-1}}{4} - \frac{\gamma^2\alpha_{t-1}F_{t-1}}{4} + \frac{9\gamma\alpha_{t-1}C_{t-1}}{2\mu} + \frac{9\kappa^2\eta^2\alpha_{t-1}D_{t-1}}{\mu\gamma} + \frac{9\kappa^2\eta^2\alpha_{t-1}E_{t-1}}{\mu\gamma}$$

When $t = \bar{t}_s$, we have:

$$B_t - B_{t-1} \leq -\frac{\mu\gamma\alpha_{t-1}B_{t-1}}{4} - \frac{\gamma^2\alpha_{t-1}F_{t-1}}{4} + \frac{9\gamma\alpha_{t-1}C_{t-1}}{2\mu} + \frac{18\kappa^2\eta^2\alpha_{t-1}D_{t-1}}{\mu\gamma}$$

$$+ \frac{18\kappa^2\eta^2\alpha_{t-1}E_{t-1}}{\mu\gamma} + \frac{9\kappa^2I\eta^2\alpha_{t-1}}{\mu\gamma}\sum_{\ell=\bar{t}_{s-1}}^{t-1}\alpha_\ell D_\ell$$

For the coefficient of the last term, we use $\alpha_t/\alpha_{\bar{t}_s-1} < 2$. We telescope from $\bar{t}_{s-1} + 1$ to $\bar{t}_s$ and have:

$$B_{\bar{t}_s} - B_{\bar{t}_{s-1}} \leq -\frac{\mu\gamma}{4}\sum_{t=\bar{t}_{s-1}}^{\bar{t}_s-1}\alpha_t B_t - \frac{\gamma^2}{4}\sum_{t=\bar{t}_{s-1}}^{\bar{t}_s-1}\alpha_t F_t + \frac{9\gamma}{2\mu}\sum_{t=\bar{t}_{s-1}}^{\bar{t}_s-1}\alpha_t C_t$$

$$+ \frac{36I\kappa^2\eta^2}{\mu\gamma}\sum_{t=\bar{t}_{s-1}}^{\bar{t}_s-1}\alpha_t D_t + \frac{18\kappa^2\eta^2}{\mu\gamma}\sum_{t=\bar{t}_{s-1}}^{\bar{t}_s-1}\alpha_t E_t \tag{45}$$

Next, by Lemma D.11, we have:

$$\mathbb{E}[h(\bar{x}_{t+1})] \leq \mathbb{E}[h(\bar{x}_t)] - \frac{\eta\alpha_t}{4}E_t - \frac{\eta\alpha_t}{2}\mathbb{E}[\|\nabla h(\bar{x}_t)\|^2] + \bar{L}^2I\eta^3\alpha_t\sum_{\ell=\bar{t}_{s-1}}^{t-1}\alpha_l^2 D_l + 2\eta\alpha_t A_t + 4\hat{L}^2\eta\alpha_t B_t + 4\eta\alpha_t G_1^2$$

We telescope from $\bar{t}_{s-1}$ to $\bar{t}_s$ to have:

$$\mathbb{E}[h(\bar{x}_{\bar{t}_s}) - h(\bar{x}_{\bar{t}_{s-1}})] \leq -\sum_{t=\bar{t}_{s-1}}^{\bar{t}_s-1}\frac{\eta\alpha_t}{4}E_t - \sum_{t=\bar{t}_{s-1}}^{\bar{t}_s-1}\frac{\eta\alpha_t}{2}\mathbb{E}[\|\nabla h(\bar{x}_t)\|^2] + 4\eta\sum_{t=\bar{t}_{s-1}}^{\bar{t}_s-1}\alpha_t G_1^2$$

$$+ \bar{L}^2I\eta^3\sum_{t=\bar{t}_{s-1}}^{\bar{t}_s-1}\alpha_t\sum_{\ell=\bar{t}_{s-1}}^{t-1}\alpha_l^2 D_l + \sum_{t=\bar{t}_{s-1}}^{\bar{t}_s-1}2\eta\alpha_t A_t + \sum_{t=\bar{t}_{s-1}}^{\bar{t}_s-1}4\hat{L}^2\eta\alpha_t B_t$$

$$\leq -\sum_{t=\bar{t}_{s-1}}^{\bar{t}_s-1}\frac{\eta\alpha_t}{4}E_t - \sum_{t=\bar{t}_{s-1}}^{\bar{t}_s-1}\frac{\eta\alpha_t}{2}\mathbb{E}[\|\nabla h(\bar{x}_t)\|^2] + 4\eta\sum_{t=\bar{t}_{s-1}}^{\bar{t}_s-1}\alpha_t G_1^2$$

$$+ \frac{\kappa^2\eta^3}{64}\sum_{t=\bar{t}_{s-1}}^{\bar{t}_s-1}\alpha_t D_t + \sum_{t=\bar{t}_{s-1}}^{\bar{t}_s-1}2\eta\alpha_t A_t + \sum_{t=\bar{t}_{s-1}}^{\bar{t}_s-1}4\hat{L}^2\eta\alpha_t B_t \tag{46}$$

In the last inequality, we use the fact that $\bar{t}_s - \bar{t}_{s-1} \leq I$, $\alpha_t < \frac{1}{16\hat{L}I}$ and $\bar{L}/\hat{L} = \kappa + 1 \leq 2\kappa$

Combine Eq. (43),Eq. (44), Eq. (45) and Eq. (46) and we have:

$$\mathbb{E}[\mathcal{G}_{\bar{t}_s}] - \mathbb{E}[\mathcal{G}_{\bar{t}_{s-1}}] \leq - \sum_{t=\bar{t}_{s-1}}^{\bar{t}_s-1} \frac{\eta\alpha_t}{2} \mathbb{E}[\|\nabla h(\bar{x}_t)\|^2] + \left(\frac{9\hat{L}^2\eta c_\omega^2\sigma^2}{8L^2} + \frac{9\eta c_\nu^2 G_2^2}{8}\right) \sum_{t=\bar{t}_{s-1}}^{\bar{t}_s-1} \alpha_t^3 - \frac{\hat{L}^2\eta}{2} \sum_{t=\bar{t}_{s-1}}^{\bar{t}_s-1} \alpha_t B_t$$

$$- \sum_{t=\bar{t}_{s-1}}^{\bar{t}_s-1} \left(\frac{9\eta\gamma^2\hat{L}^2}{2} - \frac{9\eta\gamma^2\hat{L}^2}{8} - \frac{9\eta\gamma\hat{L}^2}{8\mu}\right)\alpha_t F_t$$

$$- \sum_{t=\bar{t}_{s-1}}^{\bar{t}_s-1} \left(\frac{1}{4} - \frac{324\kappa^2\hat{L}^2\eta^2}{\mu^2\gamma^2} - \frac{9\hat{L}^2\eta^2}{2} - \frac{9\hat{L}^2\eta^2}{2}\right)\eta\alpha_t E_t$$

$$+ \left(\frac{\kappa^2}{64} + \frac{648I\kappa^2\hat{L}^2}{\mu^2\gamma^2} + 9I\hat{L}^2 + 9I\hat{L}^2\right)\eta^3 \sum_{t=\bar{t}_{s-1}}^{\bar{t}_s-1} \alpha_t D_t + 4\eta \sum_{t=\bar{t}_{s-1}}^{\bar{t}_s-1} \alpha_t G_1^2$$

By the condition that $\eta < \frac{\mu\gamma}{144\kappa\hat{L}}$. So we have:

$$\mathbb{E}[\mathcal{G}_{\bar{t}_s}] - \mathbb{E}[\mathcal{G}_{\bar{t}_{s-1}}] \leq - \sum_{t=\bar{t}_{s-1}}^{\bar{t}_s-1} \frac{\eta\alpha_t}{2} \mathbb{E}[\|\nabla h(\bar{x}_t)\|^2] + \left(\frac{9\eta c_\omega^2\sigma^2}{8\mu\gamma} + \frac{9\eta c_\nu^2 G_2^2}{8\mu\gamma}\right) \sum_{t=\bar{t}_{s-1}}^{\bar{t}_s-1} \alpha_t^3 + 4\eta \sum_{t=\bar{t}_{s-1}}^{\bar{t}_s-1} \alpha_t G_1^2$$

$$- \frac{9\eta\gamma^2\hat{L}^2}{4} \sum_{t=\bar{t}_{s-1}}^{\bar{t}_s-1} \alpha_t F_t - \frac{3\eta}{16} \sum_{t=\bar{t}_{s-1}}^{\bar{t}_s-1} \alpha_t E_t - \frac{\hat{L}^2\eta}{2} \sum_{t=\bar{t}_{s-1}}^{\bar{t}_s-1} \alpha_t B_t + C_1 I\eta^3 \sum_{t=\bar{t}_{s-1}}^{\bar{t}_s-1} \alpha_t D_t$$

(47)

where we denote $C_1 = \left(\frac{\kappa^2}{64} + \frac{648\kappa^2\hat{L}^2}{\mu^2\gamma^2} + 9\hat{L}^2 + 9\hat{L}^2\right)$. By Lemma D.10 and choose $\eta < \frac{\hat{L}^2}{\kappa c_\nu}$, we have:

$$\sum_{t=\bar{t}_{s-1}}^{\bar{t}_s-1} \alpha_t D_t \leq \frac{c_\nu^2}{128I^4\hat{L}^2} \sum_{t=\bar{t}_{s-1}}^{\bar{t}_s-1} \alpha_t B_t + \frac{\eta^2}{8I} \sum_{t=\bar{t}_{s-1}}^{\bar{t}_s-1} \alpha_t E_t + \frac{\gamma^2}{16I} \sum_{t=\bar{t}_{s-1}}^{\bar{t}_s-1} \alpha_t F_t$$

$$+ \left(\frac{c_\nu^2 G_1^2}{8I\hat{L}^2} + \frac{c_\nu^2 G_2^2}{8Ib\hat{L}^2} + \frac{c_\nu^2\zeta^2}{4I\hat{L}^2}\right) \sum_{t=\bar{t}_{s-1}}^{\bar{t}_s-1} \alpha_t^3$$

(48)

Combine Eq. (47) and Eq. (48), and use the condition that $\eta < \min\left(\frac{\hat{L}^2}{C_1^{1/2}c_\nu}, \frac{1}{C_1^{1/2}}, \frac{\hat{L}}{C_1^{1/2}}, 1\right)$, the fact that $I \geq 1$, we have:

$$\mathbb{E}[\mathcal{G}_{\bar{t}_s}] - \mathbb{E}[\mathcal{G}_{\bar{t}_{s-1}}] \leq - \sum_{t=\bar{t}_{s-1}}^{\bar{t}_s-1} \frac{\eta\alpha_t}{2} \mathbb{E}[\|\nabla h(\bar{x}_t)\|^2] + 4\eta \sum_{t=\bar{t}_{s-1}}^{\bar{t}_s-1} \alpha_t G_1^2$$

$$+ \eta\left(\frac{9\hat{L}^2 c_\omega^2\sigma^2}{8L^2} + \frac{9c_\nu^2 G_2^2}{8} + \frac{c_\nu^2 G_1^2}{8} + \frac{c_\nu^2 G_2^2}{8b} + \frac{c_\nu^2\zeta^2}{4}\right) \sum_{t=\bar{t}_{s-1}}^{\bar{t}_s-1} \alpha_t^3$$

Sum over all $s \in [S]$ (assume $T = SI + 1$ without loss of generality), we have:

$$\mathbb{E}[\mathcal{G}_T] - \mathbb{E}[\mathcal{G}_1] \leq - \sum_{t=1}^{T-1} \frac{\eta\alpha_t}{2} \mathbb{E}[\|\nabla h(\bar{x}_t)\|^2] + \left(\eta C_{\sigma,\zeta} + \frac{4\eta G_1^2}{\alpha_T^2}\right) \sum_{t=1}^{T-1} \alpha_t^3$$

For ease of notation, we denote $C_{\sigma,\zeta} = \left(\frac{9\hat{L}^2 c_\omega^2\sigma^2}{8L^2} + \frac{9c_\nu^2 G_2^2}{8} + \frac{c_\nu^2 G_1^2}{8} + \frac{c_\nu^2 G_2^2}{8b} + \frac{c_\nu^2\zeta^2}{4}\right)$. Rearranging the terms and use the fact that $\alpha_t$ is non-increasing, we have:

$$\frac{\eta\alpha_T}{2} \sum_{t=1}^{T-1} \mathbb{E}[\|\nabla h(\bar{x}_t)\|^2] \leq \mathbb{E}[\mathcal{G}_1] - \mathbb{E}[\mathcal{G}_T] + \left(\eta C_{\sigma,\zeta} + \frac{4\eta G_1^2}{\alpha_T^2}\right) \sum_{t=1}^{T-1} \alpha_t^3$$

$$\leq h(x_1) - h^* + \frac{9bM\eta A_1}{16\alpha_1} + \frac{18\eta\hat{L}^2 B_1}{\mu\gamma} + \frac{9bM\eta C_1}{16\alpha_1} + \left(\eta C_{\sigma,\zeta} + \frac{4\eta G_1^2}{\alpha_T^2}\right) \sum_{t=1}^{T-1} \alpha_t^3$$

where we use $\mathcal{G}_T \geq h^*$ ($h^*$ is the optimal value of $h$), and for the last term, we use the following fact:

$$\sum_{t=1}^{T} \alpha_t^3 = \sum_{t=1}^{T} \frac{\delta^3}{u + \sigma^2 t} \leq \sum_{t=1}^{T} \frac{\delta^3}{\sigma^2 + \sigma^2 t} = \frac{\delta^3}{\sigma^2} \sum_{t=1}^{T} \frac{1}{1+t} \leq \frac{\delta^3}{\sigma^2} \ln(T+1) = \frac{b^2 M^2}{\hat{L}^2} \ln(T+1)$$

the first inequality follows $u_t > \sigma^2$, the last inequality follows Proposition E.3. Next, we denote the initial sub-optimality as $\Delta = h(\bar{x}_1) - h^*$, and initial inner variable estimation error *i.e.* $B_1 = \frac{1}{M} \sum_{m=1}^{M} \|y_1^{(m)} - y_{x_1^{(m)}}^{(m)}\|^2 \leq \Delta_y$, and we assume $\Delta_y = O(\kappa^{-1})$, furthermore, we have:

$$A_1 = \mathbb{E}\Big[\big\|\frac{1}{M} \sum_{m=1}^{M} \big(\Phi^{(m)}(x_1^{(m)}, y_1^{(m)}; \mathcal{B}_x) - \Phi^{(m)}(x_1^{(m)}, y_1^{(m)})\big)\big\|^2\Big] \leq \frac{\sigma^2}{b_1 M}$$

and

$$C_1 = \frac{1}{M} \sum_{m=1}^{M} \mathbb{E}\|\omega_1^{(m)} - \nabla_y g^{(m)}(x_1^{(m)}, y_1^{(m)})\|^2 \leq \frac{\sigma^2}{b_1}$$

where we choose the size of the first minibatch to be $b_x = b_1$ and $b_y = b_1 M$. Then, we divide both sides by $\eta \alpha_T T/2$ to have:

$$\frac{1}{T} \sum_{t=1}^{T-1} \mathbb{E}\big[\|\nabla h(\bar{x}_t)\|^2\big] \leq \frac{2\Delta}{\eta \alpha_T T} + \frac{9 b \sigma^2}{8 b_1 T \alpha_1 \alpha_T} + \frac{36 \hat{L}^2 \Delta_y}{\kappa \mu \gamma T \alpha_T} + \frac{9 b \sigma^2}{8 b_1 T \alpha_1 \alpha_T} + \frac{2 b^2 M^2 C_{\sigma,\zeta} \ln(T)}{\hat{L}^2 T \alpha_T} + \frac{8 b^2 M^2 G_1^2 \ln(T)}{\hat{L}^2 T \alpha_T^3}$$

Note that we have:

$$\frac{1}{\alpha_t t} = \frac{(u + \sigma^2 t)^{1/3}}{\delta t} \leq \frac{u^{1/3}}{\delta t} + \frac{\sigma^{2/3}}{\delta t^{2/3}}$$

where the inequality uses the fact that $(x+y)^{1/3} \leq x^{1/3} + y^{1/3}$. In particular, when $t = 1$, we have

$$\frac{1}{\alpha_1} \leq \frac{u^{1/3} + \sigma^{2/3}}{\delta} = \frac{\hat{L}^{2/3}(\bar{u}^{1/3}(bM)^{2/3} + 1)}{(bM)^{2/3}}$$

when $t = T$, we have:

$$\frac{1}{\alpha_T T} \leq \frac{u^{1/3}}{\delta T} + \frac{\sigma^{2/3}}{\delta T^{2/3}} = \frac{\hat{L}^{2/3} \bar{u}^{1/3}}{T} + \frac{\hat{L}^{2/3}}{(bMT)^{2/3}}$$

In summary, we have:

$$\frac{1}{T} \sum_{t=1}^{T-1} \mathbb{E}\big[\|\nabla h(\bar{x}_t)\|^2\big] \leq \big(\frac{2\Delta}{\eta} + \frac{9 b \sigma^2}{8 b_1 \alpha_1} + \frac{36 \hat{L}^2 \Delta_y}{\kappa \mu \gamma} + \frac{9 b \sigma^2}{8 b_1 \alpha_1}$$

$$+ 2\ln(T)\big(\frac{b^2 M^2 C_{\sigma,\zeta}}{\hat{L}^2} + \frac{4 b^2 M^2 G_1^2}{\hat{L}^2 \alpha_T^2}\big)\big)\big(\frac{\hat{L}^{2/3} \bar{u}^{1/3}}{T} + \frac{\hat{L}^{2/3}}{(bMT)^{2/3}}\big)$$

Note that $\hat{L} = O(\kappa^2)$, $\bar{L} = O(\kappa^3)$, $c_\nu = O((bM)^{-1}\kappa^2)$ and $c_\omega = O((bM)^{-1}\kappa^2)$, $\bar{u} = O(I^{9/2}\kappa^2 + (bM)^{-3/2}\kappa^3)$, $\alpha_1^{-1} = O(I^{3/2}\kappa^2 + (bM)^{-1/2}\kappa^{7/3})$, then for $\eta$, we have:

$$\eta \leq \min\big(\frac{\mu\gamma}{144\kappa\hat{L}}, \frac{\hat{L}^2}{\kappa c_\nu}, \frac{\hat{L}^2}{C_1^{1/2} c_\nu}, \frac{1}{C_1^{1/2}}, \frac{\hat{L}}{C_1^{1/2}}, \frac{1}{2\bar{L}}, 1\big)$$

Recall that $C_1 = \big(\frac{\kappa^2}{64} + \frac{648\kappa^2\hat{L}^2}{\mu^2\gamma^2} + 9\hat{L}^2 + 9\hat{L}^2\big)$, suppose we choose $\gamma = \frac{1}{2L}$, then $C_1 = O(\kappa^8)$ and $\eta^{-1} = O(\kappa^4)$, $\mu\gamma = O(\kappa^{-1})$. recall that $C_{\sigma,\zeta} = \big(\frac{9\hat{L}^2 c_\omega^2 \sigma^2}{8\bar{L}^2} + \frac{9 c_\nu^2 G_2^2}{8} + \frac{c_\nu^2 G_1^2}{8} + \frac{c_\nu^2 G_2^2}{8b} + \frac{c_\nu^2 \zeta^2}{4}\big)$, so we have $C_{\sigma,\zeta} = O((bM)^{-2}\kappa^8)$, suppose we choose $b_1 = O(I^{3/2})$. Finally, for the coefficient of the hyper-gradient bias term $G_1^2$, we have:

$$\frac{8 b^2 M^2 \ln(T)}{\hat{L}^{1/3}\alpha_T^2}\big(\frac{\bar{u}^{1/3}}{T} + \frac{1}{(bMT)^{2/3}}\big) \leq \frac{16 b^2 M^2 \bar{u}}{T} + 16\big(\frac{b^2 M^2 \bar{u}}{T}\big)^{2/3} + 16\big(\frac{b^2 M^2 \bar{u}}{T}\big)^{1/3} + 16 = O(\kappa^3 b^2 M^2 I^{9/2})$$

Then, we have:

$$\frac{1}{T} \sum_{t=1}^{T-1} \mathbb{E}\big[\|\nabla h(\bar{x}_t)\|^2\big] = O\big(\frac{\kappa^{19/3} I^{3/2}}{T} + \frac{\kappa^{16/3}}{(bMT)^{2/3}} + \kappa^3 b^2 M^2 I^{9/2} G_1^2\big)$$

To reach an $\epsilon$-stationary point, we need $T = O(\kappa^8 (bM)^{-1} \epsilon^{-1.5})$, $I = O(\kappa^{10/9}(bM)^{-2/3}\epsilon^{-1/3})$ and $Q = O(\kappa \log(\frac{\kappa}{bM\epsilon}))$. The communication cost is $E = T/I \geq \kappa^{62/9}(bM)^{-1/3}\epsilon^{-7/6}$, the sample complexity is $Gc(f, \epsilon) = O(M^{-1}\kappa^8 \epsilon^{-1.5})$, $Gc(g, \epsilon) = O(\kappa^8 \epsilon^{-1.5})$, $Jv(g, \epsilon) = O(\kappa^8 \epsilon^{-1.5})$, $Hv(g, \epsilon) = O(\kappa^9 \epsilon^{-1.5})$

Suppose we choose $b = O(\epsilon^{-0.5})$, we have $T = O(\kappa^8 M^{-1} \epsilon^{-1}))$, $I = \kappa^{10/9} M^{-2/3}$, $Q = O(\kappa \log(\frac{\kappa}{M\epsilon}))$ and $E = \kappa^{62/9} M^{-1/3} \epsilon^{-1}$. If we instead choose $b = O(1)$, we have $T = O(\kappa^8 M^{-1} \epsilon^{-1.5})$, $I = O(\kappa^{10/9} M^{-2/3} \epsilon^{-1/3})$ and $Q = O(\kappa \log(\frac{\kappa}{M\epsilon}))$. The communication cost is $E = O(\kappa^{62/9} M^{-1/3} \epsilon^{-7/6})$. $\qquad\square$

## D.2 Proof for the FedBiO-Local Algorithm

In this section, we investigate the convergence rate for the FedBiO-Local algorithm (Algorithm 3).

### D.2.1 Lower Problem Solution Error

**Lemma D.13.** *When $\gamma < \frac{1}{L}$, when $t \neq \bar{t}_s$, we have:*

$$\frac{1}{M} \sum_{m=1}^{M} \mathbb{E}\big\|y_t^{(m)} - y_{x_t^{(m)}}^{(m)}\big\|^2 \leq (1 - \frac{\mu\gamma}{2})\frac{1}{M}\sum_{m=1}^{M} \mathbb{E}\big\|y_{t-1}^{(m)} - y_{x_{t-1}^{(m)}}^{(m)}\big\|^2 + \frac{5\kappa^2\eta^2}{\mu\gamma M}\sum_{m=1}^{M}\mathbb{E}\big\|\nu_{t-1}^{(m)}\big\|^2 + \frac{3\gamma^2\sigma^2}{b_y}$$

*when $t = \bar{t}_s$, we have:*

$$\frac{1}{M}\sum_{m=1}^{M}\mathbb{E}\big\|y_t^{(m)} - y_{x_t^{(m)}}^{(m)}\big\|^2 \leq (1 - \frac{\mu\gamma}{2})\frac{1}{M}\sum_{m=1}^{M}\mathbb{E}\big\|y_{t-1}^{(m)} - y_{x_{t-1}^{(m)}}^{(m)}\big\|^2 + \frac{10\kappa^2\eta^2}{\mu\gamma M}\sum_{m=1}^{M}\mathbb{E}\big\|\nu_{t-1}^{(m)}\big\|^2$$

$$+ \frac{10\kappa^2}{\mu\gamma M}\sum_{m=1}^{M}\mathbb{E}\big\|x_t^{(m)} - \bar{x}_t\big\|^2 + \frac{3\gamma^2\sigma^2}{b_y}$$

*Proof.* First, we have:

$$\mathbb{E}\big\|y_t^{(m)} - y_{x_t^{(m)}}^{(m)}\big\|^2 \leq (1 + \frac{\mu\gamma}{2})\mathbb{E}\big\|y_t^{(m)} - y_{x_{t-1}^{(m)}}^{(m)}\big\|^2 + (1 + \frac{2}{\mu\gamma})\mathbb{E}\big\|y_{x_t^{(m)}}^{(m)} - y_{x_{t-1}^{(m)}}^{(m)}\big\|^2$$

$$\leq (1 + \frac{\mu\gamma}{2})(1 - \mu\gamma)\mathbb{E}\big\|y_{t-1}^{(m)} - y_{x_{t-1}^{(m)}}^{(m)}\big\|^2 + (1 + \frac{2}{\mu\gamma})\mathbb{E}\big\|y_{x_t^{(m)}}^{(m)} - y_{x_{t-1}^{(m)}}^{(m)}\big\|^2 + (1 + \frac{\mu\gamma}{2})\frac{2\gamma^2\sigma^2}{b_y}$$

$$\leq (1 - \frac{\mu\gamma}{2})\mathbb{E}\big\|y_{t-1}^{(m)} - y_{x_{t-1}^{(m)}}^{(m)}\big\|^2 + \frac{3\kappa^2}{\mu\gamma}\mathbb{E}\big\|x_t^{(m)} - x_{t-1}^{(m)}\big\|^2 + \frac{3\gamma^2\sigma^2}{b_y}$$

where the second inequality is due to Proposition E.4 where we choose $\gamma < 1/L$; in the last inequality, we use $\gamma < 1/(L)$ and $\mu \leq L$, For the last term, when $t \neq \bar{t}_s$, we have:

$$\mathbb{E}\big\|y_t^{(m)} - y_{x_t^{(m)}}^{(m)}\big\|^2 \leq (1 - \frac{\mu\gamma}{2})\mathbb{E}\big\|y_{t-1}^{(m)} - y_{x_{t-1}^{(m)}}^{(m)}\big\|^2 + \frac{5\kappa^2\eta^2}{\mu\gamma}\mathbb{E}\big\|v_{t-1}^{(m)}\big\|^2 + \frac{3\gamma^2\sigma^2}{b_y}$$

Then when $t = \bar{t}_s$, we have

$$\mathbb{E}\big\|y_t^{(m)} - y_{x_t^{(m)}}^{(m)}\big\|^2 \leq (1 - \frac{\mu\gamma}{2})\mathbb{E}\big\|y_{t-1}^{(m)} - y_{x_{t-1}^{(m)}}^{(m)}\big\|^2 + \frac{10\kappa^2\eta^2}{\mu\gamma}\mathbb{E}\big\|v_{t-1}^{(m)}\big\|^2 + \frac{10\kappa^2}{\mu\gamma}\mathbb{E}\big\|x_t^{(m)} - \bar{x}_t\big\|^2 + \frac{3\gamma^2\sigma^2}{b_y}$$

Average over all clients, we get the claim in the lemma. $\qquad\square$

### D.2.2 Upper Variable Drift

**Lemma D.14.** *For any $t \neq \bar{t}_s, s \in [S]$, we have:*

$$\|x_t^{(m)} - \bar{x}_t\|^2 \leq I\eta^2 \sum_{\ell=\bar{t}_{s-1}}^{t-1} \|\nu_\ell^{(m)} - \bar{\nu}_\ell\|^2$$

*Proof.* Note from Algorithm and the definition of $\bar{t}_s$ that at $t = \bar{t}_s$ with $s \in [S]$, $x_t^{(m)} = \bar{x}_t$, for all $k$. For $t \neq \bar{t}_s$, with $s \in [S]$, we have: $x_t^{(m)} = x_{t-1}^{(m)} - \eta\nu_{t-1}^{(m)}$, this implies that: $x_t^{(m)} = x_{\bar{t}_{s-1}}^{(m)} - \sum_{\ell=\bar{t}_{s-1}}^{t-1} \eta\nu_\ell^{(m)}$ and $\bar{x}_t = \bar{x}_{\bar{t}_{s-1}} - \sum_{\ell=\bar{t}_{s-1}}^{t-1} \eta\bar{\nu}_\ell$. So for $t \neq \bar{t}_s$, with $s \in [S]$ we have:

$$\|x_t^{(m)} - \bar{x}_t\|^2 = \left\|x_{\bar{t}_{s-1}}^{(m)} - \bar{x}_{\bar{t}_{s-1}} - \left(\sum_{\ell=\bar{t}_{s-1}}^{t-1} \eta\nu_\ell^{(m)} - \sum_{\ell=\bar{t}_{s-1}}^{t-1} \eta\bar{\nu}_\ell\right)\right\|^2 \overset{(a)}{=} \left\|\sum_{\ell=\bar{t}_{s-1}}^{t-1} \eta\left(\nu_\ell^{(m)} - \bar{\nu}_\ell\right)\right\|^2$$

$$\leq I\eta^2 \sum_{\ell=\bar{t}_{s-1}}^{t-1} \|\nu_\ell^{(m)} - \bar{\nu}_\ell\|^2$$

This completes the proof. $\qquad\qquad\qquad\qquad\qquad\qquad\qquad\qquad\qquad\qquad\qquad\qquad\square$

**Lemma D.15.** *For $t \neq \bar{t}_s, s \in [S]$, we have:*

$$\frac{1}{M}\sum_{m=1}^{M} \mathbb{E}\left\|\left(\nu_t^{(m)} - \bar{\nu}_t\right)\right\|^2 \leq \frac{4\hat{L}^2}{M}\sum_{m=1}^{M} \mathbb{E}\left\|y_t^{(m)} - y_{x_t^{(m)}}^{(m)}\right\|^2 + \frac{12\bar{L}^2 I\eta^2}{M}\sum_{m=1}^{M}\sum_{\ell=\bar{t}_{s-1}}^{t-1} \mathbb{E}\|\nu_\ell^{(m)} - \bar{\nu}_\ell\|^2$$

$$+ 8\zeta^2 + 4G_1^2 + \frac{4G_2^2}{b_x}$$

*for $t = \bar{t}_s, s \in [S]$, we have:*

$$\frac{1}{M}\sum_{m=1}^{M} \mathbb{E}\left\|\left(\nu_t^{(m)} - \bar{\nu}_t\right)\right\|^2 \leq \frac{4\hat{L}^2}{M}\sum_{m=1}^{M} \mathbb{E}\left\|y_t^{(m)} - y_{x_t^{(m)}}^{(m)}\right\|^2 + 8\zeta^2 + 4G_1^2 + \frac{4G_2^2}{b_x}$$

*Proof.* For $t \in [T]$, we have:

$$\mathbb{E}\left\|\left(\nu_t^{(m)} - \bar{\nu}_t\right)\right\|^2 \overset{(a)}{\leq} 2\mathbb{E}\left\|\left(\nu_t^{(m)} - \nabla h^{(m)}(x_t^{(m)})\right) - \left(\bar{\nu}_t - \frac{1}{M}\sum_{m=1}^{M}\nabla h^{(j)}(x_t^{(j)})\right)\right\|^2$$

$$+ 2\mathbb{E}\left\|\left(\nabla h^{(m)}(x_t^{(m)}) - \frac{1}{M}\sum_{j=1}^{M}\nabla h^{(j)}(x_t^{(j)})\right)\right\|^2$$

$$\overset{(b)}{\leq} 2\mathbb{E}\left\|\nu_t^{(m)} - \nabla h^{(m)}(x_t^{(m)})\right\|^2 + 2\mathbb{E}\left\|\nabla h^{(m)}(x_t^{(m)}) - \frac{1}{M}\sum_{j=1}^{M}\nabla h^{(j)}(x_t^{(j)})\right\|^2$$

$$\leq 4\hat{L}^2\mathbb{E}\left\|y_t^{(m)} - y_{x_t^{(m)}}^{(m)}\right\|^2 + 4G_1^2 + \frac{4G_2^2}{b_x} + 2\mathbb{E}\left\|\nabla h^{(m)}(x_t^{(m)}) - \frac{1}{M}\sum_{j=1}^{M}\nabla h^{(j)}(x_t^{(j)})\right\|^2$$

$$\tag{49}$$

where the equality $(a)$ uses triangle inequality and $(b)$ follows from the application of Proposition E.2. Next, for the second term of 49 we have:

$$\sum_{m=1}^{M} \left\|\nabla h^{(m)}(x_t^{(m)}) - \frac{1}{M}\sum_{j=1}^{M}\nabla h^{(j)}(x_t^{(j)})\right\|^2$$

$$\overset{(a)}{\leq} 2\sum_{m=1}^{M} \left\|\nabla h^{(m)}(x_t^{(m)}) - \nabla h^{(m)}(\bar{x}_t)\right\|^2 + 4M\left\|\nabla h(\bar{x}_t) - \frac{1}{M}\sum_{j=1}^{M}\nabla h^{(j)}(x_t^{(j)})\right\|^2$$

$$+ 4\sum_{m=1}^{M} \left\|\nabla h^{(m)}(\bar{x}_t) - \nabla h(\bar{x}_t)\right\|^2 \overset{(b)}{\leq} 6\bar{L}^2\sum_{m=1}^{M} \left\|x_t^{(m)} - \bar{x}_t\right\|^2 + 4M\zeta^2 \tag{50}$$

where $(a)$ follows the generalized triangle inequality; $(b)$ utilizes the heterogeneity Assumption 3.5. Next for the first term, it is 0 when $t = \bar{t}_s$ and when $t \neq \bar{t}_s$, we use Lemma D.14. Substituting 50 back to 49, we get the results in the lemma. $\qquad\square$

**Lemma D.16.** *For $s \in [S]$, we have:*

$$(1 - 12\bar{L}^2 I^2 \eta^2) \sum_{t=\bar{t}_{s-1}}^{\bar{t}_s - 1} D_t \leq 4\hat{L}^2 \sum_{t=\bar{t}_{s-1}}^{\bar{t}_s - 1} B_t + 8I\zeta^2 + 4IG_1^2 + \frac{4IG_2^2}{b_x}$$

*Proof.* For ease of notation, we denote $D_t = \frac{1}{M}\sum_{m=1}^{M} \mathbb{E}\|(\nu_t^{(m)} - \bar{\nu}_t)\|^2$ and $B_t = \frac{1}{M}\sum_{m=1}^{M} \mathbb{E}\|y_t^{(m)} - y_{x_t^{(m)}}^{(m)}\|^2$. Based on Lemma D.15, we have:

$$D_t \leq 4\hat{L}^2 B_t + 12\bar{L}^2 I\eta^2 \sum_{\ell=\bar{t}_{s-1}}^{t-1} D_\ell + 8\zeta^2 + 4G_1^2 + \frac{4G_2^2}{b_x}$$

Next, we sum from $\bar{t}_{s-1} + 1$ to $\bar{t}_s - 1$, we have:

$$\sum_{t=\bar{t}_{s-1}+1}^{\bar{t}_s-1} D_t \leq 4\hat{L}^2 \sum_{t=\bar{t}_{s-1}+1}^{\bar{t}_s-1} B_t + 12\bar{L}^2 I\eta^2 \sum_{t=\bar{t}_{s-1}+1}^{\bar{t}_s-1} \sum_{\ell=\bar{t}_{s-1}}^{t-1} D_\ell + 8(I-1)\zeta^2 + 4(I-1)G^2$$

$$\leq 4\hat{L}^2 \sum_{t=\bar{t}_{s-1}+1}^{\bar{t}_s-1} B_t + 12\bar{L}^2 I^2\eta^2 \sum_{\ell=\bar{t}_{s-1}}^{\bar{t}_s-1} D_\ell + 8(I-1)\zeta^2 + 4(I-1)G_1^2 + \frac{4(I-1)G_2^2}{b_x}$$

In the second inequality, we use $t - 1 \leq \bar{t}_s - 1$, combine with the case when $t = \bar{t}_{s-1}$ in lemma D.14, we have:

$$(1 - 12\bar{L}^2 I^2 \eta^2) \sum_{t=\bar{t}_{s-1}}^{\bar{t}_s - 1} D_t \leq 4\hat{L}^2 \sum_{t=\bar{t}_{s-1}}^{\bar{t}_s - 1} B_t + 8I\zeta^2 + 4IG_1^2 + \frac{4IG_2^2}{b_x}$$

This completes the proof. $\qquad\square$

### D.2.3 Descent Lemma

**Lemma D.17.** *For all $t \in [\bar{t}_{s-1}, \bar{t}_s - 1]$, the iterates generated satisfy:*

$$\mathbb{E}\|\nabla h(\bar{x}_t) - \mathbb{E}_\xi[\bar{\nu}_t]\|^2 \leq \frac{2\hat{L}^2}{M}\sum_{m=1}^{M}\left(4\kappa^2\mathbb{E}\|x_t^{(m)} - \bar{x}_t\|^2 + 2\mathbb{E}\|y_t^{(m)} - y_{x_t^{(m)}}^{(m)}\|^2\right) + 2G_1^2$$

*Proof.* By definition of $\bar{\nu}_t$ and $\nabla h(\bar{x}_t)$, we have:

$$\mathbb{E}\|\nabla h(\bar{x}_t) - \mathbb{E}_\xi[\bar{\nu}_t]\|^2 \overset{(a)}{\leq} \frac{1}{M}\sum_{m=1}^{M}\mathbb{E}\|\mathbb{E}_\xi[\nu_t^{(m)}] - \nabla h^{(m)}(\bar{x}_t)\|^2$$

$$\leq \frac{2}{M}\sum_{m=1}^{M}\mathbb{E}\left[\|\mathbb{E}_\xi[\nu_t^{(m)}] - \mu_t^{(m)}\|^2 + \|\mu_t^{(m)} - \nabla h^{(m)}(\bar{x}_t)\|^2\right]$$

$$\overset{(b)}{\leq} \frac{\hat{L}^2}{M}\sum_{m=1}^{M}\left(\mathbb{E}\|x_t^{(m)} - \bar{x}_t\|^2 + \mathbb{E}\|y_t^{(m)} - y_{\bar{x}_t}^{(m)}\|^2\right) + 2G_1^2$$

$$\leq \frac{2\hat{L}^2}{M}\sum_{m=1}^{M}\left(\mathbb{E}\|x_t^{(m)} - \bar{x}_t\|^2 + \mathbb{E}\|y_t^{(m)} - y_{x_t^{(m)}}^{(m)} + y_{x_t^{(m)}}^{(m)} - y_{\bar{x}_t}^{(m)}\|^2\right) + 2G_1^2$$

$$\leq \frac{2\hat{L}^2}{M}\sum_{m=1}^{M}\left((1 + 2\kappa^2)\mathbb{E}\|x_t^{(m)} - \bar{x}_t\|^2 + 2\mathbb{E}\|y_t^{(m)} - y_{x_t^{(m)}}^{(m)}\|^2\right) + +2G_1^2$$

where inequality (a) follows the generalized triangle inequality; inequality (b) follows the Proposition D.3 and Proposition D.2.

$\square$

**Lemma D.18.** *For $t \neq \bar{t}_s$, the iterates generated satisfy:*

$$\mathbb{E}[h(\bar{x}_{t+1})] \leq \mathbb{E}[h(\bar{x}_t)] - \frac{\eta}{2}\mathbb{E}\|\nabla h(\bar{x}_t)\|^2 - \frac{\eta}{4}\mathbb{E}\big\|\mathbb{E}_\xi[\bar{\nu}_t]\big\|^2 + \frac{\eta^2\bar{L}G_2^2}{2b_xM} + \eta G_1^2$$

$$+ \frac{\eta\hat{L}^2}{M}\sum_{m=1}^{M}\Big(4\kappa^2I\eta^2\sum_{\ell=\bar{t}_{s-1}}^{t-1}\mathbb{E}\big\|\nu_\ell^{(m)} - \bar{\nu}_\ell\big\|^2 + 2\mathbb{E}\big\|y_t^{(m)} - y_{x_t^{(m)}}^{(m)}\big\|^2\Big)$$

*for $t = \bar{t}_s$, we have:*

$$\mathbb{E}[h(\bar{x}_{t+1})] \leq \mathbb{E}[h(\bar{x}_t)] - \frac{\eta}{2}\mathbb{E}\|\nabla h(\bar{x}_t)\|^2 - \frac{\eta}{4}\mathbb{E}\big\|\mathbb{E}_\xi[\bar{\nu}_t]\big\|^2 + \frac{\eta^2\bar{L}G_2^2}{2b_xM} + \eta G_1^2 + \frac{2\eta\hat{L}^2}{M}\sum_{m=1}^{M}\mathbb{E}\big\|y_t^{(m)} - y_{x_t^{(m)}}^{(m)}\big\|^2$$

*where the expectation is w.r.t the stochasticity of the algorithm.*

*Proof.* Using the smoothness of $f$ we have:

$$\mathbb{E}[h(\bar{x}_{t+1})] \leq \mathbb{E}[h(\bar{x}_t)] + \mathbb{E}\langle\nabla h(\bar{x}_t), \bar{x}_{t+1} - \bar{x}_t\rangle + \frac{\bar{L}}{2}\mathbb{E}\|\bar{x}_{t+1} - \bar{x}_t\|^2$$

$$\overset{(a)}{=} \mathbb{E}[h(\bar{x}_t)] - \eta\mathbb{E}\langle\nabla h(\bar{x}_t), \mathbb{E}_\xi[\bar{\nu}_t]\rangle + \frac{\eta^2\bar{L}}{2}\mathbb{E}\|\mathbb{E}_\xi[\bar{\nu}_t]\|^2 + \frac{\eta^2\bar{L}G_2^2}{2b_xM}$$

$$\overset{(b)}{=} \mathbb{E}[h(\bar{x}_t)] - \frac{\eta}{2}\mathbb{E}\|\nabla h(\bar{x}_t)\|^2 + \frac{\eta}{2}\mathbb{E}\|\nabla h(\bar{x}_t) - \mathbb{E}_\xi[\bar{\nu}_t]\|^2 - \Big(\frac{\eta}{2} - \frac{\eta^2\bar{L}}{2}\Big)\mathbb{E}\|\mathbb{E}_\xi[\bar{\nu}_t]\|^2 + \frac{\eta^2\bar{L}G_2^2}{2b_xM}$$

$$\overset{(c)}{\leq} \mathbb{E}[h(\bar{x}_t)] - \frac{\eta}{2}\mathbb{E}\|\nabla h(\bar{x}_t)\|^2 - \frac{\eta}{4}\mathbb{E}\big\|\mathbb{E}_\xi[\nu_t^{(m)}]\big\|^2 + \frac{\eta^2\bar{L}G_2^2}{2b_xM} + \eta G_1^2$$

$$+ \frac{\eta\hat{L}^2}{M}\sum_{m=1}^{M}\big(\underbrace{4\kappa^2\mathbb{E}\big\|x_t^{(m)} - \bar{x}_t\big\|^2}_{T_1} + 2\mathbb{E}\big\|y_t^{(m)} - y_{x_t^{(m)}}^{(m)}\big\|^2\big)$$

where equality $(a)$ follows from the iterate update given in Step 6 of Algorithm 2; $(b)$ uses $\langle a, b\rangle = \frac{1}{2}[\|a\|^2 + \|b\|^2 - \|a - b\|^2]$; (c) follows the assumption that $\eta < 1/2\bar{L}$. Finally, use lemma D.17 to bound $T_1$ when $t \neq \bar{t}_s$ finishes the proof. $\square$

### D.2.4 Proof of Convergence Theorem

We first denote the following potential function $\mathcal{G}(t)$:

$$\mathcal{G}_t = \mathbb{E}[h(\bar{x}_t)] + \frac{9\eta\hat{L}^2}{\mu\gamma} \times \frac{1}{M}\sum_{m=1}^{M}\mathbb{E}\big\|y_t^{(m)} - y_{x_t^{(m)}}^{(m)}\big\|^2$$

**Theorem D.19.** *Suppose we have constant $\bar{\eta} = \min\big(\frac{1}{2C_1^{1/2}}, \frac{\mu\gamma}{12\kappa\hat{L}}, \frac{1}{2\bar{L}}, \frac{1}{6I\hat{L}}\big)$, if we choose $\eta = \min\big(\bar{\eta}, \big(\frac{2\Delta}{C_\eta T}\big)^{1/3}\big)$ and $\gamma = \frac{1}{2L}$, we have:*

$$\frac{1}{T}\sum_{t=1}^{T}\|\nabla h(\bar{x}_t)\|^2 = O\left(\frac{\kappa^5}{T} + \left(\frac{\kappa^{16}}{T^2}\right)^{1/3} + \frac{\kappa^5\sigma^2}{b_y} + \frac{G_2^2}{b_xM} + G_1^2\right)$$

*To reach an $\epsilon$ stationary point, we choose the inner batch size $b_y = O(\kappa^5\epsilon^{-1})$, upper batch size $b_x = O(M^{-1}\epsilon^{-1})$ and $Q = O(\kappa\log(\frac{\kappa}{\epsilon}))$ in Eq. 11, and $T = O(\kappa^8\epsilon^{-1.5})$ number of iterations.*

*Proof.* Similar to Lemma D.16, we denote $D_t = \frac{1}{M}\sum_{m=1}^{M}\|(\nu_t^{(m)} - \bar{\nu}_t)\|^2$, $B_t = \frac{1}{M}\sum_{m=1}^{M}\|y_t^{(m)} - y_{x_t^{(m)}}^{(m)}\|^2$, additionally, we denote $E_t = \|\mathbb{E}_\xi[\bar{\nu}_t]\|^2$. First, by Lemma D.13, when $t \neq \bar{t}_s$, by the triangle inequality, we have:

$$B_t - B_{t-1} \leq -\frac{\mu\gamma}{2}B_{t-1} + \frac{10\kappa^2\eta^2}{\mu\gamma}D_{t-1} + \frac{10\kappa^2\eta^2}{\mu\gamma}E_{t-1} + \frac{10\kappa^2\eta^2 G_2^2}{\mu\gamma b_x M} + \frac{3\gamma^2\sigma^2}{b_y}$$

When $t = \bar{t}_s$, we have:

$$B_t - B_{t-1} \leq -\frac{\mu\gamma}{2}B_{t-1} + \frac{20\kappa^2\eta^2}{\mu\gamma}D_{t-1} + \frac{20\kappa^2\eta^2}{\mu\gamma}E_{t-1} + \frac{10\kappa^2 I\eta^2}{\mu\gamma}\sum_{\ell=\bar{t}_{s-1}}^{t-1}D_\ell + \frac{10\kappa^2\eta^2 G_2^2}{\mu\gamma b_x M} + \frac{3\gamma^2\sigma^2}{b_y}$$

We telescope from $\bar{t}_{s-1}+1$ to $\bar{t}_s$ and have:

$$B_{\bar{t}_s} - B_{\bar{t}_{s-1}} \leq -\frac{\mu\gamma}{2}\sum_{t=\bar{t}_{s-1}}^{\bar{t}_s-1}B_t + \frac{40\kappa^2 I\eta^2}{\mu\gamma}\sum_{t=\bar{t}_{s-1}}^{\bar{t}_s-1}D_t + \frac{20\kappa^2\eta^2}{\mu\gamma}\sum_{t=\bar{t}_{s-1}}^{\bar{t}_s-1}E_t + \frac{10 I\kappa^2\eta^2 G_2^2}{\mu\gamma b_x M} + \frac{3I\gamma^2\sigma^2}{b_y}$$
$$(51)$$

Next, by Lemma D.18, when $t \neq \bar{t}_s$, we have:

$$\mathbb{E}[h(\bar{x}_{t+1})] - \mathbb{E}[h(\bar{x}_t)] \leq -\frac{\eta}{2}\mathbb{E}\|\nabla h(\bar{x}_t)\|^2 - \frac{\eta}{4}E_t + 4\kappa^2\hat{L}^2 I\eta^3\sum_{\ell=\bar{t}_{s-1}}^{t-1}D_l + 2\eta\hat{L}^2 B_t + \frac{\eta^2\bar{L}G_2^2}{2b_x M} + \eta G_1^2$$

and when $t = \bar{t}_s$, we have:

$$\mathbb{E}[h(\bar{x}_{t+1})] - \mathbb{E}[h(\bar{x}_t)] \leq -\frac{\eta}{2}\mathbb{E}\|\nabla h(\bar{x}_t)\|^2 - \frac{\eta}{4}E_t + 2\eta\hat{L}^2 B_t + \frac{\eta^2\bar{L}G_2^2}{2b_x M} + \eta G_1^2$$

We telescope from $\bar{t}_{s-1}$ to $\bar{t}_s$ to have:

$$\mathbb{E}[h(\bar{x}_{\bar{t}_s})] - \mathbb{E}[h(\bar{x}_{\bar{t}_{s-1}})] \leq -\sum_{t=\bar{t}_{s-1}}^{\bar{t}_s-1}\frac{\eta}{2}\mathbb{E}\|\nabla h(\bar{x}_t)\|^2 - \sum_{t=\bar{t}_{s-1}}^{\bar{t}_s-1}\frac{\eta}{4}E_t + 4\kappa^2\hat{L}^2 I\eta^3\sum_{t=\bar{t}_{s-1}+1}^{\bar{t}_s-1}\sum_{\ell=\bar{t}_{s-1}}^{t-1}D_l$$
$$+ \sum_{t=\bar{t}_{s-1}}^{\bar{t}_s-1}2\hat{L}^2\eta B_t + \frac{I\eta^2\bar{L}G_2^2}{2b_x M} + I\eta G_1^2$$
$$\leq -\sum_{t=\bar{t}_{s-1}}^{\bar{t}_s-1}\frac{\eta}{2}\mathbb{E}\|\nabla h(\bar{x}_t)\|^2 - \sum_{t=\bar{t}_{s-1}}^{\bar{t}_s-1}\frac{\eta}{4}E_t + 4\kappa^2\hat{L}^2 I^2\eta^3\sum_{t=\bar{t}_{s-1}}^{\bar{t}_s-1}D_l$$
$$+ 2\hat{L}^2\eta\sum_{t=\bar{t}_{s-1}}^{\bar{t}_s-1}B_t + \frac{I\eta^2\bar{L}G_2^2}{2b_x M} + I\eta G_1^2$$
$$(52)$$

In the last inequality, we use the fact that $\bar{t}_s - \bar{t}_{s-1} \leq I$.

Next, by the definition of the potential function and combine with Eq. 51 and Eq. 52, we have:

$$\mathcal{G}_{\bar{t}_s} - \mathcal{G}_{\bar{t}_{s-1}} \leq -\frac{\eta}{2}\sum_{t=\bar{t}_{s-1}}^{\bar{t}_s-1}\mathbb{E}\|\nabla h(\bar{x}_t)\|^2 - \frac{5\eta\hat{L}^2}{2}\sum_{t=\bar{t}_{s-1}}^{\bar{t}_s-1}B_t - \frac{\eta}{4}\left(1 - \frac{720\kappa^2\eta^2\hat{L}^2}{\mu^2\gamma^2}\right)\sum_{t=\bar{t}_{s-1}}^{\bar{t}_s-1}E_t$$
$$+ \left(\frac{360}{\mu^2\gamma^2} + 4I\right)\eta^3\kappa^2\hat{L}^2 I\sum_{t=\bar{t}_{s-1}}^{\bar{t}_s-1}D_t + \frac{27I\hat{L}^2\gamma\eta\sigma^2}{b_y\mu}$$
$$+ \frac{90I\kappa^2\hat{L}^2\eta^3 G_2^2}{\mu^2\gamma^2 b_x M} + \frac{I\eta^2\bar{L}G_2^2}{2b_x M} + I\eta G_1^2$$

to bound the coefficients above, we choose $\eta \le \frac{\mu\gamma}{48\kappa\hat{L}}$. Then we have:

$$\mathcal{G}_{\bar{t}_s} - \mathcal{G}_{\bar{t}_{s-1}} \le -\frac{\eta}{2}\sum_{t=\bar{t}_{s-1}}^{\bar{t}_s-1}\mathbb{E}\|\nabla h(\bar{x}_t)\|^2 - \frac{5\eta\hat{L}^2}{2}\sum_{t=\bar{t}_{s-1}}^{\bar{t}_s-1}B_t - \frac{\eta}{8}\sum_{t=\bar{t}_{s-1}}^{\bar{t}_s-1}E_t$$

$$+\left(\frac{360}{\mu^2\gamma^2}+4I\right)\eta^3\kappa^2\hat{L}^2 I\sum_{t=\bar{t}_{s-1}}^{\bar{t}_s-1}D_t + \frac{27I\hat{L}^2\gamma\eta\sigma^2}{b_y\mu}$$

$$+\frac{90I\kappa^2\hat{L}^2\eta^3 G_2^2}{\mu^2\gamma^2 b_x M} + \frac{I\eta^2\bar{L}G_2^2}{2b_x M} + I\eta G_1^2$$

By lemma D.16, and choosing $\eta < \frac{1}{6I\hat{L}}$, we have:

$$\sum_{t=\bar{t}_{s-1}}^{\bar{t}_s-1}D_t \le 6\hat{L}^2\sum_{t=\bar{t}_{s-1}}^{\bar{t}_s-1}B_t + 18I\zeta^2 + 6IG_1^2 + \frac{6IG_2^2}{b_x}$$

Next, we denote $C_1 = \left(\frac{360}{\mu^2\gamma^2}+4I\right)\kappa^2\hat{L}^2$, and choose $\eta < \min(\frac{1}{2C_1^{1/2}}, \frac{\mu\gamma}{12\kappa\hat{L}}, \frac{1}{2\hat{L}})$ then we have:

$$\mathcal{G}_{\bar{t}_s} - \mathcal{G}_{\bar{t}_{s-1}} \le -\frac{\eta}{2}\sum_{t=\bar{t}_{s-1}}^{\bar{t}_s-1}\mathbb{E}\|\nabla h(\bar{x}_t)\|^2 - \frac{5\eta\hat{L}^2}{2}\sum_{t=\bar{t}_{s-1}}^{\bar{t}_s-1}B_t - \frac{\eta}{8}\sum_{t=\bar{t}_{s-1}}^{\bar{t}_s-1}E_t$$

$$+C_1 I\eta^3\left(6\hat{L}^2\sum_{t=\bar{t}_{s-1}+1}^{\bar{t}_s}B_t + 18I\zeta^2 + 6IG_1^2 + \frac{6IG_2^2}{b_x}\right)$$

$$+\frac{27I\hat{L}^2\gamma\eta\sigma^2}{b_y\mu} + \frac{5I\eta G_2^2}{8b_x M} + \frac{I\eta^2\bar{L}G_2^2}{2b_x} + I\eta G_1^2$$

$$\le -\frac{\eta}{2}\sum_{t=\bar{t}_{s-1}}^{\bar{t}_s-1}\mathbb{E}\|\nabla h(\bar{x}_t)\|^2 + 18C_1 I\hat{L}^2\eta^3\zeta^2 + 6C_1 I\hat{L}^2\eta^3 G_1^2 + \frac{6C_1 I\hat{L}^2\eta^3 G_2^2}{b_x} + \frac{5I\eta G_2^2}{8b_x M}$$

$$+\frac{27I\hat{L}^2\gamma\eta\sigma^2}{b_y\mu} + \frac{I\eta^2\bar{L}G_2^2}{2b_x M} + I\eta G_1^2$$

Sum over all $s \in [S]$ (assume $T = SI + 1$ without loss of generality) to obtain:

$$\frac{\eta}{2}\sum_{t=1}^{T}\mathbb{E}\|\nabla h(\bar{x}_t)\|^2 \le \mathcal{G}_1 - \mathcal{G}_T + 18C_1 T\hat{L}^2\eta^3\zeta^2 + 6C_1 T\hat{L}^2\eta^3 G_1^2 + \frac{6C_1 T\hat{L}^2\eta^3 G_2^2}{b_x}$$

$$+\frac{5T\eta G_2^2}{8b_x M} + \frac{27T\hat{L}^2\gamma\eta\sigma^2}{b_y\mu} + \frac{T\eta^2\bar{L}G_2^2}{2b_x M} + T\eta G_1^2$$

$$\le \Delta + \frac{9\eta\hat{L}^2\Delta_y}{\mu\gamma} + 18C_1 T\hat{L}^2\eta^3\zeta^2 + 6C_1 T\hat{L}^2\eta^3 G_1^2 + \frac{6C_1 T\hat{L}^2\eta^3 G_2^2}{b_x}$$

$$+\frac{5T\eta G_2^2}{8b_x M} + \frac{27T\hat{L}^2\gamma\eta\sigma^2}{b_y\mu} + \frac{T\eta^2\bar{L}G_2^2}{2b_x M} + T\eta G_1^2$$

we define $\Delta = h(x_1) - h^*$ as the initial sub-optimality of the function and $\Delta_y = \frac{1}{M}\sum_{m=1}^{M}\left\|y_1^{(m)} - y_{x_1}^{(m)}\right\|^2$ as the initial sub-optimality of the inner variable estimation, then we divide by $\eta T/2$ on both sides and have:

$$\frac{1}{T}\sum_{t=1}^{T}\mathbb{E}\|\nabla h(\bar{x}_t)\|^2 \le \underbrace{\frac{2\Delta}{\eta T} + \frac{\eta\bar{L}G_2^2}{2b_x M} + \left(36C_1\hat{L}^2\zeta^2 + 12C_1\hat{L}^2 G_1^2 + \frac{12C_1\hat{L}^2 G_2^2}{b_x}\right)\eta^2}_{T_1}$$

$$+\underbrace{\frac{18\hat{L}^2\Delta_y}{\mu\gamma T} + \frac{54\hat{L}^2\gamma\sigma^2}{b_y\mu}}_{T_2} + \underbrace{\frac{5G_2^2}{4b_x M} + G_1^2}_{T_3}$$

As shown in the inequality, we break the bound into three parts. The $T_1$ part has a structure similar to that for the single level federated learning problems. Then the $T_2$ part includes the optimization error of the lower problem, and the statistical error of sampling. Finally, the $T_3$ part includes the bias and variance of the hyper-gradient estimate.

Next, by $\eta < \frac{1}{2L}$, we have

$$\frac{1}{T}\sum_{t=1}^{T}\mathbb{E}\|\nabla h(\bar{x}_t)\|^2 \leq \frac{2\Delta}{\eta T} + \left(36C_1\hat{L}^2\zeta^2 + 12C_1\hat{L}^2G_1^2 + \frac{12C_1\hat{L}^2G_2^2}{b_x}\right)\eta^2$$
$$+ \frac{18\hat{L}^2\Delta_y}{\mu\gamma T} + \frac{54\hat{L}^2\gamma\sigma^2}{b_y\mu} + \frac{G_2^2}{4b_xM} + \frac{5G_2^2}{4b_xM} + G_1^2 \qquad (53)$$

Next, we denote constant $\bar{\eta} = \min\left(\frac{1}{2C_1^{1/2}}, \frac{\mu\gamma}{12\kappa\hat{L}}, \frac{1}{2\bar{L}}, \frac{1}{6I\hat{L}}\right)$ and $C_\eta = \left(36C_1\hat{L}^2\zeta^2 + 12C_1\hat{L}^2G_1^2 + \frac{12C_1\hat{L}^2G_2^2}{b_x}\right)$ we choose

$$\eta = \min\left(\bar{\eta}, \left(\frac{2\Delta}{C_\eta T}\right)^{1/3}\right)$$

and $\gamma = \frac{1}{2L}$, and obtain:

$$\frac{1}{T}\sum_{t=1}^{T}\mathbb{E}\|\nabla h(\bar{x}_t)\|^2 \leq \frac{2\Delta}{\bar{\eta}T} + \frac{36\kappa\hat{L}^2\Delta_y}{T} + \left(\frac{4C_\eta\Delta^2}{T^2}\right)^{1/3} + \frac{27\hat{L}^2\sigma^2}{\mu b_y L} + \frac{3G_2^2}{2b_xM} + 2G_1^2$$

Finally, since $\hat{L} = O(\kappa^2)$, $\bar{L} = O(\kappa^3)$ and and $\mu\gamma = O(\kappa^{-1})$. Suppose we choose $I = O(1)$, then $\bar{\eta} = O(\kappa^{-4})$ and $C_1 = O(\kappa^8)$, $\zeta = O(\kappa^2)$, $C_\eta = O(\kappa^{16})$, thus, we have

$$\frac{1}{T}\sum_{t=1}^{T}\|\nabla h(\bar{x}_t)\|^2 = O\left(\frac{\kappa^5}{T} + \left(\frac{\kappa^{16}}{T^2}\right)^{1/3} + \frac{\kappa^5\sigma^2}{b_y} + \frac{G_2^2}{b_xM} + G_1^2\right)$$

and to reach an $\epsilon$ stationary point, we choose the inner batch size $b_y = O(\kappa^5\epsilon^{-1})$, upper batch size $b_x = O(M^{-1}\epsilon^{-1})$ and $Q = O(\kappa\log(\frac{\kappa}{\epsilon}))$ in Eq. 11, and $T = O(\kappa^8\epsilon^{-1.5})$ number of iterations. $\qquad\square$

## E   Useful Propositions

In this section, we state some propositions useful in the proof:

**Proposition E.1** (Lemma 3 of [27]). *(generalized triangle inequality) Let $\{x_k\}, k \in K$ be $K$ vectors. Then the following are true:*

1. $\|x_i + x_j\|^2 \leq (1+a)\|x_i\|^2 + (1+\frac{1}{a})\|x_j\|^2$ *for any $a > 0$, and*

2. $\|\sum_{k=1}^{K} x_k\|^2 \leq K\sum_{k=1}^{K}\|x_k\|^2$

**Proposition E.2** (Lemma C.1 of [30]). *For a finite sequence $x^{(k)} \in \mathbb{R}^d$ for $k \in [K]$ define $\bar{x} := \frac{1}{K}\sum_{k=1}^{K} x^{(k)}$, we then have $\sum_{k=1}^{K}\|x^{(k)} - \bar{x}\|^2 \leq \sum_{k=1}^{K}\|x^{(k)}\|^2$.*

**Proposition E.3** (Lemma C.2 of [30]). *Let $a_0 > 0$ and $a_1, a_2, \ldots, a_T \geq 0$. We have*

$$\sum_{t=1}^{T}\frac{a_t}{a_0 + \sum_{i=t}^{t} a_i} \leq \ln\left(1 + \frac{\sum_{i=1}^{t} a_i}{a_0}\right).$$

**Proposition E.4.** *Suppose we have function $g(y)$, which is L-smooth and $\mu$-strongly-convex, then suppose $\gamma < \frac{1}{L}$, the progress made by one step of gradient descent is:*

$$\mathbb{E}\|y_{t+1} - y^*\|^2 \leq (1 - \mu\gamma)\|y^* - y_t\|^2 + 2\gamma^2\sigma^2$$

*where $y^*$ is the minimum of $g(y)$ and we have update rule $g(y_{t+1}) = g(y_t) - \gamma\nabla g(y_t, \xi)$, where the error of stochastic gradient estimate is bounded by $\sigma^2$.*

*Proof.* First, by the strong convexity of of function $g(y)$, we have:

$$g(y^*) \geq g(y_t) + \langle \nabla_y g(y_t), y^* - y_t \rangle + \frac{\mu}{2}\|y^* - y_t\|^2$$

$$= g(y_t) + \langle \nabla_y g(y_t), y^* - y_{t+1} \rangle + \langle \nabla_y g(y_t), y_{t+1} - y_t \rangle + \frac{\mu}{2}\|y^* - y_t\|^2$$

Then by $L$-smoothness, we have: $\frac{L}{2}\|y_{t+1} - y_t\|^2 \geq g(y_{t+1}) - g(y_t) - \langle \nabla_y g(y_t), y_{t+1} - y_t \rangle$, Combining above two inequalities and take expectation on both sides, we have

$$g(y^*) \geq \mathbb{E}g(y_{t+1}) + \mathbb{E}\langle \nabla_y g(y_t), y^* - y_{t+1} \rangle + \frac{\mu}{2}\|y^* - y_t\|^2 - \frac{L}{2}\mathbb{E}\|y_{t+1} - y_t\|^2$$

$$\geq \mathbb{E}g(y_{t+1}) + \gamma\|\nabla_y g(y_t)\|^2 + \langle \nabla_y g(y_t), y^* - y_t \rangle + \frac{\mu}{2}\|y^* - y_t\|^2 - \frac{L\gamma^2}{2}\mathbb{E}\|\nabla_y g(y_t, \xi)\|^2$$

$$\geq \mathbb{E}g(y_{t+1}) + \gamma\|\nabla_y g(y_t)\|^2 + \langle \nabla_y g(y_t), y^* - y_t \rangle + \frac{\mu}{2}\|y^* - y_t\|^2 - \frac{L\gamma^2}{2}\mathbb{E}\|\nabla_y g(y_t)\|^2 - \frac{L\gamma^2\sigma^2}{2}$$

$$\geq \mathbb{E}g(y_{t+1}) + \langle \nabla_y g(y_t), y^* - y_t \rangle + \frac{\mu}{2}\|y^* - y_t\|^2 + \left(\gamma - \frac{L\gamma^2}{2}\right)\|\nabla_y g(y_t)\|^2 - \frac{L\gamma^2\sigma^2}{2}$$

By definition of $y^*$, we have $g(y^*) \geq g(y_{t+1})$. Thus, we obtain

$$0 \geq \langle \nabla_y g(y_t), y^* - y_t \rangle + \frac{\mu}{2}\|y^* - y_t\|^2 + \left(\gamma - \frac{L\gamma^2}{2}\right)\|\nabla_y g(y_t)\|^2 - \frac{L\gamma^2\sigma^2}{2}$$

By $y_{t+1} = y_t - \gamma\nabla_y g(y_t, \xi)$, we have:

$$\mathbb{E}\|y_{t+1} - y^*\|^2 = \mathbb{E}\|y_t - \gamma\nabla_y g(y_t, \xi) - y^*\|^2 = \|y_t - y^*\|^2 - 2\gamma\langle \nabla_y g(y_t), y_t - y^* \rangle + \gamma^2\mathbb{E}\|\nabla_y g(y_t, \xi)\|^2$$

$$\leq (1 - \mu\gamma)\|y_t - y^*\|^2 - 2\gamma\left(\gamma - \frac{L\gamma^2}{2} - \frac{\gamma}{2}\right)\|\nabla_y g(y_t)\|^2 + \left(L\gamma^3 + \gamma^2\right)\sigma^2$$

Then since we choose $\gamma < \frac{1}{L}$, we obtain:

$$\mathbb{E}\|y_{t+1} - y^*\|^2 \leq (1 - \mu\gamma)\|y^* - y_t\|^2 + 2\gamma^2\sigma^2$$

This completes the proof. $\qquad\square$

**Proposition E.5.** *Suppose we have function $g(y)$, which is L-smooth and $\mu$-strongly-convex, then suppose $\gamma < \frac{1}{2L}$ and $\alpha_t < 1$, the progress made by one step of gradient descent is:*

$$\|y_{t+1} - y^*\|^2 \leq (1 - \frac{\mu\gamma\alpha_t}{2})\|y_t - y^*\|^2 - \frac{\gamma^2\alpha_t}{4}\|\omega_t\|^2$$

$$+ \frac{4\gamma\alpha_t}{\mu}\|\nabla_y g(x_t, y_t) - \mathbb{E}[w_t]\|^2 + \frac{3\gamma^2\alpha_t}{2}Var[\omega_t].$$

*where $y^*$ is the minimum of $g(y)$ and we have update rule $g(y_{t+1}) = g(y_t) - \gamma\alpha_t\omega_t$.*

*Proof.* First, Suppose we denote $\tilde{y}_{t+1} = y_t - \gamma\omega_t$, then we have $y_{t+1} = y_t + \alpha_t(\tilde{y}_{t+1} - y_t)$. By the strong convexity of of function $g(y)$, we have:

$$g(y^*) \geq g(y_t) + \langle \nabla_y g(y_t), y^* - y_t \rangle + \frac{\mu}{2}\|y^* - y_t\|^2$$

$$= g(y_t) + \mathbb{E}\langle \mathbb{E}[w_t], y^* - \tilde{y}_{t+1} \rangle + \mathbb{E}\langle \nabla_y g(y_t) - \mathbb{E}[w_t], y^* - \tilde{y}_{t+1} \rangle$$

$$- \gamma\langle \nabla_y g(y_t), \mathbb{E}[w_t] \rangle + \frac{\mu}{2}\|y^* - y_t\|^2 \qquad (54)$$

where the expectation is *w.r.t* the stochasity of $\omega_t$. Then by $L$-smoothness, we have:

$$\frac{L}{2}\mathbb{E}\|\tilde{y}_{t+1} - y_t\|^2 \geq \mathbb{E}g(\tilde{y}_{t+1}) - g(y_t) + \gamma\langle \nabla_y g(y_t), \mathbb{E}[\omega_t] \rangle \qquad (55)$$

Combining the 54 with 55, we have

$$g(y^*) \geq \mathbb{E}g(\tilde{y}_{t+1}) + \mathbb{E}\langle \mathbb{E}[w_t], y^* - \tilde{y}_{t+1}\rangle + \mathbb{E}\langle \nabla_y g(y_t) - \mathbb{E}[w_t], y^* - \tilde{y}_{t+1}\rangle$$

$$+ \frac{\mu}{2}\|y^* - y_t\|^2 - \frac{L}{2}\mathbb{E}\|\tilde{y}_{t+1} - y_t\|^2$$

$$\geq \mathbb{E}g(\tilde{y}_{t+1}) + \gamma\|\mathbb{E}[w_t]\|^2 + \langle \mathbb{E}[w_t], y^* - y_t\rangle + \mathbb{E}\langle \nabla_y g(y_t) - \mathbb{E}[w_t], y^* - \tilde{y}_{t+1}\rangle$$

$$+ \frac{\mu}{2}\|y^* - y_t\|^2 - \frac{L\gamma^2}{2}\mathbb{E}\|\omega_t\|^2$$

$$\geq \mathbb{E}g(\tilde{y}_{t+1}) + \langle \mathbb{E}[w_t], y^* - y_t\rangle + \mathbb{E}\langle \nabla_y g(y_t) - \mathbb{E}[w_t], y^* - \tilde{y}_{t+1}\rangle$$

$$+ \frac{\mu}{2}\|y^* - y_t\|^2 + \left(\gamma - \frac{L\gamma^2}{2}\right)\|\mathbb{E}[\omega_t]\|^2 - \frac{L\gamma^2}{2}Var[\omega_t]$$

where $Var$ denotes the variance. By definition of $y^*$, we have $g(y^*) \geq g(\tilde{y}_{t+1})$. Thus, we obtain

$$0 \geq \langle \mathbb{E}[w_t], y^* - y_t\rangle + \mathbb{E}\langle \nabla_y g(y_t) - \mathbb{E}[w_t], y^* - \tilde{y}_{t+1}\rangle$$

$$+ \frac{\mu}{2}\|y^* - y_t\|^2 + \left(\gamma - \frac{L\gamma^2}{2}\right)\|\mathbb{E}[\omega_t]\|^2 - \frac{L\gamma^2}{2}Var[\omega_t] \qquad (56)$$

Considering the upper bound of the second term $\langle \nabla_y g(y_t) - w_t, y^* - y_{t+1}\rangle$, we have

$$-\mathbb{E}\langle \nabla_y g(y_t) - \mathbb{E}[w_t], y^* - \tilde{y}_{t+1}\rangle$$

$$= -\langle \nabla_y g(y_t) - \mathbb{E}[w_t], y^* - y_t\rangle + \langle \nabla_y g(y_t) - \mathbb{E}[w_t], \mathbb{E}[\omega_t]\rangle$$

$$\leq \frac{1}{\mu}\|\nabla_y g(y_t) - \mathbb{E}[w_t]\|^2 + \frac{\mu}{4}\|y^* - y_t\|^2] + \frac{1}{\mu}\|\nabla_y g(y_t) - \mathbb{E}[w_t]\|^2 + \frac{\mu\gamma^2}{4}\|\mathbb{E}[\omega_t]\|^2$$

$$= \frac{2}{\mu}\|\nabla_y g(y_t) - \mathbb{E}[w_t]\|^2 + \frac{\mu}{4}\|y^* - y_t\|^2 + \frac{\mu\gamma^2}{4}\|\mathbb{E}[\omega_t]\|^2.$$

Combining with Eq. 56:

$$0 \geq \langle \mathbb{E}[w_t], y^* - y_t\rangle - \frac{2}{\mu}\|\nabla_y g(y_t) - \mathbb{E}[w_t]\|^2 + \left(\gamma - \frac{3L\gamma^2}{4}\right)\|\mathbb{E}[w_t]\|^2 + \frac{\mu}{4}\|y^* - y_t\|^2 - \frac{L\gamma^2}{2}Var[\omega_t]$$

By $y_{t+1} = y_t - \gamma\alpha_t\omega_t$, we have:

$$\mathbb{E}\|y_{t+1} - y^*\|^2 = \mathbb{E}\|y_t - \gamma\alpha_t\omega_t - y^*\|^2 = \|y_t - y^*\|^2 - 2\gamma\alpha_t\langle \mathbb{E}[\omega_t], y_t - y^*\rangle + \gamma^2\alpha_t^2\mathbb{E}[\|\omega_t\|^2]$$

$$\leq \left(1 - \frac{\mu\gamma\alpha_t}{2}\right)\|y_t - y^*\|^2 - 2\gamma\alpha_t\left(\gamma - \frac{\gamma\alpha_t}{2} - \frac{3L\gamma^2}{4}\right)\|\mathbb{E}[\omega_t]\|^2$$

$$+ \frac{4\gamma\alpha_t}{\mu}\|\nabla_y g(y_t) - \mathbb{E}[w_t]\|^2 + (L\gamma^3\alpha_t + \gamma^2\alpha_t^2)Var[\omega_t]$$

Then since we choose $\gamma < \frac{1}{2L}$, $\alpha_t < 1$, we obtain:

$$\mathbb{E}\|y_{t+1} - y^*\|^2 \leq (1 - \frac{\mu\gamma\alpha_t}{2})\|y^* - y_t\|^2 - \frac{\gamma^2\alpha_t}{4}\|\mathbb{E}[\omega_t]\|^2$$

$$+ \frac{4\gamma\alpha_t}{\mu}\|\nabla_y g(x_t, y_t) - \mathbb{E}[w_t]\|^2 + \frac{3\gamma^2\alpha_t}{2}Var[\omega_t].$$

This completes the proof. $\qquad\qquad\square$