# OpenReview forum: "Communication-Efficient Federated Bilevel Optimization with Global and Local Lower Level Problems"
_NeurIPS.cc/2023/Conference — NeurIPS 2023 poster_

### Official Review · Reviewer_Cbyn · 2023-06-30

**Soundness:** 3 good
**Presentation:** 2 fair
**Contribution:** 2 fair
**Rating:** 5
**Confidence:** 3

**Summary:**

The paper studied the federated bilevel optimization problems, in which the upper level losses are non convex and lover level losses are strongly convex. Two algorithms, named as FedBiOAcc and FedBiOAcc-Local,  were proposed for whether the lower level problems requires to be consensus or not. The authors provided both communication can computation complexity bounds for the proposed algorithms. Two real-world tasks are adopted to validate the theoretical results.

**Strengths:**

The authors studied the bilevel optimization problems under federated learning framework and provided sufficient research motivation and literature review. They proposed recovering the hyper-gradient via solving a federated optimization problem, which is novel.

**Weaknesses:**

There have been many existing works on federated/distributed bilevel optimization. The contribution of this paper might not be significant. Also, the numerical comparison needs to be more fair.

**Questions:**

* It is a good idea to estimate the hyper-gradient via solving the federated quadric problem.
   * However, during the updating iteration, both $x_t$ and $y_t$ will be changing, and thus $l$. It leaves me concern that whether $u_t$ would converge.
   * Also how would the initialization of $u_t$ affect the algorithm convergence?
   * in line 161, would $u^*$ depend on $t$?
   * BTW, would you remind me the meaning of subscript of $2$ in $\nabla l^{(m)}$ in line 149? It looks like the definition is missing.

* The paper leveraged the STORM type of variance reduction to accelerate the convergence. However, in the original STORM [1], as well as its variant SUSTAIN [2], the initial batch size is a fixed number. Why does it need $O(M^{-1}\epsilon^{-0.5})$ initial batch size?

* The numerical studies looks somehow unfair to me.
  *  In terms of communication, most of the other algorithms only transmit two vectors $x_t$ and $y_t$ but FedBiOAcc needs three vectors $x_t$, $y_t$ and $u_t$. Instead of looking at communication rounds, would the communicated bytes be a more fair metric to measure the communication loads?
  * The authors also didn't report the computation cost. Rather than the simple convergence rate (which omit lots of key factors), the computation time / number of gradient/Hessian computation would be a better metric?
  * BTW, what is the FedBio? It doesn't have any description. What's the different between FedBio and FedBioAcc?

Some minor questions
* Why doesn't the paper also include local-upper-shared-lower level problems?
* Would the results or algorithms be extended to the case with strongly convex / convex upper level problem?

[1] Momentum-based variance reduction in non-convex sgd
[2] A Near-Optimal Algorithm for Stochastic Bilevel Optimization via Double-Momentum

**Limitations:**

The authors didn't mention any limitation and potential negative societal impact.

---

> ### Author Rebuttal · Authors · 2023-08-10
>
> Thanks for spending time reviewing our manuscript. Below are our responses to your concerns and questions:
>
> Response to Weakness:
>
> Our main contributions are three folds: Firstly, we view the estimation of hyper-gradient as solving a federated quadratic problem and therefore can get the unbiased estimation of the hyper-gradient in a communication-efficient way; Secondly, by combining with the momentum-based variance reduction, we obtain the optimal convergence rate (both the sample complexity and the communication complexity) and achieves linear speed up w.r.t. the number of clients. It is non-trivial to perform analysis which requires a careful trade-off of various types of estimation errors. Finally, we also consider the special case of local lower level problem for the first time, this type of federated bilevel problems has wide application over the personalized FL setting.
>
> Response to Questions:
>
> **Q1**: The estimation error to $x_t$, $y_t$ and $u_t$ are intertwined with each other (in particular, please refer to Lemma C.6 for a bound to the estimation error of $u_t$), and each of them is chasing a moving target. However, as shown by Theorem 3.6, $x_t$ converges to a stationary point of $h(x)$, then naturally, both $y_t$ and $u_t$  also converge as $x_t$ stops updating.
>
> The initial value of $u_t$ affects the convergence, and it appears in the constant factor of the bound in Theorem 3.6. Please refer to Line 728 in the manuscript for its explicit dependence.
>
> $u^*$ depends on t, as it is a function of the current state of the upper level variable $x_t$ as defined in Eq. 4.
>
> In line 149, we use the subscripts to differentiate the samples that are used to estimate different properties. For example, In line 149,
> $\mathcal{B}_{g,2}$ represents samples used for the estimation of hessian vector product.
>
> **Q2**: The large initial batch size is needed to mitigate the effect of client-drift error, which is generated when clients perform local updates. Note that the STEM [1] method which applies STORM to the single level FL problem also requires a large initial batch size to reach the optimal convergence rate (See corollary 1 in [1]).
>
> **Q3**:  the metric of communication rounds is widely used in Federated Optimization [1] and Federated Bilevel Optimization [2,3] works for comparison of different methods. In particular, the state-of-the-art Federated Bilevel Optimization method FedNest [2] also uses communication round for comparison, and we follow the same experimental setting as their work. In fact, the communication rounds is consistent with the theoretical analysis where we hide the constant factors.
>
> We use FedBiO to refer to the algorithm without using momentum-based variance reduction, and its update rule is shown in Eq.6 of the manuscript, where we update the lower variable, upper variable and hyper-gradient estimation variable alternatively.
>
> **Minor points**: The case of local-upper problem is also interesting, and our FedBiO/FedBiOAcc can be extended to this case by not averaging the upper level variable, we leave a more detailed discussion of this case as a future work. We can follow the same analysis framework to analyze the simpler strongly-convex/convex upper level problem case, and we also leave this as a future work.
>
> References
>
> [1] Khanduri, Prashant, et al. "Stem: A stochastic two-sided momentum algorithm achieving near-optimal sample and communication complexities for federated learning." Advances in Neural Information Processing Systems 34 (2021): 6050-6061.
>
> [2] Tarzanagh, Davoud Ataee, et al. "Fednest: Federated bilevel, minimax, and compositional optimization." International Conference on Machine Learning. PMLR, 2022.
>
> [3] Yang, Yifan, Peiyao Xiao, and Kaiyi Ji. "SimFBO: Towards Simple, Flexible and Communication-efficient Federated Bilevel Learning." arXiv preprint arXiv:2305.19442 (2023).

---

> > ### Author Response · Authors · 2023-08-10
> > **Additional Discussion about the communication and computational complexity**
> >
> > In this response, we want to further discuss the **communication and computation cost of our FedBiO/FedBiOAcc compared with other baselines**.  First, we want to clarify the **DEFINITION** of one communication round is: **performing one round of global update to the upper level variable $x_t$**.
> >
> > (**Communication cost**): **FedAvg** has the lowest communication and computational cost, however it does not perform any data cleaning and fails to converge under high heterogeneity level; **FedNest** (similar for its variants FedMBO, AggITD and CommFedBiO) first performs multiple rounds of inner update (the FedINN algorithm in FedNest), then evaluates the hyper-gradient based on the Neumann Series (the FedIHGP algorithm in FedNest). Note that every round of inner update requires communication, and evaluating the Neumann series need to transmit intermediate states (whose dimension is at the order of the model parameter) for multiple rounds . **This means that FedNest and its variants do not not only transfer $x_t$ and $y_t$ per communication round**. In comparison, our method only transfers the $u_t$, $y_t$ and $x_t$ once; **Local-BSGVR** only transfers the local and upper variable at each communication round, however, it uses local hyper-gradient to estimate the global hyper-gradient, thus only works under the homogeneous assumption. **In summary**, the 'communication round' is a fair metric in terms of comparing the communication cost, in particular, FedNest and its variants have **HIGHER** per-round communication cost than our method.
> >
> > (**Computation cost**): We compare the computation cost of our method with the FedNest algorithm (similar arguments hold for its variants such as AggITD). Theoretically, our FedBiOAcc needs $O(I = M^{-1}\kappa^{5/3}\epsilon^{-0.5})$ number of gradient queries/hessian-vector product queries per communication round, while the FedNest needs $O(\kappa^4)$. Our FedBiOAcc takes a larger local steps to obtain the optimal $O(\epsilon^{-1})$ communication rate, which is consistent with the idea of FL that we use more local computation to get lower communication cost. However, in experiments, we set $I=5$ (number of local steps) for our FedBiOAcc, and set $T=5$ (number of inner update rounds), $N=5$ (number of hyper-gradient evaluation rounds) for FedNest, so the overall computation cost is similar for both methods, yet our FedBiOAcc has a faster convergence rate.

---

> > > ### Comment · Reviewer_Cbyn · 2023-08-13
> > >
> > > Thanks authors for the response and further clarification.
> > >
> > > After going through the response, I'm still having concern on the following points:
> > >
> > > 1. [Novelty] As the authors highlighted, the first contribution is viewing the estimation of hyper-gradient as solving a federated quadratic problem. But I see there is a reference in the literature [1], which leveraged similar technique for hyper-gradient estimation. Just curious whether this paper is more advanced than [1] in terms of hyper-gradient estimation with quadratic problem from either implementation or theoretical side. Other wise, the contribution for this point would be limited.
> > >
> > > 2. [Numerical Study] I still think the authors can present the results with the total communication message rather than communication rounds for better justification. As the authors discussed, we should expect more significant gain with the proposed methods. Furthermore, as Reviewer nAuw mentioned data cleaning under FL looks forced and thus it needs better motivation.
> > >
> > > [1] Chen, Xuxing, et al. "Decentralized stochastic bilevel optimization with improved per-iteration complexity." International Conference on Machine Learning. PMLR, 2023.

---

> > > > ### Author Response · Authors · 2023-08-14
> > > >
> > > > Thanks the reviewer for comments, and we would like to respond to your concerns as follows:
> > > >
> > > >  1. [Novelty.] Although [1] also estimates hyper-gradients by solving a quadratic optimization problem, they apply this idea in a **very different** way from our algorithm (ignoring the difference of 'decentralized' and 'federated'). More specifically, as shown by Algorithm 3 of [1], each round of the algorithm includes three steps: solving lower level problem, estimate hyper-gradient (as solving a quadratic problem), and solve the upper level problem. In particular, the hyper-gradient estimation involves **multiple rounds communication** for solving the quadratic problem. **This process follows that of the FedNest [2]**, the difference is that, in FedNest, the hyper-gradient is estimated by Neumann Series instead (the FedIHGP algorithm of FedNest). Note that the Neumann Series-based estimation is equivalent to the quadratic-problem based estimation in this form. **In contrast**, we perform a three-level  federated optimization, where we update the upper-level variable, lower level variable, hyper-gradient estimation variable **alternatively** and can perform multiple rounds of local updates **without communication**.  In fact, it is crucial to view the hyper-gradient estimation as a federated optimization problem in our algorithm. This makes it possible to perform efficient alternative local update as we mentioned above and also combine with the momentum-based variance reduction to achieve optimal convergence rate.
> > > >
> > > > 2. [Numerical Study.] Our algorithm indeed exhibits more significant gains compared to FedNest [2] when considering the communication amount. For the Federated Data Cleaning task, we do not need to transfer the upper level variable (which are weights for local samples), thus we denote transferring the lower level variable once as one communication unit. Then tables below compare the performance of our FedBioAcc with FedNest:
> > > >
> > > > Table 1: Comparison between FedBiOAcc and FedNest (heterogeneity level $\rho=0.95$)
> > > >
> > > > |   Comm. Amount        | 500        | 1000       | 1500       | 2000       |
> > > > |-----------|------------|------------|------------|------------|
> > > > | FedNest   | 2.303 | 2.283 | 2.241 | 2.106 |
> > > > | FedBiOAcc | **2.302** | **1.800** | **1.423** | **1.243** |
> > > >
> > > > Table 2: Comparison between FedBiOAcc and FedNest (heterogeneity level $\rho=0.4$)
> > > >
> > > > |   Comm. Amount        | 500        | 1000       | 1500       | 2000       |
> > > > |-----------|------------|------------|------------|------------|
> > > > | FedNest   | **2.302** | 1.294 | 0.733 | 0.653 |
> > > > | FedBiOAcc | **2.302** | **0.201** | **0.140** | **0.168** |
> > > >
> > > > As shown by the tables, our algorithms outperforms FedNest significantly when measured by the total amount of communication.
> > > >
> > > > 3. (In response to Reviewer nAuw's concern) **Motivation of Federated Data Cleaning.**  Recall that we perform Federated Learning because two major constraints: insufficient local data and privacy concern. This motivation also applies to the Federated Data cleaning task. We can consider the following scenario: suppose we have client A which has a large amount of noisy data, and a client B which has a well annotated clean dataset. Since annotation is expensive and takes a lot of time and effort, client B won't share its clean data with A directly. However, client B can provide a service based on our Federated Data Cleaning formulation to help A clean the data and meanwhile gets some return.
> > > >
> > > >
> > > > References
> > > >
> > > > [1] Chen, Xuxing, et al. "Decentralized stochastic bilevel optimization with improved per-iteration complexity." International Conference on Machine Learning. PMLR, 2023.
> > > >
> > > > [2] Tarzanagh, Davoud Ataee, et al. "Fednest: Federated bilevel, minimax, and compositional optimization." International Conference on Machine Learning. PMLR, 2022.

---

> > > > > ### Comment · Reviewer_Cbyn · 2023-08-15
> > > > >
> > > > > Thanks the authors response. I raise my score from 4 to 5. But I encourage the authors to further highlight the works' challenge and contribution, so that it would not be seen as a combination of several existing techniques.

---

> > > > > > ### Author Response · Authors · 2023-08-17
> > > > > >
> > > > > > Thanks for raising the score, we appreciate it !  Below, we further summarize our work's **motivation , challenges and contributions** :
> > > > > >
> > > > > > Many tasks in Federated Learning (FL) exhibit a nested structure, such as federated data cleaning and personalized federated learning. This type of problems can be formulated as the **Federated Bilevel Optimization** problems. Several algorithms are proposed to solve the federated bilevel optimization problems in the literature [1-5.] Most of them follow the idea of a **double-loop algorithm** initially proposed in FedNest [1]. In FedNest, three steps are performed in sequential for each communication round,:  solving lower level problem, estimating hyper-gradient, and solving the upper level problem. Note that the first two steps require **multiple rounds of communication** which is expensive in FL setting. As a result, the sample complexity and communication complexity is sub-optimal compared to that of the single-level FL algorithms. However, efficient single-loop algorithms [6] exist for the non-distributed setting, and these single loop algorithms **update the lower and upper variable alternatively**, which can be potentially extended to the Federated setting. So in our work, we want to study the research question that **if an efficient single loop algorithm with optimal convergence rate exists for federated bilevel algorithms?**
> > > > > >
> > > > > > **A single loop algorithm for Federated Bilevel Optimization:** A naive way of extending a single loop algorithm to FL setting is by updating lower and upper variables alternatively on each client and then perform average periodically. However, this algorithm won't converge since **the local hyper-gradient is a biased estimation of the global hyper-gradient**.  In fact, we can get an unbiased estimate of the hyper-gradient by solving a federated quadratic problem. As a result, we propose a three-level federated problem formulation, i.e. lower-level federated problem, upper level federated problem and hyper-gradient estimation by quadratic federated optimization. By optimizing the three federated problem alternatively on clients and then perform average periodically., we get an efficient single loop algorithm for federated bilevel optimization.
> > > > > >
> > > > > > **Optimal sample and communication complexity:** In single level FL problems, we can reach the optimal communication and sample complexity through variance reduction [7], and we extend this result to the federated bilevel problems. More specifically, we use variance-reduction to control the noise of **all three levels of problems**. In summary, we reach a sample complexity of $O(\epsilon^{-1.5})$, communication complexity of $O(\epsilon^{-1})$, and linear speed-up w.r.t. the number of clients $M$.
> > > > > >
> > > > > > Finally, motivated by the important personalized FL application, we also study the type of federated bilevel optimization problems where the lower level problem is not federated. Under this setting, we can estimate the hyper-gradient on clients, while the main challenge in analysis is bounding the estimation error of Neumann series-based hyper-gradient estimation. Similar to the global lower level problem setting, we show that the optimal sample and communication complexity can also be achieved for the local lower level problem case.
> > > > > >
> > > > > > Hope this makes our contribution more clear!
> > > > > >
> > > > > > References
> > > > > >
> > > > > > [1]. Tarzanagh, Davoud Ataee, et al. "Fednest: Federated bilevel, minimax, and compositional optimization." International Conference on Machine Learning. PMLR, 2022.
> > > > > >
> > > > > > [2]. P. Xiao and K. Ji. Communication-efficient federated hypergradient computation via aggregated439
> > > > > > iterative differentiation. arXiv preprint arXiv:2302.04969, 2023.
> > > > > >
> > > > > > [3]. M. Huang, D. Zhang, and K. Ji. Achieving linear speedup in non-iid federated bilevel learning.357
> > > > > > arXiv preprint arXiv:2302.05412, 2023.
> > > > > >
> > > > > > [4].  Quan Xiao, Han Shen, Wotao Yin, and Tianyi Chen. "Alternating Implicit Projected SGD and Its Efficient Variants for Equality-constrained Bilevel Optimization." arXiv preprint arXiv:2211.07096, 2022.
> > > > > >
> > > > > > [5]. H. Gao. On the convergence of momentum-based algorithms for federated stochastic bilevel342
> > > > > > optimization problems. arXiv preprint arXiv:2204.13299, 2022.
> > > > > >
> > > > > > [6]. Yang, Junjie, Kaiyi Ji, and Yingbin Liang. "Provably faster algorithms for bilevel optimization." Advances in Neural Information Processing Systems 34 (2021): 13670-13682.
> > > > > >
> > > > > > [7]. Khanduri, Prashant, et al. "Stem: A stochastic two-sided momentum algorithm achieving near-optimal sample and communication complexities for federated learning." Advances in Neural Information Processing Systems 34 (2021): 6050-6061.

---

### Official Review · Reviewer_nAuw · 2023-07-04

**Soundness:** 3 good
**Presentation:** 3 good
**Contribution:** 3 good
**Rating:** 6
**Confidence:** 3

**Summary:**

This paper proposes an algorithm named FedBiOAcc that solves the federated bilevel optimization problem. In particular, the hyper-gradient of the bilevel optimization problem is evaluated by solving a reformulated quadratic federated optimization problem. The upper-level problem, lower-level problem and the hyper-gradient estimation are optimized in an alternative manner. A rigorous convergence analysis of the algorithm is provided. Also, the paper novelly introduces a new problem of bilevel optimization with local lower-level problems and proves the same convergence rate for a FedBiOAcc-Local algorithm that adapts to this setting.

**Strengths:**

1. Reformulation of the general federated bilevel optimization into an easier-to-solve federated quadratic optimization problem is interesting and sound.
2. The convergence analysis of both algorithms in the heterogeneous case is complex and sound.
3. The paper introduces a new problem setting with local lower-level problems, which could be even more challenging to solve. The algorithm and analysis are adapted to this new setting.
4. The formulations of the data cleaning task and the hyper-representation learning task into federated bilevel optimization problems are reasonable.

**Weaknesses:**

1. The motivation of the proposed problem can be better instantiated with actual needs or examples in real-life.

**Questions:**

1. While bilevel federated optimization is technically challenging, what is the motivation to solve bilevel problems in a federated manner? There might need more concrete motivations and a clear, practical application or usage. Is there a need for distributed computing resources in such problems? Is there a requirement to ensure data privacy for bilevel optimization practical problems?
2. The convergence analysis of the paper claims linear convergence speed-ups with the number of clients $M$. Does the convergence always become faster with the number of clients? For example, it is possible that the clients are heterogeneous. Also, does the dataset size of each client affect the convergence? It is hard to believe that given a fixed total number of samples, distributing them among more clients makes learning easier/faster.
3. [Minor] Some notations used in the paper were not explicitly explained before the first appearance. For example, the $\tau$ and $\mathcal{B}$ in Line 149. Also, what do the subscripts mean for $\mathcal{B}$?

**Limitations:**

I do not find any particular limitations in the proposed method.

---

> ### Author Rebuttal · Authors · 2023-08-10
>
> Thanks for spending time reviewing our manuscript and providing insightful comments. Below are our responses to your questions:
>
> Response to Weakness:
>
> Many real-world applications in FL exhibit a nested structure, and therefore can be solved as a federated bilevel problem. Firstly, the data-cleaning task in the FL setting can be formulated as a bilevel problem, as we discussed in section 5.1 of the manuscript. Next, various formulations of the Personalized FL has a bilevel structure: in our manuscript, we consider the hyper-representation approach in Section 5.2, where all clients jointly learn a backbone network, and each client individually learns a linear classifier with its local data. Furthermore, another way of Personalized FL is by training a subnetwork on each client, meanwhile we need train a mask for each client to do the selection.
>
> Response to Questions:
>
> **Q1**: Please refer to our response to the weakness above for some real world FL applications that are bilevel.
>
> **Q2**:  The linear speed-up w.r.t. the number of clients achieves under certain conditions. Firstly, by observing the second term of the convergence bound in Theorem 3.6, we can see the number of iterations actually scales with the **product of batch size and number of clients**, which means if the total number of samples are fixed, distributing them to more clients might not improve the convergence speed, as the batch-size might be decreased. Secondly, the linear speedup conclusion still holds when the heterogeneity exists, however, note that the heterogeneity coefficients exist in the constant factors of the convergence bound in Theorem 3.6. If we take this constant factor explicitly, we have $ \frac{1}{T}\sum_{t = 1}^{T-1} \mathbb{E} \bigg[ \|\nabla h(\bar{x}_t) \|^2 \bigg] = O(\frac{\zeta^2\kappa^{16/3}}{(bMT)^{2/3}})$, and $\zeta$ denotes the heterogeneity coefficients. This equation shows that larger heterogeneity level leads to worse convergence bound, although increasing the number of clients $M$ still decrease the bound.
>
> **Q3**: $\tau$ is the step size/learning rate to solve the quadratic problem (its value choice be found in Theorem 3.6), $\mathcal{B}$ represents mini-batch of samples used at each iteration, furthermore, we use the subscripts to differentiate the samples that are used to estimate different properties. For example, In line 149, $\mathcal{B}_{g,2}$ represents samples used for the estimation of hessian vector product. Note that, to guarantee the convergence, it is essential to use independent samples to estimate different properties (gradient, hessian-vector product etc.) at each iteration.

---

> > ### Comment · Reviewer_nAuw · 2023-08-13
> > **Thanks for the rebuttal**
> >
> > Thanks for the clarifications above.
> >
> > W1 & Q1: While the data cleaning task definitely presents a nested structure, my question is more on the need to perform this task in a federated manner. Are the advantages (e.g., distributed computing, data privacy) really necessary for these tasks (e.g., data cleaning)? If not, the marriage between bilevel optimization and FL could appear forced.

---

> > > ### Author Response · Authors · 2023-08-14
> > >
> > > Thanks for your comments. We want to respond to your concerns about the motivation of Federated Data Cleaning as follows:
> > >
> > > **Motivation of Federated Data Cleaning.**  Recall that we perform Federated Learning because of two major constraints: **insufficient local data** and **privacy concern**. This motivation also applies to the Federated Data cleaning task. We can consider the following scenario: suppose we have client A which has a large amount of noisy data, and a client B which has a well annotated clean dataset. Since annotation is expensive and takes a lot of time and effort, client B won't share its clean data with A directly. However, client B can provide a service based on our Federated Data Cleaning formulation to help A clean the data and meanwhile gets some return.
> > >
> > > Besides the Federated Data Cleaning task, Personalized FL [1,2] is also a very important application of Federated Bilevel Optimization, which is motivated by the data heterogeneity in FL: since clients have different local data distributions, a personalized model is desired on top of the jointly learned global model.
> > >
> > > References:
> > >
> > > [1] Fallah, Alireza, Aryan Mokhtari, and Asuman Ozdaglar. "Personalized federated learning: A meta-learning approach." arXiv preprint arXiv:2002.07948 (2020).
> > >
> > > [2] Collins, Liam, et al. "Exploiting shared representations for personalized federated learning." International conference on machine learning. PMLR, 2021.

---

> > > > ### Comment · Reviewer_nAuw · 2023-08-21
> > > >
> > > > I would like to thank the authors for the discussion. It would be great if you could make these application scenarios more prominent (with justifications) in the revised paper. I am happy to keep my score.

---

> > > > > ### Author Response · Authors · 2023-08-21
> > > > >
> > > > > Thanks for your comment! We will add more discussion to motivate federated bilevel optimization better in the revised version.

---

### Official Review · Reviewer_MLKp · 2023-07-06

**Soundness:** 3 good
**Presentation:** 3 good
**Contribution:** 3 good
**Rating:** 7
**Confidence:** 3

**Summary:**

This paper focuses on the problem of bilevel optimization in a federated learning or FL environment. Bilevel optimization has various FL applications, and a few recent works proposed versions of bilevel optimization schemes for FL. A challenging step in bilevel optimization is the computation of the "hypergradient", and the existing schemes are able to obtain an estimate of the hypergradient, albeit a biased estimate, with a substantial communication overhead (multiple rounds of communication per server-side update). The paper reformulates the hypergradient computation as a quadratic federated optimization problem that can provide an unbiased estimate of the hypergradient, while requiring only a single round of communication for every server-side update. Based on this insight, the paper proposes a federated bilevel optimization algorithm that updates the two variables in the bilevel problem, and the hypergradient estimate alternately on each client, with an intermittent averaging step across all clients. To improve the convergence with a constant batch size, the paper utilizes momentum based variance reduction techniques for all the three variables. The theoretical analysis establishes convergence with an improved iteration complexity compared to existing federated bilevel schemes, with a significantly improved communication complexity. Next, the paper studies an alternate form of a federated bilevel problem where all clients have local lower level variables, with applications in personalized FL. This setup allows for the local hypergradient computation without any need for communication across clients. The paper presents a modification of the previous algorithm for this problem and establishes convergence. The empirical evaluation considers two applications and compares the proposed scheme against various baselines, highlighting the improved communication complexity of the proposed scheme across various problem settings.

**Strengths:**

**Critical federated hypergradient estimation as a federated optimization.**
A key strength of this paper is a simple (yet of significant practical impact) reformulation of the hypergradient estimation using a standard quadratic program. A key property that the authors leverage is the fact that the global quadratic objective can be decomposed into per-client quadratic objectives, which is not true of the global hypergradient (which cannot be decomposed into per-client hypergradients). This simple yet powerful insight is then utilized to obtain an estimate which, upon proper solution of the global least-squares problem, is unbiased, and can be efficiently updated along side the upper and lower level variables in the bilevel problem. While this global least-squares reformulation does facilitate an intuitive communication-efficient algorithm, the paper also proposes a momentum-based variance reduction scheme, and establishes a very favorable convergence rate both in terms of iterations (and thus sample complexity) and communication. The overall algorithm makes the solution of federated bilevel problems significantly more practical.

**Favorable convergence rate.**
Along with the novel federated hypergradient estimation, this paper is also able to get a favorable convergence rate across various dimensions. First, the results are able to provide convergence with a constant batch size, and establish a sample complexity of $O(\epsilon^{-1.5})$, which is very strong for bilevel optimization. Second, the analysis establishes a $(1/M)$-dependence on the sample complexity, indicating that having more parties in the FL setup allows for smaller per-client sample complexity. Finally, as I discussed earlier, the analysis establishes a $O(\epsilon^{-1})$ communication complexity which matches the rate of the best single-level federated optimization algorithm.


**Bilevel personalized FL.**
The paper explicitly studies a form of the federated bilevel optimization where each client has its own lower level problem, and I believe this general problem would encompass various personalized FL problems and applications. While the problem is easier in terms of the computation of the hypergradient, it is still an interesting problem, and the paper establishes convergence again with favorable iteration and communication complexities.


**Positioning against existing literature.**
In addition to the section on related works, the authors do a great job at discussing their proposed scheme in comparison to existing literature during the algorithm development. For example, after presenting the main idea behind the communication efficient hypergradient computation, the paper positions this work in comparison to FedNest.

**Intuitive presentation of the theoretical results.**
In addition to establishing favorable theoretical guarantees, the authors also present the theoretical results in an intuitive manner. First, they clearly present the Lyapunov function, highlighting the different terms in it corresponding to the different variable iterates. Then the relevant parts of the Appendix are referenced which makes it easy to verify the results while reading. After the main theorem is presented, the authors also do a good job of highlighting what it implies, and how it compares to other federated algorithms.



**Weaknesses:**

**Increased hyperparameter space.**
The proposed framework utilizes various hyperparameters such as the initial learning rates $\gamma, \eta, \tau$, the learning rate schedule $\lbrace \alpha_t, t \in [T] \rbrace$, the momentum parameters $c_{\omega / \nu / u}$, the aggregation frequency $I$.
As per the theoretical analyses, it can be seen that the best convergence rate of any execution will critically depend on an appropriate setting of these problem-dependent hyperparameters. Since these hyperparameters often depend on quantities that cannot be efficiently estimated (such as Lipschitz constant), the practical bilevel implementations usually utilize some form of hyperparameter search. Hyperparameter optimization is known to be a hard unsolved problem in FL because of the overall communication overhead. This makes it hard to see how the proposed federated bilevel framework can live up to its practical potential -- one can view this proposed federated bilevel framework as having shifted the communication overhead from the model training stage to the hyperparameter optimization stage, without reducing the overall communication necessary for good training convergence (which involves trying various hyperparameters and training with them).

**Questions:**

- Line 2 in algorithm 1: Should it be $q_1^{(m)}$ since it is computed using $g^{(m)}$ and $f^{(m)}$?
- Line 176-177: The ordering of the momentum terms and the corresponding variables in the "... respectively" needs to be updated.
- In theorem 1, the results provide precise dependence on $\kappa$, but I am unable to find the definition of $\kappa$ in the theorem statement or anywhere else in the main paper. Can you please clarify what this $\kappa$ is supposed to signify?

**Limitations:**

I did not find any discussion on limitations in the main paper or the supplement. However, I do not anticipate any potential negative societal impact of this work.

---

> ### Author Rebuttal · Authors · 2023-08-09
>
> Thanks for your detailed and insightful comments.
>
> Response to Weakness:
>
> We agree with the reviewer over the increased hyper-parameter space. In fact, the optimal convergence rate of the momentum-based variance reduction technique is achieved by carefully balancing the learning rate, momentum coefficients. Next, since we view the Federated Bilevel Optimization problem as three intertwined federated optimization problems, naturally, the number of hyper-parameters is roughly three times compared to that of its non-distributed single level counterpart. In practice, we can first set the base learning rate $\frac{\delta}{u^{1/3}}$ to be around 0.1, then choose the learning rate coefficient $\gamma$, $\eta$, $\tau$ respectively according to the structure of each problem, finally, we choose the momentum coefficient $c_{\omega}$, $c_{\nu}$ and $c_u$ such that the initial momentum to be around 0.9/0.99.
>
>
> Response to Questions:
>
> **Q1**: Yes, there should be a super-script, i.e. $q_1^{(m)}$;
>
> **Q2**: Yes, the order of $\omega^{(m)}_t$ and $\nu^{(m)}_t$ should be adjusted;
>
> **Q3**: $\kappa = \frac{L}{\mu}$ denotes the condition number, and $L$ is the smoothness coefficient, $\mu$ is the strong-convexity coefficient for the lower problem.

---

> > ### Comment · Reviewer_MLKp · 2023-08-21
> > **Thank you for the author response**
> >
> > Thank for the response and for the description of the hyperparameter selection process and what definition of $\kappa$. It seems that there will still be some parameters that need to be tuned (the momentum parameters probably don't need to be tuned too much). It would be good to add this discussion in the appendix, with a broader discussion on how one might approach a practical implementation of this algorithm.
> >
> > I will continue to keep my score of 7 for this paper.

---

> > > ### Author Response · Authors · 2023-08-21
> > >
> > > Thanks for your comment. We will add this discussion of hyper-parameter selection in the appendix, furthermore, we will also add some pseudo code to illustrate the  practical implementation of our algorithm.

---

### Official Review · Reviewer_5Ddx · 2023-07-06

**Soundness:** 2 fair
**Presentation:** 2 fair
**Contribution:** 2 fair
**Rating:** 4
**Confidence:** 3

**Summary:**

this paper proposed new algorithms for federated bilevel optimization. The authors apply the idea of [1] to the federated setting, which views the hypergradient computation as solving a quadratic subproblem. Combining with the STORM algorithm, the author proposed fedbioacc with a convergence guarantee based on this idea and then extend the results to a simpler case with local lower-level problems. numerical experiments on data cleaning and hyper representation are reported.

[1] Dagréou et al. "A framework for bilevel optimization that enables stochastic and global variance reduction algorithms"

**Strengths:**

1. the proposed algorithms are pure single-loop algorithms compared with other global federated bilevel algorithms.
2. the authors provide convergence guarantees for both proposed algorithms.

**Weaknesses:**

1. One of my primary concerns is that single-loop algorithms in bilevel optimization perform worse than double-loop algorithms because of the hyper gradient estimation error introduced by the inexactness of the lower-level solution. this is different from the minimax problem because the hypergradient in bilevel optimization involves Jacobian and hessian inverse computation, thus being more complex. In practice, people are using a double-loop algorithm to reduce this hyper gradient estimation error.

2. the contribution of this paper looks trivial to me, the main ideas come from STORM and [Dagréou et al]. the proposed algorithm is very complex and thus difficult to implement and tune in practice.

**Questions:**

see weakness

**Limitations:**

see weakness

---

> ### Author Rebuttal · Authors · 2023-08-10
>
> Thanks for spending time reviewing our paper and provide comments. Below are our responses to your concerns:
>
> **W1**: We agree with the reviewer that bilevel optimization is harder than the minimax optimization problems. However, our three-level perspective of the bilevel optimization can eliminate the difficulty of dealing with the Hessian Inverse: we estimate the hyper-gradient by solving a quadratic problem, and then we perform alternative update between the upper level variable, lower level variable and the hyper-gradient estimation variable. In fact, we theoretically prove the convergence of our single loop algorithm and empirically verify its efficacy through the Federated Data Cleaning task and the Federated Hyper-representation task.
>
> **W2** Our main contributions are three folds: Firstly, we view the estimation of hyper-gradient as solving a federated quadratic problem and therefore can get the unbiased estimation of the hyper-gradient in a communication-efficient way; Secondly, by combining with the momentum-based variance reduction, we obtain the optimal convergence rate (both the sample complexity and the communication complexity)  and achieves linear speed up w.r.t. the number of clients. It is non-trivial to perform analysis which requires a careful trade-off of various types of estimation errors. Finally, we also consider the special case of local lower level problem for the first time, this type of federated bilevel problems has wide application over the personalized FL setting.
>
> As for the complexity, our algorithm is well-structured and for each of the three problems: upper level, lower level and quadratic problem, we perform the same type of acceleration operation: momentum-based variance reduction. As a result, our algorithm is straightforward to implement. Furthermore,  in terms of the hyper-parameter tuning, we can first set the base learning rate $\frac{\delta}{u^{1/3}}$ to be around 0.1, then choose the learning rate coefficient $\gamma$, $\eta$, $\tau$ respectively according to the structure of each problem, finally, we choose the momentum coefficient $c_{\omega}$, $c_{\nu}$ and $c_u$ such that the initial momentum to be around 0.9/0.99.

---

> > ### Author Response · Authors · 2023-08-17
> > **Thanks for your review**
> >
> > Dear reviewer:
> >
> > Thanks for spending time reviewing our paper again! Since the author response deadline is approaching, I want to send this message to check if you have any further concerns that we can answer?

---

> > > ### Comment · Reviewer_5Ddx · 2023-08-18
> > > **Reply to the rebuttal**
> > >
> > > I’ve read the rebuttal and decided to keep the current score.

---

> > > > ### Author Response · Authors · 2023-08-21
> > > >
> > > > Thanks for your comment!  For a more detailed discussion about our motivation, contribution and challenges of our method, please check our most recent response to Reviewer Cbyn.

---

### Official Review · Reviewer_rgL1 · 2023-07-06

**Soundness:** 3 good
**Presentation:** 3 good
**Contribution:** 2 fair
**Rating:** 5
**Confidence:** 4

**Summary:**

This paper proposed a communication efficient federated bilevel algorithm in both global and local lower-level problem setting that can achieve the best theoretical communication complexity. Extensive numerical experiments are provided to test the effectiveness of the proposed algorithm.

**Strengths:**

They provide convergence analysis for the proposed algorithm. The numerical experiments are abundant and solid to verify the effectiveness of the proposed algorithm. Also, the federated bilevel with local lower-level problem is intriguing.

**Weaknesses:**

1.	The analysis technique for bilevel method that treats the Hessian-vector product as a solution of a quadratic problem was also utilized in [R1]-[R5]. I’m afraid this would alleviate some novelty of this paper. Also, some relevant literature on fully single loop bilevel algorithms like [R1]-[R2] and communication efficient federated bilevel methods [R4]-[R5] are missing. Especially, [R4] allows for any heterogeneity without Assumption 3.5 and [R5] achieves the same communication complexity as this work, could you elaborate the pros and cons of your work and [R4]-[R5]?


[R1] Junyi Li, Bin Gu, and Heng Huang. A fully single loop algorithm for bilevel optimization without hessian inverse. In Proceedings of the AAAI Conference on Artificial Intelligence, 2022.

[R2] Michael Arbel and Julien Mairal. Amortized implicit differentiation for stochastic bilevel optimization. arXiv
preprint arXiv:2111.14580, 2021.

[R3] Mathieu Dagreou, Pierre Ablin, Samuel Vaiter, and Thomas Moreau. A framework for bilevel optimization that enables stochastic and global variance reduction algorithms. In Proceedings of the Advances in Neural Information Processing Systems, 2022.

[R4] Quan Xiao, Han Shen, Wotao Yin, and Tianyi Chen. "Alternating Implicit Projected SGD and Its Efficient Variants for Equality-constrained Bilevel Optimization." arXiv preprint arXiv:2211.07096, 2022.

[R5] Yifan Yang, Peiyao Xiao, and Kaiyi Ji. "SimFBO: Towards Simple, Flexible and Communication-efficient Federated Bilevel Learning." arXiv preprint arXiv:2305.19442, 2023.

2.	The role of heterogeneity is not expressly delineated in Theorem 3.6. It would be beneficial to see a how heterogeneity influences the convergence rate.


**Questions:**

The three-level optimization perspective appears to be applicable to the local lower-level bilevel problem as well. Is there a specific reason for selecting the Neumann series approximation for the local lower-level problem, apart from the warm start strategy?

**Limitations:**

Yes

---

> ### Author Rebuttal · Authors · 2023-08-10
>
> Thanks for your insightful comments. Below are our responses to your questions and concerns:
>
> Response to weakness:
>
> **W1** : Although the idea of viewing hyper-gradient estimation as solving a quadratic problem is investigated in several papers, these paper only consider the non-distributed setting, and we are the first one to apply this idea into the federated setting. In fact, it is even more favorable to apply this idea into the FL setting than the non-distributed setting. Since the global quadratic problem is a linear combination of the client-wise quadratic problems, we can apply the standard FedAvg algorithm to get an unbiased estimation of the true hyper-gradient in a communication-efficient way, furthermore, its optimization can also be efficiently integrated with the update of upper and lower level variables to have a three-level federated optimization problem. Furthermore,  we achieve the optimal convergence rate by combining our three-level federated optimization formulation with the momentum-based variance reduction technique.
>
> **Comparison with [4]**: This work adopts a similar strategy as FedNest, where they first update the lower level variable with multiple steps and then update the upper level variable multiple steps, in contrast, our FedBiOAcc algorithm performs a simple alternative updates between the lower level variable, upper level variable and the hyper-gradient estimation variable. Furthermore, [4] adopts the recently proposed ProxSkip for the update of the lower level variable, thus does not need the heterogeneity assumption, however, the best algorithm E2-AiPOD in [4] only sub-optimal communication rate of $O(\epsilon^{-1.5})$, while our FedBiOAcc achieves the optimal $O(\epsilon^{-1})$ rate. **Comparison with [5]**: Similar to our algorithm, SimFBO also consider a three-level optimization problem, however, it only gets sample complexity of $O(\epsilon^{-2})$ which is sub-optimal, while our FedBiOAcc achieves the optimal $O(\epsilon^{-1.5})$ sample complexity.  Finally, the convergence analysis in both [4] and [5] does not include the dependence over the condition number $\kappa$, which is an important factor to affect the performance of the algorithm in practice, and they don't consider the special case of local lower level problem, which can incorporate various Personalized FL formulations.
>
> **W2**: The term related to the heterogeneity coefficients are part of the constant factors absorbed by the Big O notation. Please refer to Line 715 and 728 in the manuscript for the precise form of the dependence, approximately, we have $\frac{1}{T}\sum_{t = 1}^{T-1} \mathbb{E} \bigg[ \|\nabla h(\bar{x}_t) \|^2 \bigg] = O(\zeta^2)$, and $\zeta$ denotes the maximum of the heterogeneity coefficients defined in Assumption 3.5.
>
> **Q1**: For the case of local lower-level bilevel problem, the local hyper-gradient is an unbiased estimation of the global hyper-gradient, therefore, we don't need to solve an extra quadratic federated optimization problem to estimate the global hyper-gradient as in the case of global lower level problem. To estimate the local hyper-gradient, we can either use Neumann series or solve a quadratic problem and they both don't need extra communication. We choose the Neumann series approach due to the following reason: Neumann series does not need to keep an extra variable $u$ as the quadratic approach, furthermore,  the Neumann series approach leads to a good approximation of the hyper-gradient with only a few number of hessian-vector products in practice, thus the quadratic approach does not outperform the Neumann series approach significantly in computational efficiency.
>
> References
>
> [1]. Quan Xiao, Han Shen, Wotao Yin, and Tianyi Chen. "Alternating Implicit Projected SGD and Its Efficient Variants for Equality-constrained Bilevel Optimization." arXiv preprint arXiv:2211.07096, 2022.
>
> [2]. Yifan Yang, Peiyao Xiao, and Kaiyi Ji. "SimFBO: Towards Simple, Flexible and Communication-efficient Federated Bilevel Learning." arXiv preprint arXiv:2305.19442, 2023.

---

> > ### Comment · Reviewer_rgL1 · 2023-08-19
> >
> > Thank you for your careful response and I decided to keep my score.

---

> > > ### Author Response · Authors · 2023-08-21
> > >
> > > Thanks for your comment!  For a more detailed discussion about our motivation, contribution and challenges of our method, please check our most recent response to Reviewer Cbyn.

---

> ### Comment · Area_Chair_L11R · 2023-08-18
>
> Hi Reviewer rgL1,
>
> The author-reviewer discussion period will end very soon. Could you please respond to the author's rebuttals?  At the minimum, please reply by acknowledging that you have read them.
>
> Thanks,
> AC

---

### Decision · Program_Chairs · 2023-09-21

**Decision:**

Accept (poster)

**Comment:**

The main strength and weakness of this paper are summarized as follows.

**Strength**:

1. The proposed method is single-loop and use a small batch size (except the initial one).

2. The communication complexity and sample complexity are the best in literature, and the linear speed-up is achieved. They are even better than the most recent method, called SimFBO, in terms of sample complexity. However, **I think the authors must cite SimFBO and include it in the table 1.**

 Yifan Yang, Peiyao Xiao, and Kaiyi Ji. "SimFBO: Towards Simple, Flexible and Communication-efficient Federated Bilevel Learning." arXiv preprint arXiv:2305.19442, 2023


**Weakness**:

1. The high-level idea is not new. Solving the Hessian-vector product as a solution of a federated quadratic problem is not new. The algorithm proposed is essentially solving three intertwined federated optimization problems using STORM momentum-based variance reduction like in [Dagréou et al] for example. As pointed out by multiple reviewers, these techniques seem well-known.

2. The single-loop method seem to not perform as well as a double-loop method in practice.

3. There are too many tuning parameters and thus hard to implement.


There are also some relatively minors issues not listed above.

My overall evaluation is that the contribution of this work outweighs its limitation. After all, **this work achieves the best sample complexity and communication complexity simultaneously in literature.** This work seems to be the new SOTA given its theoretical guarantees.